# Why (and When) does Local SGD Generalize Better than SGD?

**Xinran Gu**[*]
Institute for Interdisciplinary Information Sciences
Tsinghua University
gxr21@mails.tsinghua.edu.cn

**Kaifeng Lyu**[*]
Department of Computer Science
Princeton University
klyu@cs.princeton.edu

**Longbo Huang**[†]
Institute for Interdisciplinary Information Sciences
Tsinghua University
longbohuang@tsinghua.edu.cn

**Sanjeev Arora**[†]
Department of Computer Science
Princeton University
arora@cs.princeton.edu

## Abstract

Local SGD is a communication-efficient variant of SGD for large-scale training, where multiple GPUs perform SGD independently and average the model parameters periodically. It has been recently observed that Local SGD can not only achieve the design goal of reducing the communication overhead but also lead to higher test accuracy than the corresponding SGD baseline (Lin et al., 2020b), though the training regimes for this to happen are still in debate (Ortiz et al., 2021). This paper aims to understand why (and when) Local SGD generalizes better based on Stochastic Differential Equation (SDE) approximation. The main contributions of this paper include (i) the derivation of an SDE that captures the long-term behavior of Local SGD in the small learning rate regime, showing how noise drives the iterate to drift and diffuse after it has reached close to the manifold of local minima, (ii) a comparison between the SDEs of Local SGD and SGD, showing that Local SGD induces a stronger drift term that can result in a stronger effect of regularization, e.g., a faster reduction of sharpness, and (iii) empirical evidence validating that having a small learning rate and long enough training time enables the generalization improvement over SGD but removing either of the two conditions leads to no improvement.

## 1 Introduction

As deep models have grown larger, training them with reasonable wall-clock times has led to new distributed environments and new variants of gradient-based training. Recall that Stochastic Gradient Descent (SGD) tries to solve $\min_{\boldsymbol{\theta} \in \mathbb{R}^d} \mathbb{E}_{\xi \sim \tilde{\mathcal{D}}}[\ell(\boldsymbol{\theta}; \xi)]$, where $\boldsymbol{\theta} \in \mathbb{R}^d$ is the parameter vector of the model, $\ell(\boldsymbol{\theta}; \xi)$ is the loss function for a data sample $\xi$ drawn from the training distribution $\tilde{\mathcal{D}}$, e.g., the uniform distribution over the training set. SGD with learning rate $\eta$ and batch size $B$ does the following update at each step, using a batch of $B$ independent $\xi_{t,1}, \ldots, \xi_{t,B} \sim \tilde{\mathcal{D}}$:

$$\boldsymbol{\theta}_{t+1} \leftarrow \boldsymbol{\theta}_t - \eta \boldsymbol{g}_t, \quad \text{where} \quad \boldsymbol{g}_t = \frac{1}{B} \sum_{i=1}^{B} \nabla \ell(\boldsymbol{\theta}_t; \xi_{t,i}). \tag{1}$$

*Parallel SGD* tries to improve wall-clock time when the batch size $B$ is large enough. It distributes the gradient computation to $K \geq 2$ workers, each of whom focuses on a local batch of $B_{\text{loc}} := B/K$ samples and computes the average gradient over the local batch. Finally, $\boldsymbol{g}_t$ is obtained by averaging the local gradients over the $K$ workers.

However, large-batch training leads to a significant test accuracy drop compared to a small-batch training baseline with the same number of training steps or epochs (Smith et al., 2020; Shallue et al.,

---

[*]Equal contribution
[†]Corresponding authors

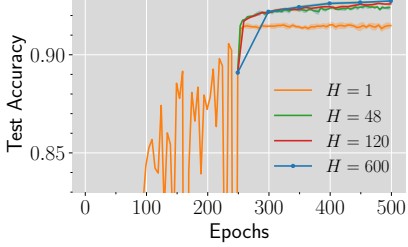 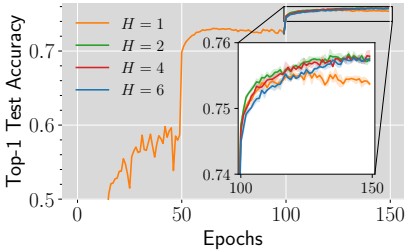

(a) CIFAR-10, $B = 4096$, ResNet-56.   (b) ImageNet, $B = 8192$, ResNet-50.

Figure 1: Post-Local SGD ($H > 1$) generalizes better than SGD ($H = 1$). We switch to Local SGD at the first learning rate decay (epoch #250) for CIFAR-10 and at the second learning rate decay (epoch #100) for ImageNet. See Appendix M.1 for training details.

2019; Keskar et al., 2017; Jastrzębski et al., 2017). Reducing this *generalization gap* is the goal of much subsequent research. It was suggested that the generalization gap arises because larger batches lead to a reduction in the level of noise in batch gradient (see Appendix A for more discussion). The *Linear Scaling Rule* (Krizhevsky, 2014; Goyal et al., 2017; Jastrzębski et al., 2017) tries to fix this by increasing the learning rate in proportion to batch size. This is found to reduce the generalization gap for (parallel) SGD, but does not entirely eliminate it.

To reduce the generalization gap further, Lin et al. (2020b) discovered that a variant of SGD, called *Local SGD* (Yu et al., 2019; Wang & Joshi, 2019; Zhou & Cong, 2018), can be used as a strong component. Perhaps surprisingly, Local SGD itself is not designed for improving generalization, but for reducing the high communication cost for synchronization among the workers, which is another important issue that often bottlenecks large-batch training (Seide et al., 2014; Strom, 2015; Chen et al., 2016; Recht et al., 2011). Instead of averaging the local gradients per step as in parallel SGD, Local SGD allows $K$ workers to train their models locally and averages the local *model parameters* whenever they finish $H$ local steps. Here every worker samples a new batch at each local step, and in this paper we focus on the case where all the workers draw samples with or without replacement from the *same* training set. See Appendix C for the pseudocode.

More specifically, Lin et al. (2020b) proposed *Post-local SGD*, a hybrid method that starts with parallel SGD (equivalent to Local SGD with $H = 1$ in math) and switches to Local SGD with $H > 1$ after a fixed number of steps $t_0$. They showed through extensive experiments that Post-local SGD significantly outperforms parallel SGD in test accuracy when $t_0$ is carefully chosen. In Figure 1, we reproduce this phenomenon on both CIFAR-10 and ImageNet.

As suggested by the success of Post-local SGD, Local SGD can improve the generalization of SGD by merely adding more local steps (while fixing the other hyperparameters), at least when the training starts from a model pre-trained by SGD. But the underlying mechanism is not very clear, and there is also controversy about when this phenomenon can happen (see Section 2.1 for a survey). The current paper tries to understand: *Why does Local SGD generalize better? Under what general conditions does this generalization benefit arise?*

Previous theoretical research on Local SGD is mainly restricted to the convergence rate for minimizing a convex or non-convex objective (see Appendix A for a survey). A related line of works (Stich, 2018; Yu et al., 2019; Khaled et al., 2020) showed that Local SGD has a slower convergence rate compared with parallel SGD after running the same number of steps/epochs. This convergence result suggests that Local SGD may implicitly regularize the model through insufficient optimization, but this does not explain why parallel SGD with early stopping, which may incur an even higher training loss, still generalizes worse than Post-local SGD.

**Our Contributions.** In this paper, we provide the first theoretical understanding on why (and when) switching from parallel SGD to Local SGD improves generalization.

1. In Section 2.2, we conduct ablation studies on CIFAR-10 and ImageNet and identify a clean setting where adding local steps to SGD consistently improves generalization: if the learning rate is small and the total number of steps is sufficient, Local SGD eventually generalizes better than the corresponding (parallel) SGD baseline.

2. In Section 3.2, we derive a special SDE that characterizes the long-term behavior of Local SGD in the small learning rate regime, as inspired by a previous work (Li et al., 2021b) that proposed this type of SDE for modeling SGD. These SDEs can track the dynamics after the iterate has reached close to a manifold of minima. In this regime, the expected gradient is near zero, but the gradient noise can drive the iterate to wander around. In contrast to the conventional SDE (3) for

SGD, where the drift and diffusion terms are connected respectively to the expected gradient and gradient noise, the SDE we derived for Local SGD has drift and diffusion terms both connected to gradient noise.

3. Section 3.3 explains the generalization improvement of Local SGD over SGD by comparing the corresponding SDEs: increasing the number of local steps $H$ strengthens the drift term of SDE while keeping the diffusion term untouched. We hypothesize that having a stronger drift term can benefit generalization.

4. As a by-product, we provide a new proof technique that can give the first quantitative approximation bound for how well Li et al. (2021b)'s SDE approximates SGD.

Back to the discussion on the generalization gap between small- and large-batch training, we remark that this gap can occur early in training when the learning rate is very large (Smith et al., 2020) and Local SGD cannot prevent this gap in this phase. Instead, our theory suggests that Local SGD can reduce the gap in late training phases after decaying the learning rate.

## 2 WHEN DOES LOCAL SGD GENERALIZE BETTER?

In our motivating example of Post-local SGD, switching from SGD to Local SGD can outperform running SGD alone (i.e., no switching) in test accuracy, but this improvement does not always arise and can depend on the choice of the switching time point. Because of this, a necessary first step for developing a theoretical understanding of Local SGD is to identify *under what general conditions* Local SGD can improve the generalization of SGD by merely adding local steps.

### 2.1 THE DEBATE ON LOCAL SGD

We first summarize a debate in the literature regarding *when* to switch from SGD to Local SGD in running Post-local SGD, which hints the conditions so that Local SGD can improve upon SGD.

**Local SGD generalizes better than SGD on CIFAR-10.** Lin et al. (2020b) empirically observed that Post-local SGD exhibits a better generalization performance than SGD. Most of their experiments are conducted on CIFAR-10 and CIFAR-100 with multiple learning rate decays, and the algorithm switches from (parallel) SGD to Local SGD right after the first learning rate decay. We refer to this particular choice of the switching time point as the *first-decay switching strategy* for short. To justify this strategy, they empirically showed that the generalization improvement can be less significant if starting Local SGD from the beginning or right after the second learning rate decay. It has also been observed by Wang & Joshi (2021) that running Local SGD from the beginning improves generalization, but the test accuracy improvement may not be large enough. A subsequent work by Lin et al. (2020a) showed that adding local steps to Extrap-SGD, a variant of SGD proposed therein, after the first learning rate decay also improves generalization, suggesting that the first-decay switching strategy can also be applied to the post-local variant of other optimizers.

**Does Local SGD exhibit the same generalization benefit on large-scale datasets?** Going beyond CIFAR-10, Lin et al. (2020b) conducted a few ImageNet experiments and showed that Post-local SGD with first-decay switching strategy still leads to better generalization than SGD. However, the improvement is sometimes marginal, e.g., $0.1\%$ for batch size $8192$. For the general case, they suggested that the time of switching should be tuned aiming at "capturing the time when trajectory starts to get into the influence basin of a local minimum" in a footnote, but no further discussion or experiments are provided to justify this guideline. Ortiz et al. (2021) conducted a more extensive evaluation on ImageNet (with a different set of hyperparameters) and concluded with the opposite: the first-decay switching strategy can hurt the validation accuracy. Instead, switching at a later time, such as the second learning rate decay, leads to a better validation accuracy than SGD.[1] To explain this phenomenon, they conjecture that switching to Local SGD has a regularization effect that is beneficial only in the short-term, so it is always better to switch as late as possible. They further conjecture that this discrepancy between CIFAR-10 and ImageNet is mainly due to the task scale. On TinyImageNet, which is a spatially downscaled subset of ImageNet, the first-decay switching strategy indeed leads to better validation accuracy.

---

[1]This generalization improvement is not mentioned explicitly in (Ortiz et al., 2021) but can be clearly seen from Figures 7 and 8 in their paper.

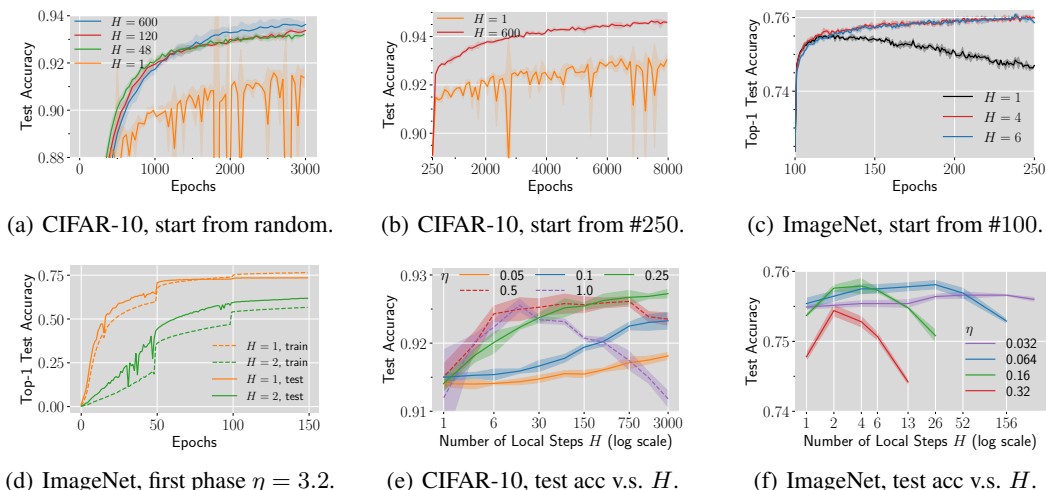

Figure 2: Ablation studies on $\eta$, $H$ and training time in the same setting as Figure 1. For (a)(d), we train from random initialization. For (b)(c)(e)(f), we start training from the checkpoints saved at the switching time points in Figure 1 (epoch #250 for CIFAR-10 and epoch #100 for ImageNet). See Appendix M.2 for training details.

## 2.2 KEY FACTORS: SMALL LEARNING RATE AND SUFFICIENT TRAINING TIME

All the above papers agree that Post-local/Local SGD improves upon SGD to some extent. However, it is in debate under what conditions the generalization benefit can consistently occur. We now conduct ablation studies to identify the key factors so that adding local steps improves the generalization of SGD. We run parallel SGD and Local SGD with the same learning rate $\eta$, local batch size $B_{\mathrm{loc}}$, and number of workers $K$. We start training from the same initialization and compare their generalization after the same number of epochs. As Post-local SGD can be viewed as Local SGD starting from an SGD-pretrained model, the initial point in our experiments can be either random or a checkpoint of SGD training. See Appendix C for implementation details and Appendix M.2 for more details about the experimental setup.

The first observation we have is that the generalization benefits can be reproduced on both CIFAR-10 and ImageNet in our setting (see Figure 1). We remark that Post-local SGD and SGD in Lin et al. (2020b); Ortiz et al. (2021) are implemented with accompanying Nesterov momentum terms. The learning rate also decays a couple of times in training with Local SGD. Nevertheless, our experiments show that the Nesterov momentum and learning rate decay are *not* necessary for Local SGD to generalize better than SGD. Our main finding after further ablation studies is summarized below:

**Finding 2.1.** *Given a sufficiently small learning rate and a sufficiently long training time, Local SGD exhibits better generalization than SGD, if the number of local steps $H$ per round is tuned properly according to the learning rate. This holds for both training from random initialization and from pre-trained models.*

Now we go through each point of our main finding. See also Appendix F for more plots.

**(1). Pretraining is not necessary.** In contrast to previous works claiming the benefits of Post-local SGD over Local SGD (Lin et al., 2020b; Ortiz et al., 2021), we observe that Local SGD with random initialization also generalizes significantly better than SGD, as long as the learning rate is small and the training time is sufficiently long (Figure 2(a)). Starting from a pretrained model may shorten the time to reach this generalization benefit to show up (Figure 2(b)), but it is not necessary.

**(2). Learning rate should be small.** We experiment with a wide range of learning rates to conclude that setting a small learning rate is necessary. The learning rate is $0.32$ for Figures 2(a) and 2(b) and is $0.16$ for Figure 2(c). As shown in Figure 2(d), Local SGD encounters optimization difficulty in the first phase where $\eta$ is large ($\eta = 3.2$), resulting in inferior final test accuracy. Even for training from a pretrained model, the generalization improvement of Local SGD disappears for large learning rates (e.g., $\eta = 1.6$ in Figure 5(d)). In contrast, if a longer training time is allowed, reducing the learning rate of Local SGD does not lead to test accuracy drop (Figure 5(c)).

**(3). Training time should be long enough.** To investigate the effect of training time, in Figures 2(b) and 2(c), we extend the training budget for the Post-local SGD experiments in Figure 1 and observe that a longer training time leads to greater generalization improvement upon SGD. On the other hand, Local SGD generalizes worse than SGD in the first few epochs of Figures 2(a) and 2(c); see Figures 5(a) and 5(b) for an enlarged view.

**(4). The number of local steps $H$ should be tuned carefully.** The number of local steps $H$ has a complex interplay with the learning rate $\eta$, but generally speaking, a smaller $\eta$ needs a higher $H$ to achieve consistent generalization improvement. For CIFAR-10 with a post-local training budget of 250 epochs (see Figure 2(e)), the test accuracy first rises as $H$ increases, and begins to fall as $H$ exceeds some threshold for relatively large $\eta$ (e.g., $\eta \geq 0.5$) while keeps growing for smaller $\eta$ (e.g., $\eta < 0.5$). For ImageNet with a post-local training budget of 50 epochs (see Figure 2(f)), the test accuracy first increases and then decreases in $H$ for all learning rates.

**Reconciling previous works.** Our finding can help to settle the debate presented in Section 2.1 to a large extent. Simultaneously requiring a small learning rate and sufficient training time poses a trade-off when learning rate decay is used with a limited training budget: switching to Local SGD earlier may lead to a large learning rate, while switching later makes the generalization improvement of Local SGD less noticeable due to fewer update steps. It is thus unsurprising that first-decay switching strategy is not always the best. The need for sufficient training time does not contradict with Ortiz et al. (2021)'s conjecture that Local SGD only has a "short-term" generalization benefit. In their experiments, the generalization improvement usually disappears right after the next learning rate decay (instead of after a fixed amount of time). We suspect that the real reason why the improvement vanishes is that the number of local steps $H$ was kept as a constant, but our finding suggests tuning $H$ after $\eta$ changes. In Figure 5(e), we reproduce this phenomenon and show that increasing $H$ after learning rate decay retains the improvement.

**Generalization performances at the optimal learning rate of SGD.** In practice, the learning rate of SGD is usually tuned to achieve the best training loss/validation accuracy within a fixed training budget. Our finding suggests that when the tuned learning rate is small and the training time is sufficient, Local SGD can offer generalization improvement over SGD. As an example, in our experiments on training from an SGD-pretrained model, the optimal learning rate for SGD is $0.5$ on CIFAR-10 (Figure 2(e)) and $0.064$ on ImageNet (Figure 2(f)). With the same learning rate as SGD, the test accuracy is improved by $1.1\%$ on CIFAR-10 and $0.3\%$ on ImageNet when using Local SGD with $H = 750$ and $H = 26$ respectively. The improvement could become even higher if the learning rate of Local SGD is carefully tuned.

## 3 Theoretical Analysis of Local SGD: The Slow SDE

In this section, we adopt an SDE-based approach to rigorously establish the generalization benefit of Local SGD in a general setting. Below, we first identify the difficulty of adapting the SDE framework to Local SGD. Then, we present our novel SDE characterization of Local SGD around the manifold of minimizers and explain the generalization benefit of Local SGD with our SDE.

**Notations.** We follow the notations in Section 1. We denote by $\eta$ the learning rate, $K$ the number of workers, $B$ the (global) batch size, $B_{\mathrm{loc}} := B/K$ the local batch size, $H$ the number of local steps, $\ell(\boldsymbol{\theta}; \zeta)$ the loss function for a data sample $\zeta$, and $\tilde{\mathcal{D}}$ the training distribution. Furthermore, we define $\mathcal{L}(\boldsymbol{\theta}) := \mathbb{E}_{\xi \sim \tilde{\mathcal{D}}}[\ell(\boldsymbol{\theta}; \xi)]$ as the expected loss, $\boldsymbol{\Sigma}(\boldsymbol{\theta}) := \mathrm{Cov}_{\xi \sim \tilde{\mathcal{D}}}[\nabla \ell(\boldsymbol{\theta}; \xi)]$ as the noise covariance of gradients at $\boldsymbol{\theta}$. Let $\{\boldsymbol{W}_t\}_{t \geq 0}$ denote the standard Wiener process. For a mapping $F : \mathbb{R}^d \to \mathbb{R}^d$, denote by $\partial F(\boldsymbol{\theta})$ the Jacobian at $\boldsymbol{\theta}$ and $\partial^2 F(\boldsymbol{\theta})$ the second order derivative at $\boldsymbol{\theta}$. Furthermore, for any matrix $\boldsymbol{M} \in \mathbb{R}^{d \times d}$, $\partial^2 F(\boldsymbol{\theta})[\boldsymbol{M}] = \sum_{i \in [d]} \langle \frac{\partial^2 F_i}{\partial \boldsymbol{\theta}^2}, \boldsymbol{M} \rangle \boldsymbol{e}_i$ where $\boldsymbol{e}_i$ is the $i$-th vector of the standard basis. We write $\partial^2(\nabla \mathcal{L})(\boldsymbol{\theta})[\boldsymbol{M}]$ as $\nabla^3 \mathcal{L}(\boldsymbol{\theta})[\boldsymbol{M}]$ for short.

**Local SGD.** We use the following formulation of Local SGD for theoretical analysis. See also Appendix C for the pseudocode. Local SGD proceeds in multiple rounds of model averaging, where each round produces a global iterate $\bar{\boldsymbol{\theta}}^{(s)}$. In the $(s+1)$-th round, every worker $k \in [K]$ starts with its local copy of the global iterate $\boldsymbol{\theta}_{k,0}^{(s)} \leftarrow \bar{\boldsymbol{\theta}}^{(s)}$ and does $H$ steps of SGD with local batches. In the $t$-th local step of the $k$-th worker, it draws a local batch of $B_{\mathrm{loc}} := B/K$ independent samples $\xi_{k,t,1}^{(s)}, \ldots, \xi_{k,t,B_{\mathrm{loc}}}^{(s)}$ from a shared training distribution $\tilde{\mathcal{D}}$ and updates as follows:

$$\boldsymbol{\theta}_{k,t+1}^{(s)} \leftarrow \boldsymbol{\theta}_{k,t}^{(s)} - \eta \boldsymbol{g}_{k,t}^{(s)}, \quad \text{where} \quad \boldsymbol{g}_{k,t}^{(s)} = \frac{1}{B_{\mathrm{loc}}} \sum_{i=1}^{B_{\mathrm{loc}}} \nabla \ell(\boldsymbol{\theta}_{k,t}^{(s)}; \xi_{k,t,i}^{(s)}), \quad t = 0, \ldots, H-1. \quad (2)$$

The local updates on different workers are independent of each other as there is no communication. After finishing the $H$ local steps, the workers aggregate the resulting local iterates $\boldsymbol{\theta}_{k,H}^{(s)}$ and assign the average to the next global iterate: $\bar{\boldsymbol{\theta}}^{(s+1)} \leftarrow \frac{1}{K} \sum_{k=1}^{K} \boldsymbol{\theta}_{k,H}^{(s)}$.

### 3.1 DIFFICULTY OF ADAPTING THE SDE FRAMEWORK TO LOCAL SGD

A widely-adopted approach to understanding the dynamics of SGD is to approximate it from a continuous perspective with the following SDE (3), which we call the *conventional SDE approximation*. Below, we discuss why it cannot be directly adopted to characterize the behavior of Local SGD.

$$\mathrm{d}\boldsymbol{X}(t) = -\nabla\mathcal{L}(\boldsymbol{X})\mathrm{d}t + \sqrt{\tfrac{\eta}{B}}\boldsymbol{\Sigma}^{1/2}(\boldsymbol{X})\mathrm{d}\boldsymbol{W}_t. \tag{3}$$

It is proved by Li et al. (2019a) that this SDE is a first-order approximation to SGD, where each discrete step corresponds to a continuous time interval of $\eta$. Several previous works adopt this SDE approximation and connect good generalization to having a large diffusion term $\sqrt{\tfrac{\eta}{B}}\boldsymbol{\Sigma}^{1/2}\mathrm{d}\boldsymbol{W}_t$ in the SDE (Jastrzębski et al., 2017; Smith et al., 2020), because a suitable amount of noise can be necessary for large-batch training to generalize well (see also Appendix A).

According to Finding 2.1, it is tempting to consider the limit $\eta \to 0$ and see if Local SGD can also be modeled via a variant of the conventional SDE. In this case the typical time length that guarantees a good SDE approximation error is $\mathcal{O}(\eta^{-1})$ discrete steps (Li et al., 2019a; 2021a). However, this time scaling is too short for the difference to appear between Local SGD and SGD. Indeed, Theorem 3.1 below shows that they closely track each other for $\mathcal{O}(\eta^{-1})$ steps.

**Theorem 3.1.** *Assume that the loss function $\mathcal{L}$ is $\mathcal{C}^3$-smooth with bounded second and third order derivatives and that $\nabla\ell(\boldsymbol{\theta};\xi)$ is bounded. Let $T > 0$ be a constant, $\bar{\boldsymbol{\theta}}^{(s)}$ be the s-th global iterate of Local SGD and $\boldsymbol{w}_t$ be the t-th iterate of SGD with the same initialization $\boldsymbol{w}_0 = \bar{\boldsymbol{\theta}}^{(0)}$ and same $\eta, B_{\mathrm{loc}}, K$. Then for any $H \le \frac{T}{\eta}$ and $\delta = \mathcal{O}(\mathrm{poly}(\eta))$, it holds with probability at least $1 - \delta$ that for all $s \le \frac{T}{\eta H}$, $\|\bar{\boldsymbol{\theta}}^{(s)} - \boldsymbol{w}_{sH}\|_2 = \mathcal{O}(\sqrt{\eta \log \frac{1}{\eta\delta}})$.*

We defer the proof to Appendix I. See also Appendix D for Lin et al. (2020b)'s attempt to model Local SGD with multiple conventional SDEs and discussions on why it does not give much insight.

### 3.2 SDE APPROXIMATION NEAR THE MINIMIZER MANIFOLD

Inspired by a recent paper (Li et al., 2021b), our strategy to overcome the shortcomings of the conventional SDE is to design a new SDE that can guarantee a good approximation for $\mathcal{O}(\eta^{-2})$ discrete steps, much longer than the $\mathcal{O}(\eta^{-1})$ discrete steps for the conventional SDE. Following their setting, we assume the existence of a manifold $\Gamma$ consisting only of local minimizers and track the global iterate $\bar{\boldsymbol{\theta}}^{(s)}$ around $\Gamma$ after it takes $\tilde{\mathcal{O}}(\eta^{-1})$ steps to approach $\Gamma$. Though the expected gradient $\nabla\mathcal{L}$ is near zero around $\Gamma$, the dynamics are still non-trivial because the noise can drive the iterate to move a significant distance in $\mathcal{O}(\eta^{-2})$ steps.

**Assumption 3.1.** *The loss function $\mathcal{L}(\cdot)$ and the matrix square root of the noise covariance $\boldsymbol{\Sigma}^{1/2}(\cdot)$ are $\mathcal{C}^\infty$-smooth. Besides, we assume that $\|\nabla\ell(\boldsymbol{\theta};\xi)\|_2$ is bounded by a constant for all $\boldsymbol{\theta}$ and $\xi$.*

**Assumption 3.2.** *$\Gamma$ is a $\mathcal{C}^\infty$-smooth, $(d - m)$-dimensional submanifold of $\mathbb{R}^d$, where any $\boldsymbol{\zeta} \in \Gamma$ is a local minimizer of $\mathcal{L}$. For all $\boldsymbol{\zeta} \in \Gamma$, $\mathrm{rank}(\nabla^2\mathcal{L}(\boldsymbol{\zeta})) = m$. Additionally, there exists an open neighborhood of $\Gamma$, denoted as $U$, such that $\Gamma = \arg\min_{\boldsymbol{\theta} \in U} \mathcal{L}(\boldsymbol{\theta})$.*

**Assumption 3.3.** *$\Gamma$ is a compact manifold.*

The smoothness assumption on $\mathcal{L}$ is generally satisfied when we use smooth activation functions, such as Swish (Ramachandran et al., 2017), softplus and GeLU (Hendrycks & Gimpel, 2016), which work equally well as ReLU in many circumstances. The existence of a minimizer manifold with $\mathrm{rank}(\nabla^2\mathcal{L}(\boldsymbol{\zeta})) = m$ has also been made as a key assumption in Fehrman et al. (2020); Li et al. (2021b); Lyu et al. (2022), where $\mathrm{rank}(\nabla^2\mathcal{L}(\boldsymbol{\zeta})) = m$ ensures that the Hessian is maximally non-degenerate on the manifold and implies that the tangent space at $\boldsymbol{\zeta} \in \Gamma$ equals the null space of $\nabla^2\mathcal{L}(\boldsymbol{\zeta})$. The last assumption is made to prevent the analysis from being too technically involved.

Our SDE for Local SGD characterizes the training dynamics near $\Gamma$. For ease of presentation, we define the following projection operators $\Phi, P_{\boldsymbol{\zeta}}$ for points and differential forms respectively.

**Definition 3.1** (Gradient Flow Projection). *Fix a point $\boldsymbol{\theta}_{\mathrm{null}} \notin \Gamma$. For $\boldsymbol{x} \in \mathbb{R}^d$, consider the gradient flow $\frac{\mathrm{d}\boldsymbol{x}(t)}{\mathrm{d}t} = -\nabla\mathcal{L}(\boldsymbol{x}(t))$ with $\boldsymbol{x}(0) = \boldsymbol{x}$. We denote the gradient flow projection of $\boldsymbol{x}$ as $\Phi(\boldsymbol{x})$. $\Phi(\boldsymbol{x}) := \lim_{t \to +\infty} \boldsymbol{x}(t)$ if the limit exists and belongs to $\Gamma$; otherwise, $\Phi(\boldsymbol{x}) = \boldsymbol{\theta}_{\mathrm{null}}$.*

**Definition 3.2.** *For any $\boldsymbol{\zeta} \in \Gamma$ and any differential form $\boldsymbol{A}\mathrm{d}\boldsymbol{W}_t + \boldsymbol{b}\mathrm{d}t$ in Itô calculus, where $\boldsymbol{A}$ is a matrix and $\boldsymbol{b}$ is a vector, we use $P_{\boldsymbol{\zeta}}(\boldsymbol{A}\mathrm{d}\boldsymbol{W}_t + \boldsymbol{b}\mathrm{d}t)$ as a shorthand for the differential form $\partial\Phi(\boldsymbol{\zeta})\boldsymbol{A}\mathrm{d}\boldsymbol{W}_t + \left(\partial\Phi(\boldsymbol{\zeta})\boldsymbol{b} + \frac{1}{2}\partial^2\Phi(\boldsymbol{\zeta})[\boldsymbol{A}\boldsymbol{A}^\top]\right)\mathrm{d}t$.*

See Øksendal (2013) for an introduction to Itô calculus. Here $P_\zeta$ equals $\Phi(\zeta + A\mathrm{d}W_t + b\mathrm{d}t) - \Phi(\zeta)$ by Itô calculus, which means that $P_\zeta$ projects an infinitesimal step from $\zeta$, so that $\zeta$ after taking the projected step does not leave the manifold $\Gamma$. It can be shown by simple calculus that $\partial\Phi(\zeta)$ equals the projection matrix onto the tangent space of $\Gamma$ at $\zeta$. We decompose the noise covariance $\Sigma(\zeta)$ for $\zeta \in \Gamma$ into two parts: the noise in the tangent space $\Sigma_\|(\zeta) := \partial\Phi(\zeta)\Sigma(\zeta)\partial\Phi(\zeta)$ and the noise in the rest $\Sigma_\Diamond(\zeta) := \Sigma(\zeta) - \Sigma_\|(\zeta)$. Now we are ready to state our SDE for Local SGD.

**Definition 3.3** (Slow SDE for Local SGD). *Given $\eta, H > 0$ and $\zeta_0 \in \Gamma$, define $\zeta(t)$ as the solution of the following SDE with initial condition $\zeta(0) = \zeta_0$:*

$$\mathrm{d}\zeta(t) = P_\zeta\Big(\underbrace{\tfrac{1}{\sqrt{B}}\Sigma_\|^{1/2}(\zeta)\mathrm{d}W_t}_{\textit{(a) diffusion}} \underbrace{-\tfrac{1}{2B}\nabla^3\mathcal{L}(\zeta)[\widehat{\Sigma}_\Diamond(\zeta)]\mathrm{d}t}_{\textit{(b) drift-I}} \underbrace{-\tfrac{K-1}{2B}\nabla^3\mathcal{L}(\zeta)[\widehat{\Psi}(\zeta)]\mathrm{d}t}_{\textit{(c) drift-II}}\Big). \qquad (4)$$

*Here $\widehat{\Sigma}_\Diamond(\zeta), \widehat{\Psi}(\zeta) \in \mathbb{R}^{d\times d}$ are defined as*

$$\widehat{\Sigma}_\Diamond(\zeta) := \sum_{i,j:(\lambda_i\neq 0)\vee(\lambda_j\neq 0)} \tfrac{1}{\lambda_i+\lambda_j}\left\langle\Sigma_\Diamond(\zeta), v_iv_j^\top\right\rangle v_iv_j^\top, \qquad (5)$$

$$\widehat{\Psi}(\zeta) := \sum_{i,j:(\lambda_i\neq 0)\vee(\lambda_j\neq 0)} \tfrac{\psi(\eta H\cdot(\lambda_i+\lambda_j))}{\lambda_i+\lambda_j}\left\langle\Sigma_\Diamond(\zeta), v_iv_j^\top\right\rangle v_iv_j^\top, \qquad (6)$$

*where $\{v_i\}_{i=1}^d$ is a set of eigenvectors of $\nabla^2\mathcal{L}(\zeta)$ that forms an orthonormal eigenbasis, and $\lambda_1,\ldots,\lambda_d$ are the corresponding eigenvalues. Additionally, $\psi(x) := \tfrac{e^{-x}-1+x}{x}$ for $x \neq 0$ and $\psi(0) = 0$.*

The use of $P_\zeta$ keeps $\zeta(t)$ on the manifold $\Gamma$ through projection. $\Sigma_\|^{\frac{1}{2}}(\zeta)$ introduces a diffusion term to the SDE in the tangent space. The two drift terms involve $\widehat{\Sigma}_\Diamond(\cdot)$ and $\widehat{\Psi}(\cdot)$, which can be intuitively understood as rescaling the entries of the noise covariance in the eigenbasis of Hessian. In the special case where $\nabla^2\mathcal{L} = \mathrm{diag}(\lambda_1,\cdots,\lambda_d) \in \mathbb{R}^{d\times d}$, we have $\widehat{\Sigma}_{\Diamond,i,j} = \tfrac{1}{\lambda_i+\lambda_j}\Sigma_{0,i,j}$. $\widehat{\Psi}_{i,j} = \tfrac{\psi(\eta H(\lambda_i+\lambda_j))}{\lambda_i+\lambda_j}\Sigma_{0,i,j}$. $\psi(x)$ is a monotonically increasing function, which goes from 0 to 1 as $x$ goes from 0 to infinity (see Figure 9)

We name this SDE as the *Slow SDE for Local SGD* because we will show that each discrete step of Local SGD corresponds to a continuous time interval of $\eta^2$ instead of an interval of $\eta$ in the conventional SDE. In this sense, our SDE is "slower" than the conventional SDE (and hence can track a longer horizon). This Slow SDE is inspired by Li et al. (2021b). Under nearly the same set of assumptions, they proved that SGD can be tracked by an SDE that is essentially equivalent to (4) with $K = 1$, namely, without the drift-II term.

$$\mathrm{d}\zeta(t) = P_\zeta\Big(\underbrace{\tfrac{1}{\sqrt{B}}\Sigma_\|^{1/2}(\zeta)\mathrm{d}W_t}_{\textit{(a) diffusion}} \underbrace{-\tfrac{1}{2B}\nabla^3\mathcal{L}(\zeta)[\widehat{\Sigma}_\Diamond(\zeta)]\mathrm{d}t}_{\textit{(b) drift-I}}\Big), \qquad (7)$$

We refer to (7) as the *Slow SDE for SGD*. We remark that the drfit-II term in (4) is novel and is the key to separate the generalization behaviors of Local SGD and SGD in theory. We will discuss this point later in Section 3.3. Now we present our SDE approximation theorem for Local SGD.

**Theorem 3.2.** *Let Assumptions 3.1 to 3.3 hold. Let $T > 0$ be a constant and $\zeta(t)$ be the solution to (4) with the initial condition $\zeta(0) = \Phi(\bar{\theta}^{(0)}) \in \Gamma$. If $H$ is set to $\tfrac{\alpha}{\eta}$ for some constant $\alpha > 0$, then for any $\mathcal{C}^3$-smooth function $g(\theta)$, $\max_{0\leq s\leq\frac{T}{H\eta^2}}\left|\mathbb{E}[g(\Phi(\bar{\theta}^{(s)}))] - \mathbb{E}[g(\zeta(sH\eta^2))]\right| = \tilde{\mathcal{O}}(\eta^{0.25})$, where $\tilde{\mathcal{O}}(\cdot)$ hides log factors and constants that are independent of $\eta$ but can depend on $g(\theta)$.*

**Theorem 3.3.** *For $\delta = \mathcal{O}(\mathrm{poly}(\eta))$, with probability at least $1 - \delta$, it holds for all $\mathcal{O}(\tfrac{1}{\alpha}\log\tfrac{1}{\eta}) \leq s \leq \tfrac{T}{\alpha\eta}$ that $\Phi(\bar{\theta}^{(s)}) \in \Gamma$ and $\|\bar{\theta}^{(s)} - \Phi(\bar{\theta}^{(s)})\|_2 = \mathcal{O}(\sqrt{\alpha\eta\log\tfrac{\alpha}{\eta\delta}})$, where $\mathcal{O}(\cdot)$ hides constants independent of $\eta$, $\alpha$ and $\delta$.*

Theorem 3.2 suggests that the trajectories of the manifold projection and the solution to the Slow SDE (4) are close to each other in the weak approximation sense. That is, $\{\Phi(\bar{\theta}^{(s)})\}$ and $\{\zeta(t)\}$ cannot be distinguished by evaluating test functions from a wide function class, including all polynomials. This measurement of closeness between the iterates of stochastic gradient algorithms and their SDE approximations is also adopted by Li et al. (2019a; 2021a); Malladi et al. (2022), but their analyses are for conventional SDEs. Theorem 3.3 further states that the iterate $\bar{\theta}^{(s)}$ keeps close to its manifold projection after the first few rounds.

**Remark 3.1.** *To connect to Finding 2.1, we remark that our theorems (1) do not require the model to be pre-trained (as long as the gradient flow starting with $\boldsymbol{\theta}^{(0)}$ converges to $\Gamma$); (2) give better bounds for smaller $\eta$; (3) characterize a long training horizon $\sim \eta^{-2}$. The need for tuning $H$ will be discussed in Section 3.3.3.*

**Technical Contribution.** The proof technique for Theorem 3.2 is novel and significantly different from the Slow SDE analysis of SGD in Li et al. (2021a). Their analysis uses advanced stochastic calculus and invokes Katzenberger's theorem (Katzenberger, 1991) to show that SGD converges to the Slow SDE in distribution, but no quantitative error bounds are provided. Also, due to the local updates and multiple aggregation steps in Local SGD, it is unclear how to extend Katzenberger's theorem to our case. To overcome this difficulty, we develop a new approach to analyze the Slow SDEs, which is based on the method of moments (Li et al., 2019a) and can provide the quantitative error bound $\tilde{\mathcal{O}}(\eta^{0.25})$ in weak approximation. See Appendix J for our proof outline. A by-product of our result is the first quantitative approximation bound for the Slow SDE approximation for SGD, which can be easily obtained by setting $K = 1$.

### 3.3 Interpretation of the Slow SDEs

In this subsection, we compare the Slow SDEs for SGD and Local SGD and provide an important insight into why Local SGD generalizes better than SGD: Local SGD strengthens the drift term in the Slow SDE, which makes the implicit regularization of stochastic gradient noise more effective.

#### 3.3.1 Interpretation of the Slow SDE for SGD.

The Slow SDE for SGD (7) consists of the diffusion and drift-I terms. The former injects noise into the dynamics in the tangent space; the latter one drives the dynamics to move along the negative gradient of $\frac{1}{2B}\langle\nabla^2\mathcal{L}(\boldsymbol{\zeta}), \widehat{\boldsymbol{\Sigma}}_\diamond(\boldsymbol{\zeta})\rangle$ projected onto the tangent space, but ignoring the dependency of $\widehat{\boldsymbol{\Sigma}}_\diamond(\boldsymbol{\zeta})$ on $\boldsymbol{\zeta}$. This can be connected to the class of semi-gradient methods which only computes a part of the gradient (Mnih et al., 2015; Sutton & Barto, 1998; Brandfonbrener & Bruna, 2020). In this view, the long-term behavior of SGD is similar to a stochastic semi-gradient method minimizing the implicit regularizer $\frac{1}{2B}\langle\nabla^2\mathcal{L}(\boldsymbol{\zeta}), \widehat{\boldsymbol{\Sigma}}_\diamond(\boldsymbol{\zeta})\rangle$ on the minimizer manifold of the original loss $\mathcal{L}$.

Though the semi-gradient method may not perfectly optimize its objective, the above argument reveals that SGD has a deterministic trend toward the region with a smaller magnitude of Hessian, which is commonly believed to correlate with better generalization (Hochreiter & Schmidhuber, 1997; Keskar et al., 2017; Neyshabur et al., 2017; Jiang et al., 2020) (see Appendix A for more discussions). In contrast, the diffusion term can be regarded as a random perturbation to this trend, which can impede optimization when the drift-I term is not strong enough.

Based on this view, we conjecture that **strengthening the drift term** of the Slow SDE can help SGD to better regularize the model, yielding a better generalization performance. More specifically, we propose the following hypothesis, which compares the generalization performances of the following generalized Slow SDEs. Note that $(\frac{1}{B}, \frac{1}{2B})$-Slow SDE corresponds to the Slow SDE for SGD (7).

**Definition 3.4.** *For $\kappa_1, \kappa_2 \geq 0$, define $(\kappa_1, \kappa_2)$-Slow SDE to be the following:*

$$\mathrm{d}\boldsymbol{\zeta}(t) = P_{\boldsymbol{\zeta}}\left(\sqrt{\kappa_1}\boldsymbol{\Sigma}_\parallel^{1/2}(\boldsymbol{\zeta})\mathrm{d}\boldsymbol{W}_t - \kappa_2\nabla^3\mathcal{L}(\boldsymbol{\zeta})[\widehat{\boldsymbol{\Sigma}}_\diamond(\boldsymbol{\zeta})]\mathrm{d}t\right). \tag{8}$$

**Hypothesis 3.1.** *Starting at a minimizer $\boldsymbol{\zeta}_0 \in \Gamma$, run $(\kappa_1, \kappa_2)$-Slow SDE and $(\kappa_1, \kappa_2')$-Slow SDE respectively for the same amount of time $T > 0$ and obtain $\boldsymbol{\zeta}(T), \boldsymbol{\zeta}'(T)$. If $\kappa_2 > \kappa_2'$, then the expected test accuracy at $\boldsymbol{\zeta}(T)$ is better than that at $\boldsymbol{\zeta}'(T)$.*

Due to the No Free Lunch Theorem, we do not claim that our hypothesis is always true, but we do believe that the hypothesis holds when training usual neural networks (e.g., ResNets, VGGNets) on standard benchmarks (e.g., CIFAR-10, ImageNet).

**Example: Training with Label Noise Regularization.** To exemplify the generalization benefit of having a larger drift term, we follow a line of theoretical works (Li et al., 2021b; Blanc et al., 2020; Damian et al., 2021) to study the case of training over-parameterized neural nets with label noise regularization. For a $C$-class classification task, the label noise regularization is as follows: every time we draw a sample from the training set, we make the true label as it is with probability $1 - p$, and replace it with any other label with equal probability $\frac{p}{C-1}$. When we use cross-entropy loss, the Slow SDE for SGD turns out to be a simple deterministic gradient flow on $\Gamma$ (instead of a semi-gradient method) for minimizing the trace of Hessian: $\mathrm{d}\boldsymbol{\zeta}(t) = -\frac{1}{4B}\nabla_\Gamma\mathrm{tr}(\nabla^2\mathcal{L}(\boldsymbol{\zeta}))\mathrm{d}t$, where $\nabla_\Gamma f$

stands for the gradient of the function $f$ projected to the tangent space of $\Gamma$. Checking the validity of our hypothesis reduces to the following question: *Is minimizing the trace of Hessian beneficial to generalization?* Many works prove positive results in concrete settings, including the line of works we just mentioned. We refer the readers to Appendix G for further discussion.

### 3.3.2 Local SGD Strengthens the Drift Term in Slow SDE.

Based on Hypothesis 3.1, we argue that **Local SGD improves generalization by strengthening the drift term of the Slow SDE**. First, it can be seen from (4) that the Slow SDE for Local SGD has an additional drfit-II term. Similar to the drift-I term of the Slow SDE for SGD, this drift-II term drives the dynamics to move along the negative semi-gradient of $\frac{K-1}{2B}\langle\nabla^2\mathcal{L}(\zeta), \widehat{\boldsymbol{\Psi}}(\zeta)\rangle$ (with the dependency of $\widehat{\boldsymbol{\Psi}}(\zeta)$ on $\zeta$ ignored). Combining it with the implicit regularizer induced by the drift-I term, we can see that the long-term behavior of Local SGD is similar to a stochastic semi-gradient method minimizing the implicit regularizer $\frac{1}{2B}\langle\nabla^2\mathcal{L}(\zeta), \widehat{\boldsymbol{\Sigma}}_\diamond(\zeta)\rangle + \frac{K-1}{2B}\langle\nabla^2\mathcal{L}(\zeta), \widehat{\boldsymbol{\Psi}}(\zeta)\rangle$ on $\Gamma$.

Comparing the definitions of $\widehat{\boldsymbol{\Sigma}}_\diamond(\zeta)$ (5) and $\widehat{\boldsymbol{\Psi}}(\zeta)$ (6), we can see that $\widehat{\boldsymbol{\Psi}}(\zeta)$ is basically a rescaling of the entries of $\widehat{\boldsymbol{\Sigma}}_\diamond(\zeta)$ in the eigenbasis of Hessian, where the rescaling factor $\psi(\eta H \cdot (\lambda_i + \lambda_j))$ for each entry is between 0 and 1 (see Figure 9 for the plot of $\psi$). When $\eta H$ is small, the rescaling factors should be close to $\psi(0) = 0$, then $\widehat{\boldsymbol{\Psi}}(\zeta) \approx \mathbf{0}$, leading to almost no additional regularization. On the other hand, when $\eta H$ is large, the rescaling factors should be close to $\psi(+\infty) = 1$, so $\widehat{\boldsymbol{\Psi}}(\zeta) \approx \widehat{\boldsymbol{\Sigma}}_\diamond(\zeta)$. We can then merge the two implicit regularizers as $\frac{K}{2B}\langle\nabla^2\mathcal{L}(\zeta), \widehat{\boldsymbol{\Sigma}}_\diamond(\zeta)\rangle$, and (4) becomes the $(\frac{1}{B}, \frac{K}{2B})$-Slow SDE, which is restated below:
$$d\zeta(t) = P_\zeta\left(\frac{1}{\sqrt{B}}\boldsymbol{\Sigma}_\|^{1/2}(\zeta)dW_t - \frac{K}{2B}\nabla^3\mathcal{L}(\zeta)[\widehat{\boldsymbol{\Sigma}}_\diamond(\zeta)]dt\right). \tag{9}$$
From the above argument we know how the Slow SDE of Local SGD (4) changes as $\eta H$ transitions from 0 to $+\infty$. Initially, when $\eta H = 0$, (4) is the same as the $(\frac{1}{B}, \frac{1}{2B})$-Slow SDE for SGD. Then increasing $\eta H$ strengthens the drift term of (4). As $\eta H \to +\infty$, (4) transitions to the $(\frac{1}{B}, \frac{K}{2B})$-Slow SDE, where the drift term becomes $K$ times larger.

According to Hypothesis 3.1, the $(\frac{1}{B}, \frac{K}{2B})$-Slow SDE generalizes better than the $(\frac{1}{B}, \frac{1}{2B})$-Slow SDE, so Local SGD with $\eta H = +\infty$ should generalize better than SGD. When $\eta H$ is chosen realistically as a finite value, the generalization performance of Local SGD interpolates between these two cases, which results in a worse generalization than $\eta H = +\infty$ but should still be better than SGD.

### 3.3.3 Theoretical Insights into Tuning the Number of Local Steps

Based on our Slow SDE approximations, we now discuss how the number of local steps $H$ affects the generalization of Local SGD. When $\eta$ is small but finite, tuning $H$ offers a trade-off between regularization strength and SDE approximation quality. Larger $\alpha := \eta H$ makes the regularization stronger in the SDE (as discussed in Section 3.3.2), but the SDE itself may lose track of Local SGD, which can be seen from the error bound $\mathcal{O}(\sqrt{\alpha\eta\log(\alpha/\eta\delta)})$ in Theorem 3.3. Therefore, we expect the test accuracy to first increase and then decrease as we gradually increase $H$. Indeed, we observe in Figures 2(e) and 2(f) that the plot of test accuracy versus $H$ is unimodal for each $\eta$.

It is thus necessary to tune $H$ for the best generalization. When $H$ is tuned together with other hyperparameters, such as learning rate $\eta$, our Slow SDE approximation recommends setting $H$ to be at least $\Omega(\eta^{-1})$ so that $\alpha := \eta H$ does not vanish in the Slow SDE. Since larger $\alpha$ gives a stronger regularization effect, the optimal $H$ should be set to the largest value so that the Slow SDE does not lose track of Local SGD. Indeed, we empirically observed that when $H$ is tuned optimally, $\alpha$ increases as $\eta$ decreases, suggesting that the optimal $H$ grows faster than $\Omega(\eta^{-1})$. See Figure 5(f).

## 4 Conclusions

In this paper, we analyze the long-term generalization behavior of Local SGD in the small learning rate regime by deriving the Slow SDE for Local SGD as a generalization of that for SGD (Li et al., 2021b). We attribute the generalization improvement over SGD to the larger drift term in the SDE for Local SGD. Our empirical validation shows that Local SGD indeed induces generalization benefits with small learning rate and long enough training time. The main limitation of our work is that our analysis does not imply any direct theoretical separation between SGD and Local SGD in test accuracy, which requires a much deeper understanding of the loss landscape and the Slow SDEs and is left for future work. Another direction for future work is to design distributed training methods that provably generalize better than SGD based on the theoretical insights obtained from Slow SDEs.

ACKNOWLEDGEMENT AND DISCLOSURE OF FUNDING

The work of Xinran Gu and Longbo Huang is supported by the Technology and Innovation Major Project of the Ministry of Science and Technology of China under Grant 2020AAA0108400 and 2020AAA0108403, the Tsinghua University Initiative Scientific Research Program, and Tsinghua Precision Medicine Foundation 10001020109. The work of Kaifeng Lyu and Sanjeev Arora is supported by funding from NSF, ONR, Simons Foundation, DARPA and SRC.

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

CONTENTS

# A  ADDITIONAL RELATED WORKS

**Optimization aspect of Local SGD.**  Local SGD is a communication-efficient variant of parallel SGD, where multiple workers perform SGD independently and average the model parameters periodically. Dating back to Mann et al. (2009) and Zinkevich et al. (2010), this strategy has been widely adopted to reduce the communication cost and speed up training in both scenarios of data center distributed training (Chen & Huo, 2016; Zhang et al., 2014; Povey et al., 2014; Su & Chen, 2015) and Federated Learning (McMahan et al., 2017; Kairouz et al., 2021). To further accelerate training, Wang & Joshi (2019) and Haddadpour et al. (2019) proposed adaptive schemes for the averaging frequency, and Basu et al. (2019) combined Local SGD with gradient compression. Motivated to theoretically understand the empirical success of Local SGD, a lot of researchers analyzed the convergence rate of Local SGD under various settings, e.g., homogeneous/heterogeneous data and convex/non-convex objective functions. Among them, Yu et al. (2019); Stich (2018); Khaled et al. (2020); Woodworth et al. (2020a) focus on the homogeneous setting where data for each worker are independent and identically distributed (IID). Li et al. (2019b); Karimireddy et al. (2020); Glasgow et al. (2022); Woodworth et al. (2020b); Wang et al. (2022) study the heterogeneous setting, where workers have non-IID data and local updates may induce "client drift" (Karimireddy et al., 2020) and hurt optimization. The error bound of Local SGD obtained by these works is typically inferior to that of SGD with the same global batch size for fixed number of iterations/epochs and becomes worse as the number of local steps increases, revealing a trade-off between less communication and better optimization. In this paper, we are interested in the generalization aspect of Local SGD in the homogeneous setting, assuming the training loss can be optimized to a small value.

**Gradient noise and generalization.**  The effect of stochastic gradient noise on generalization has been studied from different aspects, e.g., changing the order of learning different patterns Li et al. (2019a), inducing an implicit regularizer in the second-order SDE approximation Smith et al. (2021); Li et al. (2019a). Our work follows a line of works studying the effect of noise in the lens of sharpness, which is long believed to be related to generalization Hochreiter & Schmidhuber (1997); Neyshabur et al. (2017). Keskar et al. (2017) empirically observed that large-batch training leads to worse generalization and sharper minima than small-batch training. Wu et al. (2018); Hu et al. (2017); Ma & Ying (2021) showed that gradient noise destabilizes the training around sharp minima, and Kleinberg et al. (2018); Zhu et al. (2018); Xie et al. (2021); Ibayashi & Imaizumi (2021) quantitatively characterized how SGD escapes sharp minima. The most related papers are Blanc et al. (2020); Damian et al. (2021); Li et al. (2021b), which focus on the training dynamics near a manifold of minima and study the effect of noise on sharpness (see also Section 3.2). Though the mathematical definition of sharpness may be vulnerable to the various symmetries in deep neural nets (Dinh et al., 2017), sharpness still appears to be one of the most promising tools for predicting generalization (Jiang et al., 2020; Foret et al., 2021).

**Improving generalization in large-batch training.**  The generalization issue of the large-batch (or full-batch) training has been observed as early as (Bengio, 2012; LeCun et al., 2012). As mentioned in Section 1, the generalization issue of large-batch training could be due to the lack of a sufficient amount of stochastic noise. To make up the noise in large-batch training, Krizhevsky (2014); Goyal et al. (2017) empirically discovered the *Linear Scaling Rule* for SGD, which suggests enlarging the learning rate proportionally to the batch size. Jastrzębski et al. (2017) adopted an SDE-based analysis to justify that this scaling rule indeed retains the same amount of noise as small-batch training (see also Section 3.1). However, the SDE approximation may fail if the learning rate is too large (Li et al., 2021a), especially in the early phase of training before the first learning rate decay (Smith et al., 2020). Shallue et al. (2019) demonstrated that generalization gap between small- and large-batch training can also depend on many other training hyperparameters. Besides enlarging the learning rate, other approaches have also been proposed to reduce the gap, including training longer (Hoffer et al., 2017), learning rate warmup (Goyal et al., 2017), LARS (You et al., 2018), LAMB (You et al., 2020). In this paper, we focus on using Local SGD to improve generalization, but adding local steps is a generic training trick that can also be combined with others, e.g., Local LARS (Lin et al., 2020b), Local Extrap-SGD (Lin et al., 2020a).

# B    ADDITIONAL DISCUSSIONS

**Connection to the conventional wisdom that the diffusion term matters more.**   As mentioned in Section 3.1, it is believed in the literature is that a large diffusion term in the conventional SDE leads to good generalization. One may think that the diffusion term in the Slow SDE corresponds to that in the conventional SDE, and thus enlarging the diffusion term rather than the drift term should lead to better generalization. However, we note that both the diffusion and drift terms in the Slow SDEs result from the long-term effects of the diffusion term in the conventional SDE (Slow SDEs become stationary if $\Sigma = 0$). This means our view characterizes the role of gradient noise in more detail, and therefore, goes one step further on the conventional wisdom.

**Slow SDEs for neural nets with modern training techniques.**   In modern neural net training, it is common to add normalization layers and weight decay ($L^2$-regularization) for better optimization and generalization. However, these techniques lead to violations of our assumptions, e.g., no fixed point exists in the regularized loss (Li et al., 2020; Ahn et al., 2022). Still, a minimizer manifold can be expected to exist for the unregularized loss. Li et al. (2022) noted that the drift and diffusion around the manifold proceeds faster in this case, and derived a Slow SDE for SGD that captures $\mathcal{O}(\frac{1}{\eta} \log \frac{1}{\eta})$ discrete steps instead of $\mathcal{O}(\frac{1}{\eta^2})$. We believe that our analysis can also be extended to this case, and that adding local steps still results in the effect of strengthening the drift term.

# C  IMPLEMENTATION DETAILS OF PARALLEL SGD, LOCAL SGD AND POST-LOCAL SGD

In this section, we present the formal procedures for Parallel SGD, Local SGD and Post-local SGD. Given a training dataset and a data augmentation function, Algorithms 1 and 2 show the implementations of distributed samplers for sampling local batches with and without replacement. Then Algorithms 3 to 5 show the implementations of parallel SGD, Local SGD and Post-local SGD that can run with either of the samplers.

**Sampling with replacement.** Our theory analyzes parallel SGD, Local SGD and Post-local SGD when local batches are sampled with replacement (Algorithm 1). That is, local batches consist of IID samples from the same training distribution $\tilde{\mathcal{D}}$, where $\tilde{\mathcal{D}}$ serves as an abstraction of the distribution of an augmented sample drawn from the training dataset. The mathematical formulations are given in Section 1.

**Sampling without replacement.** Slightly different from our theory, we use the sampling without replacement (Algorithm 2) in our experiments unless otherwise stated. This sampling scheme is standard in practice: it is used by Goyal et al. (2017) for parallel SGD and by Lin et al. (2020b); Ortiz et al. (2021) for Post-local/Local SGD. This sampling scheme works as follows. At the beginning of every epoch, the whole training dataset is shuffled and evenly partitioned into $K$ shards. Each worker takes one shard and samples batches without replacement. When all workers pass their own shard, the next epoch begins and the whole dataset is reshuffled. An alternative view is that the workers always share the same dataset. For each epoch, they perform local steps by sampling batches of data without replacement until the dataset contains too few data to form a batch. Then another epoch starts with the dataset reloaded to the initial state.

**Discrepancy in Sampling Schemes.** We argue that this discrepancy between theory and experiments on sample schemes is minor. Though sampling without replacement is standard in practice, most previous works, e.g., Wang & Joshi (2019); Li et al. (2021a); Zhang et al. (2020), analyze sampling with replacement for technical simplicity and yields meaningful results.

Moreover, even if we change the sampling scheme to with replacement, Local SGD can still improve the generalization of SGD (by merely adding local steps). See Appendix F for the experiments. We believe that the reasons for better generalization of Local SGD with either sampling scheme are similar and leave the analysis for sampling without replacement for future work.

---

**Algorithm 1:** Distributed Sampler on $K$ Workers (Sampling with Replacement)

---

1 **Require**: shared training dataset $\mathcal{D}$, data augmentation function $\mathcal{A}(\hat{\xi})$
2 **Hyperparameters**: local batch size $B_{\text{loc}}$

3 **Function** `Sample()` **on** *worker $k$*:
4      Draw $B_{\text{loc}}$ IID samples $\hat{\xi}_1, \ldots, \hat{\xi}_{B_{\text{loc}}}$ from $\mathcal{D}$ with replacement ;
5      $\xi_b \leftarrow \mathcal{A}(\hat{\xi}_b)$ for all $1 \le b \le B_{\text{loc}}$ ;            // apply data augmentation
6      **return** $(\xi_1, \ldots, \xi_{B_{\text{loc}}})$ ;
7 **end**

---

---

**Algorithm 2:** Distributed Sampler on $K$ Workers (Sampling without Replacement)

---

1 **Require**: shared training dataset $\mathcal{D}$, data augmentation function $\mathcal{A}(\hat{\xi})$
2 **Hyperparameters**: local batch size $B_{\text{loc}}$
3 **Constant**: $N_{\text{loc}} := \left\lfloor \frac{|\mathcal{D}|}{K B_{\text{loc}}} \right\rfloor$           // number of local batches per worker per epoch
4 **Local Variables**: $c^{(k)} \leftarrow N_{\text{loc}} B_{\text{loc}}$ for worker $k$      // number of samples drawn in this epoch

5 **Function** `Sample()` **on** *worker $k$*:
6      **if** $c^{(k)} = N_{\text{loc}} B_{\text{loc}}$ **then**
         // Now start a new epoch
7          Wait until all the other workers reach this line ;           // synchronize
8          Draw a random permutation $P$ of $1, \ldots, |D|$ jointly with other workers so that the same permutation is shared among all workers ;      // reshuffle the dataset
9          $Q_j^{(k)} \leftarrow P_{(k-1)N_{\text{loc}} B_{\text{loc}} + j}$ for all $1 \le j \le N_{\text{loc}}$ ;      // partition the dataset
10          $c^{(k)} \leftarrow 0$ ;
11      **end**
12      **for** $i = 1, \ldots, B_{\text{loc}}$ **do**
13          $\hat{\xi}_i \leftarrow$ the $Q_{c^{(k)}+i}^{(k)}$-th data point of $\mathcal{D}$ ;      // sample without replacement
14          $\xi_i \leftarrow \mathcal{A}(\hat{\xi}_i)$ ;           // apply data augmentation
15      **end**
16      $c^{(k)} \leftarrow c^{(k)} + B_{\text{loc}}$ ;
17      **return** $(\xi_1, \ldots, \xi_{B_{\text{loc}}})$ ;
18 **end**

---

---

**Algorithm 3:** Parallel SGD on $K$ Workers

---

1 **Input**: loss function $\ell(\boldsymbol{\theta}; \xi)$, initial parameter $\boldsymbol{\theta}_0$
2 **Hyperparameters**: total number of iterations $T$, learning rate $\eta$, local batch size $B_{\text{loc}}$

3 **for** $t = 0, \cdots, T - 1$ **do**
4      **for** *each worker $k$* **do in parallel**
5          $(\xi_{k,t,1}, \ldots, \xi_{k,t,B_{\text{loc}}}) \leftarrow \texttt{Sample()}$ ;         // sample a local batch
6          $\boldsymbol{g}_{k,t} \leftarrow \frac{1}{B_{\text{loc}}} \sum_{i=1}^{B_{\text{loc}}} \nabla \ell(\boldsymbol{\theta}_t; \xi_{k,t,i})$ ;        // computing the local gradient
7      **end**
8      $\boldsymbol{g}_t \leftarrow \frac{1}{K} \sum_{k=1}^{K} \boldsymbol{g}_{k,t}$ ;        // all-Reduce aggregation of local gradients
9      $\boldsymbol{\theta}_{t+1} \leftarrow \boldsymbol{\theta}_t - \eta_t \boldsymbol{g}_t$ ;        // update the model
10 **end**

---

---

**Algorithm 4:** Local SGD on $K$ Workers

---

1 **Input**: loss function $\ell(\boldsymbol{\theta}; \xi)$, initial parameter $\bar{\boldsymbol{\theta}}^{(0)}$
2 **Hyperparameters**: total number of rounds $R$, number of local steps $H$ per round
3 **Hyperparameters**: learning rate $\eta$, local batch size $B_{\text{loc}}$

4 **for** $s = 0, \ldots, R - 1$ **do**
5      **for** *each worker $k$* **do in parallel**
6          $\boldsymbol{\theta}_{k,0}^{(s)} \leftarrow \bar{\boldsymbol{\theta}}^{(0)}$ ;        // maintain a local copy of the global iterate
7          **for** $t = 0, \ldots, H - 1$ **do**
8              $(\xi_{k,t,1}^{(s)}, \ldots, \xi_{k,t,B_{\text{loc}}}^{(s)}) \leftarrow \texttt{Sample()}$ ;        // sample a local batch
9              $\boldsymbol{g}_{k,t}^{(s)} \leftarrow \frac{1}{B_{\text{loc}}} \sum_{i=1}^{B_{\text{loc}}} \nabla \ell(\boldsymbol{\theta}_{k,t}^{(s)}; \xi_{k,t,i}^{(s)})$ ;       // computing the local gradient
10              $\boldsymbol{\theta}_{k,t+1}^{(s)} \leftarrow \boldsymbol{\theta}_{k,t}^{(s)} - \eta \boldsymbol{g}_{k,t}^{(s)}$ ;       // update the local model
11          **end**
12      **end**
13      $\bar{\boldsymbol{\theta}}^{(s+1)} \leftarrow \frac{1}{K} \sum_{k=1}^{K} \boldsymbol{\theta}_{k,H}^{(s)}$ ;        // all-Reduce aggregation of local iterates
14 **end**

---

---

**Algorithm 5:** Post-local SGD on $K$ Workers

---

1 **Input**: loss function $\ell(\boldsymbol{\theta}; \xi)$, initial parameter $\boldsymbol{\theta}_0$
2 **Hyperparameters**: total number of iterations $T$, learning rate $\eta$, local batch size $B_{\text{loc}}$
3 **Hyperparameters**: switching time point $t_0$, number of local steps $H$ per round
4 **Ensure**: $T - t_0$ is a multiple of $H$

5 Starting from $\boldsymbol{\theta}_0$, run Parallel SGD for $t_0$ iterations and obtain $\boldsymbol{\theta}_{t_0}$ ;
6 Starting from $\boldsymbol{\theta}_{t_0}$, run Local SGD for $\frac{1}{H}(T - t_0)$ rounds with $H$ local steps per round ;
7 **return** *the final global iterate of Local SGD* ;

---

## D  MODELING LOCAL SGD WITH MULTIPLE CONVENTIONAL SDEs

Lin et al. (2020b) tried to informally explain the success of Local SGD by adopting the argument that larger diffusion term in the conventional SDE leads to better generalization (see Section 3.1 and appendix A). Basically, they attempted to write multiple SDEs, each of which describes the $H$-step local training process of each worker in each round (from $\boldsymbol{\theta}_{k,0}^{(s)}$ to $\boldsymbol{\theta}_{k,H}^{(s)}$). The key difference between each of these SDEs and the SDE for SGD (3) is that the former one has a larger diffusion term because the workers use batch size $B_{\mathrm{loc}}$ instead of $B$:

$$\mathrm{d}\boldsymbol{X}(t) = -\nabla\mathcal{L}(\boldsymbol{X})\mathrm{d}t + \sqrt{\frac{\eta}{B_{\mathrm{loc}}}}\boldsymbol{\Sigma}^{1/2}(\boldsymbol{X})\mathrm{d}\boldsymbol{W}_t. \tag{10}$$

Lin et al. (2020b) then argue that the total amount of "noise" in the training dynamics of Local SGD is larger than that of SGD. However, it is hard to see whether it is indeed larger, since the model averaging step at the end of each round can reduce the variance in training and may cancel the effect of having larger diffusion terms.

More formally, a complete modeling of Local SGD following this idea should view the sequence of global iterates $\{\bar{\boldsymbol{\theta}}^{(s)}\}$ as a Markov process $\{\boldsymbol{X}^{(s)}\}$. Let $\mathcal{P}_{\boldsymbol{X}}(\boldsymbol{x}, B, t)$ the distribution of $\boldsymbol{X}(t)$ in (3) with initial condition $\boldsymbol{X}(0) = \boldsymbol{x}$. Then the Markov transition should be $\boldsymbol{X}^{(s+1)} = \frac{1}{K}\sum_{k=1}^{K}\boldsymbol{X}_{k,H}^{(s)}$, where $\boldsymbol{X}_{1,H}^{(s)}, \ldots, \boldsymbol{X}_{K,H}^{(s)}$ are $K$ independent samples from $\mathcal{P}_{\boldsymbol{X}}(\boldsymbol{X}^{(s)}, B_{\mathrm{loc}}, H\eta)$, i.e., sampling from (10).

Consider one round of model averaging. It is true that $\mathcal{P}_{\boldsymbol{X}}(\boldsymbol{X}^{(s)}, B_{\mathrm{loc}}, H\eta)$ may have a larger variance than the corresponding SGD baseline $\mathcal{P}_{\boldsymbol{X}}(\boldsymbol{X}^{(s)}, B, H\eta)$ because the former one has a smaller batch size. However, it is unclear whether $\boldsymbol{X}^{(s+1)}$ also has a larger variance than $\mathcal{P}_{\boldsymbol{X}}(\boldsymbol{X}^{(s)}, B, H\eta)$. This is because $\boldsymbol{X}^{(s+1)}$ is the average of $K$ samples, which means we have to compare $\frac{1}{K}$ times the variance of $\mathcal{P}_{\boldsymbol{X}}(\boldsymbol{X}^{(s)}, B_{\mathrm{loc}}, H\eta)$ with the variance of $\mathcal{P}_{\boldsymbol{X}}(\boldsymbol{X}^{(s)}, B, H\eta)$. Then it is unclear which one is larger.

In the special case where $H\eta$ is small, $\mathcal{P}_{\boldsymbol{X}}(\boldsymbol{X}^{(s)}, B_{\mathrm{loc}}, H\eta)$ is approximately equal to the following Gaussian distribution:

$$\mathcal{N}\left(\boldsymbol{X}^{(s)} - \eta H\nabla\mathcal{L}(\boldsymbol{X}^{(s)}), \frac{\eta^2 H}{B_{\mathrm{loc}}}\boldsymbol{\Sigma}(\boldsymbol{X}^{(s)})\right) \tag{11}$$

Then averaging over $K$ samples gives

$$\mathcal{N}\left(\boldsymbol{X}^{(s)} - \eta H\nabla\mathcal{L}(\boldsymbol{X}^{(s)}), \frac{\eta^2 H}{B}\boldsymbol{\Sigma}(\boldsymbol{X}^{(s)})\right), \tag{12}$$

which is exactly the same as the Gaussian approximation of the SGD baseline. This means there do exist certain cases where Lin et al. (2020b)'s argument does not give a good separation between Local SGD and SGD.

Moreover, we do not gain any further insights from this modeling since it is hard to see how model averaging interacts with the SDEs.

## E  ADDITIONAL INTERPRETATION OF THE SLOW SDEs

### E.1  UNDERSTANDING THE DIFFUSION TERM IN THE SLOW SDE

So far, we have discussed why adding local steps enlarges the drift term in the Slow SDE and why enlarging the drift term can benefit generalization. Besides this, here we remark that another way to accelerate the corresponding semi-gradient method for minimizing the implicit regularizer is to reduce the diffusion term, so that the trajectory more closely follows the drift term. More formally, we propose the following:

**Hypothesis E.1.** *Starting at a minimizer $\boldsymbol{\zeta}_0 \in \Gamma$, run $(\kappa_1, \kappa_2)$-Slow SDE and $(\kappa_1, \kappa_2')$-Slow SDE respectively for the same amount of time $T > 0$ and obtain $\boldsymbol{\zeta}(T), \boldsymbol{\zeta}'(T)$. If $\boldsymbol{\Sigma}_{\parallel} \not\equiv \boldsymbol{0}$ and $\kappa_1 < \kappa_1'$, then the expected test accuracy at $\boldsymbol{\zeta}(T)$ is better than that at $\boldsymbol{\zeta}'(T)$.*

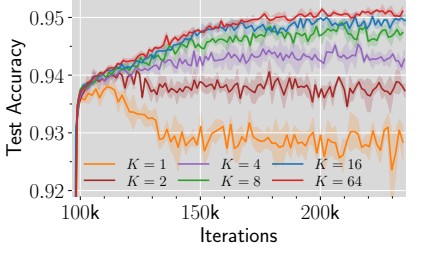 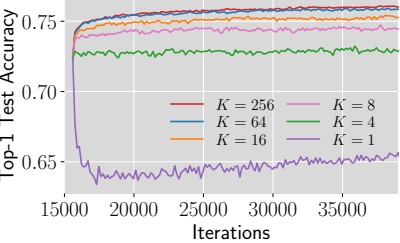

(a) CIFAR-10, $H = 600$ for $K > 1$.    (b) ImageNet, $H = 78$ for $K > 1$.

Figure 3: Reducing the diffusion term of the Slow SDE for Local SGD leads to better generalization. Test accuracy improves as we increase $K$ with fixed $\eta$ and $H$ to reduce the diffusion term while keeping the drift term untouched. See Appendix M.4 for details.

Here we exclude the case of $\boldsymbol{\Sigma}_{\parallel} \equiv \mathbf{0}$ because in this case the diffusion term in the Slow SDE is always zero. To verify Hypothesis E.1, we set the product $\alpha := \eta H$ large, keep $H, \eta$ fixed, increase the number of workers $K$, and compare the generalization performances after a fixed amount of training steps (but after different numbers of epochs). This case corresponds to the $\left(\frac{1}{KB_{\text{loc}}}, \frac{1}{2B_{\text{loc}}}\right)$-Slow SDE, so adding more workers should reduce the diffusion term. As shown in Figure 3, a higher test accuracy is indeed achieved for larger $K$.

**Implication: Enlarging the learning rate is not equally effective as adding local steps.** Given that Local SGD improves generalization by strengthening the drift term, it is natural to wonder if enlarging the learning rate of SGD would also lead to similar improvements. While it is true that enlarging the learning rate effectively increases the drift term, it also increases the diffusion term simultaneously, which can hinder the implicit regularization by Hypothesis E.1. In contrast, adding local steps does not change the diffusion term. As shown in Figure 6(a), even when the learning rate of SGD is increased, SGD still underperforms Local SGD by about $2\%$ in test accuracy.

On the other hand, in the special case of where $\boldsymbol{\Sigma}_{\parallel} \equiv \mathbf{0}$, Hypothesis E.1 does not hold, and enlarging the learning rate by $\sqrt{K}$ results in the same Slow SDE as adding local steps (see Appendix G for derivation). Then these two actions should produce the same generalization improvement, unless the learning rate is so large that Slow SDE loses track of the training dynamics. As an example of such a special case, an experiment with label noise regularization is presented in Figure 8.

### E.2 THE EFFECT OF GLOBAL BATCH SIZE ON GENERALIZATION

In this section, we discuss the effect of global batch size on the generalization of Local SGD. Given that the computation power of a single worker is limited, we consider the case where the local batch size $B_{\text{loc}}$ is fixed and the global batch size $B = KB_{\text{loc}}$ is tuned by adding or removing the workers. This scenario is relevant to the practice because one may want to know the maximum parallelism possible to train the neural net without causing generalization degradation.

For SGD, previous works have proposed the Linear Scaling Rule (LSR) (Krizhevsky, 2014; Goyal et al., 2017; Jastrzębski et al., 2017): scaling the learning rate $\eta \mapsto \kappa\eta$ linearly with the global batch size $B \mapsto \kappa B$ yields the same conventional SDE (3) under a constant epoch budget, hence leading to almost the same generalization performance as long as the SDE approximation does not fail.

We show in Theorem H.1 that the LSR does not change the Slow SDE of SGD either. Experiments in Figure 4 show that the LSR indeed holds nicely when we continue training with small learning rates from the same CIFAR-10 and ImageNet checkpoints as in Figure 2. Here we choose $K = 16$ and $K = 256$ as the base settings for CIFAR-10 and ImageNet, respectively, and then tune the learning rate to maximize the test accuracy. As shown in Figures 4(a) and 4(b), the optimal learning rate turns out to be small enough that the LSR can be applied to scale the global batch size with only a minor change in test accuracy.

Now, assuming the learning rate is scaled as LSR, we study how to tune the number of local steps $H$ for Local SGD for better generalization. A natural choice is to tune $H$ in the base settings and keep $\alpha$ unchanged via scaling $H \mapsto H/\kappa$. Then the following SDE can be derived (see Theorem H.2):

$$\mathrm{d}\boldsymbol{\zeta}(t) = P_{\boldsymbol{\zeta}}\bigg( \underbrace{\tfrac{1}{\sqrt{B}}\boldsymbol{\Sigma}_{\parallel}^{1/2}(\boldsymbol{\zeta})\mathrm{d}\boldsymbol{W}_t}_{\text{(a) diffusion (unchanged)}} \underbrace{-\tfrac{1}{2B}\nabla^3\mathcal{L}(\boldsymbol{\zeta})[\widehat{\boldsymbol{\Sigma}}_{\diamond}(\boldsymbol{\zeta})]\mathrm{d}t}_{\text{(b) drift-I (unchanged)}} \underbrace{-\tfrac{\kappa K-1}{2B}\nabla^3\mathcal{L}(\boldsymbol{\zeta})[\widehat{\boldsymbol{\Psi}}(\boldsymbol{\zeta})]\mathrm{d}t}_{\text{(c) drift-II (rescaled)}} \bigg). \quad (13)$$

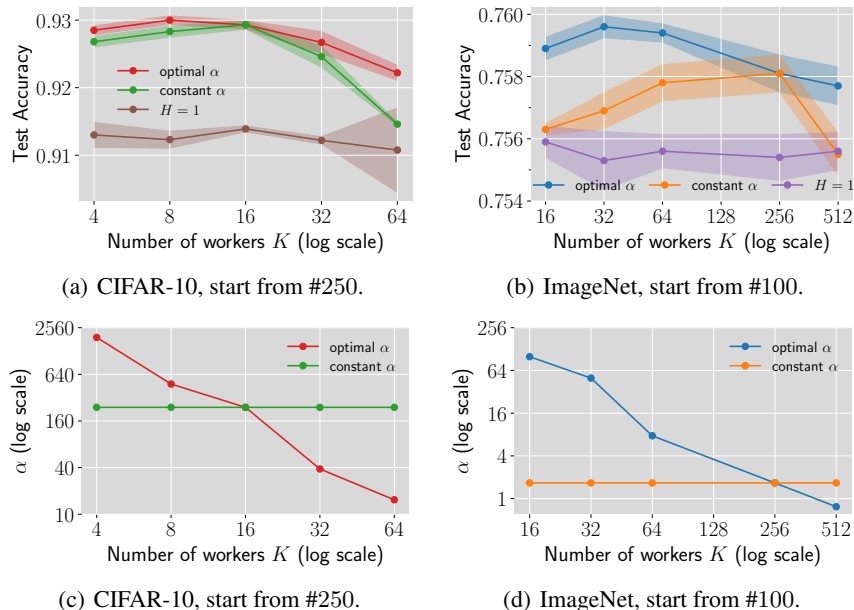

Figure 4: For training from CIFAR-10 and ImageNet checkpoints, Local SGD consistently outperforms SGD ($H = 1$) across different batch sizes $B$ (fixing $B_{\text{loc}}$ and varying $K$), where the learning rate is scaled by the LSR $\eta \propto B$. Two possible ways of tuning the number of local steps $H$ are considered: **(1).** Tune $H$ for the best test accuracy for $K = 16$ and $K = 256$ respectively on CIFAR-10 and ImageNet, then scale $H$ as $H \propto 1/B$ so that $\alpha := \eta H$ is constant; **(2).** Tune $H$ specifically for each $K$. See Appendix M.5 for training details.

Compared with (4), the drift-II term here is rescaled by a positive factor. Again, when $\alpha$ is large, we can follow the argument in Section 3.3.2 to approximate $\widehat{\boldsymbol{\Psi}}(\boldsymbol{\zeta}) \approx \widehat{\boldsymbol{\Sigma}}_\diamond(\boldsymbol{\zeta})$ and obtain the following $(\frac{1}{B}, \frac{\kappa K}{B})$-Slow SDE:

$$\mathrm{d}\boldsymbol{\zeta}(t) = P_{\boldsymbol{\zeta}}\left( \tfrac{1}{\sqrt{B}}\boldsymbol{\Sigma}_{\parallel}^{1/2}(\boldsymbol{\zeta})\mathrm{d}\boldsymbol{W}(t) - \tfrac{\kappa K}{2B}\nabla^3\mathcal{L}(\boldsymbol{\zeta})[\widehat{\boldsymbol{\Sigma}}_\diamond(\boldsymbol{\zeta})]\mathrm{d}t \right). \tag{14}$$

The drift term of the above SDE is always stronger than SGD (7), as long as there exists more than one worker after the scaling (i.e., $\kappa K > 1$). As expected from Hypothesis 3.1, we observed in the experiments that the generalization performance of Local SGD is always better than or at least comparable to SGD across different batch sizes (see Figures 4(a) and 4(b)).

Taking a closer look into the drift term in the Slow SDE (14), we can find that it scales linearly with $\kappa$. According to Hypothesis 3.1, the SDE is expected to generalize better when adding more workers ($\kappa > 1$) and to generalize worse when removing some workers ($\kappa < 1$). For the latter case, we indeed observed that the test accuracy of Local SGD drops when removing workers. For the case of adding workers, however, we also need to take into account that the LSR specifies a larger learning rate and causes a larger SDE approximation error for the same $\alpha$, which may cancel the generalization improvement brought by strengthening the drift term. In the experiments, we observed that the test accuracy does not rise when adding more workers to the base settings.

Since $\alpha$ also controls the regularization strength (Section 3.3.3), it would be beneficial to decrease $\alpha$ for large batch size so as to better trade-off between regularization strength and approximation quality. In Figures 4(c) and 4(d), we plot the optimal value of $\alpha$ for each batch size, and we indeed observed that the optimal $\alpha$ drops as we scale up $K$. Conversely, a smaller batch size (and hence a smaller learning rate) allows for using a larger $\alpha$ to enhance regularization while still keeping a low approximation error (Theorem 3.3). The test accuracy curves in Figures 4(a) and 4(b) indeed show that setting a larger $\alpha$ can compensate for the accuracy drop when reducing the batch size.

## F  ADDITIONAL EXPERIMENTAL RESULTS

In this section, we present additional experimental results to further verify our finding.

**Supplementary Plot: Training time should be long enough.** Figures 5(a) and 5(b) show enlarged views for Figures 2(a) and 2(c) respectively, showing that Local SGD can generalize worse than SGD in the first few epochs.

**Supplementary Plot: Learning rate should be small.** Figure 5(c) shows that reducing the learning rate from $0.32$ to $0.064$ does not lead to test accuracy drop for Local SGD on CIFAR-10, if the training time is allowed to be longer and the number of local steps $H$ is set properly. Figure 5(d) presents the case where, with a large learning rate, the generalization improvement of Local SGD disappears even starting from a pre-trained model.

**Supplementary Plot: Reconciling our main finding with Ortiz et al. (2021).** In Figure 5(e), the generalization benefit of Local SGD with $H = 24$ becomes less significant after the learning rate decay at epoch 226, which is consistent with the observation by Ortiz et al. (2021) that the generalization benefit of Local SGD usually disappears after the learning rate decay. But we can preserve the improvement by increasing $H$ to $900$. Here, we use Local SGD with momentum.

**Supplementary Plot: Optimal $\alpha$ gets larger for smaller $\eta$.** In Figure 5(f), we summarize the optimal $\alpha := \eta H$ that enables the highest test accuracy for each learning rate in Figure 2(f). We can see that the optimal $\alpha$ increases as we decrease the learning rate. The reason is that the approximation error bound $\mathcal{O}(\sqrt{\alpha \eta \log \frac{\alpha}{\eta \delta}})$ in Theorem 3.3 decreases with $\eta$, allowing for a larger value of $\alpha$ to better regularize the model.

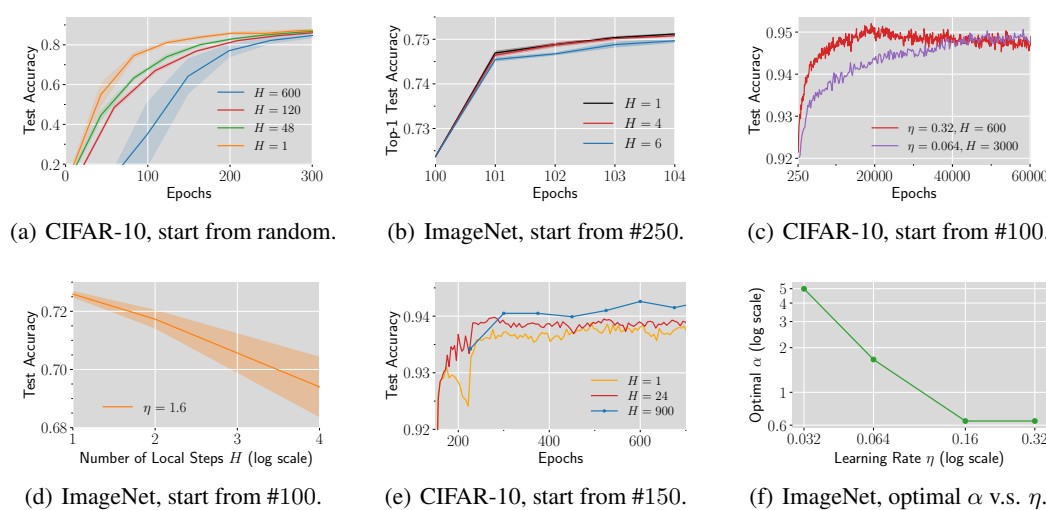

(a) CIFAR-10, start from random.    (b) ImageNet, start from #250.    (c) CIFAR-10, start from #100.

(d) ImageNet, start from #100.    (e) CIFAR-10, start from #150.    (f) ImageNet, optimal $\alpha$ v.s. $\eta$.

Figure 5: Additional experimental results about the effect of the learning rate, training time and the number of local steps. See Appendix M.2 for details.

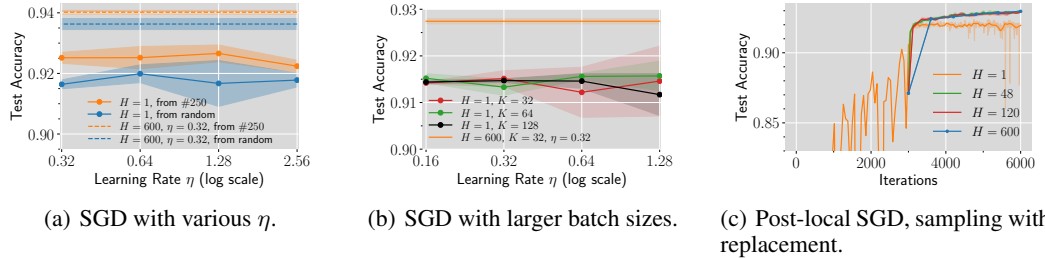

(a) SGD with various $\eta$.    (b) SGD with larger batch sizes.    (c) Post-local SGD, sampling with replacement.

Figure 6: Additional experimental results on CIFAR-10. See Appendix M.3 for details.

**SGD generalizes worse even with extensively tuned learning rates.** In Figure 6(a), we run SGD from both random initialization and the pre-trained model for another $3,000$ epochs with various learning rates and report the test accuracy. We can see that none of the SGD runs beat Local SGD with the fixed learning rate $\eta = 0.32$. Therefore, the inferior performance of SGD in Figures 2(a) and 2(b) is not due to the improper learning rate and Local SGD indeed generalizes better.

**SGD with larger batch sizes performs no better.** In Figure 6(b), we enlarge the batch size of SGD and report the test accuracy for various learning rates. We can see that SGD with larger batch sizes performs no better and none of the SGD runs outperform Local SGD with the fixed learning rate $\eta = 0.32$. This result is unsurprising since it is well established in the literature (Jastrzębski et al., 2017; Smith et al., 2020; Keskar et al., 2017) that larger batch size typically leads to worse generalization. See Appendix A for a survey of empirical and theoretical works on understanding and resolving this phenomenon.

**Sampling with or without replacement does not matter.** Note that there is a slight discrepancy in sampling schemes between our theoretical and experimental setup: the update rules (1) and (2) assume that data are sampled with replacement while most experiments use sampling without replacement (Appendix C). To eliminate the effect of this discrepancy, we conduct additional experiments on Post-local SGD using sampling with replacement (see Figure 6(c)) and Post-local SGD significantly outperforms SGD.

# G  DISCUSSIONS ON LOCAL SGD WITH LABEL NOISE REGULARIZATION

## G.1  THE SLOW SDE FOR LOCAL SGD WITH LABEL NOISE REGULARIZATION

In this subsection, we present the Slow SDE for Local SGD in the case of label noise regularization and show that Local SGD indeed induces a stronger regularization term, which presumably leads to better generalization.

**Theorem G.1** (Slow SDE for Local SGD with label noise regularization). *For a $C$-class classification task with cross-entropy loss, the slow SDE of Local SGD with label noise has the following form:*

$$d\boldsymbol{\zeta}(t) = -\frac{1}{4B}\nabla_\Gamma\left(\text{tr}(\nabla^2\mathcal{L}(\boldsymbol{\zeta})) + (K-1)\cdot\frac{\text{tr}(F(2H\eta\nabla^2\mathcal{L}(\boldsymbol{\zeta})))}{2H\eta}\right)dt, \tag{15}$$

*where $F(x) := \int_0^x \psi(y)dy$ and is interpreted as a matrix function. Additionally, $\nabla_\Gamma f$ stands for the gradient of a function $f$ projected to the tangent space of $\Gamma$.*

*Proof.* See Appendix L. $\qquad\square$

Note that the magnitude of the RHS in (15) becomes larger as $H$ increases. By letting $H$ to go to infinity, we further have the following theorem.

**Theorem G.2.** *As the number of local steps $H$ goes to infinity, the slow SDE of Local SGD with label noise (15) can be simplified as:*

$$d\boldsymbol{\zeta}(t) = -\frac{K}{4B}\nabla_\Gamma\text{tr}(\nabla^2\mathcal{L}(\boldsymbol{\zeta}))dt. \tag{16}$$

*Proof.* We obtain the corollary by simply taking the limit. By L'Hospital's rule,

$$\lim_{x\to+\infty}\frac{F(ax)}{x} = \lim_{x\to+\infty}\frac{dF(ax)}{dx} = \lim_{x\to+\infty}a\psi(ax) = a.$$

Therefore,

$$\lim_{x\to+\infty}\frac{\text{tr}(F(2H\eta\nabla^2\mathcal{L}(\boldsymbol{\zeta})))}{2H\eta} = \text{tr}(\nabla^2\mathcal{L}(\boldsymbol{\zeta})). \tag{17}$$

Substituting (17) into (15) yields (16). $\qquad\square$

As introduced in Section 3.3, the Slow SDE for SGD with label noise regularization has the following form:

$$d\boldsymbol{\zeta}(t) = -\frac{1}{4B}\nabla_\Gamma\text{tr}(\nabla^2\mathcal{L}(\boldsymbol{\zeta}))dt, \tag{18}$$

which is a deterministic flow that keeps reducing the trace of Hessian.

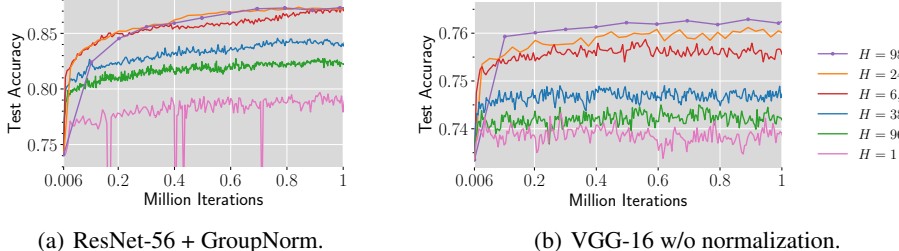

(a) ResNet-56 + GroupNorm.  (b) VGG-16 w/o normalization.

Figure 7: Local SGD with label noise regularization on CIFAR-10 without data augmentation using $K = 32$, $B_{\text{loc}} = 128$. A larger number of local steps indeed enables higher test accuracy. For both architectures, we replace ReLU with Swish. See Appendix M.6 for training details.

As the trace of Hessian can be seen as a measure for the sharpness of the local loss landscape, (18) indicates that SGD with label noise regularization has an implicit bias toward flatter minima, which presumably promotes generalization (Hochreiter & Schmidhuber, 1997; Keskar et al., 2017; Neyshabur et al., 2017). More concretely, Blanc et al. (2020) and Li et al. (2021b) connect minimizing the trace of Hessian to finding sparse or low-rank solutions for training two-layer linear nets. Damian et al. (2021) empirically showed that good generalization correlates with a smaller trace of Hessian in training ResNets with label noise. Besides, Ma & Ying (2021) connect the trace of Hessian to the smoothness of the function represented by a deep neural net.

From Theorems G.1 and G.2, we can conclude that Local SGD accelerates the process of sharpness reduction, thereby leading to better generalization. Furthermore, the regularization effect gets stronger for larger $H$ and is approximately $K$ times that of SGD. We also conduct experiments on non-augmented CIFAR-10 with label noise regularization to verify our conclusion. As shown in Figure 7, increasing the number of local steps indeed gives better generalization performance.

## G.2 THE EQUIVALENCE OF ENLARGING THE LEARNING RATE AND ADDING LOCAL STEPS

In this subsection, we explain in detail why training with label noise regularization is a special case where enlarging the learning rate of SGD can bring the same generalization benefit as adding local steps. TWhen we scale up the learning rate of SGD $\eta \mapsto \kappa\eta$ (while keeping other hyperparameters unchanged), the corresponding Slow SDE is (18) with time horizon $\kappa^2 T$ instead of $T$, where SGD tracks a continuous interval of $\kappa^2\eta^2$ per step instead of $\eta^2$. After rescaling the time horizon to $T$ so that SGD tracks a continuous interval of $\eta^2$ per step, we obtain

$$\mathrm{d}\boldsymbol{\zeta}(t) = -\frac{\kappa^2}{4B}\nabla_\Gamma \mathrm{tr}(\nabla^2 \mathcal{L}(\boldsymbol{\zeta}))\mathrm{d}t. \tag{19}$$

Let $\kappa = \sqrt{K}$ in (19) and we obtain the same Slow SDE as (16), which is for Local SGD with a large number of local steps. In Figure 8, we conduct experiments to verify that SGD indeed achieves comparable test accuracy to that of Local SGD with a large $H$ if its learning rate is scaled up by $\sqrt{K}$ that of Local SGD.

## H DERIVING THE SLOW SDE AFTER APPLYING THE LSR

In this section, we derive the Slow SDEs for SGD and Local SGD after applying the LSR in Appendix E.2. The results are formally summarized in the following theorems.

**Theorem H.1** (Slow SDE for SGD after applying the LSR). *Let Assumptions 3.1 to 3.3 hold. Assume that we run SGD with learning rate $\eta' = \kappa\eta$ and the number of workers $K' = \kappa K$ for some constant $\kappa > 0$. Let $T > 0$ be a constant and $\boldsymbol{\zeta}(t)$ be the solution to (7) with the initial condition $\boldsymbol{\zeta}(0) = \Phi(\boldsymbol{\theta}_0) \in \Gamma$. Then for any $\mathcal{C}^3$-smooth function $g(\boldsymbol{\theta})$, $\max_{0 \le s \le \frac{\kappa T}{\eta'^2}} \left| \mathbb{E}[g(\Phi(\boldsymbol{\theta}_s))] - \mathbb{E}[g(\boldsymbol{\zeta}(s\eta'^2/\kappa))] \right| = \tilde{\mathcal{O}}(\eta'^{0.25})$, where $\tilde{\mathcal{O}}(\cdot)$ hides log factors and constants that are independent of $\eta'$ but can depend on $g(\boldsymbol{\theta})$.*

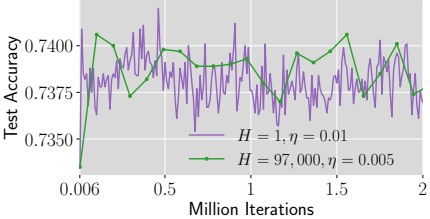

Figure 8: Local SGD with label noise regularization on CIFAR-10 without data augmentation using $K = 4$, $B_{\text{loc}} = 128$. SGD ($H = 1$) indeed achieves comparable test accuracy as Local SGD with a large $H$ when we scale up its learning rate to $\sqrt{K}$ times that of Local SGD. See Appendix M.6 for training details.

*Proof.* Replacing $B$ with $\kappa B$ in the original Slow SDE for Local SGD (7) gives the following Slow SDE:

$$\mathrm{d}\boldsymbol{\zeta}(t) = P_{\boldsymbol{\zeta}}\Big( \underbrace{\tfrac{1}{\sqrt{\kappa B}}\boldsymbol{\Sigma}_{\parallel}^{1/2}(\boldsymbol{\zeta})\mathrm{d}\boldsymbol{W}_t}_{\text{(a) diffusion}} \underbrace{- \tfrac{1}{2\kappa B}\nabla^3\mathcal{L}(\boldsymbol{\zeta})[\widehat{\boldsymbol{\Sigma}}_{\Diamond}(\boldsymbol{\zeta})]\mathrm{d}t}_{\text{(b) drift-I}} \Big). \tag{20}$$

Note that the continuous time horizon for (20) is $\kappa T$ instead of $T$ since after applying the LSR, SGD tracks a continuous interval of $\kappa^2\eta^2$ per step instead of $\eta^2$ while the total number of steps is scaled down by $\kappa$. We can then rescale the time scaling to obtain (7) that holds for $T$. □

**Theorem H.2** (Slow SDE for Local SGD after applying the LSR). *Let Assumptions 3.1 to 3.3 hold. Assume that we run Local SGD with learning rate $\eta' = \kappa\eta$, the number of workers $K' = \kappa K$, and the number of local steps $H' = \frac{\alpha}{\kappa\eta}$ for some constants $\alpha, \kappa > 0$. Let $T > 0$ be a constant and $\boldsymbol{\zeta}(t)$ be the solution to (21) with the initial condition $\boldsymbol{\zeta}(0) = \Phi(\bar{\boldsymbol{\theta}}^{(0)}) \in \Gamma$. Then for any $\mathcal{C}^3$-smooth function $g(\boldsymbol{\theta})$, $\max_{0 \leq s \leq \frac{\kappa T}{H'\eta'^2}} \left| \mathbb{E}[g(\Phi(\bar{\boldsymbol{\theta}}^{(s)}))] - \mathbb{E}[g(\boldsymbol{\zeta}(sH'\eta'^2/\kappa))] \right| = \tilde{\mathcal{O}}(\eta'^{0.25})$, where $\tilde{\mathcal{O}}(\cdot)$ hides log factors and constants that are independent of $\eta'$ but can depend on $g(\boldsymbol{\theta})$.*

$$\mathrm{d}\boldsymbol{\zeta}(t) = P_{\boldsymbol{\zeta}}\Big( \underbrace{\tfrac{1}{\sqrt{B}}\boldsymbol{\Sigma}_{\parallel}^{1/2}(\boldsymbol{\zeta})\mathrm{d}\boldsymbol{W}_t}_{\text{(a) diffusion (unchanged)}} \underbrace{- \tfrac{1}{2B}\nabla^3\mathcal{L}(\boldsymbol{\zeta})[\widehat{\boldsymbol{\Sigma}}_{\Diamond}(\boldsymbol{\zeta})]\mathrm{d}t}_{\text{(b) drift-I (unchanged)}} \underbrace{- \tfrac{\kappa K-1}{2B}\nabla^3\mathcal{L}(\boldsymbol{\zeta})[\widehat{\boldsymbol{\Psi}}(\boldsymbol{\zeta})]\mathrm{d}t}_{\text{(c) drift-II (rescaled)}} \Big). \tag{21}$$

*Proof.* Replacing $B$ with $\kappa B$ in the original Slow SDE for Local SGD (4) gives the following Slow SDE:

$$\mathrm{d}\boldsymbol{\zeta}(t) = P_{\boldsymbol{\zeta}}\Big( \underbrace{\tfrac{1}{\sqrt{\kappa B}}\boldsymbol{\Sigma}_{\parallel}^{1/2}(\boldsymbol{\zeta})\mathrm{d}\boldsymbol{W}_t}_{\text{(a) diffusion}} \underbrace{- \tfrac{1}{2\kappa B}\nabla^3\mathcal{L}(\boldsymbol{\zeta})[\widehat{\boldsymbol{\Sigma}}_{\Diamond}(\boldsymbol{\zeta})]\mathrm{d}t}_{\text{(b) drift-I}} \underbrace{- \tfrac{\kappa K-1}{2\kappa B}\nabla^3\mathcal{L}(\boldsymbol{\zeta})[\widehat{\boldsymbol{\Psi}}(\boldsymbol{\zeta})]\mathrm{d}t}_{\text{(c) drift-II}} \Big). \tag{22}$$

Note that the continuous time horizon for (22) is $\kappa T$ instead of $T$ since after applying the LSR, Local SGD tracks a continuous interval of $\kappa^2\eta^2$ per step instead of $\eta^2$ while the total number of steps is scaled down by $\kappa$. We can then rescale the time scaling to obtain (21) that holds for $T$. □

# I   PROOF OF THEOREM 3.1

This section presents the proof for Theorem 3.1. First, we introduce some notations that will be used throughout this section. For the sequence of Local SGD iterates $\{\boldsymbol{\theta}_{k,t}^{(s)} : k \in [K], 0 \leq t \leq H, s \geq 0\}$, we introduce an auxiliary sequence $\{\hat{\boldsymbol{u}}_t\}_{t \in \mathbb{N}}$, which consists of GD iterates from $\bar{\boldsymbol{\theta}}^{(0)}$:

$$\hat{\boldsymbol{u}}_0 = \bar{\boldsymbol{\theta}}^{(0)}, \qquad \hat{\boldsymbol{u}}_{t+1} \leftarrow \hat{\boldsymbol{u}}_t - \eta \nabla \mathcal{L}(\hat{\boldsymbol{u}}_t).$$

For convenience, let $\hat{\boldsymbol{u}}_t^{(s)} := \hat{\boldsymbol{u}}_{sH+t}$ and $\boldsymbol{z}_{k,sH+t} := \boldsymbol{z}_{k,t}^{(s)}$. We will use $\hat{\boldsymbol{u}}_t^{(s)}$ and $\hat{\boldsymbol{u}}_{sH+t}$, $\boldsymbol{z}_{k,t}^{(s)}$ and $\boldsymbol{z}_{k,sH+t}$ interchangeably. Recall that we have assumed that $\mathcal{L}$ is $\mathcal{C}^3$-smooth with bounded second and third order derivatives. Let $\nu_2 := \sup_{\boldsymbol{\theta} \in \mathbb{R}^d} \|\nabla^2 \mathcal{L}(\boldsymbol{\theta})\|_2$ and $\nu_3 := \sup_{\boldsymbol{\theta} \in \mathbb{R}^d} \|\nabla^3 \mathcal{L}(\boldsymbol{\theta})\|_2$. Since $\nabla \ell(\boldsymbol{\theta}; \boldsymbol{\zeta})$ is bounded, the gradient noise $\boldsymbol{z}_{k,t}^{(s)}$ is also bounded. We denote by $\sigma_{\max}$ an upper bound such that $\|\boldsymbol{z}_{k,t}^{(s)}\|_2 \leq \sigma_{\max}$ holds for all $s, k, t$.

To prove Theorem 3.1, we will show that both Local SGD iterates $\bar{\boldsymbol{\theta}}^{(s)}$ and SGD iterates $\boldsymbol{w}_{sH}$ track GD iterates $\hat{\boldsymbol{u}}_{sH}$ closely with high probability. For each client $k$, define the following sequence $\{\hat{\boldsymbol{Z}}_{k,t} : t \geq 0\}$, which will be used in the proof for bounding the overall effect of noise.

$$\hat{\boldsymbol{Z}}_{k,t} = \sum_{\tau=0}^{t-1} \left[ \prod_{l=\tau+1}^{t-1} (\boldsymbol{I} - \eta \nabla^2 \mathcal{L}(\hat{\boldsymbol{u}}_l)) \right] \boldsymbol{z}_{k,\tau}, \qquad \hat{\boldsymbol{Z}}_{k,0} = \boldsymbol{0}, \qquad \forall k \in [K].$$

The following lemma shows that $\hat{\boldsymbol{Z}}_{k,t}$ is concentrated around the origin.

**Lemma I.1** (Concentration property of $\{\hat{\boldsymbol{Z}}_{k,t}\}$). *With probability at least $1 - \delta$, the following holds simultaneously for all $k \in [K]$, $0 \leq t < \lfloor \frac{T}{\eta} \rfloor$:*

$$\|\hat{\boldsymbol{Z}}_{k,t}\|_2 \leq \hat{C}_1 \sigma_{\max} \sqrt{\frac{2T}{\eta} \log \frac{2TK}{\delta \eta}},$$

*where $\hat{C}_1 := \exp(T\nu_2)$.*

*Proof.* For each $\hat{\boldsymbol{Z}}_{k,t}$, construct a sequence $\{\hat{\boldsymbol{Z}}_{k,t,t'}\}_{t'=0}^{t}$:

$$\hat{\boldsymbol{Z}}_{k,t,t'} := \sum_{\tau=0}^{t'-1} \left( \prod_{l=\tau+1}^{t-1} (\boldsymbol{I} - \eta \nabla^2 \mathcal{L}(\hat{\boldsymbol{u}}_l)) \right) \boldsymbol{z}_{k,\tau}^{(s)}, \qquad \tilde{\boldsymbol{Z}}_{k,t,0}^{(s)} = \boldsymbol{0}.$$

Since $\|\nabla^2 \mathcal{L}(\hat{\boldsymbol{u}}_l)\|_2 \leq \nu_2$ for all $l \geq 0$, the following holds for all $0 \leq \tau < t - 1$ and $0 < t < \lfloor \frac{T}{\eta} \rfloor$:

$$\left\| \prod_{l=\tau+1}^{t-1} (\boldsymbol{I} - \eta \nabla^2 \mathcal{L}(\hat{\boldsymbol{u}}_l)) \right\|_2 \leq (1 + \rho_2 \eta)^t \leq \exp(T\nu_2) = \hat{C}_1.$$

So $\{\hat{\boldsymbol{Z}}_{k,t,t'}\}_{t'=0}^{t}$ is a martingale with $\|\hat{\boldsymbol{Z}}_{k,t,t'} - \hat{\boldsymbol{Z}}_{k,t,t'-1}\|_2 \leq \hat{C}_1 \sigma_{\max}$. Since $\hat{\boldsymbol{Z}}_{k,t} = \hat{\boldsymbol{Z}}_{k,t,t}$, by Azuma-Hoeffding's inequality,

$$\mathbb{P}(\|\hat{\boldsymbol{Z}}_{k,t}\|_2 \geq \epsilon') \leq 2 \exp \left( \frac{-\epsilon'^2}{2t \left( \hat{C}_1 \sigma_{\max} \right)^2} \right).$$

Taking union bound on all $k \in [K]$ and $0 \leq t \leq \lfloor \frac{T}{\eta} \rfloor$, we can conclude that with probability at least $1 - \delta$,

$$\|\hat{\boldsymbol{Z}}_{k,t}\|_2 \leq \hat{C}_1 \sigma_{\max} \sqrt{\frac{2T}{\eta} \log \frac{2TK}{\delta \eta}}, \qquad \forall 0 \leq t < \left\lfloor \frac{T}{\eta} \right\rfloor, k \in [K].$$

$\square$

The following lemma states that, with high probability, Local SGD iterates $\boldsymbol{\theta}_{k,t}^{(s)}$ and $\bar{\boldsymbol{\theta}}^{(s)}$ closely track the gradient descent iterates $\hat{\boldsymbol{u}}_{sH}$ for $\lfloor \frac{T}{H\eta} \rfloor$ rounds.

**Lemma I.2.** *For $\delta = \mathcal{O}(\text{poly}(\eta))$, the following inequalities hold with probability at least $1 - \delta$:*

$$\|\boldsymbol{\theta}_{k,t}^{(s)} - \hat{\boldsymbol{u}}_{sH+t}\|_2 \leq \hat{C}_3 \sqrt{\eta \log \frac{1}{\eta\delta}}, \qquad \forall k \in [K], 0 \leq s < \left\lfloor \frac{T}{H\eta} \right\rfloor, 0 \leq t \leq H,$$

*and*

$$\|\bar{\boldsymbol{\theta}}^{(s)} - \hat{\boldsymbol{u}}_{sH}\|_2 \leq \hat{C}_3 \sqrt{\eta \log \frac{1}{\eta\delta}}, \qquad \forall 0 \leq s \leq \left\lfloor \frac{T}{H\eta} \right\rfloor,$$

*where $\hat{C}_3$ is a constant independent of $\eta$ and $H$.*

*Proof.* Let $\hat{\boldsymbol{\Delta}}_{k,t}^{(s)} := \boldsymbol{\theta}_{k,t}^{(s)} - \hat{\boldsymbol{u}}_t^{(s)}$ and $\bar{\boldsymbol{\Delta}}^{(s)} := \bar{\boldsymbol{\theta}}^{(s)} - \hat{\boldsymbol{u}}_0^{(s)}$ be the differences between the Local SGD and GD iterates. According to the update rule for $\boldsymbol{\theta}_{k,t}^{(s)}$ and $\hat{\boldsymbol{u}}_t^{(s)}$,

$$\boldsymbol{\theta}_{k,t+1}^{(s)} = \boldsymbol{\theta}_{k,t}^{(s)} - \eta\nabla\mathcal{L}(\boldsymbol{\theta}_{k,t}^{(s)}) - \eta\boldsymbol{z}_{k,t}^{(s)} \tag{23}$$

$$\hat{\boldsymbol{u}}_{t+1}^{(s)} = \hat{\boldsymbol{u}}_t^{(s)} - \eta\nabla\mathcal{L}(\hat{\boldsymbol{u}}_t^{(s)}). \tag{24}$$

Subtracting (23) by (24) gives

$$\begin{aligned}
\hat{\boldsymbol{\Delta}}_{k,t+1}^{(s)} &= \hat{\boldsymbol{\Delta}}_{k,t}^{(s)} - \eta(\nabla\mathcal{L}(\boldsymbol{\theta}_{k,t}^{(s)}) - \nabla\mathcal{L}(\hat{\boldsymbol{u}}_t^{(s)})) - \eta\boldsymbol{z}_{k,t}^{(s)} \\
&= (\boldsymbol{I} - \eta\nabla^2\mathcal{L}(\hat{\boldsymbol{u}}_t^{(s)}))\hat{\boldsymbol{\Delta}}_{k,t}^{(s)} - \eta\boldsymbol{z}_{k,t}^{(s)} + \eta\hat{\boldsymbol{v}}_{k,t}^{(s)},
\end{aligned} \tag{25}$$

where $\hat{\boldsymbol{v}}_{k,t}^{(s)}$ is a remainder term with norm $\|\hat{\boldsymbol{v}}_{k,t}^{(s)}\|_2 \leq \frac{\nu_3}{2}\|\hat{\boldsymbol{\Delta}}_{k,t}^{(s)}\|_2^2$. For the $s$-th round of Local SGD, we can apply (25) $t$ times to obtain the following:

$$\begin{aligned}
\hat{\boldsymbol{\Delta}}_{k,t}^{(s)} &= \left[\prod_{\tau=0}^{t-1}(\boldsymbol{I} - \eta\nabla^2\mathcal{L}(\hat{\boldsymbol{u}}_\tau^{(s)}))\right]\hat{\boldsymbol{\Delta}}_{k,0}^{(s)} - \eta\underbrace{\sum_{\tau=0}^{t-1}\left[\prod_{l=\tau+1}^{t-1}(\boldsymbol{I} - \eta\nabla^2\mathcal{L}(\hat{\boldsymbol{u}}_l^{(s)}))\right]\boldsymbol{z}_{k,\tau}^{(s)}}_{\mathcal{T}} \\
&\quad + \eta\sum_{\tau=0}^{t-1}\prod_{l=\tau+1}^{t-1}(\boldsymbol{I} - \eta\nabla^2\mathcal{L}(\hat{\boldsymbol{u}}_l^{(s)}))\hat{\boldsymbol{v}}_{k,\tau}^{(s)}.
\end{aligned} \tag{26}$$

Here, $\mathcal{T}$ can be expressed in the following form:

$$\mathcal{T} = \hat{\boldsymbol{Z}}_{k,sH+t} - \left[\prod_{l=sH}^{sH+t-1}(\boldsymbol{I} - \eta\nabla^2\mathcal{L}(\hat{\boldsymbol{u}}_l))\right]\hat{\boldsymbol{Z}}_{k,sH}.$$

Substituting in $t = H$ and taking the average, we derive the following recursion:

$$\begin{aligned}
\bar{\boldsymbol{\Delta}}^{(s+1)} &= \frac{1}{K}\sum_{k\in[K]}\hat{\boldsymbol{\Delta}}_{k,H}^{(s)} \\
&= \left[\prod_{\tau=0}^{H-1}(\boldsymbol{I} - \eta\nabla^2\mathcal{L}(\hat{\boldsymbol{u}}_\tau^{(s)}))\right]\bar{\boldsymbol{\Delta}}^{(s)} \\
&\quad - \frac{\eta}{K}\sum_{k\in[K]}\hat{\boldsymbol{Z}}_{k,(s+1)H} + \frac{\eta}{K}\sum_{k\in[K]}\left[\prod_{l=sH}^{(s+1)H-1}(\boldsymbol{I} - \eta\nabla^2\mathcal{L}(\hat{\boldsymbol{u}}_l))\right]\hat{\boldsymbol{Z}}_{k,sH} \\
&\quad + \frac{\eta}{K}\sum_{k\in[K]}\sum_{\tau=0}^{H-1}\prod_{l=\tau+1}^{H-1}(\boldsymbol{I} - \eta\nabla^2\mathcal{L}(\hat{\boldsymbol{u}}_l^{(s)}))\hat{\boldsymbol{v}}_{k,\tau}^{(s)}.
\end{aligned} \tag{27}$$

Applying (27) $s$ times yields

$$\bar{\boldsymbol{\Delta}}^{(s)} = -\frac{\eta}{K} \sum_{k \in [K]} \hat{\boldsymbol{Z}}_{k,sH} + \frac{\eta}{K} \sum_{r=0}^{s-1} \sum_{\tau=0}^{H-1} \sum_{k \in [K]} \left[ \prod_{l=rH+\tau+1}^{sH} (\boldsymbol{I} - \eta \nabla^2 \mathcal{L}(\hat{\boldsymbol{u}}_l)) \right] \hat{\boldsymbol{v}}_{k,\tau}^{(r)}. \quad (28)$$

Substitute (28) into (26) and we have

$$\begin{aligned}
\hat{\boldsymbol{\Delta}}_{k,t}^{(s)} = &-\frac{\eta}{K} \sum_{k' \in [K]} \hat{\boldsymbol{Z}}_{k',sH} - \eta \hat{\boldsymbol{Z}}_{k,sH+t} + \eta \left[ \prod_{l=sH}^{sH+t-1} (\boldsymbol{I} - \eta \nabla^2 \mathcal{L}(\hat{\boldsymbol{u}}_l)) \right] \hat{\boldsymbol{Z}}_{k,sH} \\
&+ \frac{\eta}{K} \sum_{r=0}^{s-1} \sum_{\tau=0}^{H-1} \sum_{k' \in [K]} \left[ \prod_{l=rH+\tau+1}^{sH+t-1} (\boldsymbol{I} - \eta \nabla^2 \mathcal{L}(\hat{\boldsymbol{u}}_l)) \right] \hat{\boldsymbol{v}}_{k',\tau}^{(r)} \\
&+ \eta \sum_{\tau=0}^{t-1} \left[ \prod_{l=sH+\tau+1}^{sH+t-1} (\boldsymbol{I} - \eta \nabla^2 \mathcal{L}(\hat{\boldsymbol{u}}_l)) \right] \hat{\boldsymbol{v}}_{k,\tau}^{(s)}.
\end{aligned}$$

By Cauchy-Schwartz inequality and triangle inequality, we have

$$\begin{aligned}
\|\hat{\boldsymbol{\Delta}}_{k,t}^{(s)}\|_2 \leq &\frac{\eta}{K} \left( \sum_{k' \in [K]} \|\hat{\boldsymbol{Z}}_{k',sH}\|_2 \right) + \eta \|\hat{\boldsymbol{Z}}_{k,sH+t}\|_2 + \eta \hat{C}_1 \|\hat{\boldsymbol{Z}}_{k,sH}\|_2 \\
&+ \frac{\eta \hat{C}_1 \nu_3}{2K} \sum_{r=0}^{s-1} \sum_{\tau=0}^{H-1} \sum_{k' \in [K]} \|\hat{\boldsymbol{\Delta}}_{k',\tau}^{(r)}\|_2^2 + \frac{\eta \hat{C}_1 \nu_3}{2} \sum_{\tau=0}^{t-1} \|\hat{\boldsymbol{\Delta}}_{k,\tau}^{(r)}\|_2^2,
\end{aligned} \quad (29)$$

where $\hat{C}_1 = \exp(\nu_2 T)$.

Below we prove by induction that for $\delta = \mathcal{O}(\text{poly}(\eta))$, if

$$\|\hat{\boldsymbol{Z}}_{k,t}\|_2 \leq \hat{C}_1 \sigma_{\max} \sqrt{\frac{2T}{\eta} \log \frac{2TK}{\eta \delta}}, \quad \forall 0 \leq t < \left\lfloor \frac{T}{\eta} \right\rfloor, k \in [K], \quad (30)$$

then there exists a constant $\hat{C}_2$ such that for all $k \in [K], 0 \leq s < \lfloor \frac{T}{\eta H} \rfloor$ and $0 \leq t \leq H$,

$$\|\hat{\boldsymbol{\Delta}}_{k,t}^{(s)}\|_2 \leq \hat{C}_2 \sqrt{\eta \log \frac{2TK}{\eta \delta}}. \quad (31)$$

First, for all $k \in [K], \|\hat{\boldsymbol{\Delta}}_{k,0}^{(0)}\|_2 = 0$ and hence (31) holds. Assuming that (31) holds for all $\hat{\boldsymbol{\Delta}}_{k',\tau}^{(r)}$ where $k' \in [K], 0 \leq r < s, 0 \leq \tau \leq H$ and $r = s, 0 \leq \tau < t$, then by (29), for all $k \in [K]$, the following holds:

$$\|\hat{\boldsymbol{\Delta}}_{k,t}^{(s)}\|_2 \leq 3\hat{C}_1^2 \sigma_{\max} \sqrt{2T\eta \log \frac{2TK}{\eta \delta}} + \hat{C}_1 \hat{C}_2^2 T \eta \nu_3 \log \frac{2TK}{\eta \delta}.$$

Let $\hat{C}_2 \geq 6\hat{C}_1^2 \sigma_{\max} \sqrt{2T}$. Then for sufficiently small $\eta$, (31) holds. By Lemma I.1, (30) holds with probability at least $1 - \delta$. Furthermore, notice that $\bar{\boldsymbol{\theta}}^{(s)} - \hat{\boldsymbol{u}}_{sH} = \frac{1}{K} \sum_{k \in [K]} \hat{\boldsymbol{\Delta}}_{k,H}^{(s-1)}$. Hence we have the lemma. $\qquad \square$

The iterates of standard SGD can be viewed as the local iterates on a single client with the number of local steps $\lfloor \frac{T}{\eta} \rfloor$. Therefore, we can directly apply Lemma I.2 and obtain the following lemma about the SGD iterates $\boldsymbol{w}_t$.

**Corollary I.1.** *For $\delta = \mathcal{O}(\text{poly}(\eta))$, the following holds with probability at least $1 - \delta$:*

$$\|\boldsymbol{w}_{sH} - \hat{\boldsymbol{u}}_{sH}\|_2 \leq \hat{C}_3 \sqrt{\eta \log \frac{1}{\eta \delta}}, \qquad \forall 0 \leq s \leq \frac{T}{H\eta},$$

*where $\hat{C}_3$ is the same constant as in Lemma I.2.*

Applying Lemma I.2 and Corollary I.1 and taking the union bound, we have Theorem 3.1.

## J   PROOF OUTLINE OF MAIN THEOREMS

We adopt the general framework proposed by Li et al. (2019a) to bound the closeness of discrete algorithms and SDE solutions via the method of moments. However, their framework is not directly applicable to our case since they provide approximation guarantees for discrete algorithms with learning rate $\eta$ for $\mathcal{O}(\eta^{-1})$ steps while we want to capture Local SGD for $\mathcal{O}(\eta^{-2})$ steps. To overcome this difficulty, we treat $R_{\mathrm{grp}} := \lfloor \frac{1}{\alpha\eta^\beta} \rfloor$ rounds as a "giant step" of Local SGD with an "effective" learning rate $\eta^{1-\beta}$, where $\beta$ is a constant in $(0, 1)$, and derive the recursive formulas to compute the moments for the change in every step, every round, and every $R_{\mathrm{grp}}$ rounds. The formulation of the recursions requires a detailed analysis of the limiting dynamics of the iterate and careful control of approximation errors.

The dynamics of the iterate can be divided into two phases: the approaching phase (Phase 1) and the drift phase (Phase 2). The approaching phase only lasts for $\mathcal{O}(\log \frac{1}{\eta})$ rounds, during which the iterate is quickly driven to the minimizer manifold by the negative gradient and ends up within only $\tilde{\mathcal{O}}(\sqrt{\eta})$ from $\Gamma$ (see Appendix K.5). After that, the iterate enters the drifting phase and moves in the tangent space of $\Gamma$ while staying close to $\Gamma$ (see Appendix K.6). The closeness of the iterates (local and global) and $\Gamma$ is summarized in the following theorem.

**Theorem J.1** (Closeness of the iterates and $\Gamma$). *For $\delta = \mathcal{O}(\mathrm{poly}(\eta))$, with probability at least $1 - \delta$, for all $\mathcal{O}(\log \frac{1}{\eta}) \leq s \leq \lfloor T/(H\eta^2) \rfloor$,*

$$\Phi(\bar{\boldsymbol{\theta}}^{(s)}) \in \Gamma, \qquad \|\bar{\boldsymbol{\theta}}^{(s)} - \Phi(\bar{\boldsymbol{\theta}}^{(s)})\|_2 = \mathcal{O}\left(\sqrt{\eta \log \frac{1}{\eta\delta}}\right).$$

*Also, for all $\mathcal{O}(\log \frac{1}{\eta}) \leq s < \lfloor T/(H\eta^2) \rfloor$, $k \in [K]$ and $0 \leq t \leq H$,*

$$\|\boldsymbol{\theta}_{k,t}^{(s)} - \Phi(\bar{\boldsymbol{\theta}}^{(s)})\|_2 = \mathcal{O}\left(\sqrt{\eta \log \frac{1}{\eta\delta}}\right).$$

*Here, $\mathcal{O}(\cdot)$ hides constants independent of $\eta$ and $\delta$.*

To control the approximation errors, we also provide a high probability bound for the change of the manifold projection within $R_{\mathrm{grp}}$ rounds.

**Theorem J.2** (High probability bound for the change of manifold projection). *For $\delta = \mathcal{O}(\mathrm{poly}(\eta))$, with probability at least $1 - \delta$, for all $0 \leq s \leq \lfloor T/(H\eta^2) \rfloor - R_{\mathrm{grp}}$ and $0 \leq r \leq R_{\mathrm{grp}}$,*

$$\Phi(\bar{\boldsymbol{\theta}}^{(s)}), \Phi(\bar{\boldsymbol{\theta}}^{(s+r)}) \in \Gamma, \qquad \|\Phi(\bar{\boldsymbol{\theta}}^{(s+r)}) - \Phi(\bar{\boldsymbol{\theta}}^{(s)})\|_2 = \mathcal{O}\left(\eta^{0.5-0.5\beta}\sqrt{\log \frac{1}{\eta\delta}}\right),$$

*where $\mathcal{O}(\cdot)$ hides constants independent of $\eta$ and $\delta$.*

The proof of Theorems J.1 and J.2 is based on the analysis of the dynamics of the iterate and presented in Appendix K.7.

Utilizing Theorems J.1 and J.2, we move on to estimate the first and second moments of the change of the manifold projection every $R_{\mathrm{grp}}$ rounds. However, the randomness during training might drive the iterate far from the manifold (with a low probability, though), making the dynamics intractable. To tackle this issue, we construct a well-behaved auxiliary sequence $\{\hat{\boldsymbol{\theta}}_{k,t}^{(s)}\}$, which is constrained to the neighborhood of $\Gamma$ and equals the original sequence $\{\boldsymbol{\theta}_{k,t}^{(s)}\}$ with high probability (see Definition K.5). Then we can formulate recursions for the change of manifold projection of the auxiliary sequence using the nice properties near $\Gamma$. The estimate of moments is summarized in Theorem K.2.

Finally, based on the moment estimates, we apply the framework in Li et al. (2019a) to show that the manifold projection and the SDE solution are weak approximations of each other in Appendix K.10.

## K   PROOF DETAILS OF MAIN THEOREMS

The detailed proof is organized as follows. In Appendix K.1, we introduce the notations that will be used throughout the proof. To establish preliminary knowledge, Appendix K.2 provides explicit expression for the projection operator $\Phi(\cdot)$, and Appendix K.3 presents lemmas about gradient descent

(GD) and gradient flow (GF). Based on the preliminary knowledge, we construct a nested working zone to characterize the closeness of the iterate and $\Gamma$ in Appendix K.4. Appendices K.5 to K.10 make up the main body of the proof. Specifically, Appendices K.5 and K.6 analyze the dynamics of Local SGD iterates for phases 1 and 2, respectively. Utilizing these analyses, we provide the proof of Theorems J.1 and J.2 in Appendix K.7 and the proof of Theorem 3.3 in Appendix K.8. Then we derive the estimation for the first and second moments of one "giant step" $\Phi(\bar{\boldsymbol{\theta}}^{(s+R_{\mathrm{grp}})}) - \Phi(\bar{\boldsymbol{\theta}}^{(s)})$ in Appendix K.9. Finally, we prove the approximation theorem 3.2 in Appendix K.10.

## K.1 ADDITIONAL NOTATIONS

Let $R_{\mathrm{tot}} := \lfloor \frac{T}{H\eta^2} \rfloor$ be the total number of rounds. Denote by $\phi^{(s)}$ the manifold projection of the global iterate at the beginning of round $s$. Let $\boldsymbol{x}_{k,t}^{(s)} := \boldsymbol{\theta}_{k,t}^{(s)} - \phi^{(s)}$ be the difference between the local iterate and the manifold projection of the global iterate. Also define $\bar{\boldsymbol{x}}_{H}^{(s)} := \frac{1}{K} \sum_{k \in [K]} \boldsymbol{x}_{k,H}^{(s)}$ and $\bar{\boldsymbol{x}}_{0}^{(s)} := \frac{1}{K} \sum_{k \in [K]} \boldsymbol{x}_{k,0}^{(s)}$ which is the average of $\boldsymbol{x}_{k,t}^{(s)}$ among $K$ workers at step $0$ and $H$. Then for all $k \in [K]$, $\boldsymbol{x}_{k,0}^{(s)} = \bar{\boldsymbol{x}}_{0}^{(s)} = \bar{\boldsymbol{\theta}}^{(s)} - \phi^{(s)}$. Finally, Since $\nabla \ell(\boldsymbol{\theta}; \boldsymbol{\zeta})$ is bounded, the gradient noise $\boldsymbol{z}_{k,t}^{(s)}$ is also bounded and we denote by $\sigma_{\max}$ the upper bound such that $\|\boldsymbol{z}_{k,t}^{(s)}\|_2 \leq \sigma_{\max}, \forall s, k, t$.

We first introduce the notion of $\mu$-PL. We will later show that there exists a neighborhood of the minimizer manifold $\Gamma$ where $\mathcal{L}$ satisfies $\mu$-PL.

**Definition K.1** (Polyak-Łojasiewicz Condition). *For $\mu > 0$, we say a function $\mathcal{L}(\cdot)$ satisfies $\mu$-Polyak-Łojasiewicz condition (abbreviated as $\mu$-PL) on set $U$ if*

$$\frac{1}{2}\|\nabla \mathcal{L}(\boldsymbol{\theta})\|_2^2 \geq \mu(\mathcal{L}(\boldsymbol{\theta}) - \inf_{\boldsymbol{\theta}' \in U} \mathcal{L}(\boldsymbol{\theta}')).$$

We then introduce the definitions of the $\epsilon$-ball at a point and the $\epsilon$-neighborhood of a set. For $\boldsymbol{\theta} \in \mathbb{R}^d$ and $\epsilon > 0$, $B^{\epsilon}(\boldsymbol{\theta}) := \{\boldsymbol{\theta}' : \|\boldsymbol{\theta}' - \boldsymbol{\theta}\|_2 < \epsilon\}$ is the open $\epsilon$-ball centered at $\boldsymbol{\theta}$. For a set $\mathcal{Z} \subseteq \mathbb{R}^d$, $\mathcal{Z}^{\epsilon} := \bigcup_{\boldsymbol{\theta} \in \mathcal{Z}} B^{\epsilon}(\boldsymbol{\theta})$ is the $\epsilon$-neighborhood of $\mathcal{Z}$.

## K.2 COMPUTING THE DERIVATIVES OF THE LIMITING MAPPING

In subsection, we present lemmas that relate the derivatives of the limiting mapping $\Phi(\cdot)$ to the derivatives of the loss function $\mathcal{L}(\cdot)$. We first introduce the operator $\mathcal{V}_{\boldsymbol{H}}$.

**Definition K.2.** *For a semi-definite symmetric matrix $\boldsymbol{H} \in \mathbb{R}^{d \times d}$, let $\lambda_j$, $\boldsymbol{v}_j$ be the $j$-th eigenvalue and eigenvector and $\boldsymbol{v}_j$'s form an orthonormal basis of $\mathbb{R}^d$. Then, define the operator $\mathcal{V}_{\boldsymbol{H}} : \mathbb{R}^{d \times d} \to \mathbb{R}^{d \times d}$ as*

$$\mathcal{V}_{\boldsymbol{H}}(\boldsymbol{M}) := \sum_{i,j:\lambda_i \neq 0 \vee \lambda_j \neq 0} \frac{1}{\lambda_i + \lambda_j} \left\langle \boldsymbol{M}, \boldsymbol{v}_i \boldsymbol{v}_j^{\top} \right\rangle \boldsymbol{v}_i \boldsymbol{v}_j^{\top}, \forall \boldsymbol{M} \in \mathbb{R}^{d \times d}.$$

*Intuitively, this operator projects $\boldsymbol{M}$ to the base matrix $\boldsymbol{v}_i \boldsymbol{v}_j^{\top}$ and sums up the projections with weights $\frac{1}{\lambda_i + \lambda_j}$.*

Additionally, for $\boldsymbol{\theta} \in \Gamma$, denote by $T_{\boldsymbol{\theta}}$ and $T_{\boldsymbol{\theta}}^{\perp}$ the tangent and normal space of $\Gamma$ at $\boldsymbol{\theta}$ respectively. Lemmas K.1 to K.4 are from Li et al. (2021b). We include them to make the paper self-contained.

**Lemma K.1** (Lemma C.1 of Li et al. (2021b)). *For any $\boldsymbol{\theta} \in \Gamma$ and any $\boldsymbol{v} \in T_{\boldsymbol{\theta}}(\Gamma)$, it holds that $\nabla^2 \mathcal{L}(\boldsymbol{\theta}) \boldsymbol{v} = \boldsymbol{0}$.*

**Lemma K.2** (Lemma 4.3 of Li et al. (2021b)). *For any $\boldsymbol{\theta} \in \Gamma$, $\partial \Phi(\boldsymbol{\theta}) \in \mathbb{R}^{d \times d}$ is the projection matrix onto the tangent space $T_{\boldsymbol{\theta}}(\Gamma)$.*

**Lemma K.3** (Lemma C.4 of Li et al. (2021b)). *For any $\boldsymbol{\theta} \in \Gamma$, $\boldsymbol{u} \in \mathbb{R}^d$ and $\boldsymbol{v} \in T_{\boldsymbol{\theta}}(\Gamma)$, it holds that*

$$\partial^2 \Phi(\boldsymbol{\theta})[\boldsymbol{v}, \boldsymbol{u}] = -\partial \Phi(\boldsymbol{\theta}) \nabla^3 \mathcal{L}(\boldsymbol{\theta})[\boldsymbol{v}, \nabla^2 \mathcal{L}(\boldsymbol{\theta})^+ \boldsymbol{u}] - \nabla^2 \mathcal{L}(\boldsymbol{\theta})^+ \nabla^3 \mathcal{L}(\boldsymbol{\theta})[\boldsymbol{v}, \partial \Phi(\boldsymbol{\theta}) \boldsymbol{u}].$$

**Lemma K.4** (Lemma C.6 of Li et al. (2021b)). *For any $\boldsymbol{\theta} \in \Gamma$ and $\boldsymbol{\Sigma} \in \mathrm{span}\{\boldsymbol{u}\boldsymbol{u}^{\top} \mid \boldsymbol{u} \in T_{\boldsymbol{\theta}}^{\perp}(\Gamma)\}$,*

$$\left\langle \partial^2 \Phi(\boldsymbol{\theta}), \boldsymbol{\Sigma} \right\rangle = -\partial \Phi(\boldsymbol{\theta}) \nabla^3 \mathcal{L}(\boldsymbol{\theta})[\mathcal{V}_{\nabla^2 \mathcal{L}(\boldsymbol{\theta})}(\boldsymbol{\Sigma})].$$

**Lemma K.5.** *For all $\boldsymbol{\theta} \in \Gamma$, $\boldsymbol{u}, \boldsymbol{v} \in T_{\boldsymbol{\theta}}(\Gamma)$, it holds that*

$$\partial\Phi(\boldsymbol{\theta})\nabla^3\mathcal{L}[\boldsymbol{v}\boldsymbol{u}^\top] = \boldsymbol{0}. \tag{32}$$

*Proof.* This proof is inspired by Lemma C.4 of Li et al. (2021b). For any $\boldsymbol{\theta} \in \Gamma$, consider a parameterized smooth curve $\boldsymbol{v}(t), t \geq 0$ on $\Gamma$ such that $\boldsymbol{v}(0) = \boldsymbol{\theta}$ and $\boldsymbol{v}'(0) = \boldsymbol{v}$. Let $\boldsymbol{P}_\parallel(t) = \partial\Phi(\boldsymbol{v}(t))$, $\boldsymbol{P}_\perp(t) = \boldsymbol{I} - \partial\Phi(\boldsymbol{v}(t))$ and $\boldsymbol{H}(t) = \nabla^2\mathcal{L}(\boldsymbol{v}(t))$. By Lemma C.1 and 4.3 in Li et al. (2021b),

$$\boldsymbol{H}(t) = \boldsymbol{P}_\perp(t)\boldsymbol{H}(t).$$

Take the derivative with respect to $t$ on both sides,

$$\boldsymbol{H}'(t) = \boldsymbol{P}_\perp(t)\boldsymbol{H}'(t) + \boldsymbol{P}_\perp'(t)\boldsymbol{H}(t)$$
$$\Rightarrow \boldsymbol{P}_\parallel(t)\boldsymbol{H}'(t) = \boldsymbol{P}_\perp'(t)\boldsymbol{H}(t) = -\boldsymbol{P}_\parallel'(t)\boldsymbol{H}(t).$$

At $t = 0$, we have

$$\boldsymbol{P}_\parallel(0)\boldsymbol{H}'(0) = -\boldsymbol{P}_\parallel'(0)\boldsymbol{H}(0). \tag{33}$$

WLOG let $\boldsymbol{H}(0) = \mathrm{diag}(\lambda_1, \cdots, \lambda_d), \in \mathbb{R}^{d \times d}$, where $\lambda_i = 0$ for all $m < i \leq d$. Therefore $\boldsymbol{P}_\perp(0) = \begin{bmatrix} \boldsymbol{I}_m & \boldsymbol{0} \\ \boldsymbol{0} & \boldsymbol{0} \end{bmatrix}$, $\boldsymbol{P}_\parallel(0) = \begin{bmatrix} \boldsymbol{0} & \boldsymbol{0} \\ \boldsymbol{0} & \boldsymbol{I}_{d-m} \end{bmatrix}$. Decompose $\boldsymbol{P}_\parallel'(0)$, $\boldsymbol{H}(0)$ and $\boldsymbol{H}'(0)$ as follows.

$$\boldsymbol{P}_\parallel'(0) = \begin{bmatrix} \boldsymbol{P}_{\parallel,11}'(0) & \boldsymbol{P}_{\parallel,12}'(0) \\ \boldsymbol{P}_{\parallel,21}'(0) & \boldsymbol{P}_{\parallel,22}'(0) \end{bmatrix}, \boldsymbol{H}(0) = \begin{bmatrix} \boldsymbol{H}_{11}(0) & \boldsymbol{0} \\ \boldsymbol{0} & \boldsymbol{0} \end{bmatrix}, \boldsymbol{H}'(0) = \begin{bmatrix} \boldsymbol{H}_{11}'(0) & \boldsymbol{H}_{12}'(0) \\ \boldsymbol{H}_{21}'(0) & \boldsymbol{H}_{22}'(0) \end{bmatrix}.$$

Substituting the decomposition into (33), we have

$$\begin{bmatrix} \boldsymbol{0} & \boldsymbol{0} \\ \boldsymbol{H}_{21}'(0) & \boldsymbol{H}_{22}'(0) \end{bmatrix} = -\begin{bmatrix} \boldsymbol{P}_{\parallel,11}'(0)\boldsymbol{H}_{11}(0) & \boldsymbol{0} \\ \boldsymbol{P}_{\parallel,21}'(0)\boldsymbol{H}_{11}(0) & \boldsymbol{0} \end{bmatrix}.$$

Therefore, $\boldsymbol{H}_{22}'(0) = \boldsymbol{0}$ and

$$\boldsymbol{P}_\parallel(0)\boldsymbol{H}'(0) = -\boldsymbol{P}_\parallel'(0)\boldsymbol{H}(0) = -\begin{bmatrix} \boldsymbol{0} & \boldsymbol{0} \\ \boldsymbol{H}_{21}'(0) & \boldsymbol{0} \end{bmatrix}.$$

Any $\boldsymbol{u} \in T_{\boldsymbol{\theta}}(\Gamma)$ can be decomposed as $\boldsymbol{u} = [\boldsymbol{0}, \boldsymbol{u}_2]^\top$ where $\boldsymbol{u}_2 \in \mathbb{R}^{d-m}$. With this decomposition, we have $\boldsymbol{P}_\parallel(0)\boldsymbol{H}'(0)\boldsymbol{u} = \boldsymbol{0}$. Also, note that $\boldsymbol{H}'(0) = \nabla^3\mathcal{L}(\boldsymbol{\theta})[\boldsymbol{v}]$. Hence,

$$\partial\Phi(\boldsymbol{\theta})\nabla^3\mathcal{L}(\boldsymbol{\theta})[\boldsymbol{v}\boldsymbol{u}^T] = \boldsymbol{0}.$$

$\square$

### K.3 Preliminary Lemmas for GD and GF

In this subsection, we introduce a few useful preliminary lemmas about gradient descent and gradient flow. Before presenting the lemmas, we introduce some notations and assumptions that will be used in this subsection.

Assume that the loss function $\mathcal{L}(\boldsymbol{\theta})$ is $\rho$-smooth and $\mu$-PL in an open, convex neighborhood $U$ of a local minimizer $\boldsymbol{\theta}^*$. Denote by $\mathcal{L}^* := \mathcal{L}(\boldsymbol{\theta}^*)$ the minimum value for simplicity. Let $\epsilon'$ be the radius of the open $\epsilon'$-ball centered at $\boldsymbol{\theta}^*$ such that $B^{\epsilon'}(\boldsymbol{\theta}^*) \subseteq U$. We also define a potential function $\tilde{\Psi}(\boldsymbol{\theta}) := \sqrt{\mathcal{L}(\boldsymbol{\theta}) - \mathcal{L}^*}$.

Consider gradient descent iterates $\{\hat{\boldsymbol{u}}_t\}_{t \in \mathbb{N}}$ following the update rule $\hat{\boldsymbol{u}}_{t+1} = \hat{\boldsymbol{u}}_t - \eta\nabla\mathcal{L}(\hat{\boldsymbol{u}}_t)$. We first introduce the descent lemma for gradient descent.

**Lemma K.6** (Descent lemma for GD). *If $\hat{\boldsymbol{u}}_t \in U$ and $\eta \leq \frac{1}{\rho}$, then*

$$\frac{\eta}{2}\|\nabla\mathcal{L}(\hat{\boldsymbol{u}}_t)\|_2^2 \leq \mathcal{L}(\hat{\boldsymbol{u}}_t) - \mathcal{L}(\hat{\boldsymbol{u}}_{t+1}),$$

*and*

$$\mathcal{L}(\hat{\boldsymbol{u}}_{t+1}) - \mathcal{L}^* \leq (1 - \mu\eta)(\mathcal{L}(\hat{\boldsymbol{u}}_t) - \mathcal{L}^*).$$

*Proof.* By $\rho$-smoothness,

$$
\begin{aligned}
\mathcal{L}(\hat{\boldsymbol{u}}_{t+1}) &\leq \mathcal{L}(\hat{\boldsymbol{u}}_t) + \langle \nabla \mathcal{L}(\hat{\boldsymbol{u}}_t), \hat{\boldsymbol{u}}_{t+1} - \hat{\boldsymbol{u}}_t \rangle + \frac{\rho \eta^2}{2} \|\hat{\boldsymbol{u}}_{t+1} - \hat{\boldsymbol{u}}_t\|_2^2 \\
&= \mathcal{L}(\hat{\boldsymbol{u}}_t) - \eta(1 - \frac{\rho \eta}{2}) \|\nabla \mathcal{L}(\hat{\boldsymbol{u}}_t)\|_2^2 \\
&\leq \mathcal{L}(\hat{\boldsymbol{u}}_t) - \frac{\eta}{2} \|\nabla \mathcal{L}(\hat{\boldsymbol{u}}_t)\|_2^2
\end{aligned}
$$

By the definition of $\mu$-PL, we have

$$
\mathcal{L}(\hat{\boldsymbol{u}}_{t+1}) - \mathcal{L}^* \leq (1 - \mu\eta)(\mathcal{L}(\hat{\boldsymbol{u}}_t) - \mathcal{L}^*).
$$

$\square$

Then we prove the Lipschitzness of $\tilde{\Psi}(\boldsymbol{\theta})$.

**Lemma K.7** (Lipschitzness of $\tilde{\Psi}(\boldsymbol{\theta})$)**.** $\tilde{\Psi}(\boldsymbol{\theta})$ *is* $\sqrt{2\rho}$-*Lipschitz for* $\boldsymbol{\theta} \in U$. *That is, for any* $\boldsymbol{\theta}_1$, $\boldsymbol{\theta}_2 \in U$,

$$
|\tilde{\Psi}(\boldsymbol{\theta}_1) - \tilde{\Psi}(\boldsymbol{\theta}_2)| \leq \sqrt{2\rho} \|\boldsymbol{\theta}_1 - \boldsymbol{\theta}_2\|_2.
$$

*Proof.* Fix $\boldsymbol{\theta}_1$ and $\boldsymbol{\theta}_2$. Denote by $\boldsymbol{\theta}(t) := (1 - t)\boldsymbol{\theta}_1 + t\boldsymbol{\theta}_2$ the convex combination of $\boldsymbol{\theta}_1$ and $\boldsymbol{\theta}_2$ where $t \in [0, 1]$. Further define $f(t) := \tilde{\Psi}(\boldsymbol{\theta}(t))$. Below we consider two cases.

**Case 1.** If $\forall t \in (0, 1)$, $f(t) > 0$, then $f(t)$ is differentiable on $(0, 1)$.

$$
\begin{aligned}
|\tilde{\Psi}(\boldsymbol{\theta}_2) - \tilde{\Psi}(\boldsymbol{\theta}_1)| &= |f(1) - f(0)| \\
&= \left| \int_0^1 f'(t) \mathrm{d}t \right| \\
&= \left| \int_0^1 \left\langle \nabla \tilde{\Psi}(\boldsymbol{\theta}(t)), \boldsymbol{\theta}_2 - \boldsymbol{\theta}_1 \right\rangle \mathrm{d}t \right| \\
&= \left| \int_0^1 \frac{\langle \nabla \mathcal{L}(\boldsymbol{\theta}(t)), \boldsymbol{\theta}_2 - \boldsymbol{\theta}_1 \rangle}{\sqrt{\mathcal{L}(\boldsymbol{\theta}(t)) - \mathcal{L}^*}} \mathrm{d}t \right| \\
&\leq \|\boldsymbol{\theta}_2 - \boldsymbol{\theta}_1\|_2 \int_0^1 \frac{\|\nabla \mathcal{L}(\boldsymbol{\theta}(t))\|_2}{\sqrt{\mathcal{L}(\boldsymbol{\theta}(t)) - \mathcal{L}^*}} \mathrm{d}t.
\end{aligned}
$$

By $\rho$-smoothness of $\mathcal{L}$, for all $\boldsymbol{\theta} \in U$,

$$
\|\nabla \mathcal{L}(\boldsymbol{\theta})\|_2^2 \leq 2\rho \left( \mathcal{L}(\boldsymbol{\theta}) - \mathcal{L}^* \right).
$$

Since $\sqrt{\mathcal{L}(\boldsymbol{\theta}(t)) - \mathcal{L}^*} > 0$ for all $t \in (0, 1)$, $\frac{\|\nabla \mathcal{L}(\boldsymbol{\theta}(t))\|_2}{\sqrt{\mathcal{L}(\boldsymbol{\theta}(t)) - \mathcal{L}^*}} \leq \sqrt{2\rho}$. Therefore,

$$
|\tilde{\Psi}(\boldsymbol{\theta}_2) - \tilde{\Psi}(\boldsymbol{\theta}_1)| \leq \sqrt{2\rho_2} \|\boldsymbol{\theta}_2 - \boldsymbol{\theta}_1\|_2.
$$

**Case 2.** If $\exists t' \in (0, 1)$ such that $f(t') = 0$, then

$$
\begin{aligned}
|\tilde{\Psi}(\boldsymbol{\theta}_2) - \tilde{\Psi}(\boldsymbol{\theta}_1)| &= |f(1) - f(0)| \\
&= \left| (1 - t') \frac{f(1) - f(t')}{1 - t'} + t' \left( \frac{f(t') - f(0)}{t'} \right) \right| \\
&\leq \max \left( \frac{f(1)}{1 - t'}, \frac{f(0)}{t'} \right).
\end{aligned}
$$

Since $\boldsymbol{\theta}(t')$ minimizes $\mathcal{L}$ in an open set, $\nabla \mathcal{L}(\boldsymbol{\theta}(t')) = \mathbf{0}$. By $\rho$-smoothness of $\mathcal{L}$, for all $\boldsymbol{\theta} \in U$,

$$
\mathcal{L}(\boldsymbol{\theta}) \leq \mathcal{L}^* + \frac{\rho}{2} \|\boldsymbol{\theta} - \boldsymbol{\theta}(t')\|_2^2 \quad \Rightarrow \quad \tilde{\Psi}(\boldsymbol{\theta}) \leq \sqrt{\frac{\rho}{2}} \|\boldsymbol{\theta} - \boldsymbol{\theta}(t')\|_2.
$$

Therefore,

$$f(1) \leq \sqrt{\frac{\rho}{2}} \|\boldsymbol{\theta}_2 - \boldsymbol{\theta}(t')\|_2 = (1 - t') \sqrt{\frac{\rho}{2}} \|\boldsymbol{\theta}_2 - \boldsymbol{\theta}_1\|_2$$

$$f(0) \leq \sqrt{\frac{\rho}{2}} \|\boldsymbol{\theta}_1 - \boldsymbol{\theta}(t')\|_2 = t' \sqrt{\frac{\rho}{2}} \|\boldsymbol{\theta}_2 - \boldsymbol{\theta}_1\|_2.$$

Then we have

$$|\tilde{\Psi}(\boldsymbol{\theta}_2) - \tilde{\Psi}(\boldsymbol{\theta}_1)| \leq \sqrt{\frac{\rho}{2}} \|\boldsymbol{\theta}_2 - \boldsymbol{\theta}_1\|_2.$$

Combining case 1 and case 2, we conclude the proof. $\quad\square$

Below we introduce a lemma that relates the movement of one step gradient descent to the change of the potential function.

**Lemma K.8** (Lemma G.1 in Lyu et al. (2022)). *If $\hat{\boldsymbol{u}}_t \in U$ and $\eta \leq 1/\rho_2$ then*

$$\tilde{\Psi}(\hat{\boldsymbol{u}}_t) - \tilde{\Psi}(\hat{\boldsymbol{u}}_{t+1}) \geq \frac{\sqrt{2\mu}}{4} \eta \|\nabla \mathcal{L}(\hat{\boldsymbol{u}}_t)\|_2.$$

*Proof.*

$$\begin{aligned}
\tilde{\Psi}(\hat{\boldsymbol{u}}_t) - \tilde{\Psi}(\hat{\boldsymbol{u}}_{t+1}) &= \frac{\mathcal{L}(\hat{\boldsymbol{u}}_t) - \mathcal{L}(\hat{\boldsymbol{u}}_{t+1})}{\tilde{\Psi}(\hat{\boldsymbol{u}}_t) + \tilde{\Psi}(\hat{\boldsymbol{u}}_{t+1})} \\
&\geq \frac{\mathcal{L}(\hat{\boldsymbol{u}}_{t+1}) - \mathcal{L}(\hat{\boldsymbol{u}}_t)}{2\tilde{\Psi}(\hat{\boldsymbol{u}}_t)} \\
&\geq \frac{\eta(1 - \rho_2\eta/2)\|\nabla\mathcal{L}(\hat{\boldsymbol{u}}_t)\|_2^2}{2\tilde{\Psi}(\hat{\boldsymbol{u}}_t)},
\end{aligned}$$

where the two inequalities uses Lemma K.6. By $\mu$-PL, $\tilde{\Psi}(\hat{\boldsymbol{u}}_t) \leq \frac{1}{\sqrt{2\mu}} \|\nabla\mathcal{L}(\hat{\boldsymbol{u}}_t)\|_2$. Therefore, we have $\tilde{\Psi}(\hat{\boldsymbol{u}}_t) - \tilde{\Psi}(\hat{\boldsymbol{u}}_{t+1}) \geq \frac{\sqrt{2\mu}}{2}(1 - \eta\rho/2)\eta\|\nabla\mathcal{L}(\hat{\boldsymbol{u}}_t)\|_2 \geq \frac{\sqrt{2\mu}}{4}\eta\|\nabla\mathcal{L}(\hat{\boldsymbol{u}}_t)\|_2$. $\quad\square$

Based on Lemma K.8, we have the following lemma that bounds the movement of GD over multiple steps.

**Lemma K.9** (Bounding the movement of GD). *If $\hat{\boldsymbol{u}}_0$ is initialized such that $\|\hat{\boldsymbol{u}}_0 - \boldsymbol{\theta}^*\|_2 \leq \frac{1}{4}\sqrt{\frac{\mu}{\rho}}\epsilon'$, then for all $t \geq 0$, $\hat{\boldsymbol{u}}_t \in B^{\epsilon'}(\boldsymbol{\theta}^*)$ and*

$$\|\hat{\boldsymbol{u}}_t - \hat{\boldsymbol{u}}_0\|_2 \leq \sqrt{\frac{8}{\mu}} \tilde{\Psi}(\hat{\boldsymbol{u}}_0).$$

*Proof.* We prove the proposition by induction. When $t = 0$, it trivially holds. Assume that the proposition holds for $\hat{\boldsymbol{u}}_\tau$, $0 \leq \tau < t$. For step $t$, since $\hat{\boldsymbol{u}}_\tau \in B^{\epsilon'}(\boldsymbol{\theta}^*)$, we apply Lemma K.8 and obtain

$$\|\hat{\boldsymbol{u}}_t - \hat{\boldsymbol{u}}_0\|_2 \leq \eta \sum_{\tau=0}^{t-1} \|\nabla\mathcal{L}(\hat{\boldsymbol{u}}_\tau)\|_2 \leq \sqrt{\frac{8}{\mu}} \left( \tilde{\Psi}(\hat{\boldsymbol{u}}_0) - \tilde{\Psi}(\hat{\boldsymbol{u}}_t) \right) \leq \sqrt{\frac{8}{\mu}} \tilde{\Psi}(\hat{\boldsymbol{u}}_0).$$

Further by $\rho$-smoothness of $\mathcal{L}(\cdot)$,

$$\|\hat{\boldsymbol{u}}_t - \hat{\boldsymbol{u}}_0\|_2 \leq \sqrt{\frac{8}{\mu}} \tilde{\Psi}(\hat{\boldsymbol{u}}_0) \leq 2\sqrt{\frac{\rho}{\mu}} \|\hat{\boldsymbol{u}}_0 - \boldsymbol{\theta}^*\|_2 \leq \frac{1}{2}\epsilon'.$$

Therefore, $\|\hat{\boldsymbol{u}}_t - \boldsymbol{\theta}^*\|_2 \leq \|\hat{\boldsymbol{u}}_t - \hat{\boldsymbol{u}}_0\|_2 + \|\hat{\boldsymbol{u}}_0 - \boldsymbol{\theta}^*\|_2 < \epsilon'$, which concludes the proof. $\quad\square$

Finally, we introduce a lemma adapted from Thm. D.4 of which bounds the movement of GF. Lyu et al. (2022).

**Lemma K.10.** *Assume that $\|\boldsymbol{\theta}_0 - \boldsymbol{\theta}^*\|_2 < \sqrt{\frac{\mu}{\rho}}\epsilon'$. The gradient flow $\boldsymbol{\theta}(t) = -\frac{\mathrm{d}\mathcal{L}(\boldsymbol{\theta}(t))}{\mathrm{d}t}$ starting at $\boldsymbol{\theta}_0$ converges to a point in $U$ and*

$$\left\| \boldsymbol{\theta}_0 - \lim_{t \to +\infty} \boldsymbol{\theta}(t) \right\|_2 \leq \sqrt{\frac{2}{\mu}} \sqrt{\mathcal{L}(\boldsymbol{\theta}_0) - \mathcal{L}^*} \leq \sqrt{\frac{\rho}{\mu}} \|\boldsymbol{\theta}_0 - \boldsymbol{\theta}^*\|_2$$

*Proof.* Let $T := \inf\{t : \boldsymbol{\theta} \notin U\}$. Then for all $t < T$,

$$\frac{\mathrm{d}}{\mathrm{d}t} (\mathcal{L}(\boldsymbol{\theta}) - \mathcal{L}^*)^{1/2} = \frac{1}{2} (\mathcal{L}(\boldsymbol{\theta}) - \mathcal{L}^*)^{-1/2} \cdot \left\langle \nabla\mathcal{L}(\boldsymbol{\theta}), \frac{\mathrm{d}\boldsymbol{\theta}}{\mathrm{d}t} \right\rangle$$

$$= -\frac{1}{2} (\mathcal{L}(\boldsymbol{\theta}) - \mathcal{L}^*)^{-1/2} \|\nabla\mathcal{L}(\boldsymbol{\theta})\|_2 \|\frac{\mathrm{d}\boldsymbol{\theta}}{\mathrm{d}t}\|_2.$$

By $\mu$-PL, $\|\nabla\mathcal{L}(\boldsymbol{\theta})\|_2 \geq \sqrt{2\mu(\mathcal{L}(\boldsymbol{\theta}) - \mathcal{L}^*)}$. Hence,

$$\frac{\mathrm{d}}{\mathrm{d}t} (\mathcal{L}(\boldsymbol{\theta}) - \mathcal{L}^*)^{1/2} \leq -\frac{\sqrt{2\mu}}{2} \|\frac{\mathrm{d}\boldsymbol{\theta}}{\mathrm{d}t}\|_2.$$

Integrating both sides, we have

$$\int_0^T \|\frac{\mathrm{d}\boldsymbol{\theta}(\tau)}{\mathrm{d}\tau}\|\mathrm{d}\tau \leq \frac{2}{\sqrt{2\mu}} (\mathcal{L}(\boldsymbol{\theta}_0) - \mathcal{L}^*)^{1/2} \leq \sqrt{\frac{\rho}{\mu}} \|\boldsymbol{\theta}_0 - \boldsymbol{\theta}^*\|_2 < \epsilon',$$

where the second inequality uses $\rho$-smoothness of $\mathcal{L}$. Therefore, $T = +\infty$ and $\boldsymbol{\theta}(t)$ converges to some point in $U$. $\qquad\square$

### K.4 CONSTRUCTION OF WORKING ZONES

We construct four nested working zones $(\Gamma^{\epsilon_0}, \Gamma^{\epsilon_1}, \Gamma^{\epsilon_2}, \Gamma^{\epsilon_3})$ in the neighborhood of $\Gamma$. Later we will show that the local iterates $\boldsymbol{\theta}_{k,t}^{(s)} \in \Gamma^{\epsilon_2}$ and the global iterates $\bar{\boldsymbol{\theta}}^{(s)} \in \Gamma^{\epsilon_0}$ with high probability after $\mathcal{O}(\log \frac{1}{\eta})$ rounds. The following lemma illustrates the properties the working zones should satisfy.

**Lemma K.11** (Working zone lemma). *There exists constants $\epsilon_0 < \epsilon_1 < \epsilon_2 < \epsilon_3$ such that $(\Gamma^{\epsilon_0}, \Gamma^{\epsilon_1}, \Gamma^{\epsilon_2}, \Gamma^{\epsilon_3})$ satisfy the following properties:*

1. *$\mathcal{L}$ satisfies $\mu$-PL in $\Gamma^{\epsilon_3}$ for some $\mu > 0$.*

2. *Any gradient flow starting in $\Gamma^{\epsilon_2}$ converges to some point in $\Gamma$. Then, by Falconer (1983), $\Phi(\cdot)$ is $\mathcal{C}^\infty$ in $\Gamma^{\epsilon_2}$.*

3. *Any $\boldsymbol{\theta} \in \Gamma^{\epsilon_1}$ has an $\epsilon_1$-neighborhood $B^{\epsilon_1}(\boldsymbol{\theta})$ such that $B^{\epsilon_1}(\boldsymbol{\theta}) \subseteq \Gamma^{\epsilon_2}$.*

4. *Any gradient descent starting in $\Gamma^{\epsilon_0}$ with sufficiently small learning rate will stay in $\Gamma^{\epsilon_1}$.*

*Proof.* Let $\bar{\boldsymbol{\theta}}^{(0)}$ be initialized such that $\Phi(\bar{\boldsymbol{\theta}}^{(0)}) \in \Gamma$. Let $\mathcal{Z}$ be the set of all points on the gradient flow trajectory starting from $\bar{\boldsymbol{\theta}}^{(0)}$ and $\mathcal{Z}^\epsilon$ be the $\epsilon$-neighborhood of $\mathcal{Z}$, where $\epsilon$ is a positive constant. Since the gradient flow converges to $\phi^{(0)}$, $\mathcal{Z}$ and $\mathcal{Z}^\epsilon$ are bounded.

We construct four nested working zones. By Lemma H.3 in Lyu et al. (2022), there exists an $\epsilon_3$-neighborhood of $\Gamma$, $\Gamma^{\epsilon_3}$, such that $\mathcal{L}$ satisfies $\mu$-PL for some $\mu > 0$. Let $\mathcal{M}$ be the convex hull of $\Gamma^{\epsilon_3} \cup \mathcal{Z}^\epsilon$ and $\mathcal{M}^{\epsilon_4}$ be the $\epsilon_4$-neighborhood of $\mathcal{M}$ where $\epsilon_4$ is a positive constant. Then $\mathcal{M}^{\epsilon_4}$ is bounded.

Define $\rho_2 = \sup_{\boldsymbol{\theta} \in \mathcal{M}^{\epsilon_4}} \|\nabla^2\mathcal{L}(\boldsymbol{\theta})\|_2$ and $\rho_3 = \sup_{\mathcal{M}^{\epsilon_4}} \|\nabla^3\mathcal{L}(\boldsymbol{\theta})\|_2$. By Lemma K.10, we can construct an $\epsilon_2$-neighborhood of $\Gamma$ where $\epsilon_2 < \sqrt{\frac{\mu}{\rho_2}}\epsilon_3$ such that all GF starting in $\Gamma^{\epsilon_2}$ converges to $\Gamma$. By Falconer (1983), $\Phi(\cdot)$ is $\mathcal{C}^2$ in $\Gamma^{\epsilon_3}$. Define $\nu_1 = \sup_{\boldsymbol{\theta} \in \Gamma^{\epsilon_3}} \|\partial\Phi(\boldsymbol{\theta})\|_2$ and $\nu_2 = \sup_{\boldsymbol{\theta} \in \Gamma^{\epsilon_3}} \|\partial^2\Phi(\boldsymbol{\theta})\|_2$. We also construct an $\epsilon_1$ neighborhood of $\Gamma$, $\Gamma^{\epsilon_1}$, where $\epsilon_1 \leq \frac{1}{2}\epsilon_2 < \frac{1}{2}\sqrt{\frac{\mu}{\rho_2}}\epsilon_3$ such that all $\boldsymbol{\theta} \in \Gamma^{\epsilon_1}$ has an $\epsilon_1$ neighborhood where $\Phi$ is well defined. Finally, by Lemma K.9, there exists an $\epsilon_0$-neighborhood of $\Gamma$ where $\epsilon_0 \leq \frac{1}{4}\sqrt{\frac{\mu}{\rho_2}}\epsilon_1$ such that all gradient descent iterates starting in $\Gamma^{\epsilon_0}$ with $\eta \leq \frac{1}{\rho_2}$ will stay in $\Gamma^{\epsilon_1}$. $\qquad\square$

Note that the notions of $\mathcal{Z}^\epsilon$, $\mathcal{M}^{\epsilon_4}$, $\rho_2$, $\rho_3$, $\nu_1$, and $\nu_2$ defined in the proof will be useful in the remaining part of this section. When analyzing the limiting dynamics of Local SGD, we will show that all $\boldsymbol{\theta}_{k,t}^{(s)}$ stays in $\Gamma^{\epsilon_2}$, $\tilde{\boldsymbol{u}}_t^{(s)} \in \Gamma^{\epsilon_1}$, $\bar{\boldsymbol{\theta}}^{(s)} \in \Gamma^{\epsilon_0}$ with high probability after $\mathcal{O}(\log \frac{1}{\eta})$ rounds.

## K.5 PHASE 1: ITERATE APPROACHING THE MANIFOLD

The approaching phase can be further divided into two subphases. In the first subphase, $\bar{\boldsymbol{\theta}}^{(0)}$ is initialized such that $\boldsymbol{\phi}^{(0)} \in \Gamma$. We will show that after a constant number of rounds $s_0$, $\bar{\boldsymbol{\theta}}^{(s_0)}$ goes to the inner part of $\Gamma^{\epsilon_0}$ such that $\|\bar{\boldsymbol{\theta}}^{(s_0)} - \boldsymbol{\phi}^{(0)}\|_2 \leq c\epsilon_0$ with high probability, where $0 < c < 1$ and the constants will be specified later (see Appendix K.5.2). In the second subphase, we show that the iterate can reach within $\tilde{\mathcal{O}}(\sqrt{\eta})$ distance from $\Gamma$ after $\mathcal{O}(\log \frac{1}{\eta})$ rounds with high probability (see Appendix K.5.3).

### K.5.1 ADDITIONAL NOTATIONS

Consider an auxiliary sequence $\{\tilde{\boldsymbol{u}}_t^{(s)}\}$ where $\tilde{\boldsymbol{u}}_0^{(s)} = \bar{\boldsymbol{\theta}}^{(s)}$ and $\tilde{\boldsymbol{u}}_{t+1}^{(s)} = \tilde{\boldsymbol{u}}_t^{(s)} - \eta \nabla \mathcal{L}(\tilde{\boldsymbol{u}}_t^{(s)}), 0 \leq t \leq H - 1$. Define $\tilde{\boldsymbol{\Delta}}_{k,t}^{(s)} := \boldsymbol{\theta}_{k,t}^{(s)} - \tilde{\boldsymbol{u}}_t^{(s)}$ to be the difference between the local iterate and the gradient descent iterate. Notice that $\tilde{\boldsymbol{\Delta}}_{k,0}^{(s)} = 0$, for all $k$ and $s$.

Consider a gradient flow $\{\boldsymbol{u}(t)\}_{t \geq 0}$ with the initial condition $\boldsymbol{u}(0) = \bar{\boldsymbol{\theta}}^{(0)}$ and converges to $\boldsymbol{\phi}^{(0)} \in \Gamma$. For simplicity, let $\boldsymbol{u}_t^{(s)} := \boldsymbol{u}(s\alpha + t\eta)$ be the gradient flow after $s$ rounds plus $t$ steps. Let $s_0$ be the smallest number such that $\|\boldsymbol{u}_0^{(s_0)} - \boldsymbol{\phi}^{(0)}\|_2 \leq \frac{1}{4}\sqrt{\frac{\mu}{\rho_2}}\epsilon_0$. Note that $s_0$ is a constant independent of $\eta$.

In this subsection, the minimum value of the loss in Appendix K.3 corresponds to the loss value on $\Gamma$, i.e., $\mathcal{L}^* = \mathcal{L}(\boldsymbol{\phi}), \forall \boldsymbol{\phi} \in \Gamma$.

We also define the following sequence $\{\tilde{\boldsymbol{Z}}_{k,t}^{(s)}\}_{t=0}^H$ that will be used in the proof. Define

$$\tilde{\boldsymbol{Z}}_{k,t}^{(s)} := \sum_{\tau=0}^{t-1}\left(\prod_{l=\tau+1}^{t-1}(\boldsymbol{I} - \eta\nabla^2\mathcal{L}(\tilde{\boldsymbol{u}}_l^{(s)}))\right)\boldsymbol{z}_{k,\tau}^{(s)}, \qquad \tilde{\boldsymbol{Z}}_{k,0}^{(s)} = \boldsymbol{0}.$$

### K.5.2 PROOF FOR SUBPHASE 1

First, we have the following lemma about the concentration of $\tilde{\boldsymbol{Z}}_{k,t}^{(s)}$.

**Lemma K.12** (Concentration property of $\{\tilde{\boldsymbol{Z}}_{k,t}^{(s)}\}_{t=0}^H$). *Given $\bar{\boldsymbol{\theta}}^{(s)}$ such that $\tilde{\boldsymbol{u}}_t^{(s)} \in \Gamma^{\epsilon_3} \cup \mathcal{Z}^\epsilon$ for all $0 \leq t \leq H$, then with probability at least $1 - \delta$,*

$$\|\tilde{\boldsymbol{Z}}_{k,t}^{(s)}\|_2 \leq \tilde{C}_1 \sigma_{\max}\sqrt{2H\log\frac{2HK}{\delta}}, \qquad \forall 0 \leq t \leq H, k \in [K],$$

*where $\tilde{C}_1 := \exp(\alpha\rho_2)$.*

*Proof.* For each $\tilde{\boldsymbol{Z}}_{k,t}^{(s)}$, construct a sequence $\{\tilde{\boldsymbol{Z}}_{k,t,t'}^{(s)}\}_{t'=0}^t$:

$$\tilde{\boldsymbol{Z}}_{k,t,t'}^{(s)} := \sum_{\tau=0}^{t'-1}\left(\prod_{l=\tau+1}^{t-1}(\boldsymbol{I} - \eta\nabla^2\mathcal{L}(\tilde{\boldsymbol{u}}_l^{(s)}))\right)\boldsymbol{z}_{k,\tau}^{(s)}, \qquad \tilde{\boldsymbol{Z}}_{k,t,0}^{(s)} = \boldsymbol{0}.$$

Since $\tilde{\boldsymbol{u}}_t^{(s)} \in \Gamma^{\epsilon_3} \cup \mathcal{Z}^\epsilon$, we have $\|\nabla^2\mathcal{L}(\tilde{\boldsymbol{u}}_t^{(s)})\|_2 \leq \rho_2$ for all $0 \leq t \leq H$. Then, for all $\tau$ and $t$,

$$\left\|\prod_{l=\tau+1}^{t-1}(\boldsymbol{I} - \eta\nabla^2\mathcal{L}(\tilde{\boldsymbol{u}}_l^{(s)}))\right\|_2 \leq (1 + \rho_2\eta)^H \leq \exp(\alpha\rho_2) = \tilde{C}_1.$$

Notice that for all $0 \le t \le H$, $\{\tilde{\boldsymbol{Z}}_{k,t,t'}^{(s)}\}_{t'=0}^{t}$ is a martingale with $\|\tilde{\boldsymbol{Z}}_{k,t,t'}^{(s)} - \tilde{\boldsymbol{Z}}_{k,t,t'-1}^{(s)}\|_2 \le \tilde{C}_1 \sigma_{\max}$. By Azuma-Hoeffding's inequality,

$$\mathbb{P}(\|\tilde{\boldsymbol{Z}}_{k,t}^{(s)}\|_2 \ge \epsilon') \le 2 \exp\left(\frac{-\epsilon'^2}{2t\left(\tilde{C}_1 \sigma_{\max}\right)^2}\right) \le 2 \exp\left(\frac{-\epsilon'^2}{2H\left(\tilde{C}_1 \sigma_{\max}\right)^2}\right).$$

Taking a union bound on all $k \in [K]$ and $0 \le t \le H$, we can conclude that with probability at least $1 - \delta$,

$$\|\tilde{\boldsymbol{Z}}_{k,t}^{(s)}\|_2 \le \tilde{C}_1 \sigma_{\max} \sqrt{2H \log \frac{2HK}{\delta}}, \qquad \forall 0 \le t \le H, k \in [K].$$

$\square$

The following lemma states that the gradient descent iterates will closely track the gradient flow with the same initial point.

**Lemma K.13.** *Denote $G := \sup_{t \ge 0} \|\nabla \mathcal{L}(\boldsymbol{u}(t))\|_2$ as the upper bound of the gradient on the gradient flow trajectory. If $\|\tilde{\boldsymbol{u}}_t^{(s)} - \boldsymbol{u}_t^{(s)}\|_2 = \mathcal{O}(\sqrt{\eta})$, then for all $0 \le t \le H$, the closeness of $\tilde{\boldsymbol{u}}_t^{(s)}$ and $\boldsymbol{u}_t^{(s)}$ is bounded by*

$$\|\tilde{\boldsymbol{u}}_t^{(s)} - \boldsymbol{u}_t^{(s)}\|_2 \le \tilde{C}_1 \|\tilde{\boldsymbol{u}}_0^{(s)} - \boldsymbol{u}_0^{(s)}\|_2 + \tilde{C}_1 \eta G,$$

*where $\tilde{C}_1 = \exp(\alpha \rho_2)$.*

*Proof.* We prove by induction that

$$\|\tilde{\boldsymbol{u}}_t^{(s)} - \boldsymbol{u}_t^{(s)}\|_2 \le (1 + \rho_2 \eta)^t \|\tilde{\boldsymbol{u}}_0^{(s)} - \boldsymbol{u}_0^{(s)}\|_2 + \rho_2 \eta^2 G \sum_{\tau=0}^{t-1} (1 + \rho_2 \eta)^\tau. \tag{34}$$

When $t = 0$, (34) holds trivially. Assume that (34) holds for $0 \le \tau \le t$, then

$$\tilde{\boldsymbol{u}}_{t+1}^{(s)} - \boldsymbol{u}_{t+1}^{(s)} = \tilde{\boldsymbol{u}}_t^{(s)} - \eta \nabla \mathcal{L}(\tilde{\boldsymbol{u}}_t^{(s)}) - \left(\boldsymbol{u}_t - \int_{s\alpha+t\eta}^{s\alpha+(t+1)\eta} \nabla \mathcal{L}(\boldsymbol{u}(v)) dv\right)$$

$$= \tilde{\boldsymbol{u}}_t^{(s)} - \boldsymbol{u}_t - \eta\left(\nabla \mathcal{L}(\tilde{\boldsymbol{u}}_t^{(s)}) - \nabla \mathcal{L}(\boldsymbol{u}_t^{(s)})\right)$$

$$- \int_{s\alpha+t\eta}^{s\alpha+(t+1)\eta} \left(\nabla \mathcal{L}(\boldsymbol{u}_t^{(s)}) - \nabla \mathcal{L}(\boldsymbol{u}(v))\right) dv.$$

By smoothness of $\mathcal{L}$,

$$\|\nabla \mathcal{L}(\boldsymbol{u}_t^{(s)}) - \nabla \mathcal{L}(\boldsymbol{u}(v))\|_2 \le \rho_2 \|\boldsymbol{u}_t^{(s)} - \boldsymbol{u}(v)\|_2$$

$$\le \rho_2 \int_{s\alpha+t\eta}^{v} \|\nabla \mathcal{L}(\boldsymbol{u}(w))\|_2 dw$$

$$\le \rho_2 \eta G.$$

Since $\rho_2^2 \eta^2 G \sum_{\tau=0}^{t-1} (1 + \rho_2 \eta)^\tau \le \eta G (1 + \rho_2 \eta)^t \le \exp(\alpha \rho_2) \eta G$, then $\|\tilde{\boldsymbol{u}}_t^{(s)} - \boldsymbol{u}_t^{(s)}\|_2 = \mathcal{O}(\sqrt{\eta})$, which implies that $\tilde{\boldsymbol{u}}_t^{(s)} \in \mathcal{M}^{\epsilon_4}$. Hence, $\|\nabla \mathcal{L}(\tilde{\boldsymbol{u}}_t^{(s)}) - \mathcal{L}(\boldsymbol{u}_t^{(s)})\|_2 \le \rho_2 \|\tilde{\boldsymbol{u}}_t^{(s)} - \boldsymbol{u}_t^{(s)}\|_2$.

By triangle inequality,

$$\|\tilde{\boldsymbol{u}}_{t+1}^{(s)} - \boldsymbol{u}_{t+1}^{(s)}\|_2 \le (1 + \rho_2 \eta)\|\tilde{\boldsymbol{u}}_t^{(s)} - \boldsymbol{u}_t^{(s)}\|_2 + \rho_2 \eta^2 G$$

$$\le (1 + \rho_2 \eta)^{t+1} \|\tilde{\boldsymbol{u}}_t^{(s)} - \boldsymbol{u}_t^{(s)}\|_2 + \rho_2 \eta^2 G \sum_{\tau=0}^{t} (1 + \rho_2 \eta)^\tau,$$

which concludes the induction step. Appling $1 + \rho_2 \eta \le \exp(\rho_2 \eta)$, we have the lemma. $\square$

Utilizing the concentration probability of $\{\tilde{\boldsymbol{Z}}_{k,t}^{(s)}\}$, we can obtain the following lemma which implies that the Local SGD iterates will closely track the gradient descent iterates with high probability.

**Lemma K.14.** *Given $\bar{\boldsymbol{\theta}}^{(s)}$ such that $\tilde{\boldsymbol{u}}_t^{(s)} \in \Gamma^{\epsilon_3} \cup \mathcal{Z}^{\epsilon}$ for all $0 \le t \le H$, then for $\delta = \mathcal{O}(\mathrm{poly}(\eta))$, with probability at least $1 - \delta$, there exists a constant $\tilde{C}_3$ such that*

$$\|\boldsymbol{\theta}_{k,t}^{(s)} - \tilde{\boldsymbol{u}}_t^{(s)}\|_2 \le \tilde{C}_3 \sqrt{\eta \log \frac{1}{\eta\delta}}, \quad \forall 0 \le t \le H, k \in [K],$$

*and*

$$\|\bar{\boldsymbol{\theta}}^{(s+1)} - \tilde{\boldsymbol{u}}_H^{(s)}\|_2 \le \tilde{C}_3 \sqrt{\eta \log \frac{1}{\eta\delta}}.$$

*Proof.* Since $\tilde{\boldsymbol{u}}_t^{(s)} \in \Gamma^{\epsilon_3} \cup \mathcal{Z}^{\epsilon}$ for all $0 \le t \le H$, we have $\|\nabla^2 \mathcal{L}(\tilde{\boldsymbol{u}}_t^{(s)})\|_2 \le \rho_2$. According to the update rule for $\boldsymbol{\theta}_{k,t}^{(s)}$ and $\tilde{\boldsymbol{u}}_t^{(s)}$,

$$\boldsymbol{\theta}_{k,t+1}^{(s)} = \boldsymbol{\theta}_{k,t}^{(s)} - \eta \nabla \mathcal{L}(\boldsymbol{\theta}_{k,t}^{(s)}) - \eta \boldsymbol{z}_{k,t}^{(s)}, \tag{35}$$

$$\tilde{\boldsymbol{u}}_{t+1}^{(s)} = \tilde{\boldsymbol{u}}_t^{(s)} - \eta \nabla \mathcal{L}(\tilde{\boldsymbol{u}}_t^{(s)}). \tag{36}$$

Subtracting (36) from (35) gives

$$\tilde{\boldsymbol{\Delta}}_{k,t+1}^{(s)} = \tilde{\boldsymbol{\Delta}}_{k,t}^{(s)} - \eta(\nabla \mathcal{L}(\boldsymbol{\theta}_{k,t}^{(s)}) - \nabla \mathcal{L}(\tilde{\boldsymbol{u}}_t^{(s)})) - \eta \boldsymbol{z}_{k,t}^{(s)}$$

$$= (\boldsymbol{I} - \eta \nabla^2 \mathcal{L}(\tilde{\boldsymbol{u}}_t^{(s)}))\tilde{\boldsymbol{\Delta}}_{k,t}^{(s)} - \eta \boldsymbol{z}_{k,t}^{(s)} + \eta \tilde{\boldsymbol{v}}_{k,t}^{(s)}. \tag{37}$$

Here, $\tilde{\boldsymbol{v}}_{k,t}^{(s)} = (1 - \beta_{k,t}^{(s)})\boldsymbol{\theta}_{k,t}^{(s)} + \beta_{k,t}^{(s)}\tilde{\boldsymbol{u}}_t^{(s)}$, where $\beta_{k,t}^{(s)} \in (0,1)$ depends on $\boldsymbol{\theta}_{k,t}^{(s)}$ and $\tilde{\boldsymbol{u}}_t^{(s)}$. Therefore, $\|\tilde{\boldsymbol{v}}_{k,t}^{(s)}\|_2 \le \frac{\rho_3}{2}\|\tilde{\boldsymbol{\Delta}}_{k,t}^{(s)}\|_2^2$ if $\boldsymbol{\theta}_{k,t}^{(s)} \in \mathcal{M}^{\epsilon_4}$. Applying (37) $t$ times, we have

$$\tilde{\boldsymbol{\Delta}}_{k,t}^{(s)} = \left[\prod_{\tau=0}^{t-1}(\boldsymbol{I} - \eta \nabla^2 \mathcal{L}(\tilde{\boldsymbol{u}}_\tau^{(s)}))\right]\tilde{\boldsymbol{\Delta}}_{k,0}^{(s)} - \eta \sum_{\tau=0}^{t-1}\prod_{l=\tau+1}^{t-1}(\boldsymbol{I} - \eta \nabla^2 \mathcal{L}(\tilde{\boldsymbol{u}}_l^{(s)}))\boldsymbol{z}_{k,\tau}^{(s)}$$

$$+ \eta \sum_{\tau=0}^{t-1}\prod_{l=\tau+1}^{t-1}(\boldsymbol{I} - \eta \nabla^2 \mathcal{L}(\tilde{\boldsymbol{u}}_l^{(s)}))\tilde{\boldsymbol{v}}_{k,\tau}^{(s)}.$$

By Cauchy-Schwartz inequality, triangle inequality and the definition of $\tilde{\boldsymbol{Z}}_{k,t}^{(s)}$, if for all $0 \le \tau \le t-1$ and $k \in [K]$, $\boldsymbol{\theta}_{k,\tau}^{(s)} \in \mathcal{M}^{\epsilon_4}$, then we have

$$\|\tilde{\boldsymbol{\Delta}}_{k,t}^{(s)}\|_2 \le \eta\|\tilde{\boldsymbol{Z}}_{k,t}^{(s)}\|_2 + \frac{1}{2}\eta\rho_3\sum_{\tau=0}^{t-1}\tilde{C}_1\|\tilde{\boldsymbol{\Delta}}_{k,\tau}^{(s)}\|_2^2. \tag{38}$$

Applying Lemma K.12 and substituting in the value of $H$, we have that with probability at least $1 - \delta$,

$$\|\tilde{\boldsymbol{Z}}_{k,t}^{(s)}\|_2 \le \tilde{C}_1 \sigma_{\max}\sqrt{\frac{2\alpha}{\eta}\log\frac{2\alpha K}{\eta\delta}}, \qquad \forall k \in K, 0 \le t \le H. \tag{39}$$

Now we show by induction that for $\delta = \mathcal{O}(\mathrm{poly}(\eta))$, when (39) holds, there exists a constant $\tilde{C}_2 > 2\sigma_{\max}\sqrt{2\alpha}\tilde{C}_1$ such that $\|\tilde{\boldsymbol{\Delta}}_{k,t}^{(s)}\|_2 \le \tilde{C}_2\sqrt{\eta \log\frac{2\alpha K}{\eta\delta}}$.

When $t = 0$, $\tilde{\boldsymbol{\Delta}}_{k,0}^{(s)} = 0$. Assume that $\|\tilde{\boldsymbol{\Delta}}_{k,\tau}^{(s)}\|_2 \le \tilde{C}_2\sqrt{\eta \log\frac{2\alpha K}{\eta\delta}}$, for all $k \in [K], 0 \le \tau \le t-1$. Then for all $0 \le \tau \le t-1$, $\boldsymbol{\theta}_{k,\tau}^{(s)} \in \mathcal{M}^{\epsilon_4}$. Therefore, we can apply (38) and obtain

$$\|\tilde{\boldsymbol{\Delta}}_{k,t}^{(s)}\|_2 \le \eta\|\tilde{\boldsymbol{Z}}_{k,t}^{(s)}\|_2 + \frac{1}{2}\eta\rho_3\sum_{\tau=0}^{t-1}\tilde{C}_1\|\tilde{\boldsymbol{\Delta}}_{k,\tau}^{(s)}\|_2^2$$

$$\le \tilde{C}_1\sigma_{\max}\sqrt{2\alpha\eta \log\frac{2\alpha K}{\eta\delta}} + \frac{1}{2}\tilde{C}_1\tilde{C}_2^2\sigma_{\max}^2\alpha\rho_3\eta \log\frac{2\alpha K}{\eta\delta}.$$

Given that $\tilde{C}_2 \geq 2\sigma_{\max}\sqrt{2\alpha}\tilde{C}_1$ and $\delta = \mathcal{O}(\mathrm{poly}(\eta))$, when $\eta$ is sufficiently small, $\|\tilde{\boldsymbol{\Delta}}_{k,t}^{(s)}\|_2 \leq \tilde{C}_2\sqrt{\eta\log\frac{2\alpha K}{\eta\delta}}$.

To sum up, for $\delta = \mathcal{O}(\mathrm{poly}(\eta))$, with probability at least $1 - \delta$, $\|\tilde{\boldsymbol{\Delta}}_{k,t}^{(s)}\|_2 \leq \tilde{C}_2\sqrt{\eta\log\frac{2\alpha K}{\eta\delta}}$ for all $k \in [K], 0 \leq t \leq H$. By triangle inequality,

$$\|\bar{\boldsymbol{\theta}}^{(s+1)} - \tilde{\boldsymbol{u}}_H^{(s)}\|_2 \leq \frac{1}{K}\sum_{k\in[K]}\|\tilde{\boldsymbol{\Delta}}_{k,H}^{(s)}\|_2 \leq \tilde{C}_2\sqrt{\eta\log\frac{2\alpha K}{\eta\delta}}.$$

$\square$

The combination of Lemma K.13 and Lemma K.14 leads to the following lemma, which states that the Local SGD iterate will enter $\Gamma^{\epsilon_1}$ after $s_0$ rounds with high probability.

**Lemma K.15.** *Given $\bar{\boldsymbol{\theta}}^{(0)}$ such that $\Phi(\bar{\boldsymbol{\theta}}^{(0)}) \in \Gamma$, then for $\delta = \mathcal{O}(\mathrm{poly}(\eta))$, there exists a positive constant $\tilde{C}_4$ such that with probability at least $1 - \delta$,*

$$\|\bar{\boldsymbol{\theta}}^{(s_0)} - \boldsymbol{\phi}^{(0)}\|_2 \leq \frac{1}{4}\sqrt{\frac{\mu}{\rho_2}}\epsilon_0 + \tilde{C}_4\sqrt{\eta\log\frac{1}{\eta\delta}}.$$

*Proof.* First, we prove by induction that for $\delta = \mathcal{O}(\mathrm{poly}(\eta))$, when

$$\|\tilde{\boldsymbol{Z}}_{k,t}^{(s)}\|_2 \leq \tilde{C}_1\sigma_{\max}\sqrt{2H\log\frac{2HKs_0}{\delta}}, \qquad \forall 0 \leq t \leq H, k \in [K], 0 \leq s < s_0, \qquad (40)$$

the closeness of $\bar{\boldsymbol{\theta}}^{(s)}$ and $\boldsymbol{u}_0^{(s)}$ is bounded by

$$\|\bar{\boldsymbol{\theta}}^{(s)} - \boldsymbol{u}_0^{(s)}\|_2 \leq \sum_{l=1}^{s}\tilde{C}_1^l\left(\eta G + \tilde{C}_3\sqrt{\eta\log\frac{s_0}{\eta\delta}}\right), \qquad \forall 0 \leq s \leq s_0. \qquad (41)$$

When $s = 0$, $\bar{\boldsymbol{\theta}}^{(0)} = \boldsymbol{u}_0^{(0)}$. Assume that (41) holds for round $s$. Then by Lemma K.13, for all $0 \leq t \leq H$,

$$\begin{aligned}
\|\tilde{\boldsymbol{u}}_t^{(s)} - \boldsymbol{u}_t^{(s)}\|_2 &\leq \tilde{C}_1\|\tilde{\boldsymbol{u}}_0^{(s)} - \boldsymbol{u}_0^{(s)}\|_2 + \tilde{C}_1\eta G \\
&= \tilde{C}_1\|\bar{\boldsymbol{\theta}}_0^{(s)} - \boldsymbol{u}_0^{(s)}\|_2 + \tilde{C}_1\eta G \\
&\leq \sum_{l=1}^{s}\tilde{C}_1^{l+1}\left(\eta G + \tilde{C}_3\sqrt{\eta\log\frac{s_0}{\eta\delta}}\right) + \tilde{C}_1\eta G.
\end{aligned}$$

Therefore, for sufficiently small $\eta$, $\tilde{\boldsymbol{u}}_t^{(s)} \in \mathcal{Z}^\epsilon$, $\forall 0 \leq t \leq H$. Combing the above inequality with Lemma K.14, we have

$$\begin{aligned}
\|\bar{\boldsymbol{\theta}}^{(s+1)} - \boldsymbol{u}_0^{(s+1)}\|_2 &= \|\bar{\boldsymbol{\theta}}^{(s+1)} - \boldsymbol{u}_H^{(s)}\|_2 \\
&\leq \|\bar{\boldsymbol{\theta}}^{(s+1)} - \tilde{\boldsymbol{u}}_H^{(s)}\|_2 + \|\tilde{\boldsymbol{u}}_H^{(s)} - \boldsymbol{u}_H^{(s)}\|_2 \\
&\leq \sum_{l=1}^{s+1}\tilde{C}_1^{l+1}\left(\eta G + \tilde{C}_3\sqrt{\eta\log\frac{s_0}{\eta\delta}}\right),
\end{aligned}$$

which concludes the induction.

Therefore, when (40) holds, there exists a positive constant $\tilde{C}_4$ such that

$$\|\bar{\boldsymbol{\theta}}^{(s_0)} - \boldsymbol{u}_0^{(s_0)}\|_2 \leq \tilde{C}_4\sqrt{\eta\log\frac{1}{\eta\delta}}.$$

By definition of $\boldsymbol{u}_0^{(s_0)}$,

$$\|\bar{\boldsymbol{\theta}}^{(s_0)} - \boldsymbol{\phi}^{(0)}\|_2 \leq \frac{1}{4}\sqrt{\frac{\mu}{\rho_2}}\epsilon_0 + \tilde{C}_4\sqrt{\eta\log\frac{1}{\eta\delta}}.$$

Finally, according to Lemma K.12, (40) holds with probability at least $1 - \delta$. $\square$

### K.5.3 PROOF FOR SUBPHASE 2

In subphase 2, we show that the iterate can reach within $\tilde{\mathcal{O}}(\sqrt{\eta})$ distance from $\Gamma$ after $\mathcal{O}(\log \frac{1}{\eta})$ rounds with high probability. The following lemma manifests how the potential function $\tilde{\Psi}(\bar{\boldsymbol{\theta}}^{(s)})$ evolves after one round.

**Lemma K.16.** *Given* $\bar{\boldsymbol{\theta}}^{(s)} \in \Gamma^{\epsilon_0}$, *for* $\delta = \mathcal{O}(\mathrm{poly}(\eta))$, *with probability at least* $1 - \delta$,

$$\boldsymbol{\theta}_{k,t}^{(s)} \in \Gamma^{\epsilon_2}, \quad \tilde{\Psi}(\boldsymbol{\theta}_{k,t}^{(s)}) \leq \tilde{\Psi}(\bar{\boldsymbol{\theta}}^{(s)}) + \tilde{C}_5 \sqrt{\eta \log \frac{1}{\eta\delta}}, \quad \forall k \in [K], 0 \leq t \leq H$$

*and*

$$\bar{\boldsymbol{\theta}}^{(s+1)} \in \Gamma^{\epsilon_2}, \quad \tilde{\Psi}(\bar{\boldsymbol{\theta}}^{(s+1)}) \leq \exp(-\alpha\mu/2)\tilde{\Psi}(\bar{\boldsymbol{\theta}}^{(s)}) + \tilde{C}_5 \sqrt{\eta \log \frac{1}{\eta\delta}},$$

*where* $\tilde{C}_5$ *is a positive constant.*

*Proof.* Since $\bar{\boldsymbol{\theta}}^{(s)} \in \Gamma^{\epsilon_0}$, then for all $0 \leq t \leq H$, $\tilde{\boldsymbol{u}}_t^{(s)} \in \Gamma^{\epsilon_1}$ by the definition of the working zone. By Lemma K.6, for $\eta \leq \frac{1}{\rho_2}$,

$$\mathcal{L}(\tilde{\boldsymbol{u}}_t^{(s)}) - \mathcal{L}^* \leq (1 - \mu\eta)^t \left( \mathcal{L}(\bar{\boldsymbol{\theta}}^{(s)}) - \mathcal{L}^* \right) \leq \mathcal{L}(\bar{\boldsymbol{\theta}}^{(s)}) - \mathcal{L}^*, \quad \forall 0 \leq t \leq H.$$

Specially, for $t = H$,

$$\mathcal{L}(\tilde{\boldsymbol{u}}_H^{(s)}) - \mathcal{L}^* \leq (1 - \mu\eta)^{\frac{\alpha}{\eta}} \left( \mathcal{L}(\bar{\boldsymbol{\theta}}^{(s)}) - \mathcal{L}^* \right) \leq \exp(-\alpha\mu)(\mathcal{L}(\bar{\boldsymbol{\theta}}^{(s)}) - \mathcal{L}^*).$$

Therefore,

$$\tilde{\Psi}(\tilde{\boldsymbol{u}}_H^{(s)}) \leq \exp(-\alpha\mu/2)\tilde{\Psi}(\bar{\boldsymbol{\theta}}^{(s)}).$$

According to the proof of Lemma K.14, for $\delta = \mathcal{O}(\mathrm{poly}(\eta))$, when

$$\|\tilde{\boldsymbol{Z}}_{k,t}^{(s)}\|_2 \leq \tilde{C}_1 \sigma_{\max} \sqrt{\frac{2\alpha}{\eta} \log \frac{2\alpha K}{\eta\delta}}, \qquad \forall k \in [K], 0 \leq t \leq H, \tag{42}$$

there exists a constant $\tilde{C}_3$ such that

$$\|\boldsymbol{\theta}_{k,t}^{(s)} - \tilde{\boldsymbol{u}}_t^{(s)}\|_2 \leq \tilde{C}_3 \sqrt{\eta \log \frac{1}{\eta\delta}}, \quad \forall 0 \leq t \leq H, k \in [K],$$

and

$$\|\bar{\boldsymbol{\theta}}^{(s+1)} - \tilde{\boldsymbol{u}}_H^{(s)}\|_2 \leq \tilde{C}_3 \sqrt{\eta \log \frac{1}{\eta\delta}}.$$

Since $\tilde{\boldsymbol{u}}_t^{(s)} \in \Gamma^{\epsilon_1}, \forall 0 \leq t \leq H$, $\bar{\boldsymbol{\theta}}^{(s+1)} \in \Gamma^{\epsilon_2}$ and $\boldsymbol{\theta}_{k,t}^{(s)} \in \Gamma^{\epsilon_2}, \forall 0 \leq t \leq H, k \in [K]$.

By Lemma K.7, $\tilde{\Psi}(\cdot)$ is $\sqrt{2\rho_2}$-Lipschitz in $\mathcal{M}^{\epsilon_4}$. Therefore, when (42) holds, there exists a constant $\tilde{C}_5 := \sqrt{2\rho_2}\tilde{C}_3$ such that

$$\tilde{\Psi}(\boldsymbol{\theta}_{k,t}^{(s)}) \leq \tilde{\Psi}(\tilde{\boldsymbol{u}}_t^{(s)}) + \sqrt{2\rho_2}\|\boldsymbol{\theta}_{k,t}^{(s)} - \tilde{\boldsymbol{u}}_t^{(s)}\|_2$$

$$\leq \tilde{\Psi}(\bar{\boldsymbol{\theta}}^{(s)}) + \tilde{C}_5 \sqrt{\eta \log \frac{1}{\eta\delta}},$$

and

$$\tilde{\Psi}(\bar{\boldsymbol{\theta}}^{(s+1)}) \leq \tilde{\Psi}(\tilde{\boldsymbol{u}}_H^{(s)}) + \sqrt{2\rho_2}\|\bar{\boldsymbol{\theta}}^{(s+1)} - \tilde{\boldsymbol{u}}_H^{(s)}\|_2$$

$$\leq \exp(-\alpha\mu/2)\tilde{\Psi}(\bar{\boldsymbol{\theta}}^{(s)}) + \tilde{C}_5 \sqrt{\eta \log \frac{1}{\eta\delta}}.$$

Finally, by Lemma K.12, (42) holds with probability at least $1 - \delta$. $\quad\square$

We are thus led to the following lemma which characterizes the evolution of the potential $\tilde{\Psi}(\bar{\boldsymbol{\theta}}^{(s)})$ and $\tilde{\Psi}(\boldsymbol{\theta}_{k,t}^{(s)})$ over multiple rounds.

**Lemma K.17.** *Given* $\|\bar{\boldsymbol{\theta}}^{(0)} - \boldsymbol{\phi}^{(0)}\|_2 \leq \frac{1}{2}\sqrt{\frac{\mu}{\rho_2}}\epsilon_0$, *for* $\delta = \mathcal{O}(\text{poly}(\eta))$ *and any integer* $1 \leq R \leq R_{\text{tot}}$, *with probability at least* $1 - \delta$,

$$\bar{\boldsymbol{\theta}}^{(s)} \in \Gamma^{\epsilon_0}, \tilde{\Psi}(\bar{\boldsymbol{\theta}}^{(s)}) \leq \exp(-\alpha\mu s/2)\tilde{\Psi}(\bar{\boldsymbol{\theta}}^{(0)}) + \frac{1}{1 - \exp(-\alpha\mu/2)}\tilde{C}_5\sqrt{\eta\log\frac{R}{\eta\delta}}, \forall 0 \leq s \leq R. \tag{43}$$

*Furthermore,*

$$\bar{\boldsymbol{\theta}}_{k,t}^{(s)} \in \Gamma^{\epsilon_2}, \quad \tilde{\Psi}(\boldsymbol{\theta}_{k,t}^{(s)}) \leq \tilde{\Psi}(\bar{\boldsymbol{\theta}}^{(s)}) + \tilde{C}_5\sqrt{\eta\log\frac{R}{\eta\delta}}, \quad \forall 0 \leq t \leq H, 0 \leq s < R, k \in [K]. \tag{44}$$

*Proof.* We prove induction that for $\delta = \mathcal{O}(\text{poly}(\eta))$, when

$$\|\tilde{\boldsymbol{Z}}_{k,t}^{(s)}\|_2 \leq \tilde{C}_1\sigma_{\max}\sqrt{\frac{2\alpha}{\eta}\log\frac{2R\alpha K}{\eta\delta}}, \qquad \forall k \in [K], 0 \leq t \leq H, 0 \leq s < R, \tag{45}$$

then for all $0 \leq s \leq R$, (43) and (44) hold.

When $s = 0$, $\bar{\boldsymbol{\theta}}^{(0)} \in \Gamma^{\epsilon_0}$ and (43) trivially holds. By Lemma K.16, (44) holds. Assume that (43) and (44) hold for round $s - 1$. Then for round $s$, by Lemma K.16, $\bar{\boldsymbol{\theta}}^{(s)} \in \Gamma^{\epsilon_2}$ and

$$\Psi(\bar{\boldsymbol{\theta}}^{(s)}) \leq \exp(-\alpha\mu/2)\tilde{\Psi}(\bar{\boldsymbol{\theta}}^{(s-1)}) + \tilde{C}_5\sqrt{\eta\log\frac{R}{\eta\delta}}$$

$$\leq \exp(-\alpha\mu s/2)\tilde{\Psi}(\bar{\boldsymbol{\theta}}^{(0)}) + \frac{1}{1 - \exp(-\alpha\mu/2)}\tilde{C}_5\sqrt{\eta\log\frac{R}{\eta\delta}},$$

where the second inequality comes from the induction hypothesis. By Lemma K.10,

$$\|\bar{\boldsymbol{\theta}}^{(s)} - \boldsymbol{\phi}^{(s)}\|_2 \leq \frac{2}{\sqrt{2\mu}}\tilde{\Psi}(\bar{\boldsymbol{\theta}}^{(s)})$$

$$\leq \frac{2}{\sqrt{2\mu}}\tilde{\Psi}(\bar{\boldsymbol{\theta}}^{(0)}) + \frac{2}{\sqrt{2\mu}(1 - \exp(-\alpha\mu/2))}\tilde{C}_5\sqrt{\eta\log\frac{R}{\eta\delta}}$$

$$\leq \frac{1}{2}\epsilon_0 + \frac{2}{\sqrt{2\mu}(1 - \exp(-\alpha\mu/2))}\tilde{C}_5\sqrt{\eta\log\frac{R}{\eta\delta}}.$$

Here, the last inequality uses $\tilde{\Psi}(\bar{\boldsymbol{\theta}}^{(0)}) \leq \sqrt{\frac{\rho_2}{2}}\|\bar{\boldsymbol{\theta}}^{(s)} - \boldsymbol{\phi}^{(0)}\|_2 \leq \frac{1}{2}\sqrt{\frac{\mu}{2}}\epsilon_0$. Hence, when $\eta$ is sufficiently small, $\bar{\boldsymbol{\theta}}^{(s)} \in \Gamma^{\epsilon_0}$. Still by Lemma K.16, $\bar{\boldsymbol{\theta}}_{k,t}^{(s)} \in \Gamma^{\epsilon_2}$ and

$$\tilde{\Psi}(\boldsymbol{\theta}_{k,t}^{(s)}) \leq \tilde{\Psi}(\bar{\boldsymbol{\theta}}^{(s)}) + \tilde{C}_5\sqrt{\eta\log\frac{R}{\eta\delta}}.$$

Finally, according to Lemma K.12, (45) holds with probability at least $1 - \delta$.

$$\square$$

The following corollary is a direct consequence of Lemma K.17 and Lemma K.10.

**Corollary K.1.** *Let* $s_1 := \lceil\frac{20}{\alpha\mu}\log\frac{1}{\eta}\rceil$. *Given* $\|\bar{\boldsymbol{\theta}}^{(0)} - \boldsymbol{\phi}^{(0)}\|_2 \leq \frac{1}{2}\sqrt{\frac{\mu}{\rho_2}}\epsilon_0$, *for* $\delta = \mathcal{O}(\text{poly}(\eta))$, *with probability at least* $1 - \delta$,

$$\tilde{\Psi}(\bar{\boldsymbol{\theta}}^{(s_1)}) \leq \tilde{C}_6\sqrt{\eta\log\frac{1}{\eta\delta}}, \quad \|\bar{\boldsymbol{\theta}}^{(s_1)} - \boldsymbol{\phi}^{(s_1)}\|_2 \leq \tilde{C}_6\sqrt{\eta\log\frac{1}{\eta\delta}}, \tag{46}$$

*where* $\tilde{C}_6$ *is a constant.*

*Proof.* Substituting in $R = s_1$ to Lemma K.17 and applying $\|\bar{\boldsymbol{\theta}}^{(s_1)} - \boldsymbol{\phi}^{(s)}\|_2 \leq \sqrt{\frac{2}{\mu}}\tilde{\Psi}(\bar{\boldsymbol{\theta}}^{(s_1)})$ for $\bar{\boldsymbol{\theta}}^{(s_1)} \in \Gamma^{\epsilon_0}$, we have the lemma. $\qquad\square$

Finally, we provide a high probability bound for the change of the projection on the manifold after $s_1$ rounds $\|\boldsymbol{\phi}^{(s_1)} - \boldsymbol{\phi}^{(0)}\|_2$.

**Lemma K.18.** *Let* $s_1 := \lceil \frac{20}{\alpha\mu}\log\frac{1}{\eta}\rceil$. *Given* $\|\bar{\boldsymbol{\theta}}^{(0)} - \boldsymbol{\phi}^{(0)}\|_2 \leq \frac{1}{2}\sqrt{\frac{\mu}{\rho_2}}\epsilon_0$. *For* $\delta = \mathcal{O}(\mathrm{poly}(\eta))$, *with probability at least* $1 - \delta$,

$$\|\boldsymbol{\phi}^{(s_1)} - \boldsymbol{\phi}^{(0)}\|_2 \leq \tilde{C}_8 \log\frac{1}{\eta}\sqrt{\eta\log\frac{1}{\eta\delta}}.$$

*Proof.* From Lemma K.17, for $\delta = \mathcal{O}(\mathrm{poly}(\eta))$, when

$$\|\tilde{\boldsymbol{Z}}_{k,t}^{(s)}\|_2 \leq \tilde{C}_1\sigma_{\max}\sqrt{\frac{2\alpha}{\eta}\log\frac{2s_1\alpha K}{\eta\delta}}, \qquad \forall k \in [K], 0 \leq t \leq H, 0 \leq s < s_1, \qquad (47)$$

then $\bar{\boldsymbol{\theta}}^{(s)} \in \Gamma^{\epsilon_0}$, for all $0 \leq s \leq s_1$. By the definition of $\Gamma^{\epsilon_0}$, $\tilde{\boldsymbol{u}}_t^{(s)} \in \Gamma^{\epsilon_1}$, for all $0 \leq t \leq H, 0 \leq s \leq s_1$. By triangle inequality, $\|\boldsymbol{\phi}^{(s_1)} - \boldsymbol{\phi}^{(0)}\|_2$ can be decomposed as follows.

$$\|\boldsymbol{\phi}^{(s_1)} - \boldsymbol{\phi}^{(0)}\|_2 \leq \sum_{s=0}^{s_1-1}\|\boldsymbol{\phi}^{(s+1)} - \boldsymbol{\phi}^{(s)}\|_2$$

$$\leq \sum_{s=0}^{s_1-1}\|\Phi(\tilde{\boldsymbol{u}}_H^{(s)}) - \Phi(\tilde{\boldsymbol{u}}_0^{(s)})\|_2 + \sum_{s=0}^{s_1-1}\|\Phi(\bar{\boldsymbol{\theta}}^{(s+1)}) - \Phi(\tilde{\boldsymbol{u}}_H^{(s)})\|_2. \qquad (48)$$

By Lemma K.14, when (47) hold , then for all $0 \leq s < s_1 - 1$,

$$\|\bar{\boldsymbol{\theta}}^{(s+1)} - \tilde{\boldsymbol{u}}_H^{(s)}\|_2 \leq \tilde{C}_3\sqrt{\eta\log\frac{s_1}{\eta\delta}}.$$

This implies that $\bar{\boldsymbol{\theta}}^{(s+1)} \in B^{\epsilon_1}(\tilde{\boldsymbol{u}}_H^{(s)})$. Since for all $\boldsymbol{\theta} \in \Gamma^{\epsilon_2}$, $\|\partial\Phi(\boldsymbol{\theta})\|_2 \leq \nu_1$, then $\Phi(\cdot)$ is $\nu_1$-Lipschitz in $B^{\epsilon_1}(\tilde{\boldsymbol{u}}_H^{(s)})$. This gives

$$\|\Phi(\bar{\boldsymbol{\theta}}^{(s+1)}) - \Phi(\tilde{\boldsymbol{u}}_H^{(s)})\|_2 \leq \nu_1\|\bar{\boldsymbol{\theta}}^{(s+1)} - \tilde{\boldsymbol{u}}_H^{(s)}\|_2$$

$$\leq \nu_1\tilde{C}_3\sqrt{\eta\log\frac{s_1}{\eta\delta}}. \qquad (49)$$

Then we analyze $\|\bar{\boldsymbol{\theta}}^{(s+1)} - \tilde{\boldsymbol{u}}_H^{(s)}\|_2$. By Lemma K.9 and the definition of $\Gamma^{\epsilon_0}$ and $\Gamma^{\epsilon_1}$, there exists $\boldsymbol{\phi} \in \Gamma$ such that $\tilde{\boldsymbol{u}}_t^{(s)} \in B^{\epsilon_1}(\boldsymbol{\phi}), \forall 0 \leq t \leq H$. Therefore, we can expand $\Phi(\tilde{\boldsymbol{u}}_{t+1}^{(s)})$ as follows:

$$\Phi(\tilde{\boldsymbol{u}}_{t+1}^{(s)}) = \Phi(\tilde{\boldsymbol{u}}_t^{(s)} - \eta\nabla\mathcal{L}(\tilde{\boldsymbol{u}}_t^{(s)}))$$

$$= \Phi(\tilde{\boldsymbol{u}}_t^{(s)}) - \eta\partial\Phi(\tilde{\boldsymbol{u}}^{(s)})\nabla\mathcal{L}(\boldsymbol{u}_t^{(s)}) + \frac{\eta^2}{2}\partial^2\Phi(\hat{\boldsymbol{u}}_t^{(s)})[\nabla\mathcal{L}(\tilde{\boldsymbol{u}}_t^{(s)}), \nabla\mathcal{L}(\tilde{\boldsymbol{u}}_t^{(s)})]$$

$$= \Phi(\tilde{\boldsymbol{u}}_t^{(s)}) + \frac{\eta^2}{2}\partial^2\Phi\left(c_t^{(s)}\tilde{\boldsymbol{u}}_t^{(s)} + (1 - c_t^{(s)})\tilde{\boldsymbol{u}}_{t+1}^{(s)}\right)[\nabla\mathcal{L}(\tilde{\boldsymbol{u}}_t^{(s)}), \nabla\mathcal{L}(\tilde{\boldsymbol{u}}_t^{(s)})],$$

where $c_t^{(s)} \in (0, 1)$. Then we have

$$\|\Phi(\tilde{\boldsymbol{u}}_H^{(s)}) - \Phi(\tilde{\boldsymbol{u}}_0^{(s)})\|_2 \leq \frac{\eta^2}{2}\sum_{t=0}^{H-1}\|\partial^2\Phi((c_t^{(s)}\tilde{\boldsymbol{u}}_t^{(s)} + (1 - c_t^{(s)})\tilde{\boldsymbol{u}}_{t+1}^{(s)}))[\nabla\mathcal{L}(\tilde{\boldsymbol{u}}^{(s)}), \nabla\mathcal{L}(\tilde{\boldsymbol{u}}_t^{(s)})]\|_2$$

$$\leq \frac{\eta^2}{2}\nu_2\sum_{t=0}^{H-1}\|\nabla\mathcal{L}(\tilde{\boldsymbol{u}}_t^{(s)})\|_2^2.$$

By Lemma K.6, $\frac{\eta}{2}\|\nabla\mathcal{L}(\tilde{\boldsymbol{u}}_t^{(s)})\|_2^2 \le \mathcal{L}(\tilde{\boldsymbol{u}}_t^{(s)}) - \mathcal{L}(\tilde{\boldsymbol{u}}_{t+1}^{(s)})$. Therefore,

$$
\begin{aligned}
\|\Phi(\tilde{\boldsymbol{u}}_H^{(s)}) - \Phi(\tilde{\boldsymbol{u}}_0^{(s)})\|_2 &\le \eta\nu_2(\mathcal{L}(\tilde{\boldsymbol{u}}_0^{(s)}) - \mathcal{L}(\tilde{\boldsymbol{u}}_H^{(s)})) \\
&\le \eta\nu_2[\tilde{\Psi}(\bar{\boldsymbol{\theta}}^{(s)})]^2 \\
&\le \nu_2\eta\left[2\exp(-\alpha s\mu)\tilde{\Psi}(\bar{\boldsymbol{\theta}}^{(0)}) + \frac{\tilde{C}_5^2\eta}{(1-\exp(-\alpha\mu/2))^2}\log\frac{s_1}{\eta\delta}\right], \quad (50)
\end{aligned}
$$

where the last inequality uses Cauchy-Schwartz inequality and Lemma K.17. Summing up (50), we obtain

$$
\begin{aligned}
\sum_{s=0}^{s_1-1}\|\Phi(\tilde{\boldsymbol{u}}_H^{(s)}) - \Phi(\tilde{\boldsymbol{u}}_0^{(s)})\|_2 &\le \nu_2\eta\left[2\tilde{\Psi}(\bar{\boldsymbol{\theta}}^{(0)})\sum_{s=0}^{s_1-1}\exp(-\alpha\mu s) + \frac{s_1\tilde{C}_5^2\eta}{(1-\exp(-\alpha\mu/2))^2}\log\frac{s_1}{\eta\delta}\right] \\
&\le \tilde{C}_7\eta\log\frac{1}{\eta}\log\frac{1}{\eta\delta}, \quad (51)
\end{aligned}
$$

where $\tilde{C}_7$ is a constant. Substituting (49) and (51) into (48), for sufficiently small $\eta$, we have

$$
\begin{aligned}
\|\boldsymbol{\phi}^{(s_1)} - \boldsymbol{\phi}^{(0)}\|_2 &\le \nu_1\tilde{C}_3 s_1\sqrt{\eta\log\frac{s_1}{\eta\delta}} + \tilde{C}_7\eta\log\frac{1}{\eta}\log\frac{1}{\eta\delta} \\
&\le \tilde{C}_8\log\frac{1}{\eta}\sqrt{\eta\log\frac{1}{\eta\delta}},
\end{aligned}
$$

where $\tilde{C}_8$ is a constant. Finally, according to Lemma K.12, (47) holds with probability at least $1-\delta$. $\qquad\square$

## K.6 PHASE 2: ITERATES STAYING CLOSE TO MANIFOLD

In this subsection, we show that $\|\boldsymbol{x}_{k,t}^{(s)}\|_2 = \tilde{\mathcal{O}}(\sqrt{\eta})$ and $\|\bar{\boldsymbol{\theta}}^{(s+r)} - \bar{\boldsymbol{\theta}}^{(s)}\|_2 = \tilde{\mathcal{O}}(\eta^{0.5-0.5\beta})$, $\forall 0 \le r \le R_{\mathrm{grp}}$ with high probability.

### K.6.1 ADDITIONAL NOTATIONS

Before presenting the lemmas, we define the following martingale $\{\boldsymbol{m}_{k,t}^{(s)}\}_{t=0}^H$ that will be useful in the proof:

$$
\boldsymbol{m}_{k,t}^{(s)} := \sum_{\tau=0}^{t-1}\boldsymbol{z}_{k,\tau}^{(s)}, \quad \boldsymbol{m}_{k,0} = \mathbf{0}.
$$

We also define $\tilde{\boldsymbol{P}} : \mathbb{R}^d \to \mathbb{R}^{d\times d}$ as an extension of $\partial\Phi$:

$$
\tilde{\boldsymbol{P}}(\boldsymbol{\theta}) := \begin{cases} \partial\Phi(\boldsymbol{\theta}), & \text{if } \boldsymbol{\theta}\in\Gamma^{\epsilon_2}, \\ \mathbf{0}, & \text{otherwise.} \end{cases}
$$

Finally, we define a martingale $\{\boldsymbol{Z}_t^{(s)} : s \ge 0, 0 \le t \le H\}$:

$$
\boldsymbol{Z}_t^{(s)} := \frac{1}{K}\sum_{k\in[K]}\sum_{r=0}^{s-1}\sum_{\tau=0}^{H-1}\tilde{\boldsymbol{P}}(\bar{\boldsymbol{\theta}}^{(r)})\boldsymbol{z}_{k,t}^{(r)} + \frac{1}{K}\sum_{k\in[K]}\sum_{\tau=0}^{t-1}\tilde{\boldsymbol{P}}(\bar{\boldsymbol{\theta}}^{(s)})\boldsymbol{z}_{k,t}^{(s)}, \quad \boldsymbol{Z}_0^{(0)} = \mathbf{0}.
$$

### K.6.2 PROOF FOR THE HIGH PROBABILITY BOUNDS

A direct application of Azuma-Hoeffding's inequality yields the following lemma.

**Lemma K.19** (Concentration property of $\boldsymbol{m}_{k,t}^{(s)}$). *With probability at least $1-\delta$, the following holds:*

$$
\|\boldsymbol{m}_{k,t}^{(s)}\|_2 \le \tilde{C}_9\sqrt{\frac{1}{\eta}\log\frac{1}{\eta\delta}}, \quad \forall 0 \le t \le H, k\in[K], 0 \le s < R_{\mathrm{grp}},
$$

*where $\tilde{C}_9$ is a constant.*

*Proof.* Notice that $\|\boldsymbol{m}_{k,t+1}^{(s)} - \boldsymbol{m}_{k,t}^{(s)}\|_2 \le \sigma_{\max}$. Then by Azuma-Hoeffdings inequality,

$$\mathbb{P}(\|\boldsymbol{m}_{k,t}^{(s)}\|_2 \ge \epsilon') \le 2\exp\left(-\frac{\epsilon'^2}{2t\sigma_{\max}^2}\right).$$

Taking union bound on $K$ clients, $H$ local steps and $R_{\mathrm{grp}}$ rounds, we obtain that the following inequality holds with probability at least $1 - \delta$:

$$\|\boldsymbol{m}_{k,t}^{(s)}\|_2 \le \sigma_{\max}\sqrt{2H\log\frac{2KHR_{\mathrm{grp}}}{\delta}}, \quad \forall 0 \le t \le H, k \in [K], 0 \le s < R_{\mathrm{grp}}.$$

Substituting in $H = \frac{\alpha}{\eta}$ and $R_{\mathrm{grp}} = \lfloor\frac{1}{\alpha\eta^\beta}\rfloor$ yields the lemma. $\qquad\square$

Again applying Azuma-Hoeffding's inequality, we have the following lemma about the concentration property of $\boldsymbol{Z}_t^{(s)}$.

**Lemma K.20** (Concentration property of $\boldsymbol{Z}_t^{(s)}$). *With probability at least $1 - \delta$, the following inequality holds:*

$$\|\boldsymbol{Z}_H^{(s)}\|_2 \le \tilde{C}_{12}\eta^{-0.5-0.5\beta}\sqrt{\log\frac{1}{\eta\delta}}, \quad \forall 0 \le s < R_{\mathrm{grp}}.$$

*Proof.* Notice that $\|\boldsymbol{Z}_{t+1}^{(s)} - \boldsymbol{Z}_t^{(s)}\|_2 \le \nu_2\sigma_{\max}, \forall 0 \le t \le H-1$ and $\|\boldsymbol{Z}_0^{(s+1)} - \boldsymbol{Z}_H^{(s)}\|_2 \le \nu_2\sigma_{\max}$. By Azuma-Hoeffding's inequality,

$$\mathbb{P}(\|\boldsymbol{Z}_t^{(s)}\|_2 \ge \epsilon') \le 2\exp\left(-\frac{\epsilon'^2}{2(sH+t)\nu_2^2\sigma_{\max}^2}\right).$$

Taking union bound on $R_{\mathrm{grp}}$ rounds, we obtain that the following inequality holds with probability at least $1 - \delta$:

$$\|\boldsymbol{Z}_H^{(s)}\|_2 \le \sigma_{\max}\nu_2\sqrt{2HR_{\mathrm{grp}}\log\frac{2R_{\mathrm{grp}}}{\delta}}, \quad \forall 0 \le s < R_{\mathrm{grp}}.$$

Substituting in $H = \frac{\alpha}{\eta}$ and $R_{\mathrm{grp}} = \lfloor\frac{1}{\alpha\eta^\beta}\rfloor$ yields the lemma. $\qquad\square$

We proceed to present a direct corollary of Lemma K.17 which provides a bound for the potential function over $R_{\mathrm{grp}}$ rounds.

**Lemma K.21.** *Given $\|\bar{\boldsymbol{\theta}}^{(0)} - \boldsymbol{\phi}^{(0)}\|_2 \le C_0\sqrt{\eta\log\frac{1}{\eta}}$ where $C_0$ is a constant, then for $\delta = \mathcal{O}(\mathrm{poly}(\eta))$, with probability at least $1 - \delta$,*

$$\bar{\boldsymbol{\theta}}^{(s)} \in \Gamma^{\epsilon_0}, \quad \tilde{\Psi}(\bar{\boldsymbol{\theta}}^{(s)}) \le C_1\sqrt{\eta\log\frac{1}{\eta\delta}}, \quad \forall 0 \le s < R_{\mathrm{grp}}, \tag{52}$$

*and*

$$\bar{\boldsymbol{\theta}}_{k,t}^{(s)} \in \Gamma^{\epsilon_2}, \quad \tilde{\Psi}(\bar{\boldsymbol{\theta}}_{k,t}^{(s)}) \le C_1\sqrt{\eta\log\frac{1}{\eta\delta}}, \quad \forall 0 \le s < R_{\mathrm{grp}}, 0 \le t \le H, k \in [K], \tag{53}$$

*where $C_1$ is a constant that can depend on $C_0$.*

Furthermore,

$$\tilde{\Psi}(\bar{\boldsymbol{\theta}}^{(R_{\mathrm{grp}})}) \le \tilde{C}_{10}\sqrt{\eta\log\frac{1}{\eta\delta}},$$

where $\tilde{C}_9$ is a constant independent of $C_0$.

*Proof.* By $\rho_2$-smoothness of $\mathcal{L}$, $\tilde{\Psi}(\bar{\boldsymbol{\theta}}^{(0)}) \leq C_0\sqrt{\frac{\eta\rho_2}{2}\log\frac{1}{\eta}}$. Substituting $R_{\mathrm{grp}} = \lfloor\frac{1}{\alpha\eta^\beta}\rfloor$ and $\tilde{\Psi}(\bar{\boldsymbol{\theta}}^{(0)}) \leq C_0\sqrt{\frac{\eta\rho_2}{2}\log\frac{1}{\eta}}$ into Lemma K.17, for $\delta = \mathcal{O}(\mathrm{poly}(\eta))$, with probability at least $1-\delta$, (52) and (53) where $C_1$ is a constant that can depend on $C_0$.

Furthermore, for round $\bar{\boldsymbol{\theta}}^{(R_{\mathrm{grp}})}$,

$$\tilde{\Psi}(\bar{\boldsymbol{\theta}}^{(R_{\mathrm{grp}})}) \leq \exp(-\mathcal{O}(\eta^{-\beta})) + \frac{1}{1-\exp(-\alpha\mu/2)}\tilde{C}_5\sqrt{\eta\log\frac{R_{\mathrm{grp}}}{\eta\delta}} \leq \tilde{C}_{10}\sqrt{\eta\log\frac{1}{\eta\delta}},$$

where $\tilde{C}_9$ is a constant independent of $C_0$. $\qquad\square$

**Lemma K.22.** *Given* $\|\bar{\boldsymbol{\theta}}^{(0)} - \boldsymbol{\phi}^{(0)}\|_2 \leq C_0\sqrt{\eta\log\frac{1}{\eta}}$ *where* $C_0$ *is a constant, then for* $\delta = \mathcal{O}(\mathrm{poly}(\eta))$, *with probability at least* $1-\delta$, *for all* $0 \leq s_0 < R_{\mathrm{grp}}, 0 \leq t \leq H, k \in [K]$,

$$\|\boldsymbol{x}_{k,t}^{(s)}\|_2 \leq C_2\sqrt{\eta\log\frac{1}{\eta\delta}}, \quad \|\bar{\boldsymbol{x}}_H^{(s)}\|_2 \leq C_2\sqrt{\eta\log\frac{1}{\eta\delta}},$$

$$\|\bar{\boldsymbol{\theta}}_{k,t}^{(s)} - \bar{\boldsymbol{\theta}}^{(s)}\|_2 \leq C_2\sqrt{\eta\log\frac{1}{\eta\delta}}, \quad \|\bar{\boldsymbol{\theta}}^{(s+1)} - \bar{\boldsymbol{\theta}}^{(s)}\|_2 \leq C_2\sqrt{\eta\log\frac{1}{\eta\delta}}.$$

*where* $C_2$ *is a constant that can depend* $C_0$. *Furthermore,*

$$\|\bar{\boldsymbol{\theta}}^{(R_{\mathrm{grp}})} - \boldsymbol{\phi}^{(R_{\mathrm{grp}})}\|_2 \leq \tilde{C}_{11}\sqrt{\eta\log\frac{1}{\eta\delta}},$$

*where* $\tilde{C}_{11}$ *is a constant independent of* $C_0$.

*Proof.* Decomposing $\boldsymbol{x}_{k,t}^{(s)}$ by triangle inequality, we have

$$\|\boldsymbol{x}_{k,t}^{(s)}\|_2 \leq \|\boldsymbol{\theta}_{k,t}^{(s)} - \bar{\boldsymbol{\theta}}^{(s)}\|_2 + \|\bar{\boldsymbol{\theta}}^{(s)} - \boldsymbol{\phi}^{(s)}\|_2.$$

We first bound $\|\bar{\boldsymbol{\theta}}^{(s)} - \boldsymbol{\phi}^{(s)}\|_2$. By Lemma K.21, for $\delta = \mathcal{O}(\mathrm{poly}(\eta))$, with probability at least $1 - \frac{\delta}{2}$,

$$\tilde{\Psi}(\bar{\boldsymbol{\theta}}^{(s)}) \leq C_1\sqrt{\eta\log\frac{2}{\eta\delta}}, \forall 0 \leq s < R_{\mathrm{grp}}, \tag{54}$$

$$\tilde{\Psi}(\boldsymbol{\theta}_{k,t}^{(s)}) \leq C_1\sqrt{\eta\log\frac{2}{\eta\delta}}, \quad \forall 0 \leq s < R_{\mathrm{grp}}, 0 \leq t \leq H, \tag{55}$$

and

$$\tilde{\Psi}(\bar{\boldsymbol{\theta}}^{(R_{\mathrm{grp}})}) \leq \tilde{C}_{10}\sqrt{\eta\log\frac{2}{\eta\delta}}, \tag{56}$$

where $C_2$ is a constant that may depend on $C_0$ and $\tilde{C}_{10}$ is a constant independent of $C_0$. When (54) and (56) hold, by Lemma K.10,

$$\|\bar{\boldsymbol{\theta}}^{(s)} - \boldsymbol{\phi}^{(s)}\|_2 \leq \sqrt{\frac{2}{\mu}}\tilde{\Psi}(\bar{\boldsymbol{\theta}}^{(s)}) \leq C_1\sqrt{\frac{2\eta}{\mu}\log\frac{2}{\eta\delta}}, \tag{57}$$

$$\|\bar{\boldsymbol{\theta}}^{(R_{\mathrm{grp}})} - \boldsymbol{\phi}^{(R_{\mathrm{grp}})}\|_2 \leq \sqrt{\frac{2}{\mu}}\tilde{\Psi}(\bar{\boldsymbol{\theta}}^{(R_{\mathrm{grp}})}) \leq \tilde{C}_{10}\sqrt{\frac{2\eta}{\mu}\log\frac{2}{\eta\delta}}. \tag{58}$$

Then we bound $\|\boldsymbol{\theta}_{k,t}^{(s)} - \bar{\boldsymbol{\theta}}^{(s)}\|_2$. By the update rule, we have

$$\boldsymbol{\theta}_{k,t}^{(s)} = \bar{\boldsymbol{\theta}}^{(s)} - \eta\sum_{\tau=0}^{t-1}\nabla\mathcal{L}(\boldsymbol{\theta}_{k,\tau}^{(s)}) - \eta\sum_{\tau=0}^{t-1}\boldsymbol{z}_{k,\tau}^{(s)} = \bar{\boldsymbol{\theta}}^{(s)} - \eta\sum_{\tau=0}^{t-1}\nabla\mathcal{L}(\boldsymbol{\theta}_{k,\tau}^{(s)}) - \eta\boldsymbol{m}_{k,t}^{(s)}.$$

Still by triangle inequality, we have

$$\|\boldsymbol{\theta}_{k,t}^{(s)} - \bar{\boldsymbol{\theta}}^{(s)}\|_2 \leq \eta \sum_{\tau=0}^{t-1} \|\nabla\mathcal{L}(\boldsymbol{\theta}_{k,\tau}^{(s)})\|_2 + \eta\|\boldsymbol{m}_{k,t}^{(s)}\|_2.$$

Due to $\rho_2$-smoothness of $\mathcal{L}$, when (55) holds,

$$\|\nabla\mathcal{L}(\boldsymbol{\theta}_{k,\tau}^{(s)})\|_2 \leq \sqrt{2\rho_2}\tilde{\Psi}(\boldsymbol{\theta}_{k,\tau}^{(s)}) \leq C_1\sqrt{2\rho_2\eta\log\frac{2}{\eta\delta}}. \tag{59}$$

By Lemma K.19, with probability at least $1 - \frac{\delta}{2}$,

$$\|\boldsymbol{m}_{k,t}^{(s)}\|_2 \leq \tilde{C}_9\sqrt{\frac{1}{\eta}\log\frac{2}{\eta\delta}}, \quad \forall 0 \leq t \leq H, k \in [K], 0 \leq s < R_{\mathrm{grp}}. \tag{60}$$

Combining (59) and (60), when (55) and (56) hold simultaneously, there exists a constant $C_3$ which can depend on $C_0$ such that

$$\|\boldsymbol{\theta}_{k,t}^{(s)} - \bar{\boldsymbol{\theta}}^{(s)}\|_2 \leq C_3\sqrt{\eta\log\frac{1}{\eta\delta}}, \quad \forall k \in [K], 0 \leq t \leq H. \tag{61}$$

By triangle inequality,

$$\|\bar{\boldsymbol{\theta}}^{(s+1)} - \bar{\boldsymbol{\theta}}^{(s)}\|_2 \leq C_3\sqrt{\eta\log\frac{1}{\eta\delta}}.$$

Combining (57), (58) and (61), we complete the proof. $\qquad\square$

Then we provide high probability bounds for the movement of $\phi^{(s)}$ within $R_{\mathrm{grp}}$ rounds.

**Lemma K.23.** *Given $\|\bar{\boldsymbol{\theta}}^{(0)} - \phi^{(0)}\|_2 \leq C_0\sqrt{\eta\log\frac{1}{\eta}}$ where $C_0$ is a constant, then for $\delta = \mathcal{O}(\mathrm{poly}(\eta))$, with probability at least $1 - \delta$,*

$$\|\phi^{(s)} - \phi^{(0)}\|_2 \leq C_4\eta^{0.5-0.5\beta}\sqrt{\log\frac{1}{\eta\delta}}, \quad \forall 1 \leq s \leq R_{\mathrm{grp}}.$$

*where $C_4$ is a constant that can depend on $C_0$.*

*Proof.* By the update rule of Local SGD,

$$\boldsymbol{\theta}_{k,H}^{(s)} = \bar{\boldsymbol{\theta}}^{(s)} - \eta\sum_{t=0}^{H-1}\nabla\mathcal{L}(\boldsymbol{\theta}_{k,t}^{(s)}) - \eta\sum_{t=0}^{H-1}\boldsymbol{z}_{k,t}^{(s)}$$

Averaging among $K$ clients gives

$$\bar{\boldsymbol{\theta}}^{(s+1)} = \bar{\boldsymbol{\theta}}^{(s)} - \frac{\eta}{K}\sum_{t=0}^{H-1}\sum_{k\in[K]}\nabla\mathcal{L}(\boldsymbol{\theta}_{k,t}^{(s)}) - \frac{\eta}{K}\sum_{t=0}^{H-1}\sum_{k\in[K]}\boldsymbol{z}_{k,t}^{(s)}.$$

By Lemma K.22, for $\delta = \mathcal{O}(\mathrm{poly}(\eta))$, the following holds with probability at least $1 - \delta/3$,

$$\|\boldsymbol{\theta}_{k,t}^{(s)} - \bar{\boldsymbol{\theta}}^{(s)}\|_2 \leq C_2\sqrt{\eta\log\frac{3}{\eta\delta}}, \ \boldsymbol{\theta}_{k,t}^{(s)} \in B^{\epsilon_0}(\phi^{(s)}), \ \forall 0 \leq s < R_{\mathrm{grp}}, 0 \leq t \leq H, k \in [K], \tag{62}$$

$$\|\bar{\boldsymbol{\theta}}^{(s+1)} - \bar{\boldsymbol{\theta}}^{(s)}\|_2 \leq C_2\sqrt{\eta\log\frac{3}{\eta\delta}}, \quad \bar{\boldsymbol{\theta}}^{(s)}, \bar{\boldsymbol{\theta}}^{(s+1)} \in B^{\epsilon_0}(\phi^{(s)}), \quad \forall 0 \leq s < R_{\mathrm{grp}}. \tag{63}$$

When (62) and (63) hold, we can expand $\Phi(\bar{\boldsymbol{\theta}}^{(s+1)})$ as follows:

$$\boldsymbol{\phi}^{(s+1)} = \boldsymbol{\phi}^{(s)} + \partial\Phi(\bar{\boldsymbol{\theta}}^{(s)})(\bar{\boldsymbol{\theta}}^{(s+1)} - \bar{\boldsymbol{\theta}}^{(s)}) + \frac{1}{2}\partial^2\Phi(\tilde{\boldsymbol{\theta}}^{(s)})[\bar{\boldsymbol{\theta}}^{(s+1)} - \bar{\boldsymbol{\theta}}^{(s)}, \bar{\boldsymbol{\theta}}^{(s+1)} - \bar{\boldsymbol{\theta}}^{(s)}]$$

$$= \boldsymbol{\phi}^{(s)} \underbrace{- \frac{\eta}{K}\sum_{t=0}^{H-1}\sum_{k\in[K]}\partial\Phi(\bar{\boldsymbol{\theta}}^{(s)})\nabla\mathcal{L}(\boldsymbol{\theta}_{k,t}^{(s)})}_{\mathcal{T}_1^{(s)}} \underbrace{- \frac{\eta}{K}\partial\Phi(\bar{\boldsymbol{\theta}}^{(s)})\sum_{t=0}^{H-1}\sum_{k\in[K]}\boldsymbol{z}_{k,t}^{(s)}}_{\mathcal{T}_2^{(s)}}$$

$$\underbrace{+ \frac{1}{2}\partial^2\Phi(a^{(s)}\bar{\boldsymbol{\theta}}^{(s)} + (1-a^{(s)})\bar{\boldsymbol{\theta}}^{(s+1)})[\boldsymbol{\theta}^{(s+1)} - \boldsymbol{\theta}^{(s)}, \boldsymbol{\theta}^{(s+1)} - \boldsymbol{\theta}^{(s)}]}_{\mathcal{T}_3^{(s)}},$$

where $a^{(s)} \in (0,1)$. Telescoping from round 0 to $s-1$, we have

$$\|\boldsymbol{\phi}^{(s)} - \boldsymbol{\phi}^{(0)}\|_2 = \sum_{r=0}^{s-1}\mathcal{T}_1^{(r)} + \sum_{r=0}^{s-1}\mathcal{T}_2^{(r)} + \sum_{r=0}^{s-1}\mathcal{T}_3^{(r)}.$$

From (63), we can bound $\|\mathcal{T}_3^{(s)}\|_2$ by $\|\mathcal{T}_3^{(s)}\|_2 \leq \frac{1}{2}\nu_2 C_2^2\eta\log\frac{3}{\eta\delta}$. We proceed to bound $\|\mathcal{T}_1^{(s)}\|_2$. When (62) and (63) hold, we have

$$\partial\Phi(\bar{\boldsymbol{\theta}}^{(s)})\nabla\mathcal{L}(\boldsymbol{\theta}_{k,t}^{(s)}) = \partial\Phi(\boldsymbol{\theta}_{k,t}^{(s)})\nabla\mathcal{L}(\boldsymbol{\theta}_{k,t}^{(s)}) + \partial^2\Phi(\hat{\boldsymbol{\theta}}_{k,t}^{(s)})[\boldsymbol{\theta}_{k,t}^{(s)} - \bar{\boldsymbol{\theta}}^{(s)}, \nabla\mathcal{L}(\boldsymbol{\theta}_{k,t}^{(s)})]$$

$$= \partial^2\Phi(b_{k,t}^{(s)}\bar{\boldsymbol{\theta}}^{(s)} + (1-b_{k,t}^{(s)})\hat{\boldsymbol{\theta}}_{k,t}^{(s)})[\boldsymbol{\theta}_{k,t}^{(s)} - \bar{\boldsymbol{\theta}}^{(s)}, \nabla\mathcal{L}(\boldsymbol{\theta}_{k,t}^{(s)})],$$

where $b_{k,t}^{(s)} \in (0,1)$. By Lemma K.17, with probability at least $1 - \delta/3$, the following holds:

$$\|\nabla\mathcal{L}(\boldsymbol{\theta}_{k,t}^{(s)})\|_2 \leq \sqrt{2\rho_2}\tilde{\Psi}(\boldsymbol{\theta}_{k,t}^{(s)}) \leq C_1\sqrt{2\rho_2\eta\log\frac{3}{\eta\delta}}, \forall k\in[K], 0\leq t\leq H, 0\leq s < R_{\text{grp}}. \quad (64)$$

When (62), (63) and (64) hold simultaneously, we have for all $0 \leq s < R_{\text{grp}}$,

$$\|\mathcal{T}_1^{(s)}\|_2 \leq \frac{\eta\nu_2}{K}\sum_{t=0}^{H-1}\|\boldsymbol{\theta}_{k,t}^{(s)} - \bar{\boldsymbol{\theta}}^{(s)}\|_2\|\nabla\mathcal{L}(\boldsymbol{\theta}_{k,t}^{(s)})\|_2$$

$$\leq \frac{\alpha\nu_2\sqrt{2\rho_2}C_1 C_2}{K}\eta\log\frac{3}{\eta\delta}.$$

Finally, we bound $\|\sum_{r=0}^{s-1}\mathcal{T}_2^{(r)}\|_2$. By Lemma K.20, the following inequality holds with probability at least $1 - \delta/3$:

$$\|\boldsymbol{Z}_H^{(s)}\|_2 \leq \tilde{C}_{12}\eta^{-0.5-0.5\beta}\sqrt{\log\frac{3}{\eta\delta}}, \quad \forall 0 \leq s < R_{\text{grp}}. \quad (65)$$

When (62), (63) and (65) hold simultaneously, we have

$$\|\sum_{r=0}^{s}\mathcal{T}_2^{(r)}\|_2 = \eta\|\boldsymbol{Z}_H^{(s)}\|_2 \leq \tilde{C}_{12}\eta^{0.5-0.5\beta}\sqrt{\log\frac{3}{\eta\delta}}, \quad \forall 0 \leq s < R_{\text{grp}}$$

Combining the bounds for $\|\mathcal{T}_1^{(s)}\|_2$, $\|\sum_{r=0}^{s}\mathcal{T}_2^{(r)}\|_2$ and $\|\mathcal{T}_3^{(s)}\|_2$ and taking union bound, we obtain that for $\delta = \mathcal{O}(\text{poly}(\eta))$, the following inequality holds with probability at least $1 - \delta$:

$$\|\boldsymbol{\phi}^{(s)} - \boldsymbol{\phi}^{(0)}\|_2 \leq C_4\eta^{0.5-0.5\beta}\sqrt{\log\frac{1}{\eta\delta}}, \quad \forall 1 \leq s \leq R_{\text{grp}}.$$

where $C_4$ is a constant that can depend on $C_0$. $\qquad\square$

### K.7 SUMMARY OF THE DYNAMICS AND PROOF OF THEOREMS J.1 AND J.2

Based on the results in Appendix K.5 and Appendix K.6, we summarize the dynamics of Local SGD iterates and then present the proof of Theorems J.1 and J.2 in this subsection. For convenience, we first introduce the definition of **global step** and **$\delta$-good step**.

**Definition K.3** (Global step). *Define $\mathcal{I}$ as the index set $\{(s, t) : s \geq 0, 0 \leq t \leq H\}$ with lexicographical order, which means $(s_1, t_1) \preceq (s_2, t_2)$ if and only if $s_1 < s_2$ or $(s_1 = s_2$ and $t_1 \leq t_2)$. A global step is indexed by $(s, t)$ corresponding to the $t$-th local step at round $s$.*

**Definition K.4** ($\delta$-good step). *In the training process of Local SGD, we say the global step $(s, t) \preceq (R_{\mathrm{tot}}, 0)$ is $\delta$-good if the following inequalities hold:*

$$\|\tilde{\boldsymbol{Z}}_{k,\tau}^{(r)}\|_2 \leq \exp(\alpha\rho_2)\sigma_{\max}\sqrt{2H \log \frac{6HR_{\mathrm{tot}}K}{\delta}}, \qquad \forall k \in [K], (r, \tau) \preceq (s, t),$$

$$\|\boldsymbol{m}_{k,\tau}^{(r)}\|_2 \leq \sigma_{\max}\sqrt{2H \log \frac{6KHR_{\mathrm{tot}}}{\delta}}, \qquad \forall k \in [K], (r, \tau) \preceq (s, t),$$

$$\|\boldsymbol{Z}_H^{(r)}\|_2 \leq \sigma_{\max}\nu_2\sqrt{2HR_{\mathrm{grp}} \log \frac{2R_{\mathrm{tot}}}{\delta}}, \qquad \forall 0 \leq r < s.$$

Applying the concentration properties of $\tilde{\boldsymbol{Z}}_{k,\tau}^{(r)}, \boldsymbol{m}_{k,\tau}^{(r)}$ and $\boldsymbol{Z}_H^{(r)}$ (Lemmas K.20, K.19 and K.12) yields the following theorem.

**Theorem K.1.** *For $\delta = \mathcal{O}(\mathrm{poly}(\eta))$, with probability at least $1 - \delta$, all global steps $(s, t) \preceq (R_{\mathrm{tot}}, 0)$ are $\delta$-good.*

In the remainder of this subsection, we use $\mathcal{O}(\cdot)$ notation to hide constants independent of $\delta$ and $\eta$.

Below we present a summary of the dynamics of Local SGD when $\bar{\boldsymbol{\theta}}^{(0)}$ is initialized such that $\Phi(\bar{\boldsymbol{\theta}}^{(0)}) \in \Gamma$ and all global steps are $\delta$-good. Phase 1 lasts for $s_0 + s_1 = \mathcal{O}(\log \frac{1}{\eta})$ rounds. At the end of phase 1, the iterate reaches within $\mathcal{O}(\sqrt{\eta \log \frac{1}{\eta\delta}})$ from $\Gamma$, i.e., $\|\bar{\boldsymbol{\theta}}^{(s_0+s_1)} - \boldsymbol{\phi}^{(s_0+s_1)}\|_2 = \mathcal{O}(\sqrt{\eta \log \frac{1}{\eta\delta}})$. The change of the projection on manifold over $s_0 + s_1$ rounds, $\|\boldsymbol{\phi}^{(s_1+s_0)} - \boldsymbol{\phi}^{(0)}\|_2$, is bounded by $\mathcal{O}(\log \frac{1}{\eta}\sqrt{\eta \log \frac{1}{\eta\delta}})$.

After $s_0 + s_1$ rounds, the dynamic enters phase 2 when the iterates stay close to $\Gamma$ with $\bar{\boldsymbol{\theta}}^{(s)} \in \Gamma^{\epsilon_2}, \forall s_0 + s_1 \leq s \leq R_{\mathrm{tot}}$ and $\boldsymbol{\theta}_{k,t}^{(s)} \in \Gamma^{\epsilon_2}, \forall k \in [K], (s_0 + s_1, 0) \preceq (s, t) \preceq (R_{\mathrm{tot}}, 0)$. Furthermore, $\|\boldsymbol{x}_{k,t}^{(s)}\|_2$ and $\|\bar{\boldsymbol{x}}_H^{(s)}\|_2$ satisfy the following equations:

$$\|\boldsymbol{x}_{k,t}^{(s)}\|_2 = \mathcal{O}(\sqrt{\eta \log \frac{1}{\eta\delta}}), \qquad \forall k \in [K], 0 \leq t \leq H, s_0 + s_1 \leq s < R_{\mathrm{tot}},$$

$$\|\bar{\boldsymbol{x}}_H^{(s)}\|_2 = \mathcal{O}(\sqrt{\eta \log \frac{1}{\eta\delta}}), \qquad \forall s_0 + s_1 \leq s < R_{\mathrm{tot}}.$$

Moreover, for $s_0 + s_1 \leq s \leq R_{\mathrm{tot}} - R_{\mathrm{grp}}$, the change of the manifold projection within $R_{\mathrm{grp}}$ rounds can be bounded as follows:

$$\|\boldsymbol{\phi}^{(s+r)} - \boldsymbol{\phi}^{(s)}\|_2 = \mathcal{O}(\eta^{0.5-0.5\beta}\sqrt{\log \frac{1}{\eta\delta}}), \quad \forall 1 \leq r \leq R_{\mathrm{grp}}.$$

After combing through the dynamics of Local SGD iterates during the approaching and drift phase, we are ready to present the proof of Theorems J.1 and J.2, which are direct consequences of the lemmas in Appendix K.5 and K.6.

*Proof of Theorem J.1.* By Lemmas K.15, K.22 and Corollary K.1, for $\delta = \mathcal{O}(\mathrm{poly}(\eta))$, when all global steps are $\delta$-good, $\bar{\boldsymbol{\theta}}^{(s)} \in \Gamma^{\epsilon_2}, \forall s_0 + s_1 \leq s \leq R_{\mathrm{tot}}$ and $\boldsymbol{\theta}_{k,t}^{(s)} \in \Gamma^{\epsilon_2}, \forall k \in [K], (s_0 + s_1, 0) \preceq (s, t) \preceq (R_{\mathrm{tot}}, 0)$ and $\|\boldsymbol{x}_{k,t}^{(s)}\|_2, \|\bar{\boldsymbol{x}}_H^{(s)}\|_2$ satisfy the following equations:

$$\|\boldsymbol{x}_{k,t}^{(s)}\|_2 = \mathcal{O}(\sqrt{\eta \log \frac{1}{\eta\delta}}), \qquad \forall k \in [K], 0 \leq t \leq H, s_0 + s_1 \leq s < R_{\mathrm{tot}},$$

$$\|\bar{\boldsymbol{x}}_H^{(s)}\|_2 = \mathcal{O}(\sqrt{\eta \log \frac{1}{\eta\delta}}), \qquad \forall s_0 + s_1 \leq s < R_{\mathrm{tot}}.$$

Hence $\|\bar{\boldsymbol{x}}_0^{(R_{\text{tot}})}\|_2 = \mathcal{O}(\tilde{\Psi}(\bar{\boldsymbol{\theta}}^{(R_{\text{tot}})})) = \mathcal{O}(\|\bar{\boldsymbol{x}}_H^{(R_{\text{tot}}-1)}\|_2) = \mathcal{O}(\sqrt{\eta \log \frac{1}{\eta\delta}})$ by smoothness of $\mathcal{L}$ and Lemma K.10. According to Theorem K.1, with probability at least $1 - \delta$, all global steps are $\delta$-good, thus completing the proof. $\qquad\square$

*Proof of Theorem J.2.* By Lemma K.23, for $\delta = \mathcal{O}(\text{poly}(\eta))$, when all global steps are $\delta$-good, then $\forall s_0 + s_1 \leq s \leq R_{\text{tot}} - R_{\text{grp}}$,

$$\|\boldsymbol{\phi}^{(s+r)} - \boldsymbol{\phi}^{(s)}\|_2 = \tilde{\mathcal{O}}(\eta^{0.5-0.5\beta}), \quad \forall 0 \leq r \leq R_{\text{grp}}.$$

Also, by Lemma K.18, when all global steps are $\delta$-good, the change of projection on manifold over $s_0 + s_1$ rounds (i.e., Phase 1), $\|\boldsymbol{\phi}^{(s_0+s_1)} - \boldsymbol{\phi}^{(0)}\|_2$ is bounded by $\tilde{\mathcal{O}}(\sqrt{\eta})$. According to Theorem K.1, with probability at least $1 - \delta$, all global steps are $\delta$-good, thus completing the proof. $\qquad\square$

### K.8 Proof of Theorem 3.3

In this subsection, we explicitly derive the dependency of the approximation error on $\alpha$. The proofs are quite similar to those in Appendix K.5 and hence we only state the key proof idea for brevity. With the same method as the proofs in Appendix K.5.2, we can show that with high probability, $\|\bar{\boldsymbol{\theta}}^{(s)} - \boldsymbol{\phi}^{(s)}\|_2 \leq \frac{1}{2}\sqrt{\frac{\mu}{\rho_2}}$ after $s_0' = \mathcal{O}(1)$ rounds. Below we focus on the dynamics of Local SGD thereafter. We first remind the readers of the definition of $\{\tilde{\boldsymbol{Z}}_{k,t}^s\}$:

$$\tilde{\boldsymbol{Z}}_{k,t}^{(s)} := \sum_{\tau=0}^{t-1} \left( \prod_{l=\tau+1}^{t-1} (\boldsymbol{I} - \eta\nabla^2\mathcal{L}(\tilde{\boldsymbol{u}}_l^{(s)})) \right) \boldsymbol{z}_{k,\tau}^{(s)}, \qquad \tilde{\boldsymbol{Z}}_{k,0}^{(s)} = \boldsymbol{0}.$$

We have the following lemma that controls the norm of the matrix product $\prod_{l=\tau+1}^{t-1}(\boldsymbol{I} - \eta\nabla^2\mathcal{L}(\tilde{\boldsymbol{u}}_l^{(s)}))$.

**Lemma K.24.** *Given $\bar{\boldsymbol{\theta}}^{(s)} \in \Gamma^{\epsilon_0}$, then there exists a positive constant $C_3'$ independent of $\alpha$ such that for all $0 \leq \tau < t \leq H$,*

$$\left\| \prod_{l=\tau+1}^{t-1} (\boldsymbol{I} - \eta\nabla^2\mathcal{L}(\tilde{\boldsymbol{u}}_l^{(s)})) \right\|_2 \leq C_3'.$$

*Proof.* Since $\bar{\boldsymbol{\theta}}^{(s)} \in \Gamma^{\epsilon_0}$, then $\tilde{\boldsymbol{u}}_t^{(s)} \in \Gamma^{\epsilon_1}$ for all $0 \leq t \leq H$. We first bound the minimum eigenvalue of $\nabla^2\mathcal{L}(\tilde{\boldsymbol{u}}_t^{(s)})$. Due to the PL condition, by Lemma K.6, for $\eta \leq \frac{1}{\rho_2}$,

$$\mathcal{L}(\tilde{\boldsymbol{u}}_t^{(s)}) - \mathcal{L}^* \leq (1 - \mu\eta)^t \left( \mathcal{L}(\bar{\boldsymbol{\theta}}^{(s)}) - \mathcal{L}^* \right) \leq \exp(-\mu t\eta)(\mathcal{L}(\bar{\boldsymbol{\theta}}^{(s)}) - \mathcal{L}^*), \quad \forall 0 \leq t \leq H.$$

Therefore,

$$\tilde{\Psi}(\tilde{\boldsymbol{u}}_t^{(s)}) \leq \exp(-\mu t\eta/2)\tilde{\Psi}(\bar{\boldsymbol{\theta}}^{(s)}).$$

Let $C_1' = \rho_3\sqrt{\frac{\rho_2}{\mu}}$. By Weyl's inequality,

$$\begin{aligned}
|\lambda_{\min}(\nabla^2\mathcal{L}(\tilde{\boldsymbol{u}}_t^{(s)}))| &= |\lambda_{\min}(\nabla^2\mathcal{L}(\tilde{\boldsymbol{u}}_t^{(s)})) - \lambda_{\min}(\nabla^2\mathcal{L}(\Phi(\tilde{\boldsymbol{u}}_t^{(s)})))| \\
&\leq \rho_3\|\nabla^2\mathcal{L}(\tilde{\boldsymbol{u}}_t^{(s)}) - \nabla^2\mathcal{L}(\Phi(\tilde{\boldsymbol{u}}_t^{(s)}))\|_2 \\
&\leq \rho_3\|\tilde{\boldsymbol{u}}_t^{(s)} - \Phi(\tilde{\boldsymbol{u}}_t^{(s)})\|_2 \\
&\leq \rho_3\sqrt{\frac{2}{\mu}}\exp(-\mu t\eta/2)\tilde{\Psi}(\bar{\boldsymbol{\theta}}^{(s)}) \\
&\leq C_1'\exp(-\mu t\eta/2)\epsilon_0,
\end{aligned}$$

where the last two inequalities use Lemmas K.10 and K.7 respectively. Therefore, for all $0 \leq t \leq H$ and $0 \leq \tau \leq t - 1$,

$$\| \prod_{l=\tau+1}^{t-1} (\boldsymbol{I} - \eta \nabla^2 \mathcal{L}(\tilde{\boldsymbol{u}}_l^{(s)})) \|_2 \leq \prod_{l=\tau+1}^{t-1} (1 + \eta |\lambda_{\min} \nabla^2 \mathcal{L}(\tilde{\boldsymbol{u}}_l^{(s)})|)$$

$$\leq \prod_{l=0}^{\infty} (1 + \eta |\lambda_{\min} \nabla^2 \mathcal{L}(\tilde{\boldsymbol{u}}_l^{(s)})|)$$

$$\leq \exp(\eta \epsilon_0 C_1' \sum_{l=0}^{\infty} \exp(-\mu l \eta / 2)). \tag{66}$$

For sufficiently small $\eta$, there exists a constant $C_2'$ such that

$$\sum_{l=0}^{\infty} \exp(-\mu l \eta / 2)) = \frac{1}{1 - \exp(-\mu \eta / 2)} \leq \frac{C_2'}{\eta}. \tag{67}$$

Substituting (67) into (66), we obtain the lemma. $\square$

Based on Lemma K.24, we obtain the following lemma about the concentration property of $\tilde{\boldsymbol{Z}}_{k,t}^{(s)}$, which can be derived in the same way as Lemma K.12.

**Lemma K.25.** *Given $\bar{\boldsymbol{\theta}}^{(s)} \in \Gamma^{\epsilon_0}$ , then with probability at least $1 - \delta$,*

$$\|\tilde{\boldsymbol{Z}}_{k,t}^{(s)}\|_2 \leq C_3' \sigma_{\max} \sqrt{\frac{2\alpha}{\eta} \log \frac{2\alpha K}{\eta \delta}}, \qquad \forall 0 \leq t \leq H, k \in [K],$$

*where $C_3'$ is defined in Lemma K.24.*

The following lemma can be derived analogously to Lemma K.14 but the error bound is tighter in terms of its dependency on $\alpha$.

**Lemma K.26.** *Given $\bar{\boldsymbol{\theta}}^{(s)} \in \Gamma^{\epsilon_1}$, then for $\delta = \mathcal{O}(\mathrm{poly}(\eta))$, with probability at least $1 - \delta$, there exists a constant $C_4'$ independent of $\alpha$ such that*

$$\|\boldsymbol{\theta}_{k,t}^{(s)} - \tilde{\boldsymbol{u}}_t^{(s)}\|_2 \leq C_4' \sqrt{\alpha \eta \log \frac{\alpha}{\eta \delta}}, \quad \forall 0 \leq t \leq H, k \in [K],$$

*and*

$$\|\bar{\boldsymbol{\theta}}^{(s+1)} - \tilde{\boldsymbol{u}}_H^{(s)}\|_2 \leq C_4' \sqrt{\alpha \eta \log \frac{\alpha}{\eta \delta}}.$$

Then, similar to Lemma K.17, we can show that for $\delta = \mathcal{O}(\mathrm{poly}(\eta))$ and simultaneously all $s \geq s_0' + s_1'$ where $s_1' = \mathcal{O}(\frac{1}{\alpha} \log \frac{1}{\eta})$, it holds with probability at least $1 - \delta$ that $\|\bar{\boldsymbol{\theta}}^{(s)} - \boldsymbol{\phi}^{(s)}\|_2 = \mathcal{O}(\sqrt{\alpha \eta \log \frac{\alpha}{\eta \delta}})$. Note that to eliminate the dependency of the second term's denominator on $\alpha$ in (44), we can discuss the cases of $\alpha > c_0$ and $\alpha < c_0$ respectively where $c_0$ can be an arbitrary positive constant independent of $\alpha$. For the case of $\alpha < c_0$ group $\lceil \frac{c_0}{\alpha} \rceil$ rounds together and repeat the arguments in this subsection to analyze the closeness between Local SGD and GD iterates as well as the evolution of loss.

## K.9 COMPUTING THE MOMENTS FOR ONE "GIANT STEP"

In this subsection, we compute the first and second moments for the change of manifold projection every $R_{\mathrm{grp}}$ rounds of Local SGD. Since the randomness in training might drive the iterate out of the working zone, making the dynamic intractable, we analyze a more well-behaved sequence $\{\hat{\boldsymbol{\theta}}_{k,t}^{(s)} : (s,t) \preceq (R_{\mathrm{tot}}, 0), k \in [K]\}$ which is equal to $\{\boldsymbol{\theta}_{k,t}^{(s)}\}$ with high probability. Specifically, $\hat{\boldsymbol{\theta}}_{k,t}^{(s)}$ equal to $\boldsymbol{\theta}_{k,t}^{(s)}$ if the global step $(s,t)$ is $\eta^{100}$-good and is set as a point $\boldsymbol{\phi}_{\mathrm{null}} \in \Gamma$ otherwise. The formal definition is as follows.

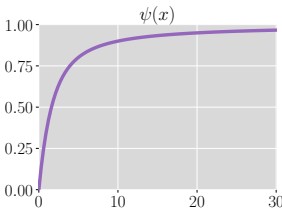

Figure 9: A plot of $\psi(x)$.

**Definition K.5** (Well-behaved sequence). *Denote by $\mathcal{E}_t^{(s)}$ the event $\{$global step $(s,t)$ is $\eta^{100}$-good$\}$. Define a well-behaved sequence $\hat{\boldsymbol{\theta}}_{k,t}^{(s)} := \boldsymbol{\theta}_{k,t}^{(s)} \mathbb{1}_{\mathcal{E}_t^{(s)}} + \boldsymbol{\phi}_{\text{null}} \mathbb{1}_{\bar{\mathcal{E}}_t^{(s)}}$, which satisfies the following update rule:*

$$\hat{\boldsymbol{\theta}}_{k,t+1}^{(s)} = \boldsymbol{\theta}_{k,t+1}^{(s)} \mathbb{1}_{\mathcal{E}_{t+1}^{(s)}} + \boldsymbol{\phi}_{\text{null}} \mathbb{1}_{\bar{\mathcal{E}}_{t+1}^{(s)}} \tag{68}$$

$$= \hat{\boldsymbol{\theta}}_{k,t}^{(s)} - \eta \nabla \mathcal{L}(\hat{\boldsymbol{\theta}}_{k,t}^{(s)}) - \eta \boldsymbol{z}_{k,t}^{(s)} \underbrace{- \mathbb{1}_{\bar{\mathcal{E}}_{t+1}^{(s)}} (\hat{\boldsymbol{\theta}}_{k,t}^{(s)} - \eta \nabla \mathcal{L}(\hat{\boldsymbol{\theta}}_{k,t}^{(s)}) - \eta \boldsymbol{z}_{k,t}^{(s)}) + \mathbb{1}_{\bar{\mathcal{E}}_{t+1}^{(s)}} \boldsymbol{\phi}_{\text{null}}}_{:=\hat{\boldsymbol{e}}_{k,t}^{(s)}}. \tag{69}$$

By Theorem K.1, with probability at least $1 - \eta^{100}$, $\hat{\boldsymbol{\theta}}_{k,t}^{(s)} = \boldsymbol{\theta}_{k,t}^{(s)}$, $\forall k \in [K], (s,t) \preceq (R_{\text{tot}}, 0)$. Similar to $\{\boldsymbol{\theta}_{k,t}^{(s)}\}$, we define the following variables with respect to $\{\hat{\boldsymbol{\theta}}_{k,t}^{(s)}\}$:

$$\hat{\boldsymbol{\theta}}_{\text{avg}}^{(s+1)} := \frac{1}{K} \sum_{k\in[K]} \hat{\boldsymbol{\theta}}_{k,H}^{(s)}, \quad \hat{\boldsymbol{\phi}}^{(s)} := \Phi(\hat{\boldsymbol{\theta}}_{\text{avg}}^{(s)}),$$

$$\hat{\boldsymbol{x}}_{k,t}^{(s)} := \hat{\boldsymbol{\theta}}_{k,t}^{(s)} - \hat{\boldsymbol{\phi}}^{(s)}, \quad \hat{\boldsymbol{x}}_{\text{avg},0}^{(s)} := \hat{\boldsymbol{\theta}}_{\text{avg}}^{(s)} - \hat{\boldsymbol{\phi}}^{(s)}, \quad \hat{\boldsymbol{x}}_{\text{avg},H}^{(s)} := \frac{1}{K} \sum_{k\in[K]} \hat{\boldsymbol{x}}_{k,H}^{(s)}.$$

Notice that $\hat{\boldsymbol{x}}_{k,0}^{(s)} = \hat{\boldsymbol{x}}_{\text{avg},0}^{(s)}$ for all $k \in [K]$. Finally, we introduce the following mapping $\boldsymbol{\Psi}(\boldsymbol{\theta})$ : $\Gamma \to \mathbb{R}^{d\times d}$, which is closely related to $\widehat{\boldsymbol{\Psi}}$ defined in Theorem 3.2.

**Definition K.6.** *For $\boldsymbol{\theta} \in \Gamma$, we define the mapping $\boldsymbol{\Psi}(\boldsymbol{\theta}) : \Gamma \to \mathbb{R}^{d\times d}$:*

$$\boldsymbol{\Psi}(\boldsymbol{\theta}) = \sum_{i,j\in[d]} \psi(\eta H(\lambda_i + \lambda_j)) \left\langle \boldsymbol{\Sigma}(\boldsymbol{\theta}), \boldsymbol{v}_i \boldsymbol{v}_j^\top \right\rangle \boldsymbol{v}_i \boldsymbol{v}_j^\top,$$

*where $\lambda_i, \boldsymbol{v}_i$ are the $i$-th eigenvalue and eigenvector of $\nabla^2 \mathcal{L}(\boldsymbol{\theta})$ and $\boldsymbol{v}_i$'s form an orthonormal basis of $\mathbb{R}^d$. Additionally, $\psi(x) := \frac{e^{-x}-1+x}{x}$ and $\psi(0) = 0$; see Figure 9 for a plot.*

**Remark K.1.** *Intuitively, $\boldsymbol{\Psi}(\boldsymbol{\theta})$ rescales the entries of $\boldsymbol{\Sigma}(\boldsymbol{\theta})$ in the eigenbasis of $\nabla^2 \mathcal{L}(\boldsymbol{\theta})$. When $\nabla^2 \mathcal{L}(\boldsymbol{\theta}) = \text{diag}(\lambda_1, \cdots, \lambda_d) \in \mathbb{R}^{d\times d}$, where $\lambda_i = 0$ for all $m < i \le d$, $\boldsymbol{\Psi}(\boldsymbol{\Sigma}_0)_{i,j} = \psi(\eta H(\lambda_i + \lambda_j))\Sigma_{0,i,j}$. Note that $\boldsymbol{\Psi}(\boldsymbol{\theta})$ can also be written as*

$$\text{vec}(\boldsymbol{\Psi}(\boldsymbol{\theta})) = \psi(\eta H(\nabla^2 \mathcal{L}(\boldsymbol{\theta}) \oplus \nabla^2 \mathcal{L}(\boldsymbol{\theta})))\text{vec}(\boldsymbol{\Sigma}(\boldsymbol{\theta})),$$

*where $\oplus$ denotes the Kronecker sum $\boldsymbol{A} \oplus \boldsymbol{B} = \boldsymbol{A} \otimes \boldsymbol{I}_d + \boldsymbol{I}_d \otimes \boldsymbol{B}$, $\text{vec}(\cdot)$ is the vectorization operator of a matrix and $\psi(\cdot)$ is interpreted as a matrix function.*

Now we are ready to present the result about the moments of $\hat{\boldsymbol{\phi}}^{(s+R_{\text{grp}})} - \hat{\boldsymbol{\phi}}^{(s)}$.

**Theorem K.2.** *For $s_0 + s_1 \le s \le R_{\text{tot}} - R_{\text{grp}}$ and $0 < \beta < 0.5$, the first and second moments of $\hat{\boldsymbol{\phi}}^{(s+R_{\text{grp}})} - \hat{\boldsymbol{\phi}}^{(s)}$ are as follows:*

$$\mathbb{E}[\hat{\boldsymbol{\phi}}^{(s+R_{\text{grp}})} - \hat{\boldsymbol{\phi}}^{(s)} \mid \hat{\boldsymbol{\phi}}^{(s)}, \mathcal{E}_0^{(s)}] = \frac{\eta^{1-\beta}}{2B} \partial^2 \Phi(\hat{\boldsymbol{\phi}}^{(s)})[\boldsymbol{\Sigma}(\hat{\boldsymbol{\phi}}^{(s)}) + (K-1)\boldsymbol{\Psi}(\hat{\boldsymbol{\phi}}^{(s)})]$$
$$+ \tilde{\mathcal{O}}(\eta^{1.5-2\beta}) + \tilde{\mathcal{O}}(\eta), \tag{70}$$

$$\mathbb{E}[(\hat{\boldsymbol{\phi}}^{(s+R_{\text{grp}})} - \hat{\boldsymbol{\phi}}^{(s)})(\hat{\boldsymbol{\phi}}^{(s+R_{\text{grp}})} - \hat{\boldsymbol{\phi}}^{(s)})^\top \mid \hat{\boldsymbol{\phi}}^{(s)}, \mathcal{E}_0^{(s)}] = \frac{\eta^{1-\beta}}{B} \boldsymbol{\Sigma}_\|(\hat{\boldsymbol{\phi}}^{(s)}) + \tilde{\mathcal{O}}(\eta^{1.5-2\beta}) + \tilde{\mathcal{O}}(\eta), \tag{71}$$

*where $\tilde{\mathcal{O}}(\cdot)$ hides log terms and constants independent of $\eta$.*

**Remark K.2.** *By Theorem K.1 and the definition of $\hat{\boldsymbol{\theta}}_{k,t}^{(s)}$, (70) and (71) still hold when we replace $\hat{\boldsymbol{\phi}}^{(s)}$ with $\boldsymbol{\phi}^{(s)}$ and replace $\hat{\boldsymbol{\phi}}^{(s+R_{\mathrm{grp}})}$ with $\boldsymbol{\phi}^{(s+R_{\mathrm{grp}})}$.*

We shall have Theorem K.2 if we prove the following theorem, which directly gives Theorem K.2 with a simple shift of index. For brevity, denote by $\Delta\hat{\boldsymbol{\phi}}^{(s)} := \hat{\boldsymbol{\phi}}^{(s)} - \hat{\boldsymbol{\phi}}^{(0)}$, $\boldsymbol{\Sigma}_0 := \boldsymbol{\Sigma}(\hat{\boldsymbol{\phi}}^{(0)})$, $\boldsymbol{\Sigma}_{0,\|} := \boldsymbol{\Sigma}_\|(\hat{\boldsymbol{\phi}}^{(0)})$.

**Theorem K.3.** *Given $\|\hat{\boldsymbol{\theta}}_{\mathrm{avg}}^{(0)} - \hat{\boldsymbol{\phi}}^{(0)}\|_2 = \mathcal{O}(\sqrt{\eta\log\frac{1}{\eta}})$, for $0 < \beta < 0.5$, the first and second moments of $\Delta\hat{\boldsymbol{\phi}}^{(R_{\mathrm{grp}})}$ are as follows:*

$$\mathbb{E}[\Delta\hat{\boldsymbol{\phi}}^{(R_{\mathrm{grp}})}] = \frac{\eta^{1-\beta}}{2B}\partial^2\Phi(\hat{\boldsymbol{\phi}}^{(0)})[\boldsymbol{\Sigma}_0 + (K-1)\boldsymbol{\Psi}(\hat{\boldsymbol{\phi}}^{(0)})] + \tilde{\mathcal{O}}(\eta^{1.5-2\beta}) + \tilde{\mathcal{O}}(\eta),$$

$$\mathbb{E}[\Delta\hat{\boldsymbol{\phi}}^{(R_{\mathrm{grp}})}\Delta\hat{\boldsymbol{\phi}}^{(R_{\mathrm{grp}})\top}] = \frac{\eta^{1-\beta}}{B}\boldsymbol{\Sigma}_{0,\|} + \tilde{\mathcal{O}}(\eta^{1.5-1.5\beta}) + \tilde{\mathcal{O}}(\eta).$$

We will prove Theorem K.3 in the remainder of this subsection. For convenience, we introduce more notations that will be used throughout the proof. Let $\boldsymbol{H}_0 := \nabla^2\mathcal{L}(\hat{\boldsymbol{\phi}}^{(0)})$. By Assumption 3.2, $\mathrm{rank}(\boldsymbol{H}_0) = m$. WLOG, assume $\boldsymbol{H}_0 = \mathrm{diag}(\lambda_1, \cdots, \lambda_d) \in \mathbb{R}^{d\times d}$, where $\lambda_i = 0$ for all $m < i \leq d$ and $\lambda_1 \geq \lambda_2 \cdots \geq \lambda_m$. By Lemma K.2, $\partial\Phi(\hat{\boldsymbol{\phi}}^{(0)})$ is the projection matrix onto the tangent space $T_{\hat{\boldsymbol{\phi}}^{(0)}}(\Gamma)$ (i.e. the null space of $\nabla^2\mathcal{L}(\hat{\boldsymbol{\phi}}^{(0)})$) and therefore, $\partial\Phi(\hat{\boldsymbol{\phi}}^{(0)}) = \begin{bmatrix} \boldsymbol{0} & \boldsymbol{0} \\ \boldsymbol{0} & \boldsymbol{I}_{d-m} \end{bmatrix}$. Let $\boldsymbol{P}_\| := \partial\Phi(\hat{\boldsymbol{\phi}}^{(0)})$ and $\boldsymbol{P}_\perp := \boldsymbol{I}_d - \boldsymbol{P}_\|$.

Let $\hat{\boldsymbol{A}}_{\mathrm{avg}}^{(s)} := \mathbb{E}[\hat{\boldsymbol{x}}_{\mathrm{avg},H}^{(s)}\hat{\boldsymbol{x}}_{\mathrm{avg},H}^{(s)\top}]$, $\hat{\boldsymbol{q}}_t^{(s)} := \mathbb{E}[\hat{\boldsymbol{x}}_{k,t}^{(s)}]$ and $\hat{\boldsymbol{B}}_t^{(s)} := \mathbb{E}[\hat{\boldsymbol{x}}_{k,t}^{(s)}\Delta\hat{\boldsymbol{\phi}}^{(s)\top}]$. The latter two notations are independent of $k$ since $\hat{\boldsymbol{\theta}}_{1,t}^{(s)}, \ldots, \hat{\boldsymbol{\theta}}_{K,t}^{(s)}$ are identically distributed. The following lemma computes the first and second moments of the change of manifold projection every round.

**Lemma K.27.** *Given $\|\hat{\boldsymbol{\theta}}_{\mathrm{avg}}^{(0)} - \hat{\boldsymbol{\phi}}^{(0)}\|_2 = \mathcal{O}(\sqrt{\eta\log\frac{1}{\eta}})$, for $0 \leq s < R_{\mathrm{grp}}$, the first and second moments of $\hat{\boldsymbol{\phi}}^{(s+1)} - \hat{\boldsymbol{\phi}}^{(s)}$ are as follows:*

$$\mathbb{E}[\hat{\boldsymbol{\phi}}^{(s+1)} - \hat{\boldsymbol{\phi}}^{(s)}] = \boldsymbol{P}_\|\hat{\boldsymbol{q}}_H^{(s)} + \partial^2\Phi(\hat{\boldsymbol{\phi}}^{(0)})[\hat{\boldsymbol{B}}_H^{(s)}] + \frac{1}{2}\partial^2\Phi(\hat{\boldsymbol{\phi}}^{(0)})[\hat{\boldsymbol{A}}_{\mathrm{avg}}^{(s)}] + \tilde{\mathcal{O}}(\eta^{1.5-\beta}), \qquad (72)$$

$$\mathbb{E}[(\hat{\boldsymbol{\phi}}^{(s+1)} - \hat{\boldsymbol{\phi}}^{(s)})(\hat{\boldsymbol{\phi}}^{(s+1)} - \hat{\boldsymbol{\phi}}^{(s)})^\top] = \boldsymbol{P}_\|\hat{\boldsymbol{A}}_{\mathrm{avg}}^{(s)}\boldsymbol{P}_\| + \tilde{\mathcal{O}}(\eta^{1.5-0.5\beta}). \qquad (73)$$

*Proof.* By Taylor expansion, we have

$$\hat{\boldsymbol{\phi}}^{(s+1)} = \Phi\left(\hat{\boldsymbol{\phi}}^{(s)} + \hat{\boldsymbol{x}}_{\mathrm{avg},H}^{(s)}\right)$$

$$= \hat{\boldsymbol{\phi}}^{(s)} + \partial\Phi(\hat{\boldsymbol{\phi}}^{(s)})\hat{\boldsymbol{x}}_{\mathrm{avg},H}^{(s)} + \frac{1}{2}\partial^2\Phi(\hat{\boldsymbol{\phi}}^{(s)})[\hat{\boldsymbol{x}}_{\mathrm{avg},H}^{(s)}\hat{\boldsymbol{x}}_{\mathrm{avg},H}^{(s)\top}] + \mathcal{O}(\|\hat{\boldsymbol{x}}_{\mathrm{avg},H}^{(s)}\|_2^3)$$

$$= \hat{\boldsymbol{\phi}}^{(s)} + \partial\Phi(\hat{\boldsymbol{\phi}}^{(0)} + \Delta\hat{\boldsymbol{\phi}}^{(s)})\hat{\boldsymbol{x}}_{\mathrm{avg},H}^{(s)} + \frac{1}{2}\partial^2\Phi(\hat{\boldsymbol{\phi}}^{(0)} + \Delta\hat{\boldsymbol{\phi}}^{(s)})[\hat{\boldsymbol{x}}_{\mathrm{avg},H}^{(s)}\hat{\boldsymbol{x}}_{\mathrm{avg},H}^{(s)\top}]$$

$$\quad + \mathcal{O}(\|\hat{\boldsymbol{x}}_{\mathrm{avg},H}^{(s)}\|_2^3)$$

$$= \hat{\boldsymbol{\phi}}^{(s)} + \boldsymbol{P}_\|\hat{\boldsymbol{x}}_{\mathrm{avg},H}^{(s)} + \partial^2\Phi(\hat{\boldsymbol{\phi}}^{(0)})[\hat{\boldsymbol{x}}_{\mathrm{avg},H}^{(s)}\Delta\hat{\boldsymbol{\phi}}^{(s)\top}] + \frac{1}{2}\partial^2\Phi(\hat{\boldsymbol{\phi}}^{(0)})[\hat{\boldsymbol{x}}_{\mathrm{avg},H}^{(s)}\hat{\boldsymbol{x}}_{\mathrm{avg},H}^{(s)\top}]$$

$$\quad + \mathcal{O}(\|\Delta\hat{\boldsymbol{\phi}}^{(s)}\|_2^2\|\hat{\boldsymbol{x}}_{\mathrm{avg},H}^{(s)}\|_2 + \|\Delta\hat{\boldsymbol{\phi}}^{(s)}\|_2\|\hat{\boldsymbol{x}}_{\mathrm{avg},H}^{(s)}\|_2^2 + \|\hat{\boldsymbol{x}}_{\mathrm{avg},H}^{(s)}\|_2^3).$$

Rearrange the terms and we obtain:

$$\hat{\boldsymbol{\phi}}^{(s+1)} - \hat{\boldsymbol{\phi}}^{(s)} = \boldsymbol{P}_\|\hat{\boldsymbol{x}}_{\mathrm{avg},H}^{(s)} + \partial^2\Phi(\hat{\boldsymbol{\phi}}^{(0)})[\hat{\boldsymbol{x}}_{\mathrm{avg},H}^{(s)}\Delta\hat{\boldsymbol{\phi}}^{(s)\top}] + \frac{1}{2}\partial^2\Phi(\hat{\boldsymbol{\phi}}^{(0)})[\hat{\boldsymbol{x}}_{\mathrm{avg},H}^{(s)}\hat{\boldsymbol{x}}_{\mathrm{avg},H}^{(s)\top}]$$

$$\quad + \mathcal{O}(\|\Delta\hat{\boldsymbol{\phi}}^{(s)}\|_2^2\|\hat{\boldsymbol{x}}_{\mathrm{avg},H}^{(s)}\|_2 + \|\Delta\hat{\boldsymbol{\phi}}^{(s)}\|_2\|\hat{\boldsymbol{x}}_{\mathrm{avg},H}^{(s)}\|_2^2 + \|\hat{\boldsymbol{x}}_{\mathrm{avg},H}^{(s)}\|_2^3). \qquad (74)$$

Moreover,

$$(\hat{\boldsymbol{\phi}}^{(s+1)} - \hat{\boldsymbol{\phi}}^{(s)})(\hat{\boldsymbol{\phi}}^{(s+1)} - \hat{\boldsymbol{\phi}}^{(s)})^\top = \boldsymbol{P}_\|\hat{\boldsymbol{x}}_{\mathrm{avg},H}^{(s)}\hat{\boldsymbol{x}}_{\mathrm{avg},H}^{(s)\top}\boldsymbol{P}_\| + \mathcal{O}(\|\Delta\hat{\boldsymbol{\phi}}^{(s)}\|_2\|\hat{\boldsymbol{x}}_{\mathrm{avg},H}^{(s)}\|_2^2). \qquad (75)$$

Noticing that $\hat{\boldsymbol{x}}_{k,H}^{(s)} \Delta \hat{\boldsymbol{\phi}}^{(s)\top}$ are identically distributed for all $k \in [K]$, we have $\mathbb{E}[\hat{\boldsymbol{x}}_{\mathrm{avg},H}^{(s)} \Delta \hat{\boldsymbol{\phi}}^{(s)\top}] = \frac{1}{K} \sum_{k \in [K]} \mathbb{E}[\hat{\boldsymbol{x}}_{k,H}^{(s)} \Delta \hat{\boldsymbol{\phi}}^{(s)\top}] = \hat{\boldsymbol{B}}_H^{(s)}$. Then taking expectation of both sides of (74) gives

$$\mathbb{E}[\hat{\boldsymbol{\phi}}^{(s+1)} - \hat{\boldsymbol{\phi}}^{(s)}] = \boldsymbol{P}_\| \hat{\boldsymbol{q}}_H^{(s)} + \partial^2 \Phi(\hat{\boldsymbol{\phi}}^{(0)})[\hat{\boldsymbol{B}}_H^{(s)}] + \frac{1}{2} \partial^2 \Phi(\hat{\boldsymbol{\phi}}^{(0)})[\hat{\boldsymbol{A}}_{\mathrm{avg}}^{(s)}]$$
$$+ \mathcal{O}(\mathbb{E}[\|\Delta\hat{\boldsymbol{\phi}}^{(s)}\|_2^2 \|\hat{\boldsymbol{x}}_{\mathrm{avg},H}^{(s)}\|_2] + \mathbb{E}[\|\Delta\hat{\boldsymbol{\phi}}^{(s)}\|_2 \|\hat{\boldsymbol{x}}_{\mathrm{avg},H}^{(s)}\|_2^2] + \mathbb{E}[\|\hat{\boldsymbol{x}}_{\mathrm{avg},H}^{(s)}\|_2^3]).$$

Again taking expectation of both sides of (75) yields

$$\mathbb{E}[(\hat{\boldsymbol{\phi}}^{(s+1)} - \hat{\boldsymbol{\phi}}^{(s)})(\hat{\boldsymbol{\phi}}^{(s+1)} - \Delta\hat{\boldsymbol{\phi}}^{(s)\top})] = \boldsymbol{P}_\| \hat{\boldsymbol{A}}_{\mathrm{avg}}^{(s)} \boldsymbol{P}_\| + \mathcal{O}(\mathbb{E}[\|\Delta\hat{\boldsymbol{\phi}}^{(s)}\|_2 \|\hat{\boldsymbol{x}}_{\mathrm{avg},H}^{(s)}\|_2^2]).$$

By Lemmas K.22 and K.23, the following holds simultaneously with probability at least $1 - \eta^{100}$:

$$\|\Delta\hat{\boldsymbol{\phi}}^{(s)}\|_2 = \tilde{\mathcal{O}}(\eta^{0.5-0.5\beta}), \quad \|\hat{\boldsymbol{x}}_{\mathrm{avg},H}^{(s)}\|_2 = \tilde{\mathcal{O}}(\eta^{0.5}).$$

Furthermore, since for all $k \in [K]$ and $(s,t) \preceq (R_{\mathrm{tot}}, 0)$, $\hat{\boldsymbol{\theta}}_{k,t}^{(s)}$ stays in $\Gamma^{\epsilon_2}$ which is a bounded set, $\|\Delta\hat{\boldsymbol{\phi}}^{(s)}\|_2$ and $\|\hat{\boldsymbol{x}}_{\mathrm{avg},H}^{(s)}\|_2$ are also bounded. Therefore, we have

$$\mathbb{E}[\|\Delta\hat{\boldsymbol{\phi}}^{(s)}\|_2^2 \|\hat{\boldsymbol{x}}_{\mathrm{avg},H}^{(s)}\|_2] = \tilde{\mathcal{O}}(\eta^{1.5-\beta}), \tag{76}$$

$$\mathbb{E}[\|\Delta\hat{\boldsymbol{\phi}}^{(s)}\|_2 \|\hat{\boldsymbol{x}}_{\mathrm{avg},H}^{(s)}\|_2^2] = \tilde{\mathcal{O}}(\eta^{1.5-0.5\beta}), \tag{77}$$

$$\mathbb{E}[\|\hat{\boldsymbol{x}}_{\mathrm{avg},H}^{(s)}\|_2^3] = \tilde{\mathcal{O}}(\eta^{1.5}), \tag{78}$$

which concludes the proof. $\qquad\square$

We compute $\hat{\boldsymbol{A}}_{\mathrm{avg}}^{(s)}$, $\hat{\boldsymbol{q}}_t^{(s)}$ and $\hat{\boldsymbol{B}}_t^{(s)}$ by solving a set of recursions, which is formulated in the following lemma. Additionally, define $\hat{\boldsymbol{A}}_t^{(s)} := \mathbb{E}[\hat{\boldsymbol{x}}_{k,t}^{(s)} \hat{\boldsymbol{x}}_{k,t}^{(s)\top}]$ and $\hat{\boldsymbol{M}}_t^{(s)} := \mathbb{E}[\hat{\boldsymbol{x}}_{k,t}^{(s)} \hat{\boldsymbol{x}}_{k,t}^{(s)}]$, $(k \neq l)$.

**Lemma K.28.** *Given* $\|\hat{\boldsymbol{\theta}}_{\mathrm{avg}}^{(0)} - \hat{\boldsymbol{\phi}}^{(0)}\|_2 = \mathcal{O}(\sqrt{\eta \log \frac{1}{\eta}})$, *for* $0 \leq s < R_{\mathrm{grp}}$ *and* $0 \leq t < H$, *we have the following recursions.*

$$\hat{\boldsymbol{q}}_{t+1}^{(s)} = \hat{\boldsymbol{q}}_t^{(s)} - \eta \boldsymbol{H}_0 \hat{\boldsymbol{q}}_t^{(s)} - \eta \nabla^3 \mathcal{L}(\boldsymbol{\phi}^{(0)})[\hat{\boldsymbol{B}}_t^{(s)}] - \frac{\eta}{2} \nabla^3 \mathcal{L}(\boldsymbol{\phi}^{(0)})[\hat{\boldsymbol{A}}_t^{(s)}] + \tilde{\mathcal{O}}(\eta^{2.5-\beta}), \tag{79}$$

$$\hat{\boldsymbol{A}}_{t+1}^{(s)} = \hat{\boldsymbol{A}}_t^{(s)} - \eta \boldsymbol{H}_0 \hat{\boldsymbol{A}}_t^{(s)} - \eta \hat{\boldsymbol{A}}_t^{(s)} \boldsymbol{H}_0 + \frac{\eta^2}{B_{\mathrm{loc}}} \boldsymbol{\Sigma}_0 + \tilde{\mathcal{O}}(\eta^{2.5-0.5\beta}), \tag{80}$$

$$\hat{\boldsymbol{M}}_{t+1}^{(s)} = \hat{\boldsymbol{M}}_t^{(s)} - \eta \boldsymbol{H}_0 \hat{\boldsymbol{M}}_t^{(s)} - \eta \hat{\boldsymbol{M}}_t^{(s)} \boldsymbol{H}_0 + \tilde{\mathcal{O}}(\eta^{2.5-0.5\beta}), \tag{81}$$

$$\hat{\boldsymbol{B}}_{t+1}^{(s)} = (\boldsymbol{I} - \eta \boldsymbol{H}_0) \hat{\boldsymbol{B}}_t^{(s)} + \tilde{\mathcal{O}}(\eta^{2.5-\beta}). \tag{82}$$

*Moreover,*

$$\hat{\boldsymbol{A}}_{\mathrm{avg}}^{(s)} = \frac{1}{K} \hat{\boldsymbol{A}}_H^{(s)} + (1 - \frac{1}{K}) \hat{\boldsymbol{M}}_H^{(s)}, \tag{83}$$

$$\hat{\boldsymbol{M}}_0^{(s+1)} = \hat{\boldsymbol{A}}_0^{(s+1)} = \boldsymbol{P}_\perp \hat{\boldsymbol{A}}_{\mathrm{avg}}^{(s)} \boldsymbol{P}_\perp + \mathcal{O}(\eta^{1.5-0.5\beta}), \tag{84}$$

$$\hat{\boldsymbol{q}}_0^{(s+1)} = \boldsymbol{P}_\perp \hat{\boldsymbol{q}}_H^{(s)} - \partial^2 \Phi(\boldsymbol{\phi}^{(0)})[\hat{\boldsymbol{B}}_H^{(s)}] - \frac{1}{2} \partial^2 \Phi(\boldsymbol{\phi}^{(0)})[\hat{\boldsymbol{A}}_{\mathrm{avg}}^{(s)}] + \tilde{\mathcal{O}}(\eta^{1.5-\beta}), \tag{85}$$

$$\hat{\boldsymbol{B}}_0^{(s+1)} = \boldsymbol{P}_\perp \hat{\boldsymbol{B}}_H^{(s)} + \boldsymbol{P}_\perp \hat{\boldsymbol{A}}_{\mathrm{avg}}^{(s)} \boldsymbol{P}_\| + \tilde{\mathcal{O}}(\eta^{1.5-\beta}). \tag{86}$$

*Proof.* We first derive the recursion for $\hat{\boldsymbol{q}}_t^{(s)}$. Recall the update rule for $\hat{\boldsymbol{\theta}}_{k,t}^{(s)}$:

$$\hat{\boldsymbol{\theta}}_{k,t+1}^{(s)} = \hat{\boldsymbol{\theta}}_{k,t}^{(s)} - \eta \nabla \mathcal{L}(\hat{\boldsymbol{\theta}}_{k,t}^{(s)}) - \eta \boldsymbol{z}_{k,t}^{(s)} + \hat{\boldsymbol{e}}_{k,t}^{(s)}.$$

Subtracting $\hat{\phi}^{(s)}$ from both sides gives

$$
\begin{aligned}
\hat{\boldsymbol{x}}_{k,t+1}^{(s)} &= \hat{\boldsymbol{x}}_{k,t}^{(s)} - \eta\nabla\mathcal{L}(\hat{\boldsymbol{\theta}}_{k,t}^{(s)}) - \eta\boldsymbol{z}_{k,t}^{(s)} + \mathcal{O}(\|\hat{e}_{k,t}^{(s)}\|_2) \\
&= \hat{\boldsymbol{x}}_{k,t}^{(s)} - \eta\left(\nabla^2\mathcal{L}(\hat{\boldsymbol{\phi}}^{(s)})\hat{\boldsymbol{x}}_{k,t}^{(s)} + \frac{1}{2}\nabla^3\mathcal{L}(\hat{\boldsymbol{\phi}}^{(s)})[\hat{\boldsymbol{x}}_{k,t}^{(s)}\hat{\boldsymbol{x}}_{k,t}^{(s)\top}] + \mathcal{O}(\|\hat{\boldsymbol{x}}_{k,t}^{(s)}\|_2^3)\right) \\
&\quad - \eta\boldsymbol{z}_{k,t}^{(s)} + \mathcal{O}(\|\hat{e}_{k,t}^{(s)}\|_2) \\
&= \hat{\boldsymbol{x}}_{k,t}^{(s)} - \eta\left(\nabla^2\mathcal{L}(\hat{\boldsymbol{\phi}}^{(0)}) + \nabla^3\mathcal{L}(\hat{\boldsymbol{\phi}}^{(0)})\Delta\hat{\boldsymbol{\phi}}^{(s)} + \mathcal{O}(\|\Delta\hat{\boldsymbol{\phi}}^{(s)}\|^2)\right)\hat{\boldsymbol{x}}_{k,t}^{(s)} \\
&\quad - \frac{\eta}{2}\left(\nabla^3\mathcal{L}(\hat{\boldsymbol{\phi}}^{(0)}) + \mathcal{O}(\|\Delta\hat{\boldsymbol{\phi}}^{(s)}\|_2)\right)[\hat{\boldsymbol{x}}_{k,t}^{(s)}\hat{\boldsymbol{x}}_{kt}^{(s)\top}] - \eta\boldsymbol{z}_{k,t}^{(s)} + \mathcal{O}(\eta\|\hat{\boldsymbol{x}}_{k,t}^{(s)}\|_2^3 + \|\hat{e}_{k,t}^{(s)}\|_2) \\
&= \hat{\boldsymbol{x}}_{k,t}^{(s)} - \eta\boldsymbol{H}_0\hat{\boldsymbol{x}}_{k,t}^{(s)} - \eta\nabla^3\mathcal{L}(\hat{\boldsymbol{\phi}}^{(0)})[\hat{\boldsymbol{x}}_{k,t}^{(s)}\Delta\hat{\boldsymbol{\phi}}^{(s)\top}] - \frac{\eta}{2}\nabla^3\mathcal{L}(\hat{\boldsymbol{\phi}}^{(0)})[\hat{\boldsymbol{x}}_{k,t}^{(s)}\hat{\boldsymbol{x}}_{k,t}^{(s)\top}] - \eta\boldsymbol{z}_{k,t}^{(s)} \\
&\quad + \mathcal{O}(\eta\|\hat{\boldsymbol{x}}_{k,t}^{(s)}\|_2^3 + \eta\|\Delta\hat{\boldsymbol{\phi}}^{(s)}\|_2\|\hat{\boldsymbol{x}}_{k,t}^{(s)}\|_2^2 + \eta\|\Delta\hat{\boldsymbol{\phi}}^{(s)}\|_2^2\|\hat{\boldsymbol{x}}_{k,t}^{(s)}\|_2 + \|\hat{e}_{k,t}^{(s)}\|_2), \quad\quad (87)
\end{aligned}
$$

where the second and third equality perform Taylor expansion. Taking expectation on both sides gives

$$
\begin{aligned}
\hat{\boldsymbol{q}}_{t+1}^{(s)} &= (\boldsymbol{I} - \eta\boldsymbol{H}_0)\hat{\boldsymbol{q}}_t^{(s)} - \eta\nabla^3\mathcal{L}(\hat{\boldsymbol{\phi}}^{(0)})[\hat{\boldsymbol{q}}_t^{(s)}] - \frac{\eta}{2}\nabla^3\mathcal{L}(\hat{\boldsymbol{\phi}}^{(0)})[\hat{\boldsymbol{A}}_t^{(s)}] \\
&\quad + \mathcal{O}\left(\eta\mathbb{E}[\|\hat{\boldsymbol{x}}_{k,t}^{(s)}\|_2^3] + \eta\mathbb{E}[\|\Delta\hat{\boldsymbol{\phi}}^{(s)}\|_2\|\hat{\boldsymbol{x}}_{k,t}^{(s)}\|_2^2] + \eta\mathbb{E}[\|\Delta\hat{\boldsymbol{\phi}}^{(s)}\|_2^2\|\hat{\boldsymbol{x}}_{k,t}^{(s)}\|_2] + \mathbb{E}[\|\hat{e}_{k,t}^{(s)}\|_2]\right).
\end{aligned}
$$

By Theorem K.1, with probability at least $1 - \eta^{100}$, $\hat{e}_{k,t}^{(s)} = \boldsymbol{0}$, $\forall k \in [K], (s,t) \preceq (R_{\mathrm{grp}}, 0)$. Also notice that both $\hat{\boldsymbol{\theta}}_{k,t}^{(s)}$ and $\boldsymbol{\phi}_{\mathrm{null}}$ belong to the bounded set $\Gamma^{\epsilon_2}$. Therefore, $\|\hat{e}_{k,t}^{(s)}\|_2$ is bounded and we have $\mathbb{E}[\|\hat{e}_{k,t}^{(s)}\|_2] = \mathcal{O}(\eta^{100})$. Combining this with (76) to (78) yields (79).

Secondly, we derive the recursion for $\hat{\boldsymbol{B}}_t^{(s)}$. Multiplying both sides of (87) by $\Delta\hat{\boldsymbol{\phi}}^{(s)\top}$ and taking expectation, we have

$$
\hat{\boldsymbol{B}}_{t+1}^{(s)} = (\boldsymbol{I} - \eta\boldsymbol{H}_0)\hat{\boldsymbol{B}}_t^{(s)} + \mathcal{O}(\eta\mathbb{E}[\|\Delta\hat{\boldsymbol{\phi}}^{(s)}\|_2\|\hat{\boldsymbol{x}}_{k,t}^{(s)}\|_2^2 + \|\Delta\hat{\boldsymbol{\phi}}^{(s)}\|_2^2\|\hat{\boldsymbol{x}}_{k,t}^{(s)}\|_2 + \|\hat{e}_{k,t}^{(s)}\|_2]).
$$

Still by Theorem K.1 and (76) to (78), we have (82).

Thirdly, we derive the recursion for $\hat{\boldsymbol{A}}_t^{(s)}$. By (87), we have

$$
\begin{aligned}
\hat{\boldsymbol{A}}_{t+1}^{(s)} &= \hat{\boldsymbol{A}}_t^{(s)} - \eta\boldsymbol{H}_0\hat{\boldsymbol{A}}_t^{(s)} - \eta\hat{\boldsymbol{A}}_t^{(s)}\boldsymbol{H}_0 + \frac{\eta^2}{B_{\mathrm{loc}}}\boldsymbol{\Sigma}_0 + \mathcal{O}(\eta^2\mathbb{E}[\|\Delta\hat{\boldsymbol{\phi}}^{(s)}\|_2 + \|\hat{\boldsymbol{x}}_{k,t}^{(s)}\|_2]) \\
&\quad + \mathcal{O}(\eta\mathbb{E}[\|\hat{\boldsymbol{x}}_{k,t}^{(s)}\|_2^3 + \|\hat{\boldsymbol{x}}_{k,t}^{(s)}\|_2^2\|\Delta\hat{\boldsymbol{\phi}}^{(s)}\|_2 + \|\hat{e}_{k,t}^{(s)}\|_2]) \\
&= (\boldsymbol{I} - \eta\boldsymbol{H}_0)\hat{\boldsymbol{A}}_t^{(s)} + \frac{\eta^2}{B_{\mathrm{loc}}}\boldsymbol{\Sigma}_0 + \tilde{\mathcal{O}}(\eta^{2.5-0.5\beta}),
\end{aligned}
$$

which establishes (80).

Fourthly, we derive the recursion for $\hat{\boldsymbol{M}}_t^{(s)}$. Multiplying both sides of (87) by $\hat{\boldsymbol{x}}_{l,t+1}^{(s)}$ and taking expectation, $l \neq k$, we obtain

$$
\begin{aligned}
\hat{\boldsymbol{M}}_{t+1}^{(s)} &= \hat{\boldsymbol{M}}_t^{(s)} - \eta\boldsymbol{H}_0\hat{\boldsymbol{M}}_t^{(s)} - \eta\hat{\boldsymbol{M}}_t^{(s)}\boldsymbol{H}_0 + \mathcal{O}(\eta\mathbb{E}[\|\hat{\boldsymbol{x}}_{k,t}^{(s)}\|_2\|\hat{\boldsymbol{x}}_{l,t}^{(s)}\|_2\|\Delta\hat{\boldsymbol{\phi}}^{(s)}\|_2]) \\
&\quad + \mathcal{O}(\eta\mathbb{E}[\|\hat{\boldsymbol{x}}_{k,t}^{(s)}\|_2^2\|\hat{\boldsymbol{x}}_{l,t}^{(s)}\|_2 + \|\hat{e}_{k,t}^{(s)}\|_2]).
\end{aligned}
$$

By a similar argument to the proof of Lemma K.27, we have

$$
\mathbb{E}[\|\hat{\boldsymbol{x}}_{k,t}^{(s)}\|_2^2\|\hat{\boldsymbol{x}}_{l,t}^{(s)}\|_2] = \tilde{\mathcal{O}}(\eta^{1.5}),
$$
$$
\mathbb{E}[\|\hat{\boldsymbol{x}}_{k,t}^{(s)}\|_2\|\hat{\boldsymbol{x}}_{l,t}^{(s)}\|_2\|\Delta\hat{\boldsymbol{\phi}}^{(s)}\|_2] = \tilde{\mathcal{O}}(\eta^{1.5-0.5\beta}),
$$

which yields (81).

Now we proceed to prove (83) to (86). By definition of $\hat{\boldsymbol{A}}_{\text{avg}}^{(s)}$,

$$
\begin{aligned}
\hat{\boldsymbol{A}}_{\text{avg}}^{(s)} &= \frac{1}{K^2}\mathbb{E}[(\sum_{k\in[K]}\hat{\boldsymbol{x}}_{k,H}^{(s)})(\sum_{k\in[K]}\hat{\boldsymbol{x}}_{k,H}^{(s)})^\top] \\
&= \frac{1}{K^2}\sum_{k\in[K]}\mathbb{E}[\hat{\boldsymbol{x}}_{k,H}^{(s)}\hat{\boldsymbol{x}}_{k,H}^{(s)\top}] + \frac{1}{K^2}\sum_{k,l\in[K],k\neq l}\mathbb{E}[\hat{\boldsymbol{x}}_{k,H}^{(s)}\hat{\boldsymbol{x}}_{l,H}^{(s)\top}] \\
&= \frac{1}{K}\hat{\boldsymbol{A}}_H^{(s)} + (1-\frac{1}{K})\hat{\boldsymbol{M}}_H^{(s)},
\end{aligned}
$$

which demonstrates (83). Then we derive (84). By definition of $\hat{\boldsymbol{x}}_{\text{avg},0}^{(s+1)}$,

$$
\begin{aligned}
\hat{\boldsymbol{x}}_{\text{avg},0}^{(s+1)} &= \hat{\boldsymbol{\phi}}^{(s)} + \hat{\boldsymbol{x}}_{\text{avg},H}^{(s)} - \Phi(\hat{\boldsymbol{\phi}}^{(s)}+\hat{\boldsymbol{x}}_{\text{avg},H}^{(s)}) \\
&= \hat{\boldsymbol{\phi}}^{(s)} + \hat{\boldsymbol{x}}_{\text{avg},H}^{(s)} - \left(\hat{\boldsymbol{\phi}}^{(s)} + \partial\Phi(\hat{\boldsymbol{\phi}}^{(s)})\hat{\boldsymbol{x}}_{\text{avg},H}^{(s)} + \mathcal{O}(\|\hat{\boldsymbol{x}}_{\text{avg},H}^{(s)}\|_2^2)\right) \\
&= \hat{\boldsymbol{x}}_{\text{avg},H}^{(s)} - \left(\boldsymbol{P}_\| + \mathcal{O}(\|\Delta\hat{\boldsymbol{\phi}}^{(s)}\|_2)\right)\hat{\boldsymbol{x}}_{\text{avg},H}^{(s)} + \mathcal{O}(\|\hat{\boldsymbol{x}}_{\text{avg},H}^{(s)}\|_2^2) \\
&= \boldsymbol{P}_\perp\hat{\boldsymbol{x}}_{\text{avg},H}^{(s)} + \mathcal{O}(\|\hat{\boldsymbol{x}}_{\text{avg},H}^{(s)}\|_2^2 + \|\hat{\boldsymbol{x}}_{\text{avg},H}^{(s)}\|_2\|\Delta\hat{\boldsymbol{\phi}}^{(s)}\|_2). \quad (88)
\end{aligned}
$$

Hence,

$$
\begin{aligned}
\hat{\boldsymbol{M}}_0^{(s+1)} = \hat{\boldsymbol{A}}_0^{(s+1)} &= \mathbb{E}[\hat{\boldsymbol{x}}_{\text{avg},0}^{(s)}\hat{\boldsymbol{x}}_{\text{avg},0}^{(s)\top}] \\
&= \boldsymbol{P}_\perp\hat{\boldsymbol{A}}_{\text{avg}}^{(s)}\boldsymbol{P}_\perp + \mathcal{O}(\mathbb{E}[\|\hat{\boldsymbol{x}}_{\text{avg},H}^{(s)}\|_2^3 + \|\hat{\boldsymbol{x}}_{\text{avg},H}^{(s)}\|_2^2\|\Delta\hat{\boldsymbol{\phi}}^{(s)}\|_2]).
\end{aligned}
$$

By (76) and (78), we obtain (84). By (74),

$$
\hat{\boldsymbol{\phi}}^{(s+1)} - \hat{\boldsymbol{\phi}}^{(s)} = \boldsymbol{P}_\|\hat{\boldsymbol{x}}_{\text{avg},H}^{(s)} + \mathcal{O}(\|\hat{\boldsymbol{x}}_{\text{avg},H}^{(s)}\|_2\|\Delta\hat{\boldsymbol{\phi}}^{(s)}\|_2 + \|\hat{\boldsymbol{x}}_{\text{avg},H}^{(s)}\|_2^2). \quad (89)
$$

Combining (88) and (89) gives

$$
\mathbb{E}[\hat{\boldsymbol{x}}_{\text{avg},0}^{(s)}(\hat{\boldsymbol{\phi}}^{(s+1)}-\hat{\boldsymbol{\phi}}^{(s)})^\top] = \boldsymbol{P}_\perp\hat{\boldsymbol{A}}_{\text{avg}}^{(s)}\boldsymbol{P}_\| + \tilde{\mathcal{O}}(\eta^{1.5-0.5\beta}).
$$

Therefore,

$$
\begin{aligned}
\hat{\boldsymbol{B}}_0^{(s+1)} = \mathbb{E}[\hat{\boldsymbol{x}}_{\text{avg},0}^{(s+1)}\Delta\hat{\boldsymbol{\phi}}^{(s+1)\top}] &= \mathbb{E}[\hat{\boldsymbol{x}}_{\text{avg},0}^{(s+1)}(\Delta\hat{\boldsymbol{\phi}}^{(s)}+\hat{\boldsymbol{\phi}}^{(s+1)}-\hat{\boldsymbol{\phi}}^{(s)})^\top] \\
&= \boldsymbol{P}_\perp\hat{\boldsymbol{B}}_H^{(s)} + \boldsymbol{P}_\perp\hat{\boldsymbol{A}}_{\text{avg}}^{(s)}\boldsymbol{P}_\| + \tilde{\mathcal{O}}(\eta^{1.5-\beta}).
\end{aligned}
$$

Finally, we apply Lemma K.27 to derive (85).

$$
\begin{aligned}
\hat{\boldsymbol{q}}_0^{(s+1)} = \mathbb{E}[\hat{\boldsymbol{x}}_{\text{avg},0}^{(s+1)}] &= \mathbb{E}[\hat{\boldsymbol{x}}_{\text{avg},H}^{(s)} - (\hat{\boldsymbol{\phi}}^{(s+1)}-\hat{\boldsymbol{\phi}}^{(s)})] \\
&= \hat{\boldsymbol{q}}_H^{(s)} - \boldsymbol{P}_\|\hat{\boldsymbol{q}}_H^{(s)} - \partial^2\Phi(\hat{\boldsymbol{\phi}}^{(0)})[\hat{\boldsymbol{B}}_H^{(s)}] - \frac{1}{2}\partial^2\Phi(\hat{\boldsymbol{\phi}}^{(0)})[\hat{\boldsymbol{A}}_{\text{avg}}^{(s)}] + \tilde{\mathcal{O}}(\eta^{1.5-\beta}) \\
&= \boldsymbol{P}_\perp\hat{\boldsymbol{q}}_H^{(s)} - \partial^2\Phi(\hat{\boldsymbol{\phi}}^{(0)})[\hat{\boldsymbol{B}}_H^{(s)}] - \frac{1}{2}\partial^2\Phi(\hat{\boldsymbol{\phi}}^{(0)})[\hat{\boldsymbol{A}}_{\text{avg}}^{(s)}] + \tilde{\mathcal{O}}(\eta^{1.5-\beta}),
\end{aligned}
$$

which concludes the proof. $\qquad\square$

With the assumption that the hessian at $\hat{\boldsymbol{\phi}}^{(0)}$ is diagonal, we have the following corollary that formulates the recursions for each matrix element.

**Corollary K.2.** *Given* $\|\hat{\boldsymbol{\theta}}_{\mathrm{avg}}^{(0)} - \hat{\boldsymbol{\phi}}^{(0)}\|_2 = \mathcal{O}(\sqrt{\eta \log \frac{1}{\eta}})$, *for* $0 \le s < R_{\mathrm{grp}}$ *and* $0 \le t < H$, *we have the following elementwise recursions.*

$$\hat{A}_{t+1,i,j}^{(s)} = (1 - (\lambda_i + \lambda_j)\eta)\hat{A}_{t,i,j}^{(s)} + \frac{\eta^2}{B_{\mathrm{loc}}}\Sigma_{0,i,j} + \tilde{\mathcal{O}}(\eta^{2.5-0.5\beta}), \tag{90}$$

$$\hat{M}_{t+1,i,j}^{(s)} = (1 - (\lambda_i + \lambda_j)\eta)\hat{M}_{t,i,j}^{(s)} + \tilde{\mathcal{O}}(\eta^{2.5-0.5\beta}), \tag{91}$$

$$\hat{B}_{t+1,i,j}^{(s)} = (1 - \lambda_i\eta)\hat{B}_{t,i,j}^{(s)} + \tilde{\mathcal{O}}(\eta^{2.5-\beta}), \tag{92}$$

$$\hat{A}_{\mathrm{avg},i,j}^{(s)} = \frac{1}{K}(\hat{A}_{H,i,j}^{(s)} - \hat{M}_{H,i,j}^{(s)}) + \hat{M}_{H,i,j}^{(s)}, \tag{93}$$

$$\hat{M}_{0,i,j}^{(s+1)} = \hat{A}_{0,i,j}^{(s+1)} = \begin{cases} \hat{A}_{\mathrm{avg},i,j}^{(s)} + \tilde{\mathcal{O}}(\eta^{1.5-0.5\beta}), & 1 \le i \le m, 1 \le j \le m, \\ \tilde{\mathcal{O}}(\eta^{1.5-0.5\beta}), & otherwise. \end{cases} \tag{94}$$

$$\hat{B}_{0,i,j}^{(s+1)} = \begin{cases} \hat{B}_{H,i,j}^{(s)} + \hat{A}_{\mathrm{avg},,i,j}^{(s)} + \tilde{\mathcal{O}}(\eta^{1.5-\beta}), & 1 \le i \le m, m < j \le d, \\ \hat{B}_{H,i,j}^{(s)} + \tilde{\mathcal{O}}(\eta^{1.5-\beta}), & 1 \le i \le m, 1 \le j \le m, \\ \tilde{\mathcal{O}}(\eta^{1.5-\beta}), & m < i \le d. \end{cases} \tag{95}$$

Having formulated the recursions, we are ready to solve out the explicit expressions. We will split each matrix into four parts and them one by on. Specifically, a matrix $M$ can be split into $\boldsymbol{P}_{\parallel}\boldsymbol{M}\boldsymbol{P}_{\parallel}$ in the tangent space of $\Gamma$ at $\hat{\boldsymbol{\phi}}^{(0)}$, $\boldsymbol{P}_{\perp}\boldsymbol{M}\boldsymbol{P}_{\perp}$ in the normal space, along with $\boldsymbol{P}_{\parallel}\boldsymbol{M}\boldsymbol{P}_{\perp}$ and $\boldsymbol{P}_{\perp}\boldsymbol{M}\boldsymbol{P}_{\parallel}$ across both spaces.

We first compute the elements of $\boldsymbol{P}_{\perp}\hat{\boldsymbol{A}}_t^{(s)}\boldsymbol{P}_{\perp}$ and $\boldsymbol{P}_{\perp}\hat{\boldsymbol{A}}_{\mathrm{avg}}^{(s)}\boldsymbol{P}_{\perp}$.

**Lemma K.29** (General formula for $\boldsymbol{P}_{\perp}\hat{\boldsymbol{A}}_t^{(s)}\boldsymbol{P}_{\perp}$ and $\boldsymbol{P}_{\perp}\hat{\boldsymbol{A}}_{\mathrm{avg}}^{(s)}\boldsymbol{P}_{\perp}$). *Let* $R_0 := \lceil \frac{10}{\lambda_m\alpha}\log\frac{1}{\eta}\rceil$. *Then for* $1 \le i \le m, 1 \le j \le m$ *and* $R_0 \le s < R_{\mathrm{grp}}$,

$$\hat{A}_{\mathrm{avg},i,j}^{(s)} = \frac{1}{(\lambda_i + \lambda_j)KB_{\mathrm{loc}}}\eta\Sigma_{0,i,j} + \tilde{\mathcal{O}}(\eta^{1.5-0.5\beta}),$$

$$\hat{A}_{t,i,j}^{(s)} = -\left(1 - \frac{1}{K}\right)\frac{(1 - (\lambda_i + \lambda_j)\eta)^t}{(\lambda_i + \lambda_j)B_{\mathrm{loc}}}\eta\Sigma_{0,i,j} + \frac{\eta}{(\lambda_i + \lambda_j)B_{\mathrm{loc}}}\Sigma_{0,i,j} + \tilde{\mathcal{O}}(\eta^{1.5-0.5\beta}).$$

*For* $s < R_0$, $\hat{A}_{t,i,j}^{(s)} = \tilde{\mathcal{O}}(\eta)$ *and* $\hat{A}_{\mathrm{avg},,i,j}^{(s)} = \tilde{\mathcal{O}}(\eta)$.

*Proof.* For $1 \le i \le m, 1 \le j \le m$, $\lambda_i > 0, \lambda_j > 0$. By (90),

$$\hat{A}_{t,i,j}^{(s)} = (1 - (\lambda_i + \lambda_j)\eta)^t\hat{A}_{0,i,j}^{(s)} + \sum_{\tau=0}^{t-1}(1 - (\lambda_i + \lambda_j)\eta)^\tau\frac{\eta^2}{B_{\mathrm{loc}}}\Sigma_{0,i,j}$$

$$+ \tilde{\mathcal{O}}(\sum_{\tau=0}^{t-1}(1 - (\lambda_i + \lambda_j)\eta)^\tau\eta^{2.5-0.5\beta})$$

$$= (1 - (\lambda_i + \lambda_j)\eta)^t\hat{A}_{0,i,j}^{(s)} + \frac{1 - (1 - (\lambda_i + \lambda_j)\eta)^t}{(\lambda_i + \lambda_j)B_{\mathrm{loc}}}\eta\Sigma_{0,i,j} + \tilde{\mathcal{O}}(\eta^{1.5-0.5\beta}),$$

where the second inequality uses $\sum_{\tau=0}^{t-1}(1 - (\lambda_i + \lambda_j)\eta)^\tau = \frac{1-(1-(\lambda_i+\lambda_j)\eta)^t}{(\lambda_i+\lambda_j)\eta} \le \frac{1}{(\lambda_i+\lambda_j)\eta}$. By (91),

$$\hat{M}_{t,i,j}^{(s)} = (1 - (\lambda_i + \lambda_j)\eta)^t\hat{M}_{0,i,j}^{(s)} + \tilde{\mathcal{O}}(\sum_{\tau=0}^{t-1}(1 - (\lambda_i + \lambda_j)\eta)^\tau\eta^{2.5-0.5\beta})$$

$$= (1 - (\lambda_i + \lambda_j)\eta)^t\hat{A}_{0,i,j}^{(s)} + \tilde{\mathcal{O}}(\eta^{1.5-0.5\beta}),$$

where the second equality uses $M_0^{(s+1)} = A_0^{(s+1)}$. By (93) and (94),

$$\hat{A}_{\text{avg},i,j}^{(s)} = \frac{1 - (1 - (\lambda_i + \lambda_j)\eta)^H}{(\lambda_i + \lambda_j)KB_{\text{loc}}}\eta\Sigma_{0,i,j} + (1 - (\lambda_i + \lambda_j)\eta)^H \hat{A}_{0,i,j}^{(s)} + \tilde{\mathcal{O}}(\eta^{1.5-0.5\beta}),$$

$$\hat{A}_{0,i,j}^{(s+1)} = \hat{A}_{\text{avg},i,j}^{(s)} + \tilde{\mathcal{O}}(\eta^{2.5-0.5\beta})$$

$$= \frac{1 - (1 - (\lambda_i + \lambda_j)\eta)^H}{(\lambda_i + \lambda_j)KB_{\text{loc}}}\eta\Sigma_{0,i,j} + (1 - (\lambda_i + \lambda_j)\eta)^H \hat{A}_{0,i,j}^{(s)} + \tilde{\mathcal{O}}(\eta^{1.5-0.5\beta}).$$

Then we obtain

$$\hat{A}_{0,i,j}^{(s)} = (1 - (\lambda_i + \lambda_j)\eta)^{sH} \hat{A}_{0,i,j}^{(0)} + \frac{1 - (1 - (\lambda_i + \lambda_j)\eta)^H}{(\lambda_i + \lambda_j)KB_{\text{loc}}}\eta\Sigma_{0,i,j} \sum_{r=0}^{s-1}(1 - (\lambda_i + \lambda_j)\eta)^{rH}$$

$$+ \tilde{\mathcal{O}}(\eta^{1.5-0.5\beta} \sum_{r=R_0}^{s-1}(1 - (\lambda_i + \lambda_j)\eta)^{rH}).$$

Notice that $|1 - (\lambda_i + \lambda_j)\eta| < 1$ and

$$(1 - (\lambda_i + \lambda_j)\eta)^H \le \exp(-(\lambda_i + \lambda_j)\eta H) = \exp(-(\lambda_i + \lambda_j)\alpha). \tag{96}$$

Therefore,

$$\sum_{r=0}^{s-1}(1 - (\lambda_i + \lambda_j)\eta)^{rH} = \frac{1 - (1 - (\lambda_i + \lambda_j)\eta)^{rH}}{1 - (1 - (\lambda_i + \lambda_j)\eta)^H} \le \frac{1}{1 - \exp(-(\lambda_i + \lambda_j)\alpha)}.$$

Then we have

$$\hat{A}_{0,i,j}^{(s)} = (1 - (\lambda_i + \lambda_j)\eta)^{sH} \hat{A}_{0,i,j}^{(0)} + \frac{1 - (1 - (\lambda_i + \lambda_j)\eta)^{sH}}{(\lambda_i + \lambda_j)KB_{\text{loc}}}\eta\Sigma_{0,i,j} + \tilde{\mathcal{O}}(\eta^{1.5-0.5\beta}).$$

Finally, we demonstrate that for $s \ge R_0$, $\hat{A}_{0,i,j}^{(s)}$ and $\hat{A}_{\text{avg},i,j}^{(s)}$ is approximately equal to $\frac{\eta}{(\lambda_i+\lambda_j)KB_{\text{loc}}}\Sigma_{0,i,j}$. By (96), when $s \ge R_0$, $(1 - (\lambda_i + \lambda_j)\eta)^{sH} = \mathcal{O}(\eta^{10})$, which gives

$$\hat{A}_{\text{avg},i,j}^{(s)} = \frac{1}{(\lambda_i + \lambda_j)KB_{\text{loc}}}\eta\Sigma_{0,i,j} + \tilde{\mathcal{O}}(\eta^{1.5-0.5\beta}),$$

$$A_{t,i,j}^{(s)} = -\left(1 - \frac{1}{K}\right)\frac{(1 - (\lambda_i + \lambda_j)\eta)^t}{(\lambda_i + \lambda_j)B_{\text{loc}}}\eta\Sigma_{0,i,j} + \frac{\eta}{(\lambda_i + \lambda_j)B_{\text{loc}}}\Sigma_{0,i,j} + \tilde{\mathcal{O}}(\eta^{1.5-0.5\beta}).$$

For $s < R_0$, since $\hat{A}_0^{(0)} = \hat{x}_{\text{avg},0}^{(s)}\hat{x}_{\text{avg},0}^{(s)\top} = \tilde{\mathcal{O}}(\eta)$, we have $\hat{A}_{\text{avg},,i,j}^{(s)} = \tilde{\mathcal{O}}(\eta)$ and $\hat{A}_{t,i,j}^{(s)} = \tilde{\mathcal{O}}(\eta)$. $\qquad\square$

Secondly, we compute $P_\parallel \hat{A}_t^{(s)} P_\perp$ and $P_\parallel \hat{A}_{\text{avg}}^{(s)} P_\perp$.

**Lemma K.30** (General formula for $P_\perp \hat{A}_t^{(s)} P_\parallel$ and $P_\perp \hat{A}_{\text{avg}}^{(s)} P_\parallel$). *For $1 \le i \le m, m < j \le d$,*

$$\hat{A}_{t,i,j}^{(s)} = \frac{1 - (1 - \lambda_i\eta)^t}{\lambda_i B_{\text{loc}}}\eta\Sigma_{0,i,j} + \tilde{\mathcal{O}}(\eta^{1.5-0.5\beta}),$$

$$\hat{A}_{\text{avg},i,j}^{(s)} = \frac{1 - (1 - \lambda_i\eta)^H}{\lambda_i KB_{\text{loc}}}\eta\Sigma_{0,i,j} + \tilde{\mathcal{O}}(\eta^{1.5-0.5\beta}).$$

*Proof.* Note that for $1 \le i \le m, m < j \le d$ and $\lambda_i > 0, \lambda_j = 0$. By (90) and (94),

$$\hat{A}_{t,i,j}^{(s)} = (1 - \lambda_i\eta)^t \hat{A}_{0,i,j}^{(s)} + \frac{1 - (1 - \lambda_i\eta)^t}{\lambda_i B_{\text{loc}}}\eta\Sigma_{0,i,j} + \tilde{\mathcal{O}}(\eta^{1.5-0.5\beta})$$

$$= \frac{1 - (1 - \lambda_i\eta)^t}{\lambda_i B_{\text{loc}}}\eta\Sigma_{0,i,j} + \tilde{\mathcal{O}}(\eta^{1.5-\beta}).$$

By (91) and (94), $\hat{M}_{t,i,j}^{(s)} = \tilde{\mathcal{O}}(\eta^{1.5-0.5\beta})$. Then,

$$\hat{A}_{\text{avg},i,j}^{(s)} = \frac{1 - (1 - \lambda_i\eta)^H}{\lambda_i KB_{\text{loc}}}\eta\Sigma_{0,i,j} + \tilde{\mathcal{O}}(\eta^{1.5-0.5\beta}).$$

$\qquad\square$

Similar to Lemma K.30, we have the following lemma for the general formula of $P_\parallel \hat{A}_t^{(s)} P_\perp$ and $P_\parallel \hat{A}_{\text{avg}}^{(s)} P_\perp$.

**Lemma K.31** (General formula for $P_\parallel \hat{A}_t^{(s)} P_\perp$ and $P_\parallel \hat{A}_{\text{avg}}^{(s)} P_\perp$). *For $m < i \leq d$ and $1 \leq j \leq m$,*

$$\hat{A}_{t,i,j}^{(s)} = \frac{1 - (1 - \lambda_j \eta)^t}{\lambda_j B_{\text{loc}}} \eta \Sigma_{0,i,j} + \tilde{\mathcal{O}}(\eta^{1.5-0.5\beta}),$$

$$\hat{A}_{\text{avg},i,j}^{(s)} = \frac{1 - (1 - \lambda_j \eta)^H}{\lambda_j K B_{\text{loc}}} \eta \Sigma_{0,i,j} + \tilde{\mathcal{O}}(\eta^{1.5-0.5\beta}).$$

Finally, we derive the general formula for $P_\parallel \hat{A}_t^{(s)} P_\parallel$ and $P_\parallel \hat{A}_{\text{avg}}^{(s)} P_\parallel$.

**Lemma K.32** (General formula for $P_\parallel \hat{A}_t^{(s)} P_\parallel$ and $P_\parallel \hat{A}_{\text{avg}}^{(s)} P_\parallel$). *For $m < i \leq d$ and $m < j \leq d$,*

$$\hat{A}_{\text{avg},i,j}^{(s)} = \frac{H \eta^2}{K B_{\text{loc}}} \Sigma_{0,i,j} + \tilde{\mathcal{O}}(\eta^{1.5-0.5\beta}),$$

$$\hat{A}_{t,i,j}^{(s)} = \hat{A}_{0,i,j}^{(s)} + \frac{t \eta^2}{B_{\text{loc}}} \Sigma_{0,i,j} + \tilde{\mathcal{O}}(\eta^{1.5-0.5\beta}).$$

*Proof.* Note that for $m < i \leq d$, $m < j \leq d$ and $\lambda_i = \lambda_j = 0$. (90) is then simplified as

$$\hat{A}_{t+1,i,j}^{(s)} = \hat{A}_{t,i,j}^{(s)} + \frac{\eta^2}{B_{\text{loc}}} \Sigma_{0,i,j} + \tilde{\mathcal{O}}(\eta^{2.5-0.5\beta}).$$

Therefore,

$$\hat{A}_{t,i,j}^{(s)} = \hat{A}_{0,i,j}^{(s)} + \frac{t \eta^2}{B_{\text{loc}}} \Sigma_{0,i,j} + \tilde{\mathcal{O}}(\eta^{1.5-0.5\beta}). \tag{97}$$

According to (91), $\hat{M}_{t,i,j}^{(s)} = \tilde{\mathcal{O}}(\eta^{1.5-0.5\beta})$ for $m < i \leq d$ and $m < j \leq d$. Combining (91), (94) and (97) yields

$$\hat{A}_{\text{avg},i,j}^{(s)} = \frac{H \eta^2}{K B_{\text{loc}}} \Sigma_{0,i,j} + \tilde{\mathcal{O}}(\eta^{1.5-0.5\beta}).$$

$\square$

Now, we move on to compute the general formula for $\hat{B}_t^{(s)}$.

**Lemma K.33** (The general formula for $P_\perp \hat{B}_t^{(s)} P_\parallel$). *Note that for $1 \leq i \leq m$ and $m < j \leq d$, when $R_0 := \lceil \frac{10}{\lambda_m \alpha} \log \frac{1}{\eta} \rceil \leq s < R_{\text{grp}}$,*

$$\hat{B}_{t,i,j}^{(s)} = \frac{(1 - \lambda_i \eta)^t}{\lambda_i K B_{\text{loc}}} \eta \Sigma_{0,i,j} + \tilde{\mathcal{O}}(\eta^{1.5-\beta}).$$

*For $s < R_0$, $\hat{B}_{t,i,j}^{(s)} = \tilde{\mathcal{O}}(\eta)$.*

*Proof.* Note that for $1 \leq i \leq m$, $\lambda_i > 0$. By (92),

$$\hat{B}_{t+1,i,j}^{(s)} = (1 - \lambda_i \eta) \hat{B}_{t,i,j}^{(s)} + \tilde{\mathcal{O}}(\eta^{2.5-\beta}).$$

Hence,

$$\hat{B}_{t,i,j}^{(s)} = (1 - \lambda_i \eta)^t \hat{B}_{0,i,j}^{(s)} + \tilde{\mathcal{O}}(\eta^{1.5-\beta}).$$

According to (95),

$$\hat{B}_{0,i,j}^{(s+1)} = \hat{B}_{H,i,j}^{(s)} + \hat{A}_{\text{avg},,i,j}^{(s)} + \tilde{\mathcal{O}}(\eta^{2.5-\beta})$$

$$= (1 - \lambda_i \eta)^H \hat{B}_{0,i,j}^{(s)} + \hat{A}_{\text{avg},i,j}^{(s)} + \tilde{\mathcal{O}}(\eta^{1.5-\beta}).$$

Then we have

$$\hat{B}_{0,i,j}^{(s)} = (1 - \lambda_i\eta)^{sH}\hat{B}_{0,i,j}^{(0)} + \hat{A}_{\text{avg},i,j}^{(s)}\sum_{r=0}^{s-1}(1 - \lambda_i\eta)^{rH} + \tilde{\mathcal{O}}(\sum_{r=0}^{s-1}(1 - \lambda_i\eta)^{rH}\eta^{1.5-\beta})$$

$$= (1 - \lambda_i\eta)^{sH}\hat{B}_{0,i,j}^{(0)} + \frac{1 - (1 - \lambda_i\eta)^{sH}}{1 - (1 - \lambda_i\eta)^H}\hat{A}_{\text{avg},,i,j}^{(s)} + \tilde{\mathcal{O}}(\eta^{1.5-\beta})$$

$$= \frac{1 - (1 - \lambda_i\eta)^{sH}}{1 - (1 - \lambda_i\eta)^H}\hat{A}_{\text{avg},,i,j}^{(s)} + \tilde{\mathcal{O}}(\eta^{1.5-\beta}).$$

where the second equality uses (96) and the last inequality uses $\hat{B}_0^{(0)} = \hat{x}_{\text{avg},0}^{(0)}\Delta\hat{\phi}^{(0)} = \mathbf{0}$. For $s \geq R_0$, $\hat{A}_{\text{avg},i,j}^{(s)} = \frac{1-(1-\lambda_i\eta)^H}{\lambda_i KB_{\text{loc}}}\eta\Sigma_{0,i,j} + \tilde{\mathcal{O}}(\eta^{1.5-0.5\beta})$, which gives

$$\hat{B}_{0,i,j}^{(s)} = \frac{\eta}{\lambda_i KB_{\text{loc}}}\Sigma_{0,i,j} + \tilde{\mathcal{O}}(\eta^{1.5-\beta}).$$

Therefore,

$$\hat{B}_{t,i,j}^{(s)} = \frac{(1 - \lambda_i\eta)^t}{\lambda_i KB_{\text{loc}}}\eta\Sigma_{0,i,j} + \tilde{\mathcal{O}}(\eta^{1.5-\beta}).$$

For $s < R_0$, $\hat{A}_{\text{avg},,i,j}^{(s)} = \tilde{\mathcal{O}}(\eta)$ and therefore, $\hat{B}_{t,i,j}^{(s)} = \tilde{\mathcal{O}}(\eta)$. $\qquad\square$

**Lemma K.34** (General formula for the elements of $\boldsymbol{P}_\perp\hat{\boldsymbol{B}}_t^{(s)}\boldsymbol{P}_\perp$ ). *For $1 \leq i \leq m$ and $1 \leq j \leq m$, , $\hat{B}_{t,i,j}^{(s)} = \tilde{\mathcal{O}}(\eta^{1.5-\beta})$.*

*Proof.* Note that for $1 \leq i \leq m$, $\lambda_i > 0$. By (92),

$$\hat{B}_{t+1,i,j}^{(s)} = (1 - \lambda_i\eta)\hat{B}_{t,i,j}^{(s)} + \tilde{\mathcal{O}}(\eta^{2.5-\beta}).$$

Hence,

$$\hat{B}_{t,i,j}^{(s)} = (1 - \lambda_i\eta)^t\hat{B}_{0,i,j}^{(s)} + \tilde{\mathcal{O}}(\eta^{1.5-\beta}).$$

By (95),

$$\hat{B}_{0,i,j}^{(s+1)} = \hat{B}_{H,i,j}^{(s)} + \tilde{\mathcal{O}}(\eta^{2.5-\beta})$$

$$= (1 - \lambda_i\eta)^H\hat{B}_{0,i,j}^{(s)} + \tilde{\mathcal{O}}(\eta^{1.5-\beta})$$

$$= (1 - \lambda_i\eta)^{sH}\hat{B}_{0,i,j}^{(0)} + \tilde{\mathcal{O}}(\sum_{r=0}^{s-1}(1 - \lambda_i\eta)^{rH}\eta^{1.5-\beta})$$

$$= (1 - \lambda_i\eta)^{sH}\hat{B}_{0,i,j}^{(0)} + \tilde{\mathcal{O}}(\eta^{1.5-\beta})$$

$$= \tilde{\mathcal{O}}(\eta^{1.5-\beta}),$$

where the last inequality uses $\hat{B}_0^{(0)} = \mathbf{0}$. $\qquad\square$

**Lemma K.35** (General formula for $\boldsymbol{P}_\parallel\hat{\boldsymbol{B}}_t^{(s)}$). *For $m < i \leq d$, $\hat{B}_{t,i,j}^{(s)} = \tilde{\mathcal{O}}(\eta^{1.5-\beta})$.*

*Proof.* Note that $\lambda_i = 0$ for $m < i \leq d$. By (92) and (95),

$$\hat{B}_{t+1}^{(s)} = \hat{B}_t^{(s)} + \tilde{\mathcal{O}}(\eta^{2.5-\beta}), \quad \hat{B}_0^{(s)} = \tilde{\mathcal{O}}(\eta^{2.5-\beta}).$$

Therefore,

$$\hat{B}_t^{(s)} = t\tilde{\mathcal{O}}(\eta^{2.5-\beta}) + \hat{B}_0^{(s)} = \tilde{\mathcal{O}}(\eta^{1.5-\beta}).$$

$\qquad\square$

Having obtained the expressions for $\hat{\boldsymbol{B}}_t^{(s)}$, $\hat{\boldsymbol{A}}_t^{(s)}$ and $\hat{\boldsymbol{A}}_{\text{avg}}^{(s)}$, we now provide explicit expressions for the first and second moments of the change of manifold projection every round in the following two lemmas.

**Lemma K.36.** *The expectation of the change of manifold projection every round is*

$$\mathbb{E}[\hat{\boldsymbol{\phi}}^{(s+1)} - \hat{\boldsymbol{\phi}}^{(s)}] = \begin{cases} \frac{H\eta^2}{2B}\partial^2\Phi(\hat{\boldsymbol{\phi}}^{(0)})[\boldsymbol{\Sigma}_0 + \boldsymbol{\Psi}(\hat{\boldsymbol{\phi}}^{(0)})] + \tilde{\mathcal{O}}(\eta^{1.5-\beta}), & R_0 < s < R_{\text{grp}} \\ \tilde{\mathcal{O}}(\eta), & s \leq R_0 \end{cases}, \quad (98)$$

*where* $R_0 := \lceil\frac{10}{\lambda_m\alpha}\log\frac{1}{\eta}\rceil$.

*Proof.* We first compute $\mathbb{E}[\hat{\boldsymbol{\phi}}^{(s+1)} - \hat{\boldsymbol{\phi}}^{(s)}]$. By (72), we only need to compute $\boldsymbol{P}_\parallel\hat{\boldsymbol{q}}_H^{(s)}$ by relating it to these matrices. Multiplying both sides of (79) by $\boldsymbol{P}_\parallel$ gives

$$\boldsymbol{P}_\parallel\hat{\boldsymbol{q}}_{t+1}^{(s)} = \boldsymbol{P}_\parallel\hat{\boldsymbol{q}}_t^{(s)} - \eta\boldsymbol{P}_\parallel\nabla^3\mathcal{L}(\hat{\boldsymbol{\phi}}^{(0)})[\hat{\boldsymbol{B}}_t^{(s)}] - \frac{\eta}{2}\boldsymbol{P}_\parallel\nabla^3\mathcal{L}(\hat{\boldsymbol{\phi}}^{(0)})[\hat{\boldsymbol{A}}_t^{(s)}] + \tilde{\mathcal{O}}(\eta^{2.5-\beta}). \quad (99)$$

Similarly, according to (85), we have

$$\boldsymbol{P}_\parallel\hat{\boldsymbol{q}}_0^{(s+1)} = -\boldsymbol{P}_\parallel\partial^2\Phi(\hat{\boldsymbol{\phi}}^{(0)})[\hat{\boldsymbol{B}}_H^{(s)}] - \frac{1}{2}\boldsymbol{P}_\parallel\partial^2\Phi(\hat{\boldsymbol{\phi}}^{(0)})[\hat{\boldsymbol{A}}_{\text{avg}}^{(s)}] + \tilde{\mathcal{O}}(\eta^{1.5-\beta}). \quad (100)$$

Combining (99) and (100) yields

$$\boldsymbol{P}_\parallel\hat{\boldsymbol{q}}_H^{(s)} = -\frac{1}{2}\boldsymbol{P}_\parallel\partial^2\Phi(\hat{\boldsymbol{\phi}}^{(0)})[\hat{\boldsymbol{A}}_{\text{avg}}^{(s-1)}] - \frac{\eta}{2}\boldsymbol{P}_\parallel\nabla^3\mathcal{L}(\hat{\boldsymbol{\phi}}^{(0)})[\sum_{t=0}^{H-1}\hat{\boldsymbol{A}}_t^{(s)}]$$
$$- \eta\boldsymbol{P}_\parallel\nabla^3\mathcal{L}(\hat{\boldsymbol{\phi}}^{(0)})[\sum_{t=0}^{H-1}\hat{\boldsymbol{B}}_t^{(s)}] - \boldsymbol{P}_\parallel\partial^2\Phi(\hat{\boldsymbol{\phi}}^{(0)})[\hat{\boldsymbol{B}}_H^{(s-1)}] + \tilde{\mathcal{O}}(\eta^{1.5-\beta}). \quad (101)$$

By Lemmas K.29, K.32 and K.30, for $s \leq R_0 = \lfloor\frac{10}{\lambda\alpha}\log\frac{1}{\eta}\rfloor$, $\hat{\boldsymbol{A}}_t^{(s)} = \tilde{\mathcal{O}}(\eta)$, $\hat{\boldsymbol{A}}_{\text{avg}}^{(s)} = \tilde{\mathcal{O}}(\eta)$ and $\hat{\boldsymbol{B}}_t^{(s)} = \tilde{\mathcal{O}}(\eta)$. Therefore, $\mathbb{E}[\hat{\boldsymbol{\phi}}^{(s+1)} - \hat{\boldsymbol{\phi}}^{(s)}] = \tilde{\mathcal{O}}(\eta)$. For $s > R_0$, $\hat{\boldsymbol{A}}_{\text{avg}}^{(s-1)} = \hat{\boldsymbol{A}}_{\text{avg}}^{(s)} + \tilde{\mathcal{O}}(\eta^{1.5-0.5\beta})$. Substituting (101) into (72) gives

$$\mathbb{E}[\hat{\boldsymbol{\phi}}^{(s+1)} - \hat{\boldsymbol{\phi}}^{(s)}] = \underbrace{\frac{1}{2}\boldsymbol{P}_\perp\partial^2\Phi(\hat{\boldsymbol{\phi}}^{(0)})[\hat{\boldsymbol{A}}_{\text{avg}}^{(s)}] + \boldsymbol{P}_\perp\partial^2\Phi(\hat{\boldsymbol{\phi}}^{(0)})[\hat{\boldsymbol{B}}_H^{(s)}]}_{\mathcal{T}_1}$$
$$\underbrace{-\eta\boldsymbol{P}_\parallel\nabla^3\mathcal{L}(\hat{\boldsymbol{\phi}}^{(0)})[\underbrace{\frac{1}{2}\sum_{t=0}^{H-1}\hat{\boldsymbol{A}}_t^{(s)} + \sum_{t=0}^{H-1}\hat{\boldsymbol{B}}_t^{(s)}}_{\mathcal{T}_3}] + \tilde{\mathcal{O}}(\eta^{1.5-\beta})}_{\mathcal{T}_2}.$$

Below we compute $\mathcal{T}_1$ and $\mathcal{T}_2$ for $s > R_0$ respectively. By Lemma K.3,

$$\boldsymbol{P}_\perp\partial^2\Phi(\hat{\boldsymbol{\phi}}^{(0)})[\boldsymbol{P}_\perp\hat{\boldsymbol{A}}_{\text{avg}}^{(s)}\boldsymbol{P}_\parallel] = \boldsymbol{P}_\perp\partial^2\Phi(\hat{\boldsymbol{\phi}}^{(0)})[\boldsymbol{P}_\parallel\hat{\boldsymbol{A}}_{\text{avg}}^{(s)}\boldsymbol{P}_\perp] = \boldsymbol{0},$$
$$\boldsymbol{P}_\perp\partial^2\Phi(\hat{\boldsymbol{\phi}}^{(0)})[\boldsymbol{P}_\parallel\hat{\boldsymbol{A}}_{\text{avg}}^{(s)}\boldsymbol{P}_\parallel] = \partial^2\Phi(\hat{\boldsymbol{\phi}}^{(0)})[\boldsymbol{P}_\parallel\hat{\boldsymbol{A}}_{\text{avg}}^{(s)}\boldsymbol{P}_\parallel].$$

By Lemma K.4,

$$\boldsymbol{P}_\perp\partial^2\Phi(\hat{\boldsymbol{\phi}}^{(0)})[\boldsymbol{P}_\perp\hat{\boldsymbol{A}}_{\text{avg}}^{(s)}\boldsymbol{P}_\perp] = \boldsymbol{0}.$$

Therefore, for $s > R_0$,

$$\boldsymbol{P}_\perp\partial^2\Phi(\hat{\boldsymbol{\phi}}^{(0)})[\hat{\boldsymbol{A}}_{\text{avg}}^{(s)}] = \frac{H\eta^2}{2KB_{\text{loc}}}\partial^2\Phi(\hat{\boldsymbol{\phi}}^{(0)})\Phi[\boldsymbol{\Sigma}_{0,\parallel}] + \tilde{\mathcal{O}}(\eta^{1.5-0.5\beta}),$$

where we apply Lemma K.32. Similarly, for $s > R_0$,

$$\boldsymbol{P}_\perp\partial^2\Phi(\hat{\boldsymbol{\phi}}^{(0)})[\hat{\boldsymbol{B}}_H^{(s)}] = \partial^2\Phi(\hat{\boldsymbol{\phi}}^{(0)})[\boldsymbol{P}_\parallel\hat{\boldsymbol{B}}_H^{(s)}\boldsymbol{P}_\parallel] = \tilde{\mathcal{O}}(\eta^{1.5-\beta}),$$

where we apply Lemma K.35. Hence,

$$\mathcal{T}_1 = \frac{H\eta^2}{2B}\partial^2\Phi(\hat{\boldsymbol{\phi}}^{(0)})[\boldsymbol{\Sigma}_{0,\parallel}] + \tilde{\mathcal{O}}(\eta^{1.5-\beta}). \quad (102)$$

We move on to show that

$$\mathcal{T}_2 = \frac{H\eta^2}{2B}\partial^2\Phi(\hat{\phi}^{(0)})[\boldsymbol{\Sigma}_0 - \boldsymbol{\Sigma}_{0,\|} + (K-1)\boldsymbol{\Psi}(\hat{\phi}^{(0)})]. \tag{103}$$

Similar to the way we compute $\hat{\boldsymbol{A}}_t^{(s)}$, $\hat{\boldsymbol{A}}_{\mathrm{avg}}^{(s)}$ and $\hat{\boldsymbol{B}}_t^{(s)}$, we compute $\mathcal{T}_2$ by splitting $\mathcal{T}_3$ into four matrices and then substituting them into the linear operator $-\eta\boldsymbol{P}_\|\nabla^3\mathcal{L}(\hat{\phi}^{(0)})[\cdot]$ one by one. First, we show that

$$\begin{aligned}
-\eta\boldsymbol{P}_\|\nabla^3\mathcal{L}(\hat{\phi}^{(0)})[\boldsymbol{P}_\perp\mathcal{T}_3\boldsymbol{P}_\perp] &= \frac{H\eta^2}{2B}\partial^2\Phi(\hat{\phi}^{(0)})[\boldsymbol{\Sigma}_{0,\perp} + (K-1)\psi(\boldsymbol{\Sigma}_{0,\perp})] \\
&\quad + \tilde{\mathcal{O}}(\eta^{1.5-\beta}),
\end{aligned} \tag{104}$$

where $\psi(\cdot)$ is interpreted as an *elementwise* matrix function here. By Lemmas K.29 and K.34, for $1 \le i \le m$, $1 \le j \le m$ and $s > R_0$,

$$\begin{aligned}
\hat{A}_{t,i,j}^{(s)} &= -\left(1 - \frac{1}{K}\right)\frac{(1 - (\lambda_i + \lambda_j)\eta)^t}{(\lambda_i + \lambda_j)B_{\mathrm{loc}}}\eta\Sigma_{0,i,j} + \frac{\eta}{(\lambda_i + \lambda_j)B_{\mathrm{loc}}}\Sigma_{0,i,j} + \tilde{\mathcal{O}}(\eta^{1.5-0.5\beta}), \\
\hat{B}_{t,i,j}^{(s)} &= \tilde{\mathcal{O}}(\eta^{1.5-\beta}).
\end{aligned}$$

Therefore,

$$\begin{aligned}
\sum_{t=0}^{H-1}\hat{A}_{t,i,j}^{(s)} &= -\left(1 - \frac{1}{K}\right)\frac{1 - (1 - (\lambda_i + \lambda_j)\eta)^H}{(\lambda_i + \lambda_j)^2 B_{\mathrm{loc}}}\Sigma_{0,i,j} + \frac{H\eta}{(\lambda_i + \lambda_j)B_{\mathrm{loc}}}\Sigma_{0.,i,j} + \tilde{\mathcal{O}}(\eta^{0.5-\beta}) \\
&= \frac{H\eta}{K(\lambda_i + \lambda_j)B_{\mathrm{loc}}}\Sigma_{0.,i,j} \\
&\quad + \left(1 - \frac{1}{K}\right)\frac{H\eta}{(\lambda_i + \lambda_j)B_{\mathrm{loc}}}\underbrace{\left[1 - \frac{1 - (1 - (\lambda_i + \lambda_j)\eta)^H}{H\eta(\lambda_i + \lambda_j)}\right]}_{\mathcal{T}_4}\Sigma_{0,i,j} + \tilde{\mathcal{O}}(\eta^{0.5-\beta}).
\end{aligned}$$

$$\sum_{t=0}^{H-1}\hat{B}_{t,i,j}^{(s)} = \tilde{\mathcal{O}}(\eta^{0.5-\beta}),$$

Then we simplify $\mathcal{T}_4$. Notice that

$$\begin{aligned}
(1 - (\lambda_i + \lambda_i)\eta)^H &= \exp(-H(\lambda_i + \lambda_j)\eta)[1 + \mathcal{O}(H\eta^2)] \\
&= \exp(-H(\lambda_i + \lambda_j)\eta) + \mathcal{O}(\eta).
\end{aligned}$$

Therefore,

$$\mathcal{T}_4 = \psi((\lambda_i + \lambda_j)H\eta) + \mathcal{O}(\eta).$$

Substituting $\mathcal{T}_4$ back into the expression for $\sum_{t=0}^{H-1}\hat{A}_{t,i,j}^{(s)}$ gives

$$\sum_{t=0}^{H-1}\hat{A}_{t,i,j}^{(s)} = \frac{H\eta}{K(\lambda_i + \lambda_j)B_{\mathrm{loc}}}\Sigma_{0.,i,j} + \left(1 - \frac{1}{K}\right)\frac{H\eta\psi((\lambda_i + \lambda_j)H\eta)}{(\lambda_i + \lambda_j)B_{\mathrm{loc}}}\Sigma_{0,i,j} + \tilde{\mathcal{O}}(\eta^{0.5-\beta}).$$

Combining the elementwise results, we obtain the following matrix form expression:

$$\begin{aligned}
-\eta\boldsymbol{P}_\|\nabla^3\mathcal{L}(\hat{\phi}^{(0)})[\boldsymbol{P}_\perp\mathcal{T}_3\boldsymbol{P}_\perp] &= -\frac{H\eta^2}{2B}\boldsymbol{P}_\|\nabla^3\mathcal{L}(\hat{\phi}^{(0)})[\mathcal{V}_{\boldsymbol{H}_0}(\boldsymbol{\Sigma}_{0,\perp} + (K-1)\psi(\boldsymbol{\Sigma}_{0,\perp}))] \\
&\quad + \tilde{\mathcal{O}}(\eta^{1.5-\beta}).
\end{aligned}$$

By Lemma K.4, we have (104).

Secondly, we show that for $s > R_0$,

$$\begin{aligned}
-\eta\boldsymbol{P}_\|\nabla^3\mathcal{L}(\hat{\phi}^{(0)})&[\boldsymbol{P}_\perp\mathcal{T}_3\boldsymbol{P}_\| + \boldsymbol{P}_\|\mathcal{T}_3\boldsymbol{P}_\perp] \\
&= \frac{H\eta^2}{B}\partial^2\Phi(\hat{\phi}^{(0)})[\boldsymbol{\Sigma}_{0,\perp,\|} + (K-1)\psi(\boldsymbol{\Sigma}_{0,\perp,\|})] + \tilde{\mathcal{O}}(\eta^{1.5-\beta}),
\end{aligned} \tag{105}$$

where $\psi(\cdot)$ is interpreted as an *elementwise* matrix function here. By symmetry of $\hat{A}_t^{(s)}$'s and $\nabla^3 \mathcal{L}(\hat{\phi}^{(0)})$,

$$\frac{1}{2}\nabla^3\mathcal{L}(\hat{\phi}^{(0)})\left[\sum_{t=0}^{H-1}\boldsymbol{P}_\perp\hat{A}_t^{(s)}\boldsymbol{P}_\parallel + \sum_{t=0}^{H-1}\boldsymbol{P}_\parallel\hat{A}_t^{(s)}\boldsymbol{P}_\perp\right] = \nabla^3\mathcal{L}(\hat{\phi}^{(0)})\left[\sum_{t=0}^{H-1}\boldsymbol{P}_\perp\hat{A}_t^{(s)}\boldsymbol{P}_\parallel\right].$$

Therefore, we only have to evaluate

$$\nabla^3\mathcal{L}(\hat{\phi}^{(0)})\left[\sum_{t=0}^{H-1}\boldsymbol{P}_\perp(\hat{A}_t^{(s)}+\hat{B}_t^{(s)})\boldsymbol{P}_\parallel + \sum_{t=0}^{H-1}\boldsymbol{P}_\parallel\hat{B}_t^{(s)}\boldsymbol{P}_\perp\right].$$

To compute the elements of $\sum_{t=0}^{H-1}\boldsymbol{P}_\perp(\hat{A}_t^{(s)}+\hat{B}_t^{(s)})\boldsymbol{P}_\parallel$, we combine Lemmas K.30 and K.33 to obtain that for $1 \le i \le m$ and $m < j \le d$,

$$\begin{aligned}
\sum_{t=0}^{H-1}\hat{A}_{t,i,j}^{(s)} &= \sum_{t=0}^{H-1}\frac{1-(1-\lambda_i\eta)^t}{\lambda_i B_{\text{loc}}}\eta\Sigma_{0,i,j} + \tilde{\mathcal{O}}(\eta^{0.5-\beta}) \\
&= \frac{H\eta}{\lambda_i B_{\text{loc}}}\Sigma_{0,i,j} - \frac{1-(1-\lambda_i\eta)^H}{\lambda_i^2 B_{\text{loc}}}\Sigma_{0,i,j} + \tilde{\mathcal{O}}(\eta^{0.5-\beta}) \\
&= \frac{H\eta}{\lambda_i B_{\text{loc}}}\left(1 - \frac{1-(1-\lambda_i\eta)^H}{\lambda_i H\eta}\right)\Sigma_{0,i,j} + \tilde{\mathcal{O}}(\eta^{0.5-\beta}) \\
&= \frac{H\eta}{\lambda_i B_{\text{loc}}}\psi(\lambda_i H\eta)\Sigma_{0,i,j} + \tilde{\mathcal{O}}(\eta^{0.5-\beta}),
\end{aligned}$$

and

$$\begin{aligned}
\sum_{t=0}^{H-1}\hat{B}_{t,i,j}^{(s)} &= \sum_{t=0}^{H-1}\frac{(1-\lambda_i\eta)^t}{\lambda_i K B_{\text{loc}}}\eta\Sigma_{0,i,j} + \tilde{\mathcal{O}}(\eta^{1.5-\beta}), \\
&= \frac{1-(1-\lambda_i\eta)^H}{\lambda_i^2 K B_{\text{loc}}}\Sigma_{0,i,j} + \tilde{\mathcal{O}}(\eta^{0.5-\beta}) \\
&= \frac{H\eta}{\lambda_i K B_{\text{loc}}}\Sigma_{0,i,j} - \frac{H\eta}{\lambda_i K B_{\text{loc}}}\left(1 - \frac{1-(1-\lambda_i\eta)^H}{\lambda_i H\eta}\right)\Sigma_{0,i,j} + \tilde{\mathcal{O}}(\eta^{0.5-\beta}) \\
&= \frac{H\eta}{\lambda_i K B_{\text{loc}}}\Sigma_{0,i,j} - \frac{H\eta}{\lambda_i K B_{\text{loc}}}\psi(\lambda_i H\eta)\Sigma_{0,i,j} + \tilde{\mathcal{O}}(\eta^{0.5-\beta}).
\end{aligned}$$

Therefore, the matrix form of $\sum_{t=0}^{H-1}\boldsymbol{P}_\perp(\hat{A}_t^{(s)}+\hat{B}_t^{(s)})\boldsymbol{P}_\parallel$ is

$$\sum_{t=0}^{H-1}\boldsymbol{P}_\perp(\hat{A}_t^{(s)}+\hat{B}_t^{(s)})\boldsymbol{P}_\parallel = \frac{H\eta}{B}\mathcal{V}_{\boldsymbol{H}_0}\left(\boldsymbol{\Sigma}_{0,\perp,\parallel} + (K-1)\psi(\boldsymbol{\Sigma}_{0,\perp,\parallel})\right) + \tilde{\mathcal{O}}(\eta^{0.5-\beta}),$$

where $\psi(\cdot)$ is interpreted as an *elementwise* matrix function here. Furthermore, by Lemma K.35, $\sum_{t=0}^{H-1}\hat{B}_t^{(s)} = \tilde{\mathcal{O}}(\eta^{0.5-\beta})$. Applying Lemma K.3, we have (105). Finally, directly applying Lemma K.5, we have

$$-\eta\boldsymbol{P}_\parallel\nabla^3\mathcal{L}(\hat{\phi}^{(0)})[\boldsymbol{P}_\parallel\mathcal{T}_3\boldsymbol{P}_\parallel] = \boldsymbol{0}. \tag{106}$$

Notice that $\psi(\boldsymbol{\Sigma}_{0,\parallel}) = \boldsymbol{0}$ where $\psi(\cdot)$ operates on each element. Combining (104), (105) and (106), we obtain (103). By (102) and (103), we have (98). $\qquad\square$

**Lemma K.37.** *The second moment of the change of manifold projection every round is*

$$\mathbb{E}[(\hat{\phi}^{(s+1)} - \hat{\phi}^{(s)})(\hat{\phi}^{(s+1)} - \hat{\phi}^{(s)})^\top] = \begin{cases} \frac{H\eta^2}{B}\boldsymbol{\Sigma}_{0,\parallel} + \tilde{\mathcal{O}}(\eta^{1.5-0.5\beta}), & R_0 \le s < R_{\text{grp}} \\ \tilde{\mathcal{O}}(\eta), & s < R_0 \end{cases},$$

*where $R_0 := \lceil\frac{10}{\lambda_m\alpha}\log\frac{1}{\eta}\rceil$.*

*Proof.* Directly apply Lemma K.32 and Lemma K.27 and we have the lemma. $\qquad\square$

With Lemmas K.36 and K.37, we are ready to prove Theorem K.3.

*Proof of Theorem K.3.* We first derive $\mathbb{E}[\Delta \hat{\phi}^{(R_{\mathrm{grp}})}]$. Recall that $R_{\mathrm{grp}} = \lfloor \frac{1}{\alpha \eta^\beta} \rfloor = \frac{1}{H\eta^{1+\beta}} + o(1)$ where $0 < \beta < 0.5$. By Lemma K.36,

$$\mathbb{E}[\hat{\phi}^{(R_{\mathrm{grp}})} - \hat{\phi}^{(0)}] = \sum_{s=0}^{R_0} \mathbb{E}[\hat{\phi}^{(s+1)} - \hat{\phi}^{(s)}] + \sum_{s=R_0+1}^{R_{\mathrm{grp}}-1} \mathbb{E}[\hat{\phi}^{(s+1)} - \hat{\phi}^{(s)}]$$

$$= \frac{\eta^{1-\beta}}{2B} \partial^2 \Phi(\hat{\phi}^{(0)})[\mathbf{\Sigma}_0 + \mathbf{\Psi}(\hat{\phi}^{(0)})] + \tilde{\mathcal{O}}(\eta^{1.5-2\beta}) + \tilde{\mathcal{O}}(\eta).$$

Then we compute $\mathbb{E}[\Delta \hat{\phi}^{(R_{\mathrm{grp}})} \Delta \hat{\phi}^{(R_{\mathrm{grp}})\top}]$.

$$\mathbb{E}\left[ \left( \sum_{s=0}^{R_{\mathrm{grp}}-1} (\hat{\phi}^{(s+1)} - \hat{\phi}^{(s)}) \right) \left( \sum_{s=0}^{R_{\mathrm{grp}}-1} (\hat{\phi}^{(s+1)} - \hat{\phi}^{(s)}) \right)^\top \right]$$

$$= \sum_{s=0}^{R_{\mathrm{grp}}-1} \mathbb{E}[(\hat{\phi}^{(s+1)} - \hat{\phi}^{(s)})(\hat{\phi}^{(s+1)} - \hat{\phi}^{(s)})^\top] + \sum_{s \neq s'} \mathbb{E}[(\hat{\phi}^{(s+1)} - \hat{\phi}^{(s)})]\mathbb{E}[(\hat{\phi}^{(s'+1)} - \hat{\phi}^{(s')})^\top]$$

$$= \frac{\eta^{1-\beta}}{B} \mathbf{\Sigma}_{0,\|} + \tilde{\mathcal{O}}(\eta) + \tilde{\mathcal{O}}(\eta^{1.5-1.5\beta}),$$

where the last inequality uses $\mathbb{E}[(\hat{\phi}^{(s+1)} - \hat{\phi}^{(s)})]\mathbb{E}[(\hat{\phi}^{(s'+1)} - \hat{\phi}^{(s')})^\top] = \tilde{\mathcal{O}}(\eta^2)$. $\square$

### K.10 PROOF OF WEAK APPROXIMATION

We are now in a position to utilize the estimate of moments obtained in previous subsections to prove the closeness of the sequence $\{\phi^{(s)}\}_{s=0}^{\lfloor T/(H\eta^2) \rfloor}$ and the SDE solution $\{\zeta : t \in [0, T]\}$ in the sense of weak approximation. Recall the SDE that we expect the manifold projection $\{\Phi(\bar{\theta}^{(s)})\}_{s=0}^{\lfloor T/(H\eta^2) \rfloor}$ to track:

$$d\zeta(t) = P_\zeta \Big( \underbrace{\frac{1}{\sqrt{B}} \mathbf{\Sigma}_{\|}^{1/2}(\zeta)dW_t}_{\text{(a) diffusion}} \underbrace{- \frac{1}{2B}\nabla^3 \mathcal{L}(\zeta)[\widehat{\mathbf{\Sigma}}_\diamond(\zeta)]dt}_{\text{(b) drift-I}} \underbrace{- \frac{K-1}{2B}\nabla^3 \mathcal{L}(\zeta)[\widehat{\mathbf{\Psi}}(\zeta)]dt}_{\text{(c) drift-II}} \Big), \qquad (107)$$

According to Lemma K.3 and Lemma K.4, the drift term in total can be written as the following form:

$$(b) + (c) = \frac{1}{2B}\partial^2 \Phi(\zeta)[\mathbf{\Sigma}(\zeta) + (K-1)\mathbf{\Psi}(\zeta)].$$

Then by definition of $P_\zeta$, (107) is equivalent to the following SDE:

$$d\zeta(t) = \frac{1}{\sqrt{B}}\partial\Phi(\zeta)\mathbf{\Sigma}^{1/2}(\zeta)dW_t + \frac{1}{2B}\partial^2 \Phi(\zeta)\left[\mathbf{\Sigma}(\zeta) + (K-1)\mathbf{\Psi}(\zeta)\right]dt. \qquad (108)$$

Therefore, we only have to show that $\phi^{(s)}$ closely tracks $\{\zeta(t)\}$ satisfying Equation (108). By Lemma K.11, there exists an $\epsilon_3$ neighborhood of $\Gamma$, $\Gamma^{\epsilon_3}$, where $\Phi(\cdot)$ is $\mathcal{C}^\infty$-smooth. Due to compactness of $\Gamma$, $\Gamma^{\epsilon_3}$ is bounded and the mappings $\partial^2 \Phi(\cdot)$, $\partial \Phi(\cdot)$, $\mathbf{\Sigma}^{1/2}(\cdot)$, $\mathbf{\Sigma}(\cdot)$ and $\mathbf{\Psi}(\cdot)$ are all Lipschitz in $\Gamma^{\epsilon_3}$. By Kirszbraun theorem, both the drift and diffusion term of (108) can be extended as Lipschitz functions on $\mathbb{R}^d$. Therefore, the solution to the extended SDE exists and is unique. We further show that the solution, if initialized as a point on $\Gamma$, always stays on the manifold almost surely.

As a preparation, we first show that $\Gamma$ has no boundary.

**Lemma K.38.** *Under Assumptions 3.1 to 3.3, $\Gamma$ has no boundary.*

*Proof.* We prove by contradiction. If $\Gamma$ has boundary $\partial\Gamma$, WLOG, for a point $p \in \partial\Gamma$, let the Hessian at $p$ be diagonal with the form $\nabla^2 \mathcal{L}(p) = \mathrm{diag}(\lambda_1, \cdots, \lambda_d)$ where $\lambda_i > 0$ for $1 \leq i \leq m$ and $\lambda_i = 0$ for $m < i \leq d$.

Denote by $\boldsymbol{x}_{i:j} := (x_i, x_{i+1}, \cdots, x_j)$ $(i \leq j)$ the $(j - i + 1)$-dimensional vector formed by the $i$-th to $j$-th coordinates of $\boldsymbol{x}$. Since $\frac{\partial(\nabla \mathcal{L}(\boldsymbol{p}))}{\partial \boldsymbol{p}_{1:m}} = \mathrm{diag}(\lambda_1, \cdots, \lambda_m)$ is invertible, by the implicit function theorem, there exists an open neighborhood $V$ of $\boldsymbol{p}_{m+1:d}$ such that $\nabla \mathcal{L}(\boldsymbol{v}) = \boldsymbol{0}$, $\forall \boldsymbol{v} \in V$. Then, $\mathcal{L}(\boldsymbol{v}) = \mathcal{L}(\boldsymbol{p}) = \min_{\boldsymbol{\theta} \in U} \mathcal{L}(\boldsymbol{\theta})$ and hence $V \subset \Gamma$, which contradicts with $\boldsymbol{p} \in \partial\Gamma$. $\qquad\square$

Therefore, $\Gamma$ is a closed manifold (i.e., compact and without boundary). Then we have the following lemma stating that $\Gamma$ is invariant for (108).

**Lemma K.39.** *Let $\boldsymbol{\zeta}(t)$ be the solution to (108) with $\boldsymbol{\zeta}(0) \in \Gamma$, then $\boldsymbol{\zeta}(t) \in \Gamma$ for all $t \geq 0$. In other words, $\Gamma$ is invariant for (108).*

*Proof.* According to Filipović (2000) and Du & Duan (2007), for a closed manifold $\mathcal{M}$ to be viable for the SDE $\mathrm{d}\boldsymbol{X}(t) = F(\boldsymbol{X}(t))\mathrm{d}t + \boldsymbol{B}(\boldsymbol{X}(t))\mathrm{d}\boldsymbol{W}_t$ where $F : \mathbb{R}^d \to \mathbb{R}^d$ and $\boldsymbol{B} : \mathbb{R}^d \to \mathbb{R}^d$ are locally Lipschitz, we only have to verify the following Nagumo type consistency condition:

$$\mu(\boldsymbol{x}) := F(\boldsymbol{x}) - \frac{1}{2} \sum_j \mathrm{D}[B_j(\boldsymbol{x})]B_j(\boldsymbol{x}) \in T_{\boldsymbol{x}}(\mathcal{M}), \quad B_j(\boldsymbol{x}) \in T_{\boldsymbol{x}}(\mathcal{M}),$$

where $\mathrm{D}[\cdot]$ is the Jacobian operator and $B_j(\boldsymbol{x})$ denotes the $j$-th column of $\boldsymbol{B}(\boldsymbol{x})$.

In our context, since for $\boldsymbol{\phi} \in \Gamma$, $\partial\Phi(\boldsymbol{\phi})$ is a projection matrix onto $T_{\boldsymbol{\phi}}(\Gamma)$, each column of $\partial\Phi(\boldsymbol{\phi})\boldsymbol{\Sigma}^{1/2}(\boldsymbol{\phi})$ belongs to $T_{\boldsymbol{\phi}}(\Gamma)$, verifying the second condition. Denote by $\boldsymbol{P}_{\perp}(\boldsymbol{\phi}) := \boldsymbol{I}_d - \partial\Phi(\boldsymbol{\phi})$ the projection onto the normal space of $\Gamma$ at $\boldsymbol{\phi}$. To verify the first condition, it suffices to show that $\boldsymbol{P}_{\perp}(\boldsymbol{\phi})\mu(\boldsymbol{\phi}) = \boldsymbol{0}$. We evaluate $\sum_j \boldsymbol{P}_{\perp}(\boldsymbol{\phi})\mathrm{D}[B_j(\boldsymbol{\phi})]B_j(\boldsymbol{\phi})$ as follows.

$$\sum_j \boldsymbol{P}_{\perp}(\boldsymbol{\phi})\mathrm{D}[B_j(\boldsymbol{\phi})]B_j(\boldsymbol{\phi}) = \frac{1}{B} \sum_j \mathrm{D}[\partial\Phi(\boldsymbol{\phi})\boldsymbol{\Sigma}_j^{1/2}(\boldsymbol{\phi})]\partial\Phi(\boldsymbol{\phi})\boldsymbol{\Sigma}_j^{1/2}(\boldsymbol{\phi})$$

$$= \frac{1}{B}\boldsymbol{P}_{\perp}(\boldsymbol{\phi}) \sum_j \partial^2\Phi(\boldsymbol{\phi})[\boldsymbol{\Sigma}_j^{1/2}(\boldsymbol{\phi}), \partial\Phi(\boldsymbol{\phi})\boldsymbol{\Sigma}_j^{1/2}(\boldsymbol{\phi})]$$

$$= -\frac{1}{B}\nabla^2\mathcal{L}(\boldsymbol{\phi})^+\nabla^3\mathcal{L}(\boldsymbol{\phi})[\boldsymbol{\Sigma}_{\parallel}(\boldsymbol{\phi})], \tag{109}$$

where the last inequality uses Lemma K.3. Again applying Lemma K.3, we have

$$\boldsymbol{P}_{\perp}(\boldsymbol{\phi})F(\boldsymbol{\phi}) = -\frac{1}{2B}\nabla^2\mathcal{L}(\boldsymbol{\phi})^+\nabla^3\mathcal{L}(\boldsymbol{\phi})[\boldsymbol{\Sigma}_{\parallel}(\boldsymbol{\phi})]. \tag{110}$$

Combining (109) and (110), we can verify the first condition. $\qquad\square$

In order to establish Theorem 3.2, it suffices to prove the following theorem, which captures the closeness of $\boldsymbol{\phi}^{(s)}$ and $\boldsymbol{\zeta}(t)$ every $R_{\mathrm{grp}}$ rounds.

**Theorem K.4.** *If $\|\bar{\boldsymbol{\theta}}^{(0)} - \boldsymbol{\phi}^{(0)}\|_2 = \mathcal{O}(\sqrt{\eta \log \frac{1}{\eta}})$ and $\boldsymbol{\zeta}(0) = \boldsymbol{\phi}^{(0)} \in \Gamma$, then for $R_{\mathrm{grp}} = \lfloor\frac{1}{\alpha\eta^{0.75}}\rfloor$ every test function $g \in \mathcal{C}^3$,*

$$\max_{n=0,\cdots,\lfloor T/\eta^{0.75}\rfloor} \left| \mathbb{E}g(\boldsymbol{\phi}^{(nR_{\mathrm{grp}})}) - \mathbb{E}g(\boldsymbol{\zeta}(n\eta^{0.75})) \right| \leq C_g\eta^{0.25}(\log \tfrac{1}{\eta})^b,$$

*where $C_g > 0$ is a constant independent of $\eta$ but can depend on $g(\cdot)$ and $b > 0$ is a constant independent of $\eta$ and $g(\cdot)$.*

### K.10.1 PRELIMINARIES AND ADDITIONAL NOTATIONS

We first introduce a general formulation for stochastic gradient algorithms (SGAs) and then specify the components of this formulation in our context. Consider the following SGA:

$$\boldsymbol{x}_{n+1} = \boldsymbol{x}_n + \eta_{\mathrm{e}}\boldsymbol{h}(\boldsymbol{x}_n, \boldsymbol{\xi}_n),$$

where $\boldsymbol{x}_n \in \mathbb{R}^d$ is the parameter, $\eta_{\mathrm{e}}$ is the learning rate, $\boldsymbol{h}(\cdot, \cdot)$ is the update which depends on $\boldsymbol{x}_n$ and a random vector $\boldsymbol{\xi}_n$ sampled from some distribution $\Xi(\boldsymbol{x}_n)$. Also, consider the following Stochastic Differential Equation (SDE).

$$\mathrm{d}\boldsymbol{X}(t) = \boldsymbol{b}(\boldsymbol{X}(t))\mathrm{d}t + \boldsymbol{\sigma}(\boldsymbol{X}(t))\mathrm{d}\boldsymbol{W}_t,$$

where $\boldsymbol{b}(\cdot) : \mathbb{R}^d \to \mathbb{R}^d$ is the drift function and $\boldsymbol{\sigma}(\cdot) : \mathbb{R}^{d \times d} \to \mathbb{R}^{d \times d}$ is the diffusion matrix.

Denote by $\mathcal{P}_{\boldsymbol{X}}(\boldsymbol{x}, s, t)$ the distribution of $\boldsymbol{X}(t)$ with the initial condition $\boldsymbol{X}(s) = \boldsymbol{x}$. Define

$$\tilde{\boldsymbol{\Delta}}(\boldsymbol{x}, n) := \boldsymbol{X}_{(n+1)\eta_{\mathrm{e}}} - \boldsymbol{x}, \qquad \text{where } \boldsymbol{X}_{(n+1)\eta_{\mathrm{e}}} \sim \mathcal{P}_{\boldsymbol{X}}(\boldsymbol{x}, n\eta_{\mathrm{e}}, (n+1)\eta_{\mathrm{e}}),$$

which characterizes the update in one step.

In our context, we view the change of manifold projection over $R_{\mathrm{grp}} := \lfloor \frac{1}{\alpha \eta^{1-\beta}} \rfloor (\beta \in (0, 0.5))$ rounds as one "giant step". Hence the $\phi^{(nR_{\mathrm{grp}})}$ corresponds to the discrete time random variable $\boldsymbol{x}_n$ corresponds to and $\boldsymbol{\zeta}(t)$ corresponds to the continuous time random variable $\boldsymbol{X}_t$. According to Theorem K.2, we set

$$\eta_{\mathrm{e}} = \eta^{1-\beta}, \quad \boldsymbol{b}(\boldsymbol{\zeta}) = \frac{1}{2B} \partial^2 \Phi(\boldsymbol{\zeta}) \left[ \boldsymbol{\Sigma}(\boldsymbol{\zeta}) + (K-1)\boldsymbol{\Psi}(\boldsymbol{\zeta}) \right], \quad \boldsymbol{\sigma}(\boldsymbol{\zeta}) = \frac{1}{\sqrt{B}} \partial \Phi(\boldsymbol{\zeta}) \boldsymbol{\Sigma}^{1/2}(\boldsymbol{\zeta}).$$

Due to compactness of $\Gamma$, $\boldsymbol{b}(\cdot)$ and $\boldsymbol{\sigma}(\cdot)$ are Lipschitz on $\Gamma$.

As for the update in one step, $\tilde{\boldsymbol{\Delta}}(\cdot, \cdot)$ is defined in our context as:

$$\tilde{\boldsymbol{\Delta}}(\boldsymbol{\phi}, n) := \boldsymbol{\zeta}_{(n+1)\eta_{\mathrm{e}}} - \boldsymbol{\phi}, \qquad \text{where } \boldsymbol{\zeta}_{(n+1)\eta_{\mathrm{e}}} \sim \mathcal{P}_{\boldsymbol{\zeta}}(\boldsymbol{\phi}, n\eta_{\mathrm{e}}, (n+1)\eta_{\mathrm{e}}) \text{ and } \boldsymbol{\phi} \in \Gamma.$$

For convenience, we further define

$$\begin{aligned}
\boldsymbol{\Delta}^{(n)} &:= \hat{\boldsymbol{\phi}}^{((n+1)R_{\mathrm{grp}})} - \hat{\boldsymbol{\phi}}^{(nR_{\mathrm{grp}})}, & \tilde{\boldsymbol{\Delta}}^{(n)} &:= \tilde{\boldsymbol{\Delta}}(\hat{\boldsymbol{\phi}}^{(R_{\mathrm{grp}})}, n), \\
\boldsymbol{b}^{(n)} &:= \boldsymbol{b}(\hat{\boldsymbol{\phi}}^{(nR_{\mathrm{grp}})}), & \boldsymbol{\sigma}^{(n)} &:= \boldsymbol{\sigma}(\hat{\boldsymbol{\phi}}^{(nR_{\mathrm{grp}})}).
\end{aligned}$$

We use $C_{g,i}$ to denote constants that can depend on the test function $g$ and independent of $\eta_{\mathrm{e}}$. The following lemma relates the moments of $\tilde{\boldsymbol{\Delta}}(\boldsymbol{\phi}, n)$ to $\boldsymbol{b}(\boldsymbol{\phi})$ and $\boldsymbol{\sigma}(\boldsymbol{\phi})$.

**Lemma K.40.** *There exists a positive constant $C_0$ independent of $\eta_{\mathrm{e}}$ and $g$ such that for all $\boldsymbol{\phi} \in \Gamma$,*

$$|\mathbb{E}[\tilde{\Delta}_i(\boldsymbol{\phi}, n)] - \eta_{\mathrm{e}} b_i(\boldsymbol{\phi})| \leq C_0 \eta_{\mathrm{e}}^2, \qquad \forall 1 \leq i \leq d,$$

$$|\mathbb{E}[\tilde{\Delta}_i(\boldsymbol{\phi}, n)\tilde{\Delta}_j(\boldsymbol{x}, n)] - \eta_{\mathrm{e}} \sum_{l=1}^{d} \sigma_{i,l}(\boldsymbol{\phi})\sigma_{l,j}(\boldsymbol{\phi})| \leq C_0 \eta_{\mathrm{e}}^2, \qquad \forall 1 \leq i, j \leq d,$$

$$\mathbb{E}\left[ \left| \prod_{s=1}^{6} \tilde{\Delta}_{i_s}(\boldsymbol{\phi}, n) \right| \right] \leq C_0 \eta_{\mathrm{e}}^3, \qquad \forall 1 \leq i_1, \cdots, i_6 \leq d.$$

*The lemma below states that the expectation of the test function is smooth with respect to the initial value.*

*Proof.* Noticing that (i) the solution to (108) always stays on $\Gamma$ almost surely if its initial value $\boldsymbol{\zeta}(0)$ belongs to $\Gamma$, (ii) $\boldsymbol{b}(\cdot)$ and $\boldsymbol{\sigma}(\cdot)$ are $\mathcal{C}^\infty$ and (iii) $\Gamma$ is compact, we can directly apply Lemma B.3 in Malladi et al. (2022) and Lemma 26 in Li et al. (2019a) to obtain the above lemma. $\square$

The following lemma states that the expectation of $g(\boldsymbol{\zeta}(t))$ for $g \in \mathcal{C}^3$ is smooth with respect to the initial value of the SDE solution.

**Lemma K.41.** *Let $s \in [0, T]$, $\boldsymbol{\phi} \in \Gamma$ and $g \in \mathcal{C}^3$. For $t \in [s, T]$, define*

$$u(\boldsymbol{\phi}, s, t) := \mathbb{E}_{\boldsymbol{\zeta}_t \sim \mathcal{P}_{\boldsymbol{\zeta}}(\boldsymbol{\phi}, s, t)}[g(\boldsymbol{\zeta}_t)].$$

*Then $u(\cdot, s, t) \in \mathcal{C}^3$ uniformly in $s, t$.*

*Proof.* A slight modification of Lemma B.4 in Malladi et al. (2022) will give the above lemma. $\square$

### K.10.2   PROOF OF THE APPROXIMATION IN OUR CONTEXT

For $\beta \in (0, 0.5)$, define $\gamma_1 := \frac{1.5 - 2\beta}{1 - \beta}, \gamma_2 := \frac{1}{1 - \beta}$, and then $1 < \gamma_1 < 1.5$, $1 < \gamma_2 < 2$. We introduce the following lemma which serves as a key step to control the approximation error. Specifically, this lemma bounds the difference in one step change between the discrete process and the continuous one as well as the product of higher orders.

**Lemma K.42.** *If $\|\bar{\boldsymbol{\theta}}^{(0)} - \boldsymbol{\phi}^{(0)}\|_2 = \mathcal{O}(\sqrt{\eta \log \frac{1}{\eta}})$, then there exist positive constants $C_1$ and $b$ independent of $\eta_{\mathrm{e}}$ and $g$ such that for all $0 \leq n < \lfloor T/\eta_{\mathrm{e}} \rfloor$,*

1.

$$|\mathbb{E}[\Delta_i^{(n)} - \tilde{\Delta}_i^{(n)} \mid \mathcal{E}_0^{(nR_{\mathrm{grp}})}| \leq C_1 \eta_{\mathrm{e}}^{\gamma_1} (\log \frac{1}{\eta_{\mathrm{e}}})^b + C_1 \eta_{\mathrm{e}}^{\gamma_2} (\log \frac{1}{\eta_{\mathrm{e}}})^b, \qquad \forall 1 \leq i \leq d,$$

$$|\mathbb{E}[\Delta_i^{(n)} \Delta_j^{(n)} - \tilde{\Delta}_i^{(n)} \tilde{\Delta}_j^{(n)} \mid \mathcal{E}_0^{(nR_{\mathrm{grp}})}| \leq C_1 \eta_{\mathrm{e}}^{\gamma_1} (\log \frac{1}{\eta_{\mathrm{e}}})^b + C_1 \eta_{\mathrm{e}}^{\gamma_2} (\log \frac{1}{\eta_{\mathrm{e}}})^b, \quad \forall 1 \leq i, j \leq d.$$

2.

$$\mathbb{E}\left[ \left| \prod_{s=1}^{6} \Delta_{i_s}^{(n)} \right| \mid \mathcal{E}_0^{(nR_{\mathrm{grp}})} \right] \leq C_1^2 \eta_{\mathrm{e}}^{2\gamma_1} (\log \frac{1}{\eta_{\mathrm{e}}})^{2b}, \qquad \forall 1 \leq i_1, \cdots, i_6 \leq d,$$

$$\mathbb{E}\left[ \left| \prod_{s=1}^{6} \tilde{\Delta}_{i_s}^{(n)} \right| \mid \mathcal{E}_0^{(nR_{\mathrm{grp}})} \right] \leq C_1^2 \eta_{\mathrm{e}}^{2\gamma_1} (\log \frac{1}{\eta_{\mathrm{e}}})^{2b}, \qquad \forall 1 \leq i_1, \cdots, i_6 \leq d.$$

*Proof.* According to Appendix K.7, we have

$$\mathbb{E}\left[ \left| \prod_{s=1}^{6} \Delta_{i_s}^{(n)} \right| \mid \mathcal{E}_0^{(nR_{\mathrm{grp}})} \right] = \tilde{\mathcal{O}}(\eta^{3-3\beta}).$$

Since $\gamma_1 < 1.5$ and $\gamma_2 < 2$, we can utilize Theorem K.3 and conclude that there exist positive constants $C_2$ and $b$ independent of $\eta_{\mathrm{e}}$ and $g$ such that

$$\left| \mathbb{E}[\Delta_i^{(n)} - \eta_{\mathrm{e}} b_i^{(n)} \mid \mathcal{E}_0^{(nR_{\mathrm{grp}})}] \right| \leq C_2 \eta_{\mathrm{e}}^{\gamma_1} (\log \frac{1}{\eta_{\mathrm{e}}})^b + C_2 \eta_{\mathrm{e}}^{\gamma_2} (\log \frac{1}{\eta_{\mathrm{e}}})^b, \forall 1 \leq i \leq d,$$

(111)

$$\left| \mathbb{E}[\Delta_i^{(n)} \Delta_j^{(n)} - \eta_{\mathrm{e}} \sum_{l=1}^{d} \sigma_{i,l}^{(n)} \sigma_{l,j}^{(n)} \mid \mathcal{E}_0^{(nR_{\mathrm{grp}})}] \right| \leq C_2 \eta_{\mathrm{e}}^{\gamma_1} (\log \frac{1}{\eta_{\mathrm{e}}})^b + C_2 \eta_{\mathrm{e}}^{\gamma_2} (\log \frac{1}{\eta_{\mathrm{e}}})^b, \forall 1 \leq i, j \leq d,$$

(112)

$$\mathbb{E}\left[ \left| \prod_{s=1}^{6} \Delta_{i_s}^{(n)} \right| \mid \mathcal{E}_0^{(nR_{\mathrm{grp}})} \right] \leq C_2^2 \eta_{\mathrm{e}}^{2\gamma_1} (\log \frac{1}{\eta_{\mathrm{e}}})^{2b}, \quad \forall 1 \leq i_1, \cdots, i_6 \leq d. \quad (113)$$

Combining (111) - (113) with Lemma K.40 gives the above lemma. $\qquad\square$

**Lemma K.43.** *For a test function $g \in \mathcal{C}^3$, let $u_{l,n}(\boldsymbol{\phi}) := u(\boldsymbol{\phi}, l\eta_{\mathrm{e}}, n\eta_{\mathrm{e}}) = \mathbb{E}_{\boldsymbol{\zeta}_t \sim \mathcal{P}_{\boldsymbol{\zeta}}(\boldsymbol{\phi}, l\eta_{\mathrm{e}}, n\eta_{\mathrm{e}})}[g(\boldsymbol{\zeta}_t)]$. If $\|\bar{\boldsymbol{\theta}}^{(0)} - \boldsymbol{\phi}^{(0)}\|_2 = \mathcal{O}(\sqrt{\eta \log \frac{1}{\eta}})$, then for all $0 \leq l \leq n-1$ and $1 \leq n \leq \lfloor T/\eta_{\mathrm{e}} \rfloor$,*

$$\left| \mathbb{E}[u_{l+1,n}(\hat{\boldsymbol{\phi}}^{(lR_{\mathrm{grp}})} + \boldsymbol{\Delta}^{(l)}) - u_{l+1,n}(\hat{\boldsymbol{\phi}}^{(lR_{\mathrm{grp}})} + \tilde{\boldsymbol{\Delta}}^{(l+1)}) \mid \hat{\boldsymbol{\phi}}^{(lR_{\mathrm{grp}})}] \right| \leq C_{g,1}(\eta_{\mathrm{e}}^{\gamma_1} + \eta_{\mathrm{e}}^{\gamma_2}) \log(\frac{1}{\eta_{\mathrm{e}}})^b,$$

*where $C_{g,1}$ is a positive constant independent of $\eta$ and $\hat{\boldsymbol{\phi}}^{(lR_{\mathrm{grp}})}$ but can depend on $g$.*

*Proof.* By Lemma K.41, $u_{l,n}(\boldsymbol{\phi}) \in \mathcal{C}^3$ for all $l$ and $n$. That is, there exists $K(\cdot) \in G$ such that for all $l, n$, $u_{l,n}(\boldsymbol{\phi})$ and its partial derivatives up to the third order are bounded by $K(\boldsymbol{\phi})$.

By the law of total expectation and triangle inequality,

$$\left| \mathbb{E}[u_{l+1,n}(\hat{\boldsymbol{\phi}}^{(lR_{\mathrm{grp}})} + \boldsymbol{\Delta}^{(l)}) - u_{l+1,n}(\hat{\boldsymbol{\phi}}^{(lR_{\mathrm{grp}})} + \tilde{\boldsymbol{\Delta}}^{(l)})] \mid \hat{\boldsymbol{\phi}}^{(lR_{\mathrm{grp}})} \right|$$

$$\leq \underbrace{\left| \mathbb{E}[u_{l+1,n}(\hat{\boldsymbol{\phi}}^{(lR_{\mathrm{grp}})} + \boldsymbol{\Delta}^{(l)}) - u_{l+1,n}(\hat{\boldsymbol{\phi}}^{(lR_{\mathrm{grp}})} + \tilde{\boldsymbol{\Delta}}^{(l)}) \mid \hat{\boldsymbol{\phi}}^{(lR_{\mathrm{grp}})}, \mathcal{E}_0^{(lR_{\mathrm{grp}})}] \right|}_{\mathcal{A}_1}$$

$$+ \underbrace{\eta^{100} \mathbb{E}[|u_{l+1,n}(\hat{\boldsymbol{\phi}}^{(lR_{\mathrm{grp}})} + \boldsymbol{\Delta}^{(l)})| \mid \hat{\boldsymbol{\phi}}^{(lR_{\mathrm{grp}})}, \bar{\mathcal{E}}_0^{(lR_{\mathrm{grp}})}]}_{\mathcal{A}_2}$$

$$+ \underbrace{\eta^{100} \mathbb{E}[|u_{l+1,n}(\hat{\boldsymbol{\phi}}^{(lR_{\mathrm{grp}})} + \tilde{\boldsymbol{\Delta}}^{(l)})| \mid \hat{\boldsymbol{\phi}}^{(lR_{\mathrm{grp}})}, \bar{\mathcal{E}}_0^{(lR_{\mathrm{grp}})}]}_{\mathcal{A}_3}.$$

We first bound $\mathcal{A}_2$ and $\mathcal{A}_3$. Since $\hat{\phi}^{(lR_{\mathrm{grp}})} \in \Gamma$, both $\hat{\phi}^{(lR_{\mathrm{grp}})} + \mathbf{\Delta}^{(l)}$ and $\hat{\phi}^{(lR_{\mathrm{grp}})} + \tilde{\mathbf{\Delta}}^{(l)}$ belong to $\Gamma$. Due to compactness of $\Gamma$ and smoothness of $u_{l+1,n}(\cdot)$ on $\Gamma$, there exist a positive constant $C_{g,2}$ such that $\mathcal{A}_2 + \mathcal{A}_3 \leq C_{g,2}\eta^{100}$.

We proceed to bound $\mathcal{A}_1$. Expanding $u_{l+1,n}(\cdot)$ at $\hat{\phi}^{(lR_{\mathrm{grp}})}$ and by triangle inequality,

$$
\mathcal{A}_1^{(s)} \leq \underbrace{\sum_{i=1}^{d} \left| \mathbb{E}[\frac{\partial u_{l+1,n}}{\partial \phi_i}(\hat{\phi}^{(lR_{\mathrm{grp}})}) \left( \Delta_i^{(l)} - \tilde{\Delta}_i^{(l)} \right) \mid \hat{\phi}^{(lR_{\mathrm{grp}})}, \mathcal{E}_0^{(lR_{\mathrm{grp}})} \right|}_{\mathcal{B}_1}
$$

$$
+ \underbrace{\frac{1}{2} \sum_{1 \leq i,j \leq d} \left| \mathbb{E}[\frac{\partial^2 u_{l+1,n}}{\partial \phi_i \partial \phi_j}(\hat{\phi}^{(lR_{\mathrm{grp}})}) \left( \Delta_i^{(l)}\Delta_j^{(l)} - \tilde{\Delta}_i^{(l)}\tilde{\Delta}_j^{(l)} \right) \mid \hat{\phi}^{(lR_{\mathrm{grp}})}, \mathcal{E}_0^{(lR_{\mathrm{grp}})} ] \right|}_{\mathcal{B}_2}
$$

$$
+ |\mathcal{R}| + |\tilde{\mathcal{R}}|,
$$

where the remainders $\mathcal{R}$ and $\tilde{\mathcal{R}}$ are

$$
\mathcal{R} = \frac{1}{6} \sum_{1 \leq i,j,p \leq d} \mathbb{E}[\frac{\partial^3 u_{l+1,n}}{\partial \phi_i \partial \phi_j \partial \phi_p}(\hat{\phi}^{(lR_{\mathrm{grp}})} + \theta\mathbf{\Delta}^{(l)})\Delta_i^{(l)}\Delta_j^{(l)} \mid \hat{\phi}^{(lR_{\mathrm{grp}})}, \mathcal{E}_0^{(lR_{\mathrm{grp}})}],
$$

$$
\tilde{\mathcal{R}} = \frac{1}{6} \sum_{1 \leq i,j,p \leq d} \mathbb{E}[\frac{\partial^3 u_{l+1,n}}{\partial \phi_i \partial \phi_j \partial \phi_p}(\hat{\phi}^{(lR_{\mathrm{grp}})} + \tilde{\theta}\tilde{\mathbf{\Delta}}^{(l)})\tilde{\Delta}_i^{(l)}\tilde{\Delta}_j^{(l)}\tilde{\Delta}_p^{(l)} \mid \hat{\phi}^{(lR_{\mathrm{grp}})}, \mathcal{E}_0^{(lR_{\mathrm{grp}})}],
$$

for some $\theta, \tilde{\theta} \in (0,1)$. Since $\hat{\phi}^{(lR_{\mathrm{grp}})}$ belongs to $\Gamma$ which is compact, there exists a constant $C_{g,3}$ such that for all $1 \leq i,j \leq d, 0 \leq l \leq n-1, 1 \leq n \leq \lfloor T/\eta_{\mathrm{e}} \rfloor$,

$$
|\frac{\partial u_{l+1,n}}{\partial \phi_i}(\hat{\phi}^{(lR_{\mathrm{grp}})})| \leq C_{g,3}, \qquad |\frac{\partial^2 u_{l+1,n}}{\partial \phi_i \partial \phi_j}(\hat{\phi}^{(lR_{\mathrm{grp}})})| \leq C_{g,3}.
$$

By Lemma K.42,

$$
\mathcal{B}_1 \leq dC_{g,3}C_1(\eta_{\mathrm{e}}^{\gamma_1} + \eta_{\mathrm{e}}^{\gamma_2})(\log \frac{1}{\eta_{\mathrm{e}}})^b, \qquad \mathcal{B}_2 \leq \frac{d^2}{2}C_{g,3}C_1(\eta_{\mathrm{e}}^{\gamma_1} + \eta_{\mathrm{e}}^{\gamma_2})(\log \frac{1}{\eta_{\mathrm{e}}})^b.
$$

Now we bound the remainders. By Cauchy-Schwartz inequality,

$$
\left| \mathbb{E}[\frac{\partial^3 u_{l+1,n}}{\partial \phi_i \partial \phi_j \partial \phi_p}(\hat{\phi}^{(lR_{\mathrm{grp}})} + \theta\mathbf{\Delta}^{(l)})\Delta_i^{(l)}\Delta_j^{(l)}\Delta_p^{(l)} \mid \hat{\phi}^{(lR_{\mathrm{grp}})}, \mathcal{E}_0^{(lR_{\mathrm{grp}})}] \right|
$$

$$
\leq \left( \mathbb{E}\left[ \left( \frac{\partial^3 u_{l+1,n}}{\partial \phi_i \partial \phi_j \partial \phi_p}(\hat{\phi}^{(lR_{\mathrm{grp}})} + \theta\mathbf{\Delta}^{(l)}) \right)^2 \mid \hat{\phi}^{(lR_{\mathrm{grp}})}, \mathcal{E}_0^{(nR_{\mathrm{grp}})} \right] \right)^{1/2} \times
$$

$$
\left( \mathbb{E}[(\Delta_i^{(l)}\Delta_j^{(l)}\Delta_p^{(l)})^2 \mid \hat{\phi}^{(lR_{\mathrm{grp}})}, \mathcal{E}_0^{(nR_{\mathrm{grp}})}] \right)^{1/2}.
$$

Since $\hat{\phi}^{(lR_{\mathrm{grp}})}$ and $\hat{\phi}^{(lR_{\mathrm{grp}})} + \mathbf{\Delta}^{(l)}$ both belong to $\Gamma$ which is compact, there exists a constant $C_{g,4}$ such that for all $1 \leq i,j,p \leq d, 0 \leq l \leq n-1$ and $1 \leq n \leq \lfloor T/\eta_{\mathrm{e}} \rfloor$,

$$
\left( \frac{\partial^3 u_{l+1,n}}{\partial \phi_i \partial \phi_j \partial \phi_p}(\hat{\phi}^{(lR_{\mathrm{grp}})} + \theta\mathbf{\Delta}^{(l)}) \right)^2 \leq C_{g,4}^2.
$$

Combining the above inequality with Lemma K.42, we have

$$
\left| \mathbb{E}[\frac{\partial^3 u_{l+1,n}}{\partial \phi_i \partial \phi_j \partial \phi_p}(\hat{\phi}^{(lR_{\mathrm{grp}})} + \theta\mathbf{\Delta}^{(l)})\Delta_i^{(l)}\Delta_j^{(l)}\Delta_p^{(l)} \mid \hat{\phi}^{(lR_{\mathrm{grp}})}, \mathcal{E}_0^{(lR_{\mathrm{grp}})}] \right| \leq C_{g,4}C_1\eta_{\mathrm{e}}^{\gamma_1} \log(\frac{1}{\eta_{\mathrm{e}}})^b.
$$

Hence, for all $1 \leq n \leq \lfloor T/\eta_{\mathrm{e}} \rfloor, 0 \leq l \leq n-1$,

$$
|\mathcal{R}| \leq \frac{d^3}{6}C_{g,4}C_1\eta_{\mathrm{e}}^{\gamma_1} \log(\frac{1}{\eta_{\mathrm{e}}})^b.
$$

Similarly, we can show that there exists a constant $C_{g,5}$ such that for all $1 \leq n \leq \lfloor T/\eta_{\mathrm{e}} \rfloor, 0 \leq l \leq n-1$,

$$
|\tilde{\mathcal{R}}| \leq \frac{d^3}{6}C_{g,5}C_1\eta_{\mathrm{e}}^{\gamma_1} \log(\frac{1}{\eta_{\mathrm{e}}})^b.
$$

Combining the bounds on $\mathcal{A}_1$ to $\mathcal{A}_3$, we have the lemma. $\square$

Finally, we prove Theorem K.4.

*Proof.* For $0 \leq l \leq n$, define the random variable $\hat{\boldsymbol{\zeta}}_{l,n}$ which follows the distribution $\mathcal{P}_{\zeta}(\hat{\boldsymbol{\phi}}^{(lR_{\text{grp}})}, l, n)$ conditioned on $\hat{\boldsymbol{\phi}}^{(lR_{\text{grp}})}$. Therefore, $\mathbb{P}(\hat{\boldsymbol{\zeta}}_{n,n} = \hat{\boldsymbol{\phi}}^{(nR_{\text{grp}})}) = 1$ and $\hat{\boldsymbol{\zeta}}_{0,n} \sim \boldsymbol{\zeta}_{n\eta_{\text{e}}}$. Denote by $u(\boldsymbol{\phi}, s, t) := \mathbb{E}_{\boldsymbol{\zeta}_t \sim \mathcal{P}_{\zeta}(\phi, s, t)}[g(\boldsymbol{\zeta}_t)]$ and $\mathcal{T}_{l+1,n} := u_{l+1,n}(\hat{\boldsymbol{\phi}}^{(lR_{\text{grp}})} + \boldsymbol{\Delta}^{(l)}, (l+1)\eta_{\text{e}}, n\eta_{\text{e}}) - u_{l+1,n}(\hat{\boldsymbol{\phi}}^{(lR_{\text{grp}})} + \tilde{\boldsymbol{\Delta}}^{(l)}, (l+1)\eta_{\text{e}}, n\eta_{\text{e}})$.

$$
\left| \mathbb{E}[g(\boldsymbol{\phi}^{(nR_{\text{grp}})})] - \mathbb{E}[g(\boldsymbol{\zeta}(n\eta_{\text{e}}))] \right|
$$

$$
\leq \left| \mathbb{E}[g(\hat{\boldsymbol{\zeta}}_{n,n}) - g(\hat{\boldsymbol{\zeta}}_{0,n}) \mid \mathcal{E}_0^{(nR_{\text{grp}})}] \right| + \mathcal{O}(\eta^{100})
$$

$$
\leq \sum_{l=0}^{n-1} \left| \mathbb{E}[g(\hat{\boldsymbol{\zeta}}_{l+1,n}) - g(\hat{\boldsymbol{\zeta}}_{l,n}) \mid \mathcal{E}_0^{(nR_{\text{grp}})}] \right| + \mathcal{O}(\eta^{100})
$$

$$
= \sum_{l=0}^{n-1} \left| \mathbb{E}[u(\hat{\boldsymbol{\phi}}^{((l+1)R_{\text{grp}})}, (l+1)\eta_{\text{e}}, n\eta_{\text{e}}) - u(\hat{\boldsymbol{\zeta}}_{l,l+1}, (l+1)\eta_{\text{e}}, n\eta_{\text{e}}) \mid \mathcal{E}_0^{(nR_{\text{grp}})}] \right| + \mathcal{O}(\eta^{100})
$$

$$
= \sum_{l=0}^{n-1} \left| \mathbb{E}[\mathcal{T}_{l+1,n} \mid \mathcal{E}_0^{(nR_{\text{grp}})}] \right| + \mathcal{O}(\eta^{100}).
$$

Noticing that $\mathbb{E}[\mathcal{T}_{l+1,n} \mid \mathcal{E}_0^{(nR_{\text{grp}})}] = \mathbb{E}[\mathbb{E}[\mathcal{T}_{l+1,n} \mid \hat{\boldsymbol{\phi}}^{(lR_{\text{grp}})}, \mathcal{E}_0^{(lR_{\text{grp}})}] \mid \mathcal{E}_0^{(nR_{\text{grp}})}]$, we can apply Lemma K.43 and obtain that for all $0 \leq n \leq \lfloor T/\eta_{\text{e}} \rfloor$,

$$
\left| \mathbb{E}[g(\boldsymbol{\phi}^{(nR_{\text{grp}})})] - \mathbb{E}[g(\boldsymbol{\zeta}(n\eta_{\text{e}}))] \right| \leq n C_{g,1} (\eta_{\text{e}}^{\gamma_1} + \eta_{\text{e}}^{\gamma_2})(\log \tfrac{1}{\eta_{\text{e}}})^b
$$

$$
\leq T C_{g,1} (\eta_{\text{e}}^{\gamma_1 - 1} + \eta_{\text{e}}^{\gamma_2 - 1})(\log \tfrac{1}{\eta_{\text{e}}})^b.
$$

Notice that $\eta_{\text{e}}^{\gamma_1} + \eta_{\text{e}}^{\gamma_2} = \eta^{0.5-\beta} + \eta^{\beta}$ and $T, C_{g,1}$ are both constants that are independent of $\eta_{\text{e}}$. Let $\beta = 0.25$ and we have Theorem K.4. $\qquad\square$

Having established Theorem K.4, we are thus led to prove Theorem 3.2.

*Proof of Theorem 3.2.* Denote by $s_{\text{cls}} = s_0 + s_1 = \mathcal{O}(\log \tfrac{1}{\eta})$, which is the time the global iterate $\bar{\boldsymbol{\theta}}^{(s)}$ will reach within $\tilde{\mathcal{O}}(\eta)$ from $\Gamma$ with high probability. Define $\tilde{\boldsymbol{\zeta}}(t)$ to be the solution to the limiting SDE (108) conditioned on $\mathcal{E}_0^{(s_{\text{cls}})}$ and $\tilde{\boldsymbol{\zeta}}(0) = \boldsymbol{\phi}^{(s_{\text{cls}})}$. By Theorem K.4, we have

$$
\max_{n=0,\cdots,\lfloor T/\eta^{0.75} \rfloor} \left| \mathbb{E}[g(\boldsymbol{\phi}^{(nR_{\text{grp}}+s_{\text{cls}})}) - g(\tilde{\boldsymbol{\zeta}}(n\eta^{0.75})) \mid \boldsymbol{\phi}^{(s_{\text{cls}})}, \mathcal{E}_0^{(s_{\text{cls}})}] \right| \leq C_g \eta^{0.25}(\log \tfrac{1}{\eta})^b,
$$

where $R_{\text{grp}} = \lfloor \frac{1}{\alpha\eta^{0.75}} \rfloor$. Noticing that (i) $g \in \mathcal{C}^3$ (ii) $\boldsymbol{b}, \boldsymbol{\sigma} \in \mathcal{C}^{\infty}$ and (iii) $\boldsymbol{\zeta}(t), \tilde{\boldsymbol{\zeta}}(t) \in \Gamma, t \in [0, \infty)$ almost surely, we can conclude that given $\mathcal{E}_0^{(s_{\text{cls}})}$,

$$
\|\boldsymbol{\zeta}(t) - \tilde{\boldsymbol{\zeta}}(t)\|_2 = \tilde{\mathcal{O}}(\sqrt{\eta}), \quad \forall t \in [0, T].
$$

Then there exists positive constant $b'$ independent of $\eta$ and $g$, and $C_g'$ which is independent of $\eta$ but can depend on $g$ such that

$$
\max_{n=0,\cdots,\lfloor T/\eta^{0.75} \rfloor} \left| \mathbb{E}[g(\boldsymbol{\phi}^{(nR_{\text{grp}}+s_{\text{cls}})}) - g(\boldsymbol{\zeta}(n\eta^{0.75} + s_{\text{cls}}H\eta^2))] \right| \leq C_g' \eta^{0.25}(\log \tfrac{1}{\eta})^{b'}.
$$

We can view the random variable pairs $\{(\boldsymbol{\phi}^{(nR_{\text{grp}}+s_{\text{cls}})}, \boldsymbol{\zeta}_{n\eta^{0.75}+s_{\text{cls}}\alpha\eta}) : n = 0, \cdots, \lfloor T/\eta^{0.75} \rfloor\}$ as reference points and then approximate the value of $g(\boldsymbol{\phi}^{(s)})$ and $g(\boldsymbol{\zeta}(sH\eta^2))$ with the value at the nearest reference points. By Lemmas K.18 and K.23, for $0 \leq r \leq R_{\text{grp}}$ and $0 \leq s \leq R_{\text{tot}} - r$,

$$
\mathbb{E}[\|\boldsymbol{\phi}^{(s+r)} - \boldsymbol{\phi}^{(s)}\|_2] = \tilde{\mathcal{O}}(\eta^{0.375}).
$$

Since the values of $\boldsymbol{\phi}^{(s)}$ and $\boldsymbol{\zeta}$ are restricted to a bounded set, $g(\cdot)$ is Lipschitz on that set. Therefore, we have the theorem. $\qquad\square$

## L  DERIVING THE SLOW SDE FOR LABEL NOISE REGULARIZATION

In this section, we formulate how label noise regularization works and provide a detailed derivation of the theoretical results in Appendix G.

Consider training a model for $C$-class classification on dataset $\mathcal{D} = \{(\boldsymbol{x}_i, y_i)\}_{i=1}^N$, where $\boldsymbol{x}_i$ denotes the input and $y_i \in [C]$ denotes the label. Denote by $\Delta_+^{C-1}$ the $(C-1)$-open simplex. Let $f(\boldsymbol{\theta}; \boldsymbol{x}) \in \Delta_+^{C-1}$ be the model output on input $\boldsymbol{x}$ with parameter $\boldsymbol{\theta}$, whose $j$-th coordinate $f_j(\boldsymbol{\theta}; \boldsymbol{x})$ stands for the probability of $\boldsymbol{x}$ belonging to class $j$. Let $\ell(\boldsymbol{\theta}; \boldsymbol{x}, y)$ be the cross entropy loss given input $\boldsymbol{x}$ and label $y$, i.e, $\ell(\boldsymbol{\theta}; \boldsymbol{x}, y) = -\log f_y(\boldsymbol{\theta}; \boldsymbol{x})$.

Adding label noise means replacing the true label $y$ with a fresh noisy label $\hat{y}$ every time we access the sample. Specifically, $\hat{y}$ is set as the true label $y$ with probability $1 - p$ and as any other label with probability $\frac{p}{C-1}$, where $p$ is the fixed corruption probability. The training loss is defined as $\mathcal{L}(\boldsymbol{\theta}) = \frac{1}{N} \sum_{i=1}^N \mathbb{E}[\ell(\boldsymbol{\theta}; \boldsymbol{x}_i, \hat{y}_i)]$, where the expectation is taken over the stochasticity of $\hat{y}_i$. Notice that given a sample $(\boldsymbol{x}, y)$,

$$\mathbb{E}[\ell(\boldsymbol{\theta}; \boldsymbol{x}, \hat{y})] = -(1-p) \log f_y(\boldsymbol{\theta}; \boldsymbol{x}) - \frac{p}{C-1} \sum_{j \neq y} \log f_j(\boldsymbol{\theta}; \boldsymbol{x}). \tag{114}$$

By the property of cross-entropy loss, (114) attains its global minimum if and only if $f_j = \frac{p}{C-1}$, for all $j \in [C], j \neq y$ and $f_y = 1 - p$. Due to the large expressiveness of modern deep learning models, there typically exists a set $\mathcal{S}^* := \{\boldsymbol{\theta} \mid f_i(\boldsymbol{\theta}) = \mathbb{E}[\hat{y}_i], \forall i \in [N]\}$ such that all elements of $\mathcal{S}^*$ minimizes $\mathcal{L}(\boldsymbol{\theta})$. Then, the manifold $\Gamma$ is a subset of $\mathcal{S}^*$. The following lemma relates the noise covariance $\boldsymbol{\Sigma}(\boldsymbol{\theta}) := \frac{1}{N} \sum_{i \in [N]} \mathbb{E}[(\nabla \ell(\boldsymbol{\theta}; \boldsymbol{x}_i, \hat{y}_i) - \nabla \mathcal{L}(\boldsymbol{\theta}))(\nabla \ell(\boldsymbol{\theta}; \boldsymbol{x}_i, \hat{y}_i) - \nabla \mathcal{L}(\boldsymbol{\theta}))^\top]$ to the hessian $\nabla^2 \mathcal{L}(\boldsymbol{\theta})$ for all $\boldsymbol{\theta} \in \mathcal{S}^*$.

**Lemma L.1.** *If $f(\boldsymbol{\theta}; \boldsymbol{x}_i, \hat{y}_i)$ is $\mathcal{C}^2$-smooth on $\mathbb{R}^d$ given any $i \in [N]$, $\hat{y}_i \in [C]$ and $\mathcal{S}^* \neq \varnothing$, then for all $\boldsymbol{\theta} \in \mathcal{S}^*$, $\boldsymbol{\Sigma}(\boldsymbol{\theta}) = \nabla^2 \mathcal{L}(\boldsymbol{\theta})$.*

*Proof.* Since $\mathcal{L}(\cdot)$ is $\mathcal{C}_2$-smooth, $\nabla \mathcal{L}(\boldsymbol{\theta}) = \boldsymbol{0}$ for all $\boldsymbol{\theta} \in \mathcal{S}^*$. To prove the above lemma, it suffices to show that $\forall i \in [N]$, $\mathbb{E}[\nabla \ell(\boldsymbol{\theta}; \boldsymbol{x}_i, \hat{y}_i) \nabla \ell(\boldsymbol{\theta}; \boldsymbol{x}_i, \hat{y}_i)^\top] = \nabla^2 \mathcal{L}(\boldsymbol{\theta})$. W.L.O.G, let $y = 1$ and therefore for all $\boldsymbol{\theta} \in S^*$,

$$f_1(\boldsymbol{\theta}; \boldsymbol{x}) = 1 - p =: a_1,$$
$$f_j(\boldsymbol{\theta}; \boldsymbol{x}) = \frac{p}{C-1} =: a_2, \forall j > 1, j \in [C].$$

Additionally, let $h(x) := -\log(x), x \in \mathbb{R}^+$. The stochastic gradient $\nabla \ell(\boldsymbol{\theta}; \boldsymbol{x}, \hat{y})$ follows the distribution:

$$\nabla \ell(\boldsymbol{\theta}; \boldsymbol{x}, \hat{y}) = \begin{cases} h'(a_1) \frac{\partial f_1}{\partial \boldsymbol{\theta}} & \text{w.p. } 1 - p, \\ h'(a_2) \frac{\partial f_j}{\partial \boldsymbol{\theta}}, & \text{w.p. } \frac{p}{C-1}, \forall j \in [C], j > 1. \end{cases}$$

Then the covariance of the gradient noise is:

$$\mathbb{E}[\nabla \ell(\boldsymbol{\theta}; \boldsymbol{x}, \hat{y}) \nabla \ell(\boldsymbol{\theta}; \boldsymbol{x}, \hat{y})^\top] = (1-p)(h'(a_1))^2 \frac{\partial f_1(\boldsymbol{\theta}^*)}{\partial \boldsymbol{\theta}^*} \left( \frac{\partial f_1(\boldsymbol{\theta}^*)}{\partial \boldsymbol{\theta}^*} \right)^\top$$
$$+ \frac{p(h'(a_2))^2}{C-1} \sum_{j>1} \frac{\partial f_j(\boldsymbol{\theta}^*)}{\partial \boldsymbol{\theta}^*} \left( \frac{\partial f_j(\boldsymbol{\theta}^*)}{\partial \boldsymbol{\theta}^*} \right)^\top.$$

And the hessian is:

$$\nabla^2 \mathcal{L}(\boldsymbol{\theta}) = \underbrace{(1-p)h'(a_1) \frac{\partial^2 f_1}{\partial \boldsymbol{\theta}^2} + \frac{ph'(a_2)}{C-1} \sum_{j>1} \frac{\partial^2 f_j}{\partial \boldsymbol{\theta}^2}}_{\mathcal{T}}$$
$$+ (1-p)h''(a_1) \frac{\partial f_1}{\partial \boldsymbol{\theta}} \left( \frac{\partial f_1}{\partial \boldsymbol{\theta}} \right)^\top + \frac{ph''(a_2)}{C-1} \sum_{j>1} \frac{\partial f_j}{\partial \boldsymbol{\theta}} \left( \frac{\partial f_j(\boldsymbol{\theta})}{\partial \boldsymbol{\theta}} \right)^\top.$$

Since $\sum_{j \in [C]} f_i = 1$,

$$\frac{\partial^2 f_1}{\partial \boldsymbol{\theta}^2} = -\sum_{j>1} \frac{\partial^2 f_j}{\partial \boldsymbol{\theta}^2}. \tag{115}$$

Also, notice that $h'(x) = -\frac{1}{x}$. Therefore,

$$(1-p)h'(a_1) = \frac{ph'(a_2)}{C-1}. \tag{116}$$

Substituting (115) and (116) into the expression of $\mathcal{T}$ gives $\mathcal{T} = \mathbf{0}$, which simplifies $\nabla^2 \mathcal{L}(\boldsymbol{\theta})$ as the following form:

$$\nabla^2 \mathcal{L}(\boldsymbol{\theta}) = (1-p)h''(a_1)\frac{\partial f_1}{\partial \boldsymbol{\theta}} \left(\frac{\partial f_j(\boldsymbol{\theta})}{\partial \boldsymbol{\theta}}\right)^\top + \frac{ph''(a_2)}{C-1}\sum_{j>1}\frac{\partial f_j}{\partial \boldsymbol{\theta}} \left(\frac{\partial f_j(\boldsymbol{\theta})}{\partial \boldsymbol{\theta}}\right)^\top.$$

Again notice that $h''(x) = h'(x)$ for all $x \in \mathbb{R}^+$. Therefore, $\nabla^2 \mathcal{L}(\boldsymbol{\theta}) = \boldsymbol{\Sigma}(\boldsymbol{\theta})$. □

With the property $\boldsymbol{\Sigma}(\boldsymbol{\theta}) = \nabla^2 \mathcal{L}(\boldsymbol{\theta})$, we are ready to prove Theorem G.1.

*Proof of Theorem G.1.* Recall the general form of the slow SDE:

$$d\boldsymbol{\zeta}(t) = \frac{1}{\sqrt{B}}\partial \Phi(\boldsymbol{\zeta})\boldsymbol{\Sigma}^{1/2}(\boldsymbol{\zeta})d\boldsymbol{W}(t) + \frac{1}{2B}\partial^2 \Phi(\boldsymbol{\zeta})\left[\boldsymbol{\Sigma}(\boldsymbol{\zeta}) + (K-1)\boldsymbol{\Psi}(\boldsymbol{\zeta})\right]dt, \tag{117}$$

where $\boldsymbol{\Psi}$ is defined in Definition K.6. Since for $\boldsymbol{\zeta} \in \Gamma$, $\boldsymbol{\Sigma}(\boldsymbol{\zeta}) = \nabla^2 \mathcal{L}(\boldsymbol{\zeta})$, then

$$\partial \Phi(\boldsymbol{\zeta})\boldsymbol{\Sigma}^{1/2}(\boldsymbol{\zeta}) = \mathbf{0}. \tag{118}$$

Now we show that

$$\partial^2 \Phi(\boldsymbol{\zeta})[\boldsymbol{\Sigma}(\boldsymbol{\zeta})] = -\nabla_\Gamma \text{tr}(\nabla^2 \mathcal{L}(\boldsymbol{\zeta})). \tag{119}$$

Since $\nabla^2 \mathcal{L}(\boldsymbol{\zeta}) = \boldsymbol{\Sigma}(\boldsymbol{\zeta})$, $\mathcal{V}_{\nabla^2 \mathcal{L}(\boldsymbol{\zeta})}[\boldsymbol{\Sigma}] = \frac{1}{2}\boldsymbol{I}$. By Lemma K.4,

$$\partial^2 \Phi(\boldsymbol{\zeta})[\boldsymbol{\Sigma}(\boldsymbol{\zeta})] = -\frac{1}{2}\partial \Phi(\boldsymbol{\zeta})\nabla^3 \mathcal{L}(\boldsymbol{\zeta})[\boldsymbol{I}] = -\frac{1}{2}\nabla_\Gamma \text{tr}(\nabla^2 \mathcal{L}(\boldsymbol{\zeta})).$$

Finally, we show that

$$\partial^2 \Phi(\boldsymbol{\zeta})[\boldsymbol{\Psi}(\boldsymbol{\zeta})] = -\nabla_\Gamma \frac{1}{2H\eta}\text{tr}(F(2H\eta\nabla^2 \mathcal{L}(\boldsymbol{\zeta}))). \tag{120}$$

Define $\hat{\psi}(x) := x\psi(x) = e^{-x} - 1 + x$. By definition of $\boldsymbol{\Psi}(\boldsymbol{\zeta})$, when $\boldsymbol{\Sigma}(\boldsymbol{\zeta}) = \nabla^2 \mathcal{L}(\boldsymbol{\zeta})$, $\boldsymbol{\Psi}(\boldsymbol{\zeta}) = \hat{\psi}(2\eta H\nabla^2 \mathcal{L}(\boldsymbol{\zeta}))$, where $\hat{\psi}(\cdot)$ is interpreted as a matrix function. Since $\psi(2\eta H\nabla^2 \mathcal{L}(\boldsymbol{\zeta})) \in \text{span}\{\boldsymbol{u}\boldsymbol{u}^\top \mid \boldsymbol{u} \in T_{\boldsymbol{\zeta}}^\perp(\Gamma)\}$, by Lemma K.4,

$$\partial^2 \Phi(\boldsymbol{\zeta})[\boldsymbol{\Psi}(\boldsymbol{\zeta})] = -\frac{1}{2}\partial \Phi(\boldsymbol{\zeta})\text{tr}\psi(2\eta H\nabla^2 \mathcal{L}(\boldsymbol{\zeta})).$$

By the chain rule, we have (120). Combining (118),(119) and (120) gives the theorem. □

# M EXPERIMENTAL DETAILS

In this section, we specify the experimental details that are omitted in the main text. Our experiments are conducted on CIFAR-10 (Krizhevsky et al., 2009) and ImageNet Russakovsky et al. (2015). Our code is available at `https://github.com/hmgxr128/Local-SGD`. Our implementation of ResNet-56 (He et al., 2016) and VGG-16 (Simonyan & Zisserman, 2015) is based on the high-starred repository by Wei Yang[2] and we use the implementation of ResNet-50 from torchvision 0.3.1. We run all CIFAR-10 experiments with $B_{\text{loc}} = 128$ on 8 NVIDIA Tesla P100 GPUs while ImageNet experiments are run on 8 NVIDIA A5000 GPUS with $B_{\text{loc}} = 32$. All ImageNet experiments are trained with ResNet-50.

We generally adopt the following training strategies. We do not add any momentum unless otherwise stated. We follow the suggestions by Jia et al. (2018) and do not add weight decay to the bias and learnable parameters in the normalization layers. For all models with BatchNorm layers, we go through 100 batches of data with batch size $B_{\text{loc}}$ to estimate the running mean and variance before evaluation. Experiments on both datasets follow the standard data augmentation pipeline in He et al. (2016) except the label noise experiments. Additionally, we use FFCV (Leclerc et al., 2022) to accelerate data loading for ImageNet training.

Slightly different from the update rule of Local SGD in Section 1, we use sampling without replacement unless otherwise stated. See Appendix C for implementation details and discussion.

## M.1 POST-LOCAL SGD EXPERIMENTS IN SECTION 1

**CIFAR-10 experiments.** We simulate 32 clients with $B = 4096$. We follow the linear scaling rule and linear learning rate warmup strategy suggested by Goyal et al. (2017). We first run 250 epochs of SGD with the learning rate gradually ramping up from 0.1 to 3.2 for the first 50 epochs. Resuming from the model obtained at epoch 250, we run Local SGD with $\eta = 0.32$. Note that we conduct grid search for the initial learning rate among $\{0.005, 0.01, 0.05, 0.1, 0.15, 0.2\}$ and choose the learning rate with which parallel SGD ($H = 1$) achieves the best test accuracy. We also make sure that the optimal learning rate resides in the middle of the set. The weight decay $\lambda$ is set as $5 \times 10^{-4}$. As for the initialization scheme, we follow Lin et al. (2020b) and Goyal et al. (2017). Specifically, we use Kaiming Normal (He et al., 2015) for the weights of convolutional layers and initialize the weights of fully-connected layers by a Gaussian distribution with mean zero and standard deviation 0.01. The weights for normalization layers are initialized as one. All bias parameters are initialized as zero. We report the mean and standard deviation over 5 runs.

**ImageNet experiments.** We simulate 256 workers with $B = 8192$. We follow the linear scaling rule and linear learning rate warmup strategy suggested by Goyal et al. (2017). We first run 100 epochs of SGD where the learning rate linearly ramps up from 0.5 to 16 for the first 5 epochs and then decays by a factor of 0.1 at epoch 50. Resuming from epoch 100, we run Local SGD with $\eta = 0.16$. Note that we conduct grid search for the initial learning rate among $\{0.05, 0.1, 0.5, 1\}$ and choose the learning rate with which parallel SGD ($H = 1$) achieves the best test accuracy. We also make sure that the optimal learning rate resides in the middle of the set. The weight decay $\lambda$ is set as $1 \times 10^{-4}$ and we do not add any momentum. The initialization scheme follows the implementation of torchvision 0.3.1. We report the mean and standard deviation over 3 runs.

## M.2 EXPERIMENTAL DETAILS FOR FIGURES 2 AND 5

**CIFAR-10 experiments.** We use ResNet-56 for all CIFAR-10 experiments in the two figures. We simulate 32 workers with $B = 4096$ and set the weight decay as $5 \times 10^{-4}$. For Figures 2(a) and 2(b), we set $\eta = 0.32$, which is the same as the learning rate after decay in Figure 1(a). For Figure 2(a), we adopt the same initialization scheme introduced in the corresponding paragraph in Appendix M.1. For Figures 2(b), 2(e) and 5(c), we use the model at epoch 250 in Figure 1(a) as the pre-trained model. Additionally, we use a training budget of 250 epochs for Figure 2(e). In Figure 5(e), we use Local SGD with momentum 0.9, where the momentum buffer is kept locally and never averaged. We run SGD with momentum 0.9 for 150 epochs to obtain the pre-trained model, where the learning

---

[2]https://github.com/bearpaw/pytorch-classification

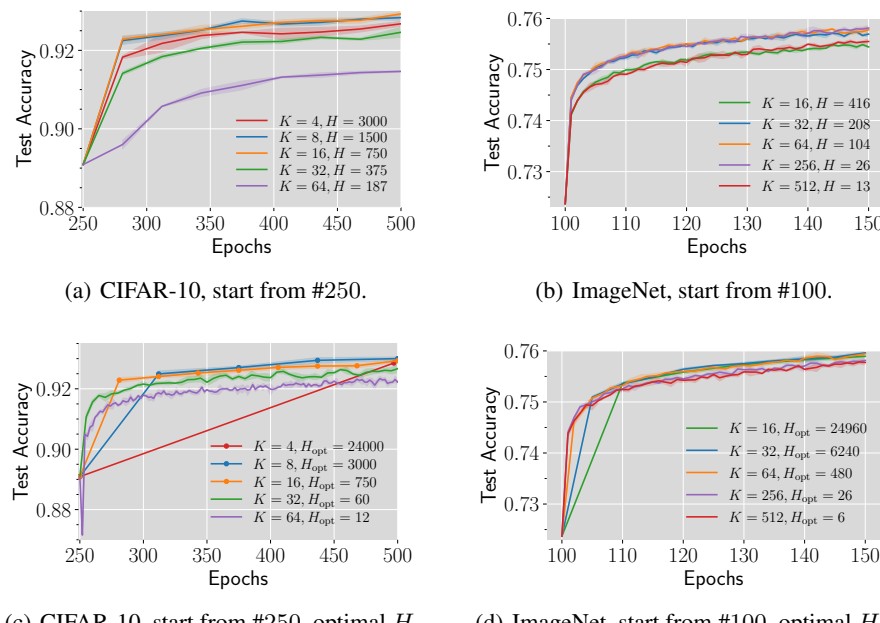

Figure 10: The learning curves for experiments in Figure 4.

rate ramps up from 0.05 to 1.6 linearly in the first 150 epochs. Note that we conduct grid search for the initial learning rate among $\{0.01, 0.05, 0.1, 0.15, 0.2\}$ and choose the learning rate with which parallel SGD ($H = 1$) achieves the highest test accuracy. We also make sure that the optimal learning rate resides in the middle of the set. Resuming from epoch 150, we run Local SGD $H = 1$ (i.e., SGD) and 24 with $\eta = 0.16$ and decay $\eta$ by 0.1 at epoch 226. For Local SGD $H = 900$, we resume from the model at epoch 226 of $H = 24$ with $\eta = 0.016$. We report the mean and standard deviation over 3 runs for Figures 2(a), 2(b) and 5(c), and over 5 runs for Figure 2(e).

**ImageNet experiments.** We simulate 256 clients with $B = 8192$ and set the weight decay as $1 \times 10^{-4}$. In Figure 2(d), both Local SGD and SGD start from the same random initialization. We warm up the learning rate from 0.1 to 3.2 in the first 5 epochs and decay the learning rate by a factor of 0.1 at epochs 50 and 100. For Figures 2(c), 2(f) and 5(d), we use the model at epoch 100 in Figure 1(b) as the pre-trained model. In Figure 2(c), we set the learning rate as 0.16, which is the same as the learning rate after epoch 100 in Figure 1(b). Finally, in Figures 2(c), 2(f), 5(b) and 5(d), we report the mean and average over 3 runs.

## M.3 DETAILS FOR EXPERIMENTS IN FIGURE 6

For all experiments in Figure 6, we train a ResNet-56 model on CIFAR-10. We report mean test accuracy over three runs and the shaded area reflects the standard deviation. For Figure 6(a), we use the same setup as Figures 2(a) and 2(b) for training from random initialization and from a pre-trained model respectively except the learning rate. For Figure 6(b), we resume from the model obtained at epoch 250 in Figure 1(a) and train for another 250 epochs. For Figure 6(c), we follow the same procedure as Figure 1(a) except that we use sampling with replacement. We also ensure that the total numbers of iterations in Figures 1(a) and 6(c) are the same.

## M.4 DETAILS FOR EXPERIMENTS ON THE EFFECT OF THE DIFFUSION TERM

**CIFAR-10 experiments.** The model we use is ResNet-56. For Figure 3(a), we first run SGD with batch size 128 and learning rate $\eta = 0.5$ for 250 epochs to obtain the pre-trained model. The initialization scheme is the same as the corresponding paragraph in Appendix M.1. Resuming from epoch 250 with $\eta = 0.05$, we run Local SGD with $K = 16$ until epoch 6000 and run all other setups for the same number of iterations. We report the mean and standard deviation over 3 runs.

**ImageNet experiments.** For Figures 3(b) and 4(b), we start from the model obtained at epoch 100 in Figure 1(b). In Figure 3(b), we run Local SGD with $K = 256$ for another 150 epochs with $\eta = 0.032$. We run all other setups for the same number of iterations with the same learning rate.

## M.5   Details for Experiments on the Effect of Global Batch Size

**CIFAR-10 experiments.**   The model we use is ResNet-56. We resume from the model obtained in Figure 1(a) at epoch 250 and train for another 250 epochs. The local batch size for all runs is $B_{\text{loc}} = 128$. We first make grid search of $\eta$ for SGD with $K = 16$ among $\{0.04, 0.08, 0.16, 0.32, 0.64\}$ and find that the final test accuracy varies little across different learning rates (within $0.1\%$). Then we choose $\eta = 0.32$. For the green curve in Figure 4(a), we search for the optimal $H$ for $K = 16$ and keep $\alpha$ fixed when scaling $\eta$ with $K$. For the red curve in Figure 4(a), we search for the optimal $H$ for each $K$ among $\{6, 12, 60, 120, 300, 750, 1500, 3000, 6000, 12000, 24000\}$ and also make sure that $H$ does not exceed the total number of iterations for 250 epochs. The learning curves for constant and optimal $\alpha$ are visualized in Figures 10(a) and 10(c) respectively. We report the mean and standard deviation over three runs.

**ImageNet experiments.**   We start from the model obtained at epoch 100 in Figure 1(b) and train for another 50 epochs. The local batch size for all runs is $B_{\text{loc}} = 32$. We first make grid search among $\{0.032, 0.064, 0.16, 0.32\}$ for $H = 1$ to achieve the best test accuracy and choose $H = 0.064$. For the orange curve in Figure 4(b), we search $H$ among $\{2, 4, 6, 13, 26, 52, 78, 156\}$ for $K = 256$ to achieve the optimal test accuracy and the keep $\alpha$ constant as we scale $\eta$ with $K$. To obtain the optimal $H$ for each $K$, we search among $\{6240, 7800, 10400, 12480, 15600, 20800, 24960, 31200\}$ for $K = 16$, $\{1600, 3120, 4160, 5200, 6240, 7800, 10400\}$ for $K = 32$, $\{312, 480, 520, 624, 800, 975, 1040, 1248, 1560, 1950\}$ for $K = 64$, and $\{1, 2, 3, 6, 13\}$ for $K = 512$. The learning curves for constant and optimal $\alpha$ are visualized in Figures 10(b) and 10(d) respectively. We report the mean and standard deviation over three runs.

## M.6   Details for Experiments on Label Noise Regularization

For all label noise experiments, we do not use data augmentation, use sampling with replacement, and set the corruption probability as $0.1$. We simulate 32 workers with $B = 4096$ in Figure 7 and 4 workers with $B = 512$ in Figure 8. We use ResNet-56 with GroupNorm with the number of groups 8 for Figure 7(a) and VGG-16 without normalization for Figures 7(b) and 8. Below we list the training details for ResNet-56 and VGG-16 respectively.

**ResNet-56.**   As for the model architecture, we replace the batch normalization layer in Yang's implementation with group normalization such that the training loss is independent of the sampling order. We also use Swish activation (Ramachandran et al., 2017) in place of ReLU to ensure the smoothness of the loss function. We generate the pre-trained model by running label noise SGD with corruption probability $p = 0.1$ for 500 epochs ($6,000$ iterations). We initialize the model by the same strategy introduced in the first paragraph of Appendix M.1. Applying the linear warmup scheme proposed by Goyal et al. (2017), we gradually ramp up the learning rate $\eta$ from $0.1$ to $3.2$ for the first 20 epochs and multiply the learning rate by $0.1$ at epoch 250. All subsequent experiments in Figure 7(a) (a) use learning rate $0.1$. The weight decay $\lambda$ is set as $5 \times 10^{-4}$ . Note that adding weight decay in the presence of normalization accelerates the limiting dynamics and will not affect the implicit regularization on the original loss function (Li et al., 2022).

**VGG-16.**   We follow Yang's implementation of the model architecture except that we replace maximum pooling with average pooling and use Swish activation (Ramachandran et al., 2017) to make the training loss smooth. We initialize all weight parameters by Kaiming Normal and all bias parameters as zero. The pre-trained model is obtained by running label noise SGD with total batch size 4096 and corruption probability $p = 0.1$ for 6000 iterations. We use a linear learning rate warmup from $0.1$ to $0.5$ in the first 500 iterations. All runs in Figures 7(b) and 8 resume from the model obtained by SGD with label noise. In Figure 7(b), we use learning rate $\eta = 0.1$. In Figure 8, we set $\eta = 0.005$ for $H = 97,000$ and $\eta = 0.01$ for SGD ($H = 1$). The weight decay $\lambda$ is set as zero.

