# OpenReview forum: "Why (and When) does Local SGD Generalize Better than SGD?"
_ICLR.cc/2023/Conference — ICLR 2023 poster_

### Official Review · Reviewer_QgEL · 2022-10-24

**Confidence:** 4
**Correctness:** 3
**Technical Novelty And Significance:** 3
**Empirical Novelty And Significance:** 2
**Recommendation:** 6

**Clarity, Quality, Novelty And Reproducibility:**

The paper's writing is fine except for some unsupported or confusing statements:

>The local batches on different workers are independent with each other as there is no communication.

1. They would have been independent even if there was communication, i.e., $H=1$. The independence comes from sampling in the model described in section 1, not from lack of communication.

>All the above papers agree that local SGD* generalizes better than SGD to some extent.

2. *Post-local SGD, not local SGD. One explicit take-away from [Lin et al.](https://arxiv.org/pdf/1808.07217.pdf) is that Post-local SGD is better than both Local SGD and mini-batch SGD. Perhaps this confusion arises because the authors switch between the perspectives of pre-training and switching optimizers. Both are reasonable perspectives, but the writing should be consistent. I understand that some of the experiments in this paper (figure 2) claim that pre-training is not essential to show the benefit of local SGD. Thus the difference between local and post-local SGD is not significant. But as I mention above, there are issues with how hyper-parameters are tuned, making this conclusion questionable.

>Simultaneously requiring a small learning rate and sufficient training time poses a trade-off when learning rate decay is used with a limited training budget: switching to Local SGD earlier may lead to a large learning rate, while switching later may result in insufficient training time.

3. This sentence is confusing. Perhaps replacing the phrase "insufficient training time" with something like "local-SGD makes fewer steps making the generalization-improvement less noticeable" should make it clearer.


The experiments appear reproducible, as the setup is similar to the cited papers. Some of the theoretical techniques are novel and could be of independent interest. I encourage the authors to highlight this further.

**Strength And Weaknesses:**

#### **Unfair hyper-parameter tuning**
The paper almost treats the step size as a part of the problem and not a part of the algorithm/optimizer. This is why the authors make unfair comparisons and come up with questionable conclusions. For instance, why should one consider the same step size while changing the number of local steps in figure 1? It is well understood that at least in the convex setting, the optimal step size for local SGD inversely depends on the number of local steps $H$ (see the [optimal rate for local SGD]((https://arxiv.org/abs/2111.03741)) in the homogeneous setting in [this paper](https://arxiv.org/abs/2002.07839)).

The relationship might be different in the non-convex setting (as hinted by figure 2(e)-(f)). Thus the correct thing to do is to tune the step size separately for each $H$. Most of the experiments in [Lin et al.](https://arxiv.org/pdf/1808.07217.pdf) use the step-size schedule of dense/large mini-batch SGD for local SGD, thus giving mini-batch SGD the benefit of the doubt. The authors seem to be doing the opposite here, and I conjecture that dense/large mini-batch SGD will be better than the local SGD baselines in figure 2(a) if it uses the correct hyper-parameters. I also expect that $H=1$ might be optimal for larger step sizes in figures 2(e)-(f).

Ideally, the authors must first fix a training budget and tune the step size to obtain the best validation loss/accuracy for the "final model" of each optimizer **separately**. This would make the optimal step size a function of $H$ and the training budget. Finally, it is important to clarify the stopping criterion for optimization; I couldn't find it in the appendix.


#### **Missing comparison to Large Mini-batch SGD**
There is a lot of work trying to understand the optimization/generalization properties of local SGD in the convex setting. But the theory has been disappointing in showing the benefit of local SGD over large mini-batch SGD and single-machine SGD ([Woodworth et al.](https://arxiv.org/abs/2102.01583)). In fact, in the heterogeneous convex setting, which is the setting of the experiments, large mini-batch SGD is almost always better than local SGD [Woodworth et al.](https://arxiv.org/abs/2006.04735). Thus it is important to include large mini-batch SGD in all the experiments and use the correct step size for it (usual [scaling rules often fail](https://arxiv.org/abs/1811.03600)). The relevant setting, then, is the one where post-local SGD is better than both variants of mini-batch SGD. I expect that for small $H$, large-mini-batch SGD would be the correct algorithm to use, but for very large $H$ post-local SGD would be better. It is also important to compare against the SDE approximation of large mini-batch SGD in section 3. However, it might be challenging because the larger optimal step size might hurt the approximation error and make the comparison with the slow-SDE inconclusive. In particular, note that when $\eta$ is very small local SGD is expected to behave as large mini-batch SGD. Can the authors comment on this theoretically from the SDE perspective?


#### **Inconsistent optimization setup**
The optimization setup introduced in section 1 and used throughout section 3 is that of stochastic optimization. However, all the experiments are performed using SGD with multiple passes, i.e., sampling without replacement, along with hyper-parameter tuning on the validation data set. Furthermore, the data is split between the machines for parallel training. While this is fine for the mini-batch algorithms, it introduces heterogeneity in client updates for local SGD; check [this](https://arxiv.org/abs/2006.04735) and [this](https://arxiv.org/abs/2206.04723). For a large number of local steps, this could cause a [client drift](http://proceedings.mlr.press/v130/charles21a.html) and have a regularization effect for local SGD. Several conflating factors make it hard to reconcile the experiments with each other and the theoretical guarantees. To make things worse, the writing doesn't acknowledge these nuances in optimization.

> Given a sufficiently small learning rate and a sufficiently long training time, Local SGD exhibits better generalization than SGD if the number of local steps H per round is tuned correctly according to the learning rate. This holds for both training from random initialization and from pre-trained models.

In light of the above comments, the above finding, which seems to be the paper's main contribution, is a bit vague. It is **not interesting** to identify which hyper-parameters make local SGD look good. Ideally, different optimizers should be compared after "convergence," as the main goal is generalization here. Even if the training budget is fixed apriori and the authors want to consider the effect of different training budgets, they must change the step sizes accordingly. The interesting question is why post-local SGD generalizes better than mini-batch SGD, even if it potentially uses a non-optimal step size ([Lin et al.](https://arxiv.org/pdf/1808.07217.pdf)). The paper does not answer this question.

#### **Highlight connections to the optimization literature**
As mentioned above, several connections to the optimization literature about local SGD must be mentioned in the main paper. This includes observations in figure 2(e)-(f), the client drift in the heterogeneous setting, min-max optimal algorithms in the homogeneous setting, etc. Note that many of these papers are generalization guarantees as they directly optimize on distribution $\tilde{\mathcal{D}}$, using stochastic first-order oracles and thus offer similar insights as in section 3.

#### **Theoretical results**
The theoretical results and some of the developed techniques are novel. But the authors should improve the writing in section three, as the main result of the section doesn't come off clearly.

I have already mentioned the missing comparison to large mini-batch SGD. I expect that for very small $\eta = O(1/H)$, local-SGD and large mini-batch SGD should behave similarly, as the local gradients are all computed at very similar points (for context see [this paper](https://arxiv.org/pdf/1910.06378.pdf)). Thus, the interesting regime is where local SGD (or its approximation) can be shown to improve over both variants of mini-batch SGD. Can the current theoretical results even highlight such a regime because $\eta = \theta(1/H)$ in theorem 3.2? Also, what can the authors conclude about large mini-batch SGD, assuming hypothesis 3.1?

There are two levels of approximation in section 3.2, one more approximation leading up to (9), and these might not hold very well for the optimal step size for local SGD, which can be larger than $1/H$ (at least in the convex setting). Even If the optimal step size is smaller than $1/H$, then large mini-batch SGD might have comparable performance. On top of this, the generalization benefit would only hold in the regime where hypothesis 3.1 is correct. Thus it is a bit hard to make sense of the theoretical result here. The writing will benefit from a discussion of these different regimes.

Finally, I'm not too fond of using the section to explain finding 2.1 (due to the above reasons). The theoretical results would be more interesting as a standalone contribution after more polishing.





**Summary Of The Paper:**

**Note**: Throughout the review, I will make a distinction between dense mini-batch SGD, i.e., mini-batch SGD with batch size $B$ and large mini-batch SGD with batch size $B\cdot H$, where $H$ is the number of local steps, for the local-SGD algorithm with the exact computation and communication cost, and $B$ is the baseline batch-size used in this paper. Thus, dense mini-batch SGD is the same as parallel SGD in this paper's nomenclature.

**Summary**: The paper studies the generalization behavior of Local SGD and its variant Post-local SGD ([Lin et al.](https://arxiv.org/pdf/1808.07217.pdf)). Specifically, it tries to understand and justify the better generalization performance of post-local SGD over dense mini-batch SGD. This problem is interesting, and while previous work explored this question, there is no satisfying, empirical, or theoretical explanation.

Authors first reproduce some experiments from previous works ([Lin et al.](https://arxiv.org/pdf/1808.07217.pdf), [Ortiz et al.](https://arxiv.org/abs/2110.08133?context=cs.CV)) showing that Post-local SGD, indeed has a generalization benefit over dense mini-batch SGD on CIFAR-10 and ImageNet. Then it compares local-SGD runs with different local steps but the same constant learning rate and shows that both local SGD and post-local SGD have a generalization benefit over dense mini-batch SGD if the learning rate is small enough. Further, the authors show that the optimal learning rate for local SGD depends on the number of local steps $H$. This observation is not surprising, given the optimization literature on local SGD in both convex and non-convex settings. Finally, the paper presents some theoretical results using the stochastic differential approximation of local SGD to justify the empirical behavior.


**Summary Of The Review:**

The paper tries to answer an interesting question using new empirical and theoretical insights. However, I believe the comparisons between different optimizers are unfair due to incorrect hyperparameter tuning. As a result, the paper doesn't study what it claims to study. Some important baselines are also missing from the experiments. The theoretical result is not a standalone contribution but is used to explain the empirical findings. Since the finding seems less valuable, it is also unclear if the developed theory helps understand local SGD any better. More importantly, there is a mismatch between the settings of the experiments and the theory. The theoretical techniques might be of independent interest, but they don't constitute the paper's main contribution, the way it is written currently. Thus, I don't recommend accepting the work in its current form.

---

> ### Author Response · Authors · 2022-11-15
> **Response to Reviewer QgEL**
>
> We sincerely thank the reviewer for acknowledging that our theoretical results and techniques are novel. We also thank the reviewer for helpful suggestions on writing and for pointing out some missing references. We have polished our draft according to the suggestions.
>
> Below we first address the reviewer’s main questions on hyperparameter tuning and then address the other concerns. We follow the reviewer's terminology: dense mini-batch SGD is parallel SGD with batch size $B$, and large mini-batch SGD is parallel SGD with batch size $B \cdot H$, where $H$ is the number of local steps.
>
> **Main Question 1**: The comparisons between Local SGD and dense mini-batch SGD (equivalent to Local SGD with $H=1$) are unfair because the learning rate (LR) is fixed for different H. Intuitions from convex optimization suggest that dense mini-batch SGD with a larger LR will be better than Local SGD in some of the experiments (Figures 1, 2(a), 2(e), 2(f)).
>
> **A**:
> 1. Our comparison is fair for our purpose, but the reviewer may have misunderstood it. We aim to understand under what conditions (and why) we can improve generalization **by merely increasing the number of local steps $H$** (while keeping the other hyperparameters unchanged). In other words, given a set of hyperparameters of SGD that is either tuned or not tuned, we want to know whether Local SGD with the same hyperparameters except $H>1$ generalizes better than SGD. For this reason, we always keep LR unchanged when changing $H$. In our experiments, e.g., Figure 1, we tune LR for SGD and then directly use the same LR for Post-local SGD (see Appendix K.1 for details). Tuning LR for $H > 1$ can only make Local SGD generalize better but not worse.
>
> 2. The reviewer's intuition about the large LR is not applicable to training deep neural nets. For making the training loss small in the end, it is necessary to use small LR either all through training or at least decay the LR late in training. In our experiments, Figures 2(f) and 3(d) show that SGD outperforms Local SGD when $\eta=1.6$ but SGD attains the best test accuracy for $\eta=0.064$, where Local SGD generalizes better. In Figure 2(e), the best LR for SGD is $0.5$ and Local SGD with the same LR also generalizes better.
>
> 3. To address the reviewer's doubt about Figure 2(a), we conduct additional experiments for SGD with various learning rates and none of them beat Local SGD (see Figure 4 (a)).
>
> **Main Question 2**: The interesting question is why Post-local SGD generalizes better than dense mini-batch SGD, even if it potentially uses a non-optimal LR (Lin et al., 2020). The paper does not answer this question because it only identifies which hyperparameters make Local SGD look good.
>
> **A**: Our empirical and theoretical findings can indeed help to explain the generalization benefit of Post-local SGD (Lin et al., 2020).
> As clarified above, the reviewer may have missed the point that we transfer the hyperparameters of SGD to Post-local SGD, not the other way around. Our findings should be interpreted as follows: if LR of SGD is small enough and the training time budget is sufficient, then switching to Local SGD is beneficial for generalization.
> It is **not possible** to show that Post-Local SGD **always** generalizes better than dense mini-batch SGD because Local SGD is already known to perform worse in some cases (see Section 2.1). To explain the success of Post-Local SGD, we have to identify which hyperparameters of SGD lead to worse performance than Local SGD.

---

> > ### Author Response · Authors · 2022-11-15
> > **Response to Reviewer QgEL (for Major Concerns)**
> >
> > **Major Concern 1**: Comparison to large mini-batch SGD is missing but it is a strong baseline in convex optimization.
> >
> > **A**: For achieving good generalization of deep neural nets, large mini-batch SGD is not a strong baseline because of the following reasons.
> >
> > 1. Large mini-batch SGD (bs=$HB$) is no better than the dense mini-batch SGD (bs=$B$) baseline since it is well-established in the literature that the generalization performance typically degrades for larger batch sizes. See Appendix A in the paper for a survey of empirical and theoretical works on understanding and resolving this phenomenon.
> > We also conduct additional experiments for mini-batch SGD for various total batch sizes. Unsurprisingly, Figure 4 (b) shows that large mini-batch SGD performs no better than dense mini-batch SGD and worse than Local SGD.
> >
> > 2. We also want to emphasize that all these convex optimization results are analyzing the convergence of the training loss, but our paper is studying a fundamentally different question: how do different training algorithms (SGD v.s. Local SGD) pick different global minimizers with different generalization behaviors? Note that the generalization benefit of Local SGD exists even when the training loss is already very small (e.g., the late phase of training in Post-Local SGD), so analyzing loss convergence is not enough.
> >
> > **Major Concern 2**: The theoretical setup assumes independent gradient noise. However, in experiments, (D1) the training data is split between workers, so the datasets are heterogeneous among different workers; (D2) SGD is performed by sampling without replacement.
> >
> > **A**: You are right about the discrepancy between the theoretical setup and experiments, but we argue it is a minor issue.
> >
> > 1. We first clarify the training procedure of Local SGD in our experiments because the reviewer could have misunderstood it. Following Lin et al. (2020), Ortiz et al. (2021), we split data between machines but this is done repeatedly for **every epoch**. An alternative view is that the workers always share the same dataset. For each epoch, they perform local steps by sampling batches of data without replacement until the dataset contains too few data to form a batch. Then another epoch starts with the dataset reloaded to the initial state.
> >
> > 2. The data heterogeneity issue (D1) **does not exist** in our experiment setting because every data point has an equal probability to be sampled by each worker in every epoch.
> >
> > 3. To address the reviewer's concern about (D2), we conduct additional experiments for sampling with replacement and show that Local SGD is still better (see Figure 4(c)). We believe that the reasons for better generalization of Local SGD with either sampling scheme are similar and leave the analysis for sampling without replacement for future work.
> >
> > 4. In fact, most SGD analyses in the literature (e.g., Li et al. (2021), Zhang et al. (2020), Wang et al. (2019)) focus on sampling with replacement for the sake of technical simplicity, yet they lead to meaningful results to practical training methods without replacement.

---

> > > ### Author Response · Authors · 2022-11-15
> > > **Response to Reviewer QgEL (for Minor Concerns)**
> > >
> > > **Minor Concern 1**: The observation that the optimal learning rate depends on the number of local steps $H$ is not surprising, given the optimization literature on local SGD in both convex and non-convex settings.
> > >
> > > **A**: We study the effect of $H$ on generalization when fixing LR and the total number of steps, not the optimal LR for convergence given a fixed $H$. Our empirical and theoretical results suggest that for small LR, using $H>1$ (Local SGD) improves upon $H=1$ (SGD), and that the optimal $H$ is larger for smaller LR.
> > > To the best of our knowledge, none of the existing convergence bounds can be applied here to show that using $H > 1$ (Local SGD) instead of $H = 1$ (SGD) could ever be beneficial for optimization, because these bounds always get worse (or at least stay unchanged) as $H$ increases.
> > >
> > >
> > > **Minor Concern 2**: The stopping criteria of SGD and Local SGD are not specified.
> > >
> > > **A**: For both SGD and Local SGD, we stop training after a fixed number of epochs. This number is picked to be large enough so that SGD can achieve a satisfying test accuracy in the end. In Figure 1, we run 500 epochs for CIFAR-10 and 150 epochs for ImageNet. For Figures 2(a)-(c), we let all algorithms run for a large number of epochs to observe the long-term behavior. For Figure 2(d)-(f), we use the same training budget as that of Figure 1.
> > >
> > > **Minor Concern 3**: The small LR regime $\eta = O(1/H)$ is not interesting because Local SGD should behave similarly as large mini-batch SGD (Karimireddy et al., 2020), hence no generalization benefit.
> > >
> > > **A**:
> > > In the small LR regime, Local SGD does generalize better than large mini-batch SGD. As shown in experiments Figure 4 (b), the test accuracies are ordered as large mini-batch SGD $\approx$ dense mini-batch SGD <  Local SGD. See also our response to Major Concern 1.
> > > As mentioned in our response to Major Concern 1, our paper is studying the generalization behavior of Local SGD. The analyses of Karimireddy et al. (2020) are for the convergence rate of the training loss. But the same level of training loss does not imply similar generalization performance.
> > >
> > > **References**
> > > - Wang, J., & Joshi, G. (2019). Adaptive communication strategies to achieve the best error-runtime trade-off in local-update SGD. Proceedings of Machine Learning and Systems.
> > >
> > > - Li, Z., Malladi, S., & Arora, S. (2021). On the validity of modeling sgd with stochastic differential equations (sdes). Advances in Neural Information Processing Systems.
> > >
> > > - Zhang, J., Karimireddy, S. P., Veit, A., Kim, S., Reddi, S., Kumar, S., & Sra, S. (2020). Why are adaptive methods good for attention models?  Advances in Neural Information Processing Systems.
> > >
> > > - Karimireddy, S.P., Kale, S., Mohri, M., Reddi, S., Stich, S. &amp; Suresh, A.T.. (2020). SCAFFOLD: Stochastic Controlled Averaging for Federated Learning. Proceedings of the 37th International Conference on Machine Learning.

---

> ### Author Response · Authors · 2022-11-27
> **We believe the main concerns have been addressed**
>
> In summary, our responses to the main concerns are as follows:
>
> 1. The reviewer suspects that our hyperparameter tuning is unfair for Local SGD and SGD. However, we indeed tune the LR of SGD and study why and when the generalization of SGD can be improved by merely adding local steps (while keeping other hyperparameters unchanged).
>
> 2. The reviewer recommends that we compare Local SGD with the large mini-batch SGD because it is a strong baseline for convex optimization. We have conducted new experiments showing that the generalization of large mini-batch SGD in deep learning is no better than dense mini-batch SGD and worse than Local SGD.
>
> 3.  The reviewer points out that our theory assumes independent gradient noise but our experiments use heterogeneous datasets across different workers and sample batches without replacement. However, we indeed use the same dataset for different workers. We argue that the discrepancy between sampling with and without replacement is minor, as is standard in previous analyses (Wang et al., 2019; Li et al., 2021; Zhang et al., 2020). The analysis for sampling without replacement is left for future work.
>
> We believe that our response has addressed the reviewer’s concerns, but we are more than happy to answer any further questions.
>
> **References**
> - Wang, J., & Joshi, G. (2019). Adaptive communication strategies to achieve the best error-runtime trade-off in local-update SGD. Proceedings of Machine Learning and Systems.
> - Li, Z., Malladi, S., & Arora, S. (2021). On the validity of modeling sgd with stochastic differential equations (sdes). Advances in Neural Information Processing Systems.
> - Zhang, J., Karimireddy, S. P., Veit, A., Kim, S., Reddi, S., Kumar, S., & Sra, S. (2020). Why are adaptive methods good for attention models? Advances in Neural Information Processing Systems.

---

> ### Author Response · Authors · 2022-11-30
> **Reminder to Reviewer QgEL**
>
> Dear reviewer,
>
> Thank you for your effort in reviewing our paper! We wonder whether you have gotten a chance to read our response and revision? We believe that the score 5 does not accurately reflect the quality of our work. If our response has resolved your concerns, could you please kindly raise your score? Or if you have any further questions, please let us know. We will be more than happy to answer them.
>
> Thanks!

---

> > ### Comment · Reviewer_QgEL · 2022-12-07
> > **Response (1/2)**
> >
> > First, I apologize for the late response and thank the authors for a detailed rebuttal. I am convinced with the happy with the explanation provided for major concern 2 and minor concern 2. Now let me address main question 1:
> >
> > > Our comparison is fair for our purpose, but the reviewer may have misunderstood it. We aim to understand under what conditions (and why) we can improve generalization by merely increasing the number of local steps $H$ (while keeping the other hyperparameters unchanged).
> >
> > I understood what the paper is trying to do correctly the first time.
> >
> > > In other words, given a set of hyperparameters of SGD that is either tuned or not tuned, we want to know whether Local SGD with the same hyperparameters except  $H>1$ generalizes better than SGD.
> >
> > I just **don't think this is an interesting problem** to study. Almost never in practice would one tune the local steps of a their method for some fixed learning rate. From a systems perspective the choice of using local SGD comes from the difference in computation and communication times. Thus more often than not, the ideal communication graph is given to us, and the goal is to look for the best algorithms given those constraints. This is precisely why large mini-batch and local SGD are the correct algorithms to compare with each other: they use the same number of oracle queries and communication rounds. Thus the title of Lin et al. and the experiments within.
> >
> > > The reviewer's intuition about the large LR is not applicable to training deep neural nets. For making the training loss small in the end, it is necessary to use small LR either all through training or at least decay the LR late in training....
> >
> > This is a reasonable point, and I think I concede that with a fixed step-size tuning for optimal step-size separately for both the algorithms, local SGD will indeed improve over SGD. Unfortunately, it requires looking at different figures to conclude this. And what I really wanted was a single graph where this can be seen for different algorithms using their optimal constant step-size. From a practical point of view, on bigger data-sets a warmup and decay phase is almost inevitable to get the best test accuracy. And the comparison is much harder to do fairly there, which is why Lin et al. use the optimal base learning rate for MB-SGD and use it for post-local SGD with the same warm-up and decay schedule.
> >
> > > To address the reviewer's doubt about Figure 2(a), we conduct additional experiments for SGD with various learning rates and none of them beat Local SGD (see Figure 4 (a)).
> >
> > These look good.
> >
> > Going to main question 2.
> >
> > > Our empirical and theoretical findings can indeed help to explain the generalization benefit of Post-local SGD (Lin et al., 2020). As clarified above, the reviewer may have missed the point that we transfer the hyperparameters of SGD to Post-local SGD, not the other way around. Our findings should be interpreted as follows: if LR of SGD is small enough and the training time budget is sufficient, then switching to Local SGD is beneficial for generalization.
> >
> > This is indeed helpful. If one looks at the ideal step-size for dense or large MB-SGD and then uses the same step size for post local SGD with some switching time and consistently shows that local SGD outperforms MB-SGD, then that is an interesting observation. In hindsight, I think the paper's framing of this observation is incorrect. Instead of saying when learning rates are "small" local SGD has an improvement, the paper should rather say "in regimes when even the optimal learning rate for MB-SGD is small local SGD can offer improvement without needing to tune further". I believe re-writing the main claims of the paper and emphasizing this will indeed improve the paper.  As already mentioned in my original review my concern was that the paper views learning rate and local steps as separate hyperparameters. But this explanation doesn't require that view anymore. Essentially the paper has identified regimes where even after giving MB-SGD the benefit of doubt one can hope for local SGD to lead to better generalizing solutions. **I will increase my score conditioned on the authors re-writing and re-emphasizing their findings with this view**. I don't believe it hurts the original story in the paper either.

---

> > ### Comment · Reviewer_QgEL · 2022-12-07
> > **Response (2/2)**
> >
> > >  It is not possible to show that Post-Local SGD always generalizes better than dense mini-batch SGD because Local SGD is already known to perform worse in some cases (see Section 2.1). To explain the success of Post-Local SGD, we have to identify which hyperparameters of SGD lead to worse performance than Local SGD.
> >
> > This sentence doesn't make sense to me given post-local SGD is not a fixed algorithm and depending on the switching times interpolates between dense MB-SGD and local SGD.
> >
> > > Large mini-batch SGD (bs=$HB$) is no better than the dense mini-batch SGD (bs=$B$) baseline since it is well-established in the literature that the generalization performance typically degrades for larger batch sizes. See Appendix A in the paper for a survey of empirical and theoretical works on understanding and resolving this phenomenon. We also conduct additional experiments for mini-batch SGD for various total batch sizes. Unsurprisingly, Figure 4 (b) shows that large mini-batch SGD performs no better than dense mini-batch SGD and worse than Local SGD.
> >
> > I am well aware of these papers, but the first statement isn't true for all batch sizes. Yes, it does make sense when communication is very expensive, and one needs to pick a very large $B$. Not all experiments in this paper are in that regime.
> >
> > > We also want to emphasize that all these convex optimization results are analyzing the convergence of the training loss, but our paper is studying a fundamentally different question: how do different training algorithms (SGD v.s. Local SGD) pick different global minimizers with different generalization behaviors? Note that the generalization benefit of Local SGD exists even when the training loss is already very small (e.g., the late phase of training in Post-Local SGD), so analyzing loss convergence is not enough.
> >
> > This is false. As mentioned in my original paper, most of these papers are taking the stochastic optimization view, i.e., single pass SGD without replacement sampling from the true distribution. Thus, the results are generalization guarantees.
> >
> > > In the small LR regime, Local SGD does generalize better than large mini-batch SGD. As shown in experiments Figure 4 (b), the test accuracies are ordered as large mini-batch SGD  dense mini-batch SGD < Local SGD. See also our response to Major Concern 1. As mentioned in our response to Major Concern 1, our paper is studying the generalization behavior of Local SGD. The analyses of Karimireddy et al. (2020) are for the convergence rate of the training loss. But the same level of training loss does not imply similar generalization performance.
> >
> > Again, instead of the small learning rate regime, look at data sets/tasks where a small learning rate is optimal for MB-SGD, and adding local steps without additional tuning can still improve generalization. I will re-emphasize that's the correct view to take here.

---

> ### Author Response · Authors · 2022-12-09
> **Response to Reviewer QgEL’s new comments (1/2)**
>
> We sincerely thank the reviewer for helpful suggestions and for raising the score from 5 to 6. Below we respond to the reviewer’s new comments in detail.
>
> **New Comment 1**: The paper should rephrase the empirical finding. Instead of saying when learning rates are "small" local SGD has an improvement, the paper should rather say "in regimes when even the optimal learning rate for MB-SGD is small local SGD can offer improvement without needing to tune further".
>
> **A**: We thank the reviewer for the constructive suggestion. As the reviewer’s new comment came after the deadline for updating the rebuttal revision, we are unable to update the paper right now, but we promise to highlight the reviewer’s rephrasing of our main finding in the next revision of our paper.
>
> Instead of replacing our current statement, we will include it as a remark right below our finding because our finding is a stronger improvement guarantee that holds even if the learning rate of SGD is not optimally tuned. Note that this is also of high practical interest to study because tuning hyperparameters to the optimal values could be time-consuming and less realistic in large-scale training.
>
> The remark we will add is stated below
>
> **Remark**: In practice, the learning rate of SGD is usually tuned for the best training loss/validation accuracy given a fixed training budget. In this case, our finding suggests that If the tuned learning rate is small and the training time is long enough, then Local SGD can offer improvement by merely adding more local steps. With the learning rate tuned specifically for Local SGD, the improvement could become even more significant.
>
> In the text following the remark, we will use Figures 2(e), 2(f), and 3(d) to exemplify this view, as we have done in the initial rebuttal.
>
> **New Comment 2**: The number of local steps $H$ is usually determined from a system perspective. The problem of concern is to seek the best algorithm given the computation graph.
>
> **A**: We agree that the case with given $H$ is a scenario of interest, but we believe that our setting with tunable $H$ is also important. As pointed out by Lin et al. (2020), there are two main scenarios in parallel training: (1) the communication-restricted setting and (2) the regime of poor generalization of large-batch SGD. They proposed Post-local SGD to address the generalization issue of large batch training in the second scenario. Our paper follows this line and studies why the generalization performance can be improved by merely adding local steps. We are also very interested in the first scenario, where the optimal learning rate should be figured out for each $H$. But to the best of our knowledge, the optimal choice for the best generalization is poorly understood in theory even for the standard SGD ($H=1$). Therefore, we leave the study of the first scenario for future work.
>
>
> **New Comment 3 (minor)**: The sentence “It is not possible to show that Post-Local SGD always generalizes better than dense mini-batch SGD because Local SGD is already known to perform worse in some cases (see Section 2.1)” does not make sense since the flexibility in the choice of switching time allows Post-local SGD to interpolate between SGD and Local SGD.
>
> **A**: We would like to clarify our point that Post-local SGD does not always generalize better than SGD.  As discussed in Section 2.1, Post-local SGD can perform worse if the switching time is not set properly. In the view that Post-local SGD = Local SGD from a pre-trained model, the switching time of Post-local SGD determines the learning rate, training time, and initial point of Local SGD. To understand how to set an appropriate switching time for Post-local SGD, it is thus important to study the training regime where switching from SGD to Local SGD is beneficial.
>
> **New Comment 4 (minor)**: Not all experiments in this paper are in the regime where enlarging the batch size leads to no better test accuracy.
>
> **A**: We would like to clarify that all our experiments are in the regime where the total batch size $B$ is large and further enlarging $B$ leads to no better test accuracy. Specifically, all our CIFAR-10 experiments in Figures 1 to 3 use $B=4096$ and as shown in Figures 4 (b), further increasing B does not lead to better test accuracy. All our ImageNet experiments use $B=8192$. Figure 1 in Goyal et al. (2017) suggests that further increasing the batch size leads to test accuracy drop.

---

> ### Author Response · Authors · 2022-12-09
> **Response to Reviewer QgEL’s new comments (2/2)**
>
> **New Comment 5 (minor)**: The convex optimization papers mentioned in the original review already give generalization guarantees since most of them consider single-pass SGD without replacement sampling from the true distribution.
>
> **A**: We would like to clarify that we are studying overparameterized neural nets where the number of model parameters is much larger than the training set size and the training loss is highly non-convex. This is highlighted in our theory by assuming a manifold of minimizers instead of just one. The generalization behavior of commonly used optimization algorithms (e.g., SGD) in the overparameterized setup is fundamentally different from that in the convex optimization setup, because the implicit regularization effects of training algorithms can greatly influence the generalization. Therefore, the results in the convex optimization papers do not suffice to explain the generalization of overparameterized neural nets.
>
> -------------
> We will be more than happy to revise our paper if you have any further suggestions. If our response addresses your comments to a satisfactory level, we would like to ask you to kindly raise your score.
>
> **References**
>
> Goyal, Priya, et al. "Accurate, large minibatch sgd: Training imagenet in 1 hour." arXiv preprint arXiv:1706.02677 (2017).

---

### Official Review · Reviewer_KHJA · 2022-10-26

**Confidence:** 3
**Correctness:** 4
**Technical Novelty And Significance:** 3
**Empirical Novelty And Significance:** Not applicable
**Recommendation:** 8

**Clarity, Quality, Novelty And Reproducibility:**

The claims are clear and also there is a clear relation between the claim and the provided theoretical analysis. They provide novel and simpler ways to analyze the proposed SDE compared to existing theoretical results by adding a mild assumption.


**Strength And Weaknesses:**

They provide novel and simpler ways to analyze the proposed SDE compared to existing theoretical results by adding a mild assumption.


**Summary Of The Paper:**

Local SGD (LSGD) is a communication-efficient variant of SGD for large-scale training, where multiple GPUs perform SGD independently and average the model parameters periodically. It has been shown previously that LSGD generalizes better when with a small enough learning rate and sufficient training time. To analyze this observation theoretically,  this paper introduces a new SDE that approximates LSGD in continuous time and compares it with the SDE of SGD. This comparison shows that LSGD has a stronger drift term which results in a stronger effect of regularizer and that leads to a faster reduction of sharpness.

**Summary Of The Review:**

comments
1- Although remark 3.1 says that the analysis holds when SGDL starts out of zero loss manifold, however in the thm 3.2 it is needed that \zeta(0) should be in the manifold. Where does this discrepancy come from?

2- Since thm 3.2 needs that \zeta(0) should be in the manifold, isn’t true that the presented analysis is more related to the Post LSGD?

3- It would be more justified if you add some empirical results to the main body of the paper about your hypothesis 3.1.

Minor comments:
For Fig 2, what is \eta_1 and shouldn’t that be 0.32? Also, plots (e) and (f) don’t have proper legend i.e. not clear what are the values corresponding to each color.

---

> ### Author Response · Authors · 2022-11-15
> **Response to Reviewer KHJA**
>
> We sincerely thank the reviewer for the positive review. We respond to the reviewer's comments as follows.
>
> **Major comment 1**: Remark 3.1 says that the analysis holds when Local SGD starts out of the minimizer manifold but Theorem 3.2 requires $\zeta(0)$ to be on the manifold. Where does this discrepancy come from?
>
> **A**: In Theorem 3.2, Local SGD also starts out of the minimizer manifold. We would like to clarify that our theory uses a random process $\\{\zeta(t)\\}$, which stays on the manifold by definition, to approximate the trajectory of **the gradient flow projections** of Local SGD iterates. Theorem 3.2 sets the initial point of $\\{\zeta(t)\\}$ (i.e., $\zeta(0)$) as the gradient flow projection of the starting point of Local SGD and provides weak approximation guarantees. Theorem 3.3 further states that Local SGD iterates are close to their gradient flow projections.
>
> **Major comment 2**: Is the analysis more related to Post-local SGD since Theorem 3.2 requires $\zeta(0)$ to be in the manifold?
>
> **A**: No. As clarified in the previous paragraph, $\\{\zeta(t)\\}$ approximates **the gradient flow projections** of  Local SGD iterates, not the iterates themselves. Theorem 3.2 only requires that the gradient flow projection of the starting point of Local SGD is on the manifold $\Gamma$. That is, the gradient flow starting from the starting point converges to $\Gamma$. As discussed in Remark 3.1, this requirement is rather mild since $\bar{\theta}_0$ can be arbitrarily away from $\Gamma$, which connects to the observation that pre-training is not necessary for Local SGD to yield generalization benefits.
>
> **Major comment 3**: It would be more justified if some empirical results are added to the main body of the paper about Hypothesis 3.1.
>
> **A**: We thank the reviewer for the helpful suggestion. Hypothesis 3.1 cannot be validated directly because the exact computation of the SDEs can be time-consuming. However, we can check if Hypothesis 3.1 correctly predicts the relative order of the generalization performances of two different algorithms that can be approximated by the $(\kappa_1, \kappa_2)$-Slow SDE and $(\kappa_1, \kappa’_2)$-Slow SDE respectively. Besides predicting the relative order for Local SGD v.s. SGD, Hypothesis 3.1 can predict that longer training time leads to better generalization in training with label noise regularization (“Example” in Section 3.3). The experiments in Figure 5 validate this prediction. More specifically, the diffusion term is zero in this case, and training for $M$ times more step corresponds to making $\kappa_2$ $M$ times larger (after a time rescaling to make the continuous time constant). This enables Hypothesis 3.1 to predict the relative order of the generalization of training with shorter and longer training time.
>
> **Minor comment 1**: In Figure 2 (a), $\eta_1$ is not clearly specified and Figures 2 (e)-(f) don’t have proper legend.
>
> **A**: In Figure 2 (a), $\eta_1$ is the learning rate before the first learning rate decay (from epoch 1 to 50). In Figure 2 (e)-(f), different colors stand for different values of learning rates. We are sorry for the confusion and have made it more clear in the updated draft.

---

### Official Review · Reviewer_ELvh · 2022-10-27

**Confidence:** 2
**Correctness:** 3
**Technical Novelty And Significance:** 3
**Empirical Novelty And Significance:** 3
**Recommendation:** 8

**Clarity, Quality, Novelty And Reproducibility:**

The paper provides novel theoretical and empirical results, which are clear and easy to follow.

**Strength And Weaknesses:**

Strength:
1. This paper uses the derivation of an SDE to explain why and under what condition local SGD generalizes better than SGD.
2. Empirical results are also provided to show that small learning rate + long enough training time + carefully tuned $H$ results in generalization improvement over SGD.
3. The theoretical and empirical results provides some general guidelines and intuition of using post-local SGD, which is easy to follow.

Weakness (concerns and questions):
1. I recommend to show the detailed algorithms of SGD, local SGD, and post-local SGD in the paper, appendix is also fine, so that it will be easier for the readers to refer to these algorithms, especially for those who are not familiar with local SGD and post-local SGD. I believe post-local SGD is much less well-known to the general readers.
2. The theoretical and empirical analysis focuses on learning rate, total number of steps, and $H$. I wonder if the SDE analysis could also show how the global batch sizes affects the generalization of local SGD.

**Summary Of The Paper:**

This paper uses the derivation of an SDE to explain why and under what condition local SGD generalizes better than SGD. Empirical results are also provided to show that small learning rate + long enough training time + carefully tuned $H$ results in generalization improvement over SGD.

**Summary Of The Review:**

This paper uses the derivation of an SDE to explain why and under what condition local SGD generalizes better than SGD. Empirical results are also provided to show that small learning rate + long enough training time + carefully tuned $H$ results in generalization improvement over SGD. I wonder if the batch size also plays an important role in the SDE analysis.

---

> ### Author Response · Authors · 2022-11-15
> **Response to Reviewer ELvh**
>
> We sincerely thank the reviewer for acknowledging that our theoretical and empirical results are novel and clear. We have included the detailed algorithms for SGD, Local SGD and Post-local SGD in Appendix B.
>
> **Q**: Can the SDE analysis show the effects of global batch size on the generalization of Local SGD?
>
> **A**: We thank the reviewer for pointing out this direction to explore the implications of our theory.
>
> 1. One way to change the global batch size is to change the number of workers while fixing all other hyperparameters. It has been discussed at the end of Section 3 and Appendix F that in this case, adding more workers reduces the diffusion term of the Slow SDE for Local SGD and hence improves generalization.
>
> 2. Another case to highlight is to increase the number of workers $K$ and apply the Linear Scaling Rule given a fixed epoch budget. Specifically, we scale the learning rate $\eta$ proportionally to $B$ and the number of local steps $H$ proportionally to $1/B$, and fix other hyperparameters. By simple calculation, it can be shown that the Linear Scaling Rule does not change the SDE approximation of SGD after time rescaling. By simple calculation, it can be shown that the Linear Scaling Rule does not change the SDE approximation of SGD after time rescaling. In contrast, the drift term of the Slow SDE for Local SGD gets stronger when we increase the global batch size. Therefore, Local SGD will generalize better. We further conduct experiments on CIFAR-10 and ImageNet to verify this prediction. We have included the discussion of this case in Appendix F.

---

> > ### Comment · Reviewer_ELvh · 2022-11-17
> > **reply acknowledged**
> >
> > Thanks for the authors' clarification. I will keep the positive score.
> > However, since I'm no expert in SDE, I will have to rely on the other reviewers' option to make the final decision, though I do think this is a good work providing new insights in local SGD.

---

### Decision · Program_Chairs · 2023-01-20

**Decision:**

Accept: poster

**Justification For Why Not Higher Score:**

I had a very brief look at the paper, and in my quick judgement, the paper is not of sufficient standard to warrant a higher score.

**Justification For Why Not Lower Score:**

All reviewers recommended acceptance, some with a clear margin.

**Metareview: Summary, Strengths And Weaknesses:**

This is a clear acceptance decision as all reviewers suggested acceptance, and with some margin. Congratulations to the authors! Summary of the strengths and weaknesses is not necessary in this very clear case.

[Disclaimer: I did not have the chance to read the paper in detail myself, I am basing my recommendation on the views of the reviewers and the discussion.]

**Note From Pc:**

if the above contains the word "oral" or "spotlight" please see: "oral" presentation means -> notable-top-5% and "spotlight" means -> notable-top-25%. As stated in our emails, we are disassociating presentation type from AC recommendations